



# On modeling the Southern Ocean Phytoplankton Functional Types

Svetlana N. Losa[1, 2], Stephanie Dutkiewicz[3], Martin Losch[1], Julia Oelker[4], Mariana A. Soppa[1], Scarlett Trimborn[1], Hongyan Xi[1], and Astrid Bracher[1, 4]

[1]Alfred Wegener Institute Helmholtz Centre for Polar and Marine Research, Bremerhaven, Germany
[2]Shirshov Institute of Oceanology, Russian Academy of Sciences, Moscow, Russia
[3]Massachusetts Institute of Technology, Cambridge, Massachusetts, USA
[4]Institute of Environmental Physics (IUP), University of Bremen, Bremen, Germany

**Correspondence:** Svetlana Losa (Svetlana.Losa@awi.de)

**Abstract.** This study highlights recent advances and challenges of applying coupled physical-biogeochemical modeling for investigating the distribution of the key phytoplankton groups in the Southern Ocean, an area of strong interest for understanding biogeochemical cycling and ecosystem functioning under present climate change. Our simulations of the phenology of various Phytoplankton Functional Types (PFTs) are based on a version of the Darwin biogeochemical model coupled to the

Massachusetts Institute of Technology (MIT) general circulation model (Darwin-MITgcm). The ecological module version was adapted for the Southern Ocean by: 1) improving coccolithophores abundance relative to the original model by introducing a high affinity for nutrients and an ability to escape grazing control for coccolithophores; 2) including two different (small *vs.* large) size classes of diatoms; and 3) accounting for two distinct life stages for *Phaeocystis* (single cell *vs.* colonial). This new model configuration describes best the competition and co-occurrence of the PFTs in the Southern Ocean. It improves

significantly relative to an older version the agreement of the simulated abundance of the coccolithophores and diatoms with *in situ* scanning electron microscopy observations in the Subantarctic Zone as well as with *in situ* diatoms and haptophytes (including coccolithophores and *Phaeocystis*) chlorophyll $a$ concentrations within the Patagonian Shelf and along the Western Antarctic Peninsula obtained by diagnostic pigment analysis. The modeled Southern Ocean PFT dominance also agrees well with satellite-based PFT information.

## 1   Introduction

The Southern Ocean is one of the most important regions in regulating climate via the sinking of $CO_2$ and, at the same time, is a region with the dynamics evidently altered by present climate change. The climatic changes in the Southern Ocean environmental conditions affect the spatial distribution of phytoplankton (Deppeler and Davidson, 2017). The phenology and dominance of different phytoplankton functional types (PFTs) sustaining the marine food web affect the diversity of higher

trophic levels. Playing distinct roles in biogeochemical cycling, PFTs also determine how and on which spatial and temporal scales the ocean mediates climate. The term "key Plankton Functional Types" was proposed for the ocean biogeochemical modelling community in the study by Le Quéré et al. (2005). The specific biogeochemical role of particular oceanic plankton (bacteria, phyto- and zoo-plankton) was one of the criteria chosen for distinguishing key plankton types among/across different





size (pico-, nano-, micro-), trophic and taxonomic groups. For instance, the importance of diatoms, the main phytoplankton

silicifiers, is related to the silica cycling and their high efficiency of carbon export through grazing, direct sinking of single

cells, and through mass sedimentation events (Le Quéré et al., 2005). The extent to which diatoms influence the ocean's

biological carbon pump depends on the silicification determined by size (from nano- to micro-) diversity within this functional

type (Tréguer et al., 2018). Coccolithophores, the main planktonic calcifiers, make a major contribution to the total content of

particulate inorganic carbon in the oceans (Milliman, 1993; Ackleson et al., 1988; Rost and Riebesell, 2004) through production

and release of calcium carbonate plates (coccoliths), and, therefore, also impact the alkalinity of the ocean. *Phaeocystis* as a

dimethyl sulfide producer alters the atmospheric sulfur cycle and similar to diatoms forms prominent blooms in the Southern

Ocean and supports export production. In the global ocean, the pico-phytoplankton group of N-fixers regenerates nutrients

and, therefore, influence the marine recycled production (Waterbury et al., 1986; Morán et al., 2004). The biogeochemical role

of pico-autotrophs including small eukaryotes and photosynthetic bacteria (*Prochlorococcus*, *Synechococcus*) is defined as a

substantial contributor to primary production in the oligotrophic regions and High Nutrients Low Chlorophyll areas.

The other two criteria considered by Le Quéré et al. (2005) for defining the key plankton functional types were directly

related to modeling requirements: 1) the biomass and distribution of the key planktonic type should be controlled by distinct

environmental conditions (hydrography, nutrients) and physiological parameters; 2) the dynamics of particular planktonic type

should impact distinctly the distribution of others (Le Quéré et al., 2005). For instance, while never known to be dominant in

the Southern Ocean, pico-phytoplankton occupying its particular ecological niche might alter the distribution extent of larger

phytoplankton groups. Thus, the term "functional" contains at least three aspects: 1) how the plankton type relates to the

biogeochemical processes; 2) how it adapts to the environmental conditions; and 3) how it alters biogeography of others. The

set of key Plankton Functional Types (PFTs) defined by Le Quéré et al. (2005) includes also heterotrophic bacteria, and three

groups of zooplankton. However, in the ocean color community that has been dealing with detecting photosynthetic plankton

– phytoplankton – the meaning of the abbreviation PFT has been transformed to Phytoplankton Functional Types.

The recognized need of understanding the variability and distribution of different PFTs has motivated some studies on

developing satellite based algorithms for PFTs retrievals (Sathyendranath, 2014; Mouw et al., 2017; Bracher et al., 2017) com-

plementary to the scarce number of available *in situ* observations (Peloquin et al., 2013; Buitenhuis et al., 2013b; Leblanc et al.,

2012; O'Brien et al., 2013; Vogt et al., 2012). In this respect, we have to emphasize that different satellite retrieval algorithms

as well as *in situ* measurements use various approaches not necessarily directly relating PFTs to its biogeochemical role or

functional type, but rather its morphological and/or optical features (Bracher et al., 2017). This fact provoked a discussion on

differences between differentiations among phytoplankton used in observations and assumed in biogeochemical models. In the

review by Bracher et al. (2017), the term "Phytoplankton Group" was introduced allowing to use, and emphasize, if required,

other classifications, for instance phytoplankton size class or taxonomic group. Hereafter, with the abbreviation PFT we refer

to Phytoplankton Functional Types and use the term "Phytoplankton Group" (PG) as a synonyme.

The studies by Le Quéré et al. (2005) and Follows et al. (2007) initiated efforts in numerical biogeochemical modeling

with key PFTs resolved explicitly given the thee aforementioned aspects of functionality. Some of the biogeochemical models

developed for the global ocean represent only two (diatoms and other nano-phytoplankton) or three (plus diazotroph/N-fixer)



PFTs (Hashioka et al., 2013; Vogt et al., 2013; Bopp et al., 2013; Hauck et al., 2013; Schourup-Kristensen et al., 2014;
Laufkötter et al., 2015) and only implicitly account for calcification (except for the study by Buitenhuis et al. 2013a where
coccolithophores were considered as a distinct phytoplankton group). The NASA ocean biogeochemical model (NOBM, Gregg
and Casey 2007; Gregg and Rousseaux 2016) includes four PFTs: diatoms, coccolithophores, cyanobacteria and chlorophytes.
The Darwin ecological model (Follows et al., 2007; Ward et al., 2012; Dutkiewicz et al., 2015) allows to represent more
than three (up to several thousands) phytoplankton groups (including the key types proposed by Le Quéré et al. 2005) in
accordance to their functional type, size class (Ward et al., 2012, 2017) and photophysiological/optical properties (Hickman
et al., 2010; Dutkiewicz et al., 2018). Since the Darwin model offers the highest potential to simulate globally relevant PFTs,
we use this model in our study to set up such a configuration that would allow best to describe the key Southern Ocean
phytoplankton groups but still keeping tractable computationally with only a handful of PFTs. The environmental conditions
related to phytoplankton growth are simulated by the Massachusetts Institute of Technology (MIT) ocean general circulation
model (MITgcm) coupled with a sea-ice model. We investigate: 1) whether this coupled sea-ice – ocean – biogeochemical
model is skillful enough to predict the distribution of diatoms, coccolithophores and *Phaeocystis* over the various dynamical
regions of the Southern Ocean; 2) the prerequisite for plausible simulations of their co-occurrence and temporal evolution;
and 3) how this model data on the Southern Ocean PFTs can complement available *in situ* and satellite observations with a
perspective to obtain long-term information. When determining the model requirements in diversity of assumed PFT traits we
hypothesise on biological factors explaining the observed Southern Ocean PFT biogeography.

The paper is organized as follows. Section 2 describes the numerical model set up, experimental design and observations (*in situ* and satellite retrievals) exploited for model evaluation, Section 3 presents the results and Section 4 the discussion. Section 5 concludes with summary and outlook.

## 2 Method

### 2.1 Numerical Modeling

#### 2.1.1 Biogeochemistry

The Darwin biogeochemical model represents ocean biogeochemical cycling of phosphorus (P), nitrogen (N), carbon (C), silicon (Si) and iron (Fe). Chlorophyll "*a*" (Chla) and carbon are decoupled given the Geider et al. (1998) photophysiological model. The version of the Darwin model used in our study simulates, among total 42 biogeochemical compartments describ-
ing these biogeochemical cycles, two types of zooplankton and six phytoplankton groups. These phytoplankton groups are analogues of diatoms, nano-phytoplankton, prochlorophytes, other pico-phytoplankton (including pico-eukaryotes), nitrogen fixing phytoplankton (including *Trichodesmium*) and coccolithophores. Starting from the setup of Dutkiewicz et al. (2015), to adapt the Darwin model for realistic simulations of the Southern Ocean biogeochemistry and phytoplankton dynamics and diversity, the following steps were performed:





Diatoms were represented by two distinct size classes (as two different model variables): small and "slightly silicified and fast growing" at lower latitudes (introduced instead of "other pico"); large and "strongly silicified slowly growing cells" at high latitudes (Queguiner, 2013);

Assumed coccolithophores physiology accounted for high affinity for nutrients (Paasche, 2001) and ability to escape grazing control (Huskin et al., 2000; Nejstgaard et al., 1997; Losa et al., 2006);

Other nano-phytoplanktonplankton (referred to as "other large" in the original Dutkiewicz et al. 2015) was presented by *Phaeocystis antarctica*. After performing several sensitivity experiments with different physiological traits assumed for the various phytoplankton types and allowing to simulate realistically co-occurrence of coccolithophores and *Phaeocystis*, two phases of the *Phaeocystis* life stages (colonies and solitary cells) were introduced following the parameterization used in Popova et al. (2007) and Kaufman et al. (2017).

Thus, the following six PFTs were considered, large and small diatoms, *Phaeocystis* and coccolithophores, *Proclorococcus*-like and N-fixers. In the current model configuration, instead of exploiting the radiative transfer model accounting explicitly for absorption and scattering of spectrally resolved light as in the version by Dutkiewicz et al. (2015), we used a simplified (because of computational limitations) parametrization of the light in terms of shortwave irradiance ($I$) penetrated over depth ($z$ from vertical model grid level $h_k$ to $h_{k+1}$):

$$I = \int_{h_k}^{h_{k+1}} I_k * e^{-(k_w + k_c Phy + k_{cdom} CDOM)z} dz, \tag{1}$$

where, $k_w$ is the light attenuation coefficient due to water absorption (0.04 m$^{-1}$), $k_c$ (m$^2$ mmol$^{-1}$) denotes attenuation due to phytoplankton ($Phy$, mmol m$^{-3}$), $k_{CDOM}$ represents attenuation by the coloured dissolved organic matter (CDOM, mmol m$^{-3}$). $k_{CDOM}$ is estimated as following:

$$k_{CDOM} = \int_{\lambda_{i=1}}^{\lambda_{i=K}} c_{cdom}(\lambda_0) * e^{-s_{cdom}(\lambda_i - \lambda_0)} d\lambda, \tag{2}$$

here, $c_{cdom}(\lambda_0)$ is the CDOM – specific absorption at reference waveband $\lambda_0$, $s_{cdom}$ is the CDOM spectral slope (the values are taken from the study by Kitidis et al., 2006). As a result of integrating CDOM absorption over the spectral range from $\lambda_1$ = 400 nm to $\lambda_K$ = 700 nm, $k_{CDOM}$ is equal to 10 m$^2$ mmolP$^{-1}$.

Table 1 summarizes specific traits for the simulated PFTs, which is presented by the following physiological parameters: the maximum photosynthetic rate ($P^C_{max}$, day$^{-1}$); the photoinhibition parameter ($\beta$); the growth half-saturation constant ($k_{sat}$,

mmol m$^{-3}$); the biomineralizing function ($mfunc$). These main differences between traits specified alter the growth ($\mu_j$) of the particular phytoplankton (j = 1, 2,..., 6), grazing of phytoplankton by small or micro-zooplanktons ($Gr_{jk}$, k = 1, 2) given the palatability factor ($r_{j,k}$) and sinking ($w_{sink}$, $mday^{-1}$).



*Growth* is parameterized following Geider et al. (1998):

$$\mu_j = P_{mj}^C (1 - e^{-\frac{\alpha_j I \theta_j}{P_{mj}^C}}) \cdot f(\beta), \tag{3}$$

$$P_{mj}^C = P_{max_j}^C \gamma_T \gamma_\eta, \tag{4}$$

$$\alpha_j = \phi_{max_j} \overline{a_j^*}, \tag{5}$$

$$\gamma_\eta = min(\eta_{ji}), \eta_{ji} = f(k_{sat_i}) \tag{6}$$

here $P_{mj}^C$ is the light saturated photosynthesis rate; $\gamma_T$ and $\gamma_\eta$ denote the functions of the growth rate on temperature and limiting nutrients ($\eta_{ji}$, $i = P, N, Si, Fe$), respectively; $\alpha_j$ is the initial slope of the photosynthesis *vs.* irradiance (P-I, Platt et al. 1980) curve, which is a product of the phytoplankton-specific light absorption (considered spectrally averaged, $\overline{a_j^*}$) and the maximum quantum yield of carbon fixation ($\phi_{max_j}$); $\theta_j$ is the simulated chlorophyll to carbon ratio. The $P_{mj}^C$ and $k_{sat_i}$ parameters are specified with the use of empirical allometric relationships (Ward et al., 2012, 2017). Opposed to the study by Gregg and Casey (2007), Gregg and Rousseaux (2016), the $\gamma_T$ function was considered the same for diatom, coccolithophores, *Phaeocystis* and prokaryotes.

*Grazing* is formulated as a Holling II function:

$$Gr_{jk} = g_{max_{jk}} \gamma_T \frac{r_{jk} Phy_j}{G_k} \frac{G_k^2}{G_k^2 + \kappa_{sat_k}^2}, \tag{7}$$

$$G_k = \sum_j r_{jk} Phy_j, \tag{8}$$

where $g_{max_{jk}}$ is the zooplankton maximum grazing rate on phytoplankton, and $\kappa_{sat_k}$ is the half-saturation constant for grazing. *Sinking* is expressed given the phytoplankton-specific sinking rate $w_{sink_j}$ as

$$Phy_{sink} = \frac{\partial w_{sink_j} Phy_j}{\partial z}. \tag{9}$$

The described biogeochemical model configurations given the parameters in Table 1 was exploited for a reference experiment (REF). In an additional experiment (PHAEO), two *Phaeocystis* life stages were considered as a function of iron availability (Bender et al. 2018): if the iron concentration is less than the iron half saturation constant ($k_{satFe}$), *Phaeocystis* is assumed to be present as solitary cells with the mortality rate and grazing pressure being higher by 1.3 and 1.25, respectively, than those cells in a colonial form. Following Popova et al. (2007), we considered *Phaeocystis* sinking rate ($w_{sink}$) dependent of available nutrients, but in our case it was limited to iron concentration as following

$$w_{sink}(Phaeo) = w_{sink}(1 - Fe/(Fe + k_{satFe})), \tag{10}$$

$$k_{satFe}(Phaeo_{cell}) = k_{satFe}(Phaeo) * 0.8. \tag{11}$$

Note, that in the model *Phaeocystis* is still the same variable/array, but the assumed morphology and, therefor, physiology different for the different life stages of *Phaeocystis*.



### 2.1.2 Physics

The biogeochemical model was coupled to the MITgcm circulation model (MITgcm Group, 2012) in a configuration based on a cubed-sphere grid (Menemenlis et al., 2008) with mean horizontal spacing of 18 km and 50 vertical levels with the resolution ranging from 10 m near the surface to 450 m in the deep ocean (as in Taylor et al. 2013). The model was forced by 3-hourly atmospheric conditions from the Japanese 55-year reanalysis (JRA55, Kobayashi et al. 2015) over the time of 1999 – 2012 following a spin up period 1992 – 1998. Our configuration of the MITgcm was similar to the configuration used by Taylor et al. (2013) to examine the mechanisms behind the phytoplankton bloom in the Antarctic seasonal ice zone.

The available light and light penetration depend on present ice conditions:

$$Q_{sw,net} = Q_{sw,water} * (1 - c) + Q_{sw,ice} * c, \tag{12}$$

$$Q_{sw,ice} = 0.3 * Q_{sw} * (1 - \alpha_{ice}) e^{-1.5 h_{ice}}, \tag{13}$$

where c is the fractional ice cover, $h_{ice}$ is the ice thickness, $\alpha_{ice}$ is the sea ice albedo. $Q_{sw}$ denotes the incoming shortwave radiation, $Q_{sw,net}$ is the net downward shortwave heat flux as a result of water and ice fractions ($Q_{sw,water}$ and $Q_{sw,ice}$, respectively). However, the light does not penetrate if the ice is covered by snow.

### 2.1.3 Biogeochemical tracers initialisation

To initialise (in 1992) the biogeochemical model variables evolved in cycling N, C, Fe, Si (including inorganic and organic particular and dissolved pools) and Chla, we used the results of the study by Taylor et al. (2013), since these variables are presented in both models, Darwin and the Regulated Ecosystem Model (REcoM, Schartau et al. 2007) used in Taylor et al. (2013). The model variables describing the phosphorus cycle were initialised given N-based variables and the Redfield N:P ratio. The REcoM-based phytoplankton and zooplankton biomasses from Taylor et al. (2013) were distributed equally between six and two Darwin PFTs and zooplankton groups, respectively.

## 2.2 Evaluation with observational data

### 2.2.1 *In situ* observations

As independent information on phytoplankton distribution, we used Chla of diatoms, haptophytes and prokaryotes from a global data set available at https://doi.pangaea.de/10.1594/PANGAEA.875879 (Soppa et al., 2017). The data was derived from *in situ* high precision liquid chromatography (HPLC) pigment measurements using the Diagnostic Pigment Analysis (DPA) following Vidussi et al. (2001) and Uitz et al. (2006) and modified as in Hirata et al. (2011) and Brewin et al. (2015) and adapted to a much larger data set. For more details, we refer the reader to the study by Losa et al. (2017, Supplementary Material, Section 1). Table 2 contains the information about evaluated phytoplankton groups as classified in the model and observations. It is worth mentioning that the DPA allows also to retrieve other phytoplankton groups – like dinoflagellates, cryptophytes and green algae – however, they were not included into the referred data set and not considered in our study.



Figure 1 shows the locations of the available *in situ* data in the Southern Ocean. To support discussions on the biogeography of the observed and simulated PFTs, this figure also depicts several biogeochemical provinces (Longhurst, 1998) distributed over the Southern Ocean: Austral Polar Province (APLR), Antarctic Province (ANTA), Subantarctic Water Ring Province (SANT), South Subtropical Convergence Province (SSTC), Humbold Current Coastal Province (CHIL), Southwest Atlantic

180 Shelves Province (FKLD), Eastern Africa Coastal Province (EAFR), Australia-Indonesia Coastal Province (AUSW), East Australian Coastal Province (AUSE). The system of the Southern Ocean fronts (Orsi et al., 1995) – the Sub-Antarctic Front (SAF), the Polar Front (PF), the Southern Antarctic Circumpolar Current Front (SACCF) and the Southern Boundary of ACC (SBDY) - is also shown. Simulated PFTs were additionally compared to measurements reported by Smith et al. (2017) which were obtained by scanning electron microscopy.

185 **2.2.2 Remote sensing**

Model results were evaluated based on phytoplankton dominating groups from the climatological monthly mean satellite derived product PHYSAT. This product based on the multispectral Sea-Viewing Wide Field-of-View Sensor (SeaWiFS) information was developed by Alvain et al. (2008) and available on http://log.cnrs.fr/Physat-2?lang=fr. We also used the Chla retrieved for three PFTs with differential optical absorption spectroscopy (PhytoDOAS, Bracher et al. 2009; Sadeghi et al.

190 2012) given hyperspectral information from the Scanning Imaging Absorption Spectrometer for Atmospheric Chartography (SCIAMACHY, Bracher et al. 2017; https://doi.org/10.1594/PANGAEA.870486) and the SynSenPFT Chla data product (Losa et al. 2017, https://doi.org/10.1594/PANGAEA.875873). The SynSenPFT product is a combination of PhytoDOAS Chla for diatoms and coccolithophores and OC-PFT Chla for diatoms and haptophytes derived with an abundance-based approach (Hirata et al. 2011, further refined in Soppa et al. 2014) applied to multi-spectral satellite based total Chla from the Ocean

195 Colour Climate Change Initiative (OC-CCI version 2, https://rsg.pml.ac.uk/thredds/catalog-cci.html, OC-CCI 2015). While PhytoDOAS product from the SCIAMACHY sensor are only available at 0.5° horizontal resolution as monthly mean Chla, the SynSenPFT allows by the combination of PhytoDOAS with OC-PFT multispectral data at a daily 4 km resolution. Respectively, PhytoDOAS and PHYSAT satellite products were derived based on phytoplankton absorption properties captured by the satellite sensors and distinguished by the retrieval algorithms either as a particular PFT optical imprint ("finger print") in

200 case of available hyperspectral information (in PhytoDOAS) or as anomalies in a multispectral signal (in PHYSAT). Thus, the PhytoDOAS allows to retrieve quantitatively major PFTs (coccolithophores, diatoms, cyanobacteria), while PHYSAT provides information about the dominant phytoplankton among five PGs: prokaryotes (presented by *Prochloroccocus* and *Synechococcus*-like SCL), diatoms, haptophytes in general and *Phaeocystis* in particular.



## 3 Results and Discussions

### 3.1 Improved diatom phenology: diversity within diatoms

Figure 2 shows PFT dominance retrieved by PHYSAT and simulated by Darwin-MITgcm Dutkiewicz et al. (2015) version for July and January. It is seen that, in comparison with PHYSAT, modeled diatoms were misrepresented in the Southern Ocean during austral summer. In July simulated diatoms were dominant in the Atlantic section of subantarctic zone, opposed to observed dominance of haptophytes. This disagreement between simulations and satellite observations indicates model deficiencies in representing PFT phenology and distribution. Generally, the model overestimated the dominance of small non-silicified phytoplankton. A series of Darwin-MITgcm experiments, with different model configurations with respect to assumed PFTs and their traits described by various physiological parameters, have been conducted for the global ocean and evaluated for the Southern Ocean over the period of 1999 – 2012. The detailed protocol of the experiments can be found in the Supplementary Material (Section S1). The best simulations, in terms of agreement with observed phytoplankton composition (Trimborn et al., 2015) and dominance (PHYSAT, Alvain et al. 2008) and diatom phenology (Soppa et al., 2016b), were performed with prescribing two size classes of diatoms (large and small, Queguiner 2013; Tréguer et al. 2018) given the parameters in Table 1 but initially without considering *Phaeocystis* in two distinct phases. Figure 2 illustrates the model phytoplankton dominance obtained for July 2003 and January 2004 for this reference (REF) experiment. One can notice a significant improvement in the model PFT dominance agreement with the PHYSAT product as compared to those previously reported (Dutkiewicz et al., 2015). These results suggest that the too early (relative to observations) appearance of diatom blooms in the Southern Ocean simulated by most (global ocean) biogeochemical models (Vogt et al. 2013; as well in the Darwin model set up published in the study by Dutkiewicz et al. 2015) could be explained by lack of inclusion of the size diversity in diatoms (Tréguer et al., 2018).

### 3.2 Co-occurrence of coccolithophores and diatoms

This augmentation of the biogeochemical module by two size classes of diatoms appeared to be a prerequisite for the realistic simulation of the abundance of diatoms and coccolithophores in the Subantarctic Zone. Figure 3 shows the Southern Ocean spatial distribution of Chla for small and large diatoms, coccolithophores and "other large" (*Phaeocystis*-like) phytoplankton in January 2004, which is in agreement with the observations by Signorini et al. (2006), Balch et al. (2016) and Smith et al. (2017). The figure illustrates the co-occurrence of model simulated diatoms and coccolithophores as observed north of the Subantarcic Front (Smith et al., 2017). The results are supported by the PhytoDOAS PFT retrievals from SCIAMACHY hyperspectral information (Bracher et al. 2017, https://doi.org/10.1594/PANGAEA.870486) and SynSenPFT product (Losa et al. 2017, https://doi.org/10.1594/PANGAEA.875873), as presented for the related time frame and region in Losa et al. (2018, https://oceanopticsconference.org/extended/Losa_Svetlana.pdf). However, it is worth emphasizing that model representation of co-existence/competition within the haptophytes group (coccolithophores vs. *Phaeocystis*) still remained a challenge: small changes in the Darwin model physiological parameters led to loss of either *Phaeocystis* or coccolithophores. For instance, in experiment REF to the end of the considered period of time, coccolithophores did not survive.



### 3.3 Differentiation among haptophytes: coccolithophores *vs. Phaeocystis*

To cope with the aforementioned "chaoticity" (Popova et al., 1997) of the system leading to two different states either with coccolithophores or with *Phaeocystis ant.* existent, in experiment PHAEO we introduced additional differences in the traits of

these two PFTs. In particular, we considered two distinct life stages of *Phaeocystis ant.* (colonies and solitary cells) in which its morphological features and physiology depend on iron availability (Bender et al., 2018)). To illustrate the simulated Southern Ocean phytoplankton compositions, we calculated zonally averaged ratios of individual PFT biomass to summed biomass over all simulated PFTs for the following four sectors of the Southern Ocean: the Atlantic Ocean sector (AOS, $60^o$W – $18^o$E), the Indian Ocean sector (IOS, $18^o$E – $120^o$E), Australian sector (AST, $120^o$E – $180^o$E) and the Pacific Ocean sector (POS, $180^o$E

– $60^o$W). Figure 4 presents these meridional distributions of the different PFTs in February 2008 for experiments with and without considering *Phaeocystis* in two different life stages (we refer to these experiments as PHAEO and REF, respectively). One can see that in experiment REF, "other large" (including *Phaeocystis*) outcompetes coccolithophores leading to too low concentrations of coccolithophores south of the Polar Front, while small diatoms exist there in both experiments (however, in different percentages). Experiment PHAEO reveals more plausible meridional distributions of the phytoplankton composition

with respect to the coccolithophores fractions gradually increasing in the direction to the north of the Subantarctic Front, where it reaches ~30% of the total biomass. This result agrees with the estimates of Smith et al. (2017) obtained for January – February – March. Seasonal variations of the PHAEO phytoplankton compositions are shown in the Supplementary Material (Section S4, Figure S8). As seen in this figure, the fraction of coccolithophores is higher in austral winter.

Differences in the biomass distribution between coccolithophores, *Phaeocystis* and diatoms influence zooplankton abun-

dances as prescribed by the model assumptions and parameterisations (Figure 4, blue curve). In this respect, we have to emphasize that the realistic coccolithophores distribution was obtained as a result of reduced grazing pressure on this PFT among other circumstances. This contradicts the study by Nissen et al. (2018), who reported on an increased grazing of coccolithophores as a factor controlling the coccolithophores phenology in the Southern Ocean. However, our results are supported by studies by Nejstgaard et al. (1997), Huskin et al. (2000) and Losa et al. (2006). In the study by Losa et al. (2006) on optimised

biogeochemical parameters the authors showed that the coccolithophores bloom was associated with low grazing pressure. Based on their laboratory experiments, Nejstgaard et al. (1997) and Huskin et al. (2000) concluded that coccolithophores (due to its "stony" structure) do not influence the microzooplankton growth. High affinity of coccolithophores for nutrients (for phosphate and iron to a larger extent than for nitrogen, Paasche 2001) makes them strongly competitive in environmental conditions with declining nutrient concentrations (Paasche, 2001; Iglesias-Rodríguez et al., 2002), for instance under strong ocean

stratifications or nutrient consumption by other PFTs (see Figure 5).

Figure 5 depicts the Chla spatial distribution for diatoms, *Phaeocystis* and coccolithophores for January and March 2004. This figure clearly shows the co-occurrence of diatoms and coccolithophores in the Subantarctic Zone north of the Subantarctic Front, as well as co-existence of simulated diatoms and *Phaeocystis* south of the Polar Front. Figure 5 also presents the spatial distribution of silicon, dissolved iron and phosphate in March 2004 (agreed well with the World Ocean Atlas, Garcia et al.

2014). The regions with high iron concentrations (in the Ross Sea, along the Western Antarctic Peninsula, around the Falkland,



South Georgia and South Sandwich, Crozet and Kerquen Islands) indicates the area of *Phaeocystis ant.* potential existence in colony form. Thus Figure 5 shows that the simulated abundance of coccolithophores north of the Subtropical Front (STF) – where phosphate occurs in very low concentrations – is explained by the introduced high affinity of this PFT to phosphate (small half-saturation rate in $\gamma_\eta$ function) allowing coccolithophores to grow in nutrient depleting conditions. However, in the region

between the Subtropical and Subantarctic Fronts the occurrence of coccolithophores is more evidently linked to low grazing pressure on this PFT, due to its much lower palatability for zooplankton in comparison with small diatoms or *Phaeocystis* presented by single solitary cells. As reported by Smith et al. (2017), the simulated coccocolithophores is distributed over the silica-depleted area. However, in the model world this PFT is not directly constrained by this nutrient, opposed to diatoms.

Figure 6 illustrates the implication of the differences in the distribution among the haptophytes on the carbon cycling as

carbon distributed into different inorganic and organic, particulate and, consequently, dissolved pools. Shown are the partic-ulate inorganic carbon (PIC) and particulate organic matter (POC) for the experiments REF and PHAEO in March 2004. As seen, opposed to REF in the experiment PHAEO we obtained higher concentration of POC south of the Southern Antarctic Circumpolar Current Front but less POC and higher PIC concentrations north of the Subantarctic Front, which is in agreement with (Balch et al., 2005, 2016).

### 3.4 Experiment PHAEO: General evaluation

#### 3.4.1 PHAEO dominance against PHYSAT and SynSenPFT

Similar to Figure 2, Figure 7 depicts model phytoplankton dominance obtained for June - August 2003, December 2003 and January – February 2004 but for the experiment PHAEO. As for the experiment REF, PFT dominance agrees with PHYSAT data product better than the model results of Dutkiewicz et al. (2015). However, the results differ from the REF dominance: In

July and January, the extent of diatom dominance around the Antarctic is wider, further to the north and around the Kerguelen Islands; less extension of the diatom dominance in the Atlantic Section north of SAF, showing even larger agreement to PHYSAT.

Figure 8 presents the spatial distribution monthly means of simulated surface Chla for coccolithophores and diatoms over the region from 30°S to 70°S and from 70°W to 120°E as shown in the study by Smith et al. (2017). These model results

are compared with Chla obtained for the same domain and time period with SynSenPFT algorithm (Losa et al., 2017). The simulated coccolithophores distribution reveals a smaller "belt" in comparison with SynSenPFT with well-agreeing northern boundaries, but not extending further south of the Polar Front. For diatoms, modeled Chla exceeds SynSenPFT estimates south of the Antarctic Circumpolar Current front. In this respect, it is worth emphasizing that SynSenPFT product at the latitudes higher than 60°S is mostly influenced by OC-PFT estimates because of much less available SCIAMACHY information. (see

Supplementary Material, Section S2). OC-PFT retrievals (Losa et al., 2017) contain information generally on haptophytes (not specifically on coccolithophores). Moreover, PhytoDOAS coccolithophores retrievals are based on coccolithophores specific absorption spectrum that is, indeed, very similar to the specific absorption spectrum of *Phaeocystis*. As seen from Figures 5 and 4 model simulations support the evidence of *Phaeocystis* dominance among haptophytes at these latitudes. SynSenPFT




diatom Chla is known to be underestimated for the Antarctic Province (see SynSenPFT PVR 2016, Soppa et al. 2016a).
However, diatom Chla estimates obtained with PhytoDOAS are higher (see Supplementary Material, Section S2) despite the
low coverage of the product.

### 3.4.2 PHAEO against *in situ* observations

For a more precise comparison of the PHAEO model simulations with *in situ* information, Figures 9 – 10 and 11 show
spatial distributions of model surface Chla for diatoms, haptophytes (including coccolithophores and *Phaeocystis*) and pico-
phytoplankton (cyanobacteria or prokaryotes) as (several selected) model snapshots with *in situ* observations available within
a time window ± 1 week. The *in situ* Chla is depicted also at the same figure panels as coloured circles. A complete series
of model snapshots depicting the spatial distribution of Chla for diatoms (large + small), haptophytes and prokaryotes against
*in situ* observations (if available) has been collected over the time period from August 2000 to April 2012 and compiled
as three video files (registered via AV-Portal of the German National Library of Science and Technology (TIB, see Video
supplement). These movies also nicely illustrate the PFT phenology. In Figure 9 and 10, the *in situ* observations (indicated
by circles) match well the representing model Chla of diatoms and haptophytes in the area close to the Antarctic Peninsula
and in the Southwest Atlantic Shelves biogeochemical province (FKLD, Longhurst, 1998), which illustrates a good agreement
between the model and observations. In the Ross Sea, however, the model performance is less accurate for both diatoms and
haptophytes (Figure 10, first panel, see also Vogt et al. 2012). In general, for the entire Southern Ocean the mean absolute
deviation of model Chla from observations (mean absolute error, MAE) is 0.74 mg m$^{-3}$ and 0.22 mg m$^{-3}$ for diatoms and
haptophytes, respectively. The statistical estimates were obtained based on the model and observation match-ups within ± 1
week. Moreover, the model does not explicitly represent sea-ice algae and, therefore, might work less well in the region around
the sea-ice.

As additional information on the agreement between model and observations, in the Supplementary Material Figures S9 and
S10 present frequency distributions of diatoms and haptophytes Chla for the simulations and measurements and the frequency
distribution of the model and data differences. The latter shows that statistical criteria, such as mean absolute error (MAE)
and root mean squared error (RMSE) give statistical meaningful metrics with respect to "model minus *in situ* Chla data" and
the evaluation does not necessarily require a logarithmic transformation, as it is often done in ocean colour product validation
(Brewin et al., 2010; Losa et al., 2017).

Tables 3 and 4 present the statistics of the model to *in situ* data comparison for diatoms and haptophytes Chla at several
Longhurst's biogeochemical provinces (Longhurst 1998, see Figure 1). The worst statistics were obtained for diatoms in the
Atlantic Sector of the ANTA province, where the simulated diatom Chla is systematically overestimated by ~0.5 mg m$^{-3}$. With
respect to the agreement between model and observed *in situ* Chla for pico-phytoplankton depicted in Figures 11 and Figure
S11 (in Supplementary Material) one can conclude that their frequency distributions differ, and the frequency distribution of the
differences confirms that MAE and RMSE given absolute (Table 5) or logarithmically transformed values can hardly provide
satisfactory estimates. Nevertheless, it is worth emphasizing that the largest differences between model and observed *in situ*
pico-phytoplankton are located along the Antarctic Peninsula. In this respect, we have to point out that all the statistics are





presented for a qualitative assessment of the model rather than for a quantitative estimates of model uncertainties, since the representation error (Janjić et al., 2018) related to the differences in spatial and temporal scales considered and sampled by the
model *vs.* observations as well as to the mismatch in grouping phytoplankton (Bracher et al., 2017) are still quite large.

## 4  Concluding remarks and outlook

Leveraging satellite estimates and *in situ* observations allowed us to define the trait requirements for capturing phytoplankton biogeography in the Southern Ocean. We set up a model for simulating the distribution of key Southern Ocean PFTs: diatoms, coccolithophores and *Phaeocystis*. The observed co-occurrence of two different phytoplankton groups, coccolithophores and
diatoms, in the Subantarctic Zone (Queguiner, 2013; Smith et al., 2017) was clearly simulated by the Darwin-MITgcm model adjusted for the Southern Ocean and in a reasonable agreement with PHYSAT, PhytoDOAS and SynSenPFT satellite products.

The Southern Ocean adjustment of the Darwin-MITgcm required introducing two size classes of diatoms (small *vs.* large) and improved coccolithophores physiology (which indeed might support a biochemical/physiological hypothesis on the coccolithophores distribution in the Southern Ocean). The simulated biogeography of coccolithophores was not controlled by
temperature itself, which was also found in the study by Smith et al. (2017). Neither it was directly explained by silica depleting, but phosphate as well as by low palatability of this PFT for grazers. The simulation of co-occurrence of coccolithophores and *Phaeocystis* required additional model developments. Thus, as a first trial, the Darwin model was augmented to account for changes in assumed life stage of *Phaeocystis ant.* (Popova et al., 2007; Kaufman et al., 2017) subject to iron availability (Bender et al., 2018). This parameterization of morphological shifts did indeed allow for co-existence of the two types
of haptophytes. However, there is still room for improvement. For instance, for specifying more precisely the differences in photophysiology and related optical imprints (Moisan and Mitchell, 2018) for *Phaeocystis* in cells and colonies phases. Below we highlight several crucial points that have to be considered in future and, if addressed, could potentially guide further model developments and lead to improved PFTs distribution simulations:

*Phytoplankton growth: equation* (6) ($\phi_{max}$). The differences in the phytoplankton growth are presented mostly by the
variety of the assumed maximum photosynthesis rate ($P^C_{max_j}$) and chlorophyll to carbon ratio $\theta_j$, resulting in slower growth of coccolithophores than for diatoms and *Phaeocystis*, which determines the simulated PFT phenology and competition. The initial slope of the P-I curve ($\alpha$), opposed to the study by Hickman et al. (2010) and Dutkiewicz et al. (2015), was considered identical for all PFTs. The use of PFT specific absorption spectra (Eq. (6)) when setting up the PFT traits allows the initial slope of the P-I curve ($\alpha$) being distinct for particular phytoplankton. However,
realistic representation of the $\alpha$ parameter would also require some differences in the maximum quantum yield of carbon fixation ($\phi_{max}$) specification (Hiscock et al., 2008). This would further improve the model performance (for instance, *Phaeocystis ant.* dominance in the Ross Sea) and would probably bring the assumed $P^C_{max_j}$ values (which are currently, to some extent, overestimated) closer to measurements (*Tables S5, S6 Supplementary Material, Section S4*). However, the $\phi_{max}$ is measured given $\alpha$ and phytolankton specific absorption. That means that in the model world we have to
differentiate between the $\alpha$ parameter for distinct PFTs.



*Prokaryotes:* Even though the *Prochlorophytes*-analogue is not present/dominant in the Southern Ocean, accounting for this pico-phytoplankton is a prerequisite for the simulation of the northern edge of coccolithophores distribution south of the STF (in the SSTC bgc province). In this respect the assumption on photoinhibition for this PFT as well as for other PFTs might need a careful revision.

*Remineralisation and other parameterised processes:* The simulated distribution, competition and co-occurrence of the key Southern Ocean PFTs are generally discussed in terms of differentiating the PFT traits via the specification of phytoplankton growth (with different light acclimation strategies and affinities to nutrients) and palatibilities for zooplankton grazing. However, there are other processes altering the model PFT dynamics. For instance (a model based evidence, not shown), augmenting the model by CDOM affects the remineralisation processes affecting the nutrients distribution and

therefore the spatial and temporal distribution of PFTs competing for the available resources. One should also think of better representation of algae/sea-ice interaction.

Present-day satellite retrieval algorithms allow to detect biomass (and dominance) of some PFTs including haptophytes in general (OC-PFT, Hirata et al. 2011), coccolithophores in particular (PhytoDOAS, Sadeghi et al. 2012), diatoms and cyanobacteria/prokaryotes (OC-PFT; PhytoDOAS, Bracher et al. 2009). Several efforts have been made in deriving more specifically

green algae and dinoflagellates (see Xi et al. 2018; Hirata et al. 2011; Shang et al. 2014). Even though there is still the mismatch between the phytoplankton grouping used in numerical models, satellite algorithms and *in situ* observations, the information from these different sources becomes closer and, indeed, can be considered complementary.

However, when combining or comparing models and observational information, we have to keep in mind representation errors and limitations of approaches used for deriving PFT information from *in situ* and satellite observations. Generally,

a temporal and spatial scale mismatch exists between *in situ* or satellite observations and model output depending on the model discretization. *In situ* measurements in the Southern Ocean are sparse in space and time and only provide a fraction of the information obtained by the model. Satellite observations cover larger areas frequently but only cloud-free scenes which leads to a temporal bias in the often cloud-covered Southern Ocean. In addition, they are limited to only observe the first optical depth, which often limits the detection of the chlorophyll maximum. The development of algorithms for deriving

PFT information requires a large *in situ* dataset with homogeneous temporal and spatial distribution. Nevertheless, scientific cruises in the Southern Ocean are often carried out close to the continents/ice shelf or in regions with high phytoplankton concentration (Figure 1). The diagnostic pigment analysis used to estimate PFTs from HPLC pigments assumes that different PFTs have different marker pigments, but it is known that they can also have pigments in common (Hirata et al. 2011). This ambiguity leads to uncertainties in the *in situ* database which is, on the one hand, needed as fundamental input for the

algorithms of PFT retrievals and, on the other hand, used for direct comparison with model output here. Concerning spectral based methods applied to either *in situ* or satellite data, it is difficult to distinguish the specific absorption spectra of PFTs (e.g., coccolithophores and *Phaeocystis*). These and more limitations are well discussed by Sathyendranath (2014) and Bracher et al. (2017).

The evidence that biogeochemical modelling can capture phytoplankton specific traits in the way that it considers differ-

ent aspects of differentiation among phytoplankton groups – biogeochemical role; allometric, photophysiological and optical parameters; accounting for carbon and Chla decoupling – makes the coupled ocean/biogeochemical models a very valuable and skilful instrument that combines the knowledge from *in situ* measurements and remote sensing by exploiting various PFT retrievals principles used (separately) in these observations and relates it to the environmental conditions. Further extension/progress could be expected given the radiative transfer model coupled to the biogeochemical model Gregg and Rousseaux

(2016) and Dutkiewicz et al. (2018) allowing to simulate spectrally resolved water leaving radiance and therefore providing perspectives to assimilate explicitly multi- and hyper-spectral satellite information and correct if necessary the simulated PFTs as the main water optical constituents.

## 5   Supplementary Material

The supplementary material contains a protocol of prior Darwin sensitivity experiments with differently prescribed phyto-

plankton traits (Section S1); PhytoDOAS diatoms Chla over the Great Calcite Belt (Section S2); seasonal variation of the meridional distribution of zonally averaged phytoplankton composition for four sections of the Southern Ocean (Section S3); *in situ* and laboratory measurement information on the photophysiological parameters of diatoms and *Phaeocystis* (Section S4); and additional information on model evaluation against *in situ* (Section 5). Additionally there are three video files registered via AV-Portal of the German National Library of Science and Technology (TIB), Hannover (doi to be provided).

*Code and data availability.*   The model data were generated with the MIT Darwin Project Biogeochemical, Ecosystem, and OpticalModel. The code is part of the MITgcm, which is available fromhttp://mitgcm.org. The specific version of the code used in this study, as well as initial fields can be provided opon request from the corresponding author (svetlana.losa@awi.de). Information about observational data availability is provided in the text (via specified URL).

*Video supplement.*   The following three video files are available via AV Portal of German National Library of Science and Technology

(TIB, Hannover): Simulated distribution of diatom chlorophyll concentration in the Southern Ocean, https://doi.org/10.5446/42871; Simulated distribution of haptophytes chlorophyll concentration in the Southern Ocean, https://doi.org/10.5446/42873; Simulated distribution of prokaryotes chlorophyll concentration in the Southern Ocean, https://doi.org/10.5446/42872.

*Competing interests.*   The authors declare that they have no conflict of interest.





*Author contributions.* AB: funding and PhySyn project lead; SL: conception and design of the experiments, compilation of the first draft
of the manuscript and updates according critical revisions by all the co-authors; SD, ML, SL: model development and set up; SD, ST and SL: PFT traits; MS, HX: *in situ* PFT Chla data processing and interpretation; AB and JO: satellite retrievals; all co-authors: substantial participation in analysis and discussion of the results and their implications.

*Acknowledgement.* The authors are thankful to the MITgcm group for the MITgcm code, Anna Hickman for the previous efforts in the Darwin model developments, and all the scientists and crews involved in *in situ* HPLC data collection and analyses for providing their pigment

data. The authors thank Eva Alvarez for fruitful and inspiring discussions within the joint project "IPSO" ("Improving the prediction of photophysiology in the Southern Ocean by accounting for iron limitation, optical properties and spectral satellite data information"). The study was supported by the Deutsche Forschungsgemeinschaft (DFG, German Research Foundation) in the framework of the priority program 1158 "Antarctic Research with Comparative Investigations in Arctic Ice Areas" by the "PhySyn" project (Br2913/3-1); and Helmholtz Climate Initiative REKLIM (Regional Climate Change), a joint research project of the Helmholtz Association of German Research Centres (HGF). The

contribution of SL is partly made in the framework of the state assignment of the Federal Agency for Scientific Organizations (FASO) Russia (theme 0149-2019-0015). The model simulations were obtained with resources provided by the North-German Supercomputing Alliance (HLRN).





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





**Table 1.** Biogeochemical model internal parameters/traits settings with *Phaeo* for *Phaeocystis*, *Prochlor* for *Prochlorococcus*, Nfixer for nitrogen fixing PFT and coccolith for coccolithophores

| Param\PFTs | Large diatom | *Phaeo* | small diatoms | *Prochlor* | Nfixer | coccolith |
|---|---|---|---|---|---|---|
| $P_{max}^C$ | 1.79 | 1.59 | 2.16 | 1.09 | 0.31 | 1.23 |
| $\beta$ | | | | 1.25 | | |
| $k_{sat_N}$ | 0.451 | 0.106 | 0.053 | 0.007 | 0.0 | 0.086 |
| $k_{sat_P}$ | 0.028 | 0.007 | 0.003 | 0.0004 | 0.004 | 0.0054 |
| $k_{sat_{Fe}}$, $*10^{-3}$ | 0.028 | 0.007 | 0.0033 | 0.0005 | 0.0124 | 0.0054 |
| $r_{j,k=1}$ | 0.8 | 0.78 | 0.2 | 0.2 | 0.6 | 0.58 |
| $r_{j,k=2}$ | 0.16 | 0.156 | 1.0 | 1.0 | 0.12 | 0.12 |
| $w_{sink}$ | 0.77 | 0.23 | 0.10 | 0.03 | 0 | 0.23 |
| $mfunc$ | silicified | | silicified | | | calcifier |





**Table 2.** Table of specification of the key phytoplankton groups (PG) given their biomineralizing function (BF) and size class as referred in the observations (PSC) and model (SCM).

| PG | PSC in situ | SCM | BF |
|---|---|---|---|
| **Diatoms** | micro/nano | "large"/"small" | silicified |
| large cells | micro | "large" | strong silicified |
| smal cells | nano | "small euk" | weak silicified |
| **Haptophytes** | nano/pico | "other large" | |
| *Phaeocystis* | nano/pico | "other large" | |
| coccolithophores | nano | "large" | calcifier |
| **Prokaryotes** | pico | | |
| N-fixer | pico | small | |
| *Prochlorococcus* | pico | "other small" | |

pico < 2 $\mu$m; 2$\mu$m < nano < 20 $\mu$m; micro > 20 $\mu$m





**Table 3.** Diatoms: model *vs. in situ* statistics at Longhurst's provinces

| criteria\bioms | APLR | ANTA | SANT | SSTC | CHIL | FKLD | EAFR | AUSW | AUSE |
|---|---|---|---|---|---|---|---|---|---|
| MAE ($mgChla\ m^{-3}$) | 1.00 | 0.75 | 0.27 | 0.08 | 0.27 | 0.15 | 0.21 | 0.01 | 0.03 |
| RMSE ($mgChla\ m^{-3}$) | 1.90 | 1.44 | 0.41 | 0.13 | 0.42 | 0.21 | 0.42 | 0.02 | 0.05 |
| RMSE unbiased | 1.88 | 1.34 | 0.39 | 0.13 | 0.35 | 0.18 | 0.40 | 0.02 | 0.05 |
| bias ($mgChla\ m^{-3}$) | -0.29 | 0.52 | -0.12 | 0 | -0.22 | -0.11 | -0.13 | 0 | 0 |
| N | 1287 | 235 | 402 | 102 | 24 | 12 | 6 | 7 | 19 |

MAE – mean absolute error ($mgChla\ m^{-3}$); RMSE — root mean squared error ($mgChla\ m^{-3}$); bioms are the Longhurst's biogeochemical provinces (Longhurst, 1998): Austral Polar Province (APLR), Antarctic Province (ANTA), Subantarctic Water Ring Province (SANT), South Subtropical Convergence Province (SSTC), Humbold Current Coastal Province (CHIL), Southwest Atlantic Shelves Province (FKLD), Eastern Africa Coastal Province (EAFR), Australia-Indonesia Coastal Province (AUSW), East Australian Coastal Province (AUSE).



**Table 4.** Haptophytes: model *vs. in situ* statistics at Longhurst's provinces

| criteria\bioms | APLR | ANTA | SANT | SSTC | CHIL | FKLD | EAFR | AUSW | AUSE |
|---|---|---|---|---|---|---|---|---|---|
| MAE ($mgChla\ m^{-3}$) | 0.24 | 0.22 | 0.16 | 0.14 | 0.19 | 0.13 | 0.57 | 0.10 | 0.48 |
| RMSE ($mgChla\ m^{-3}$) | 0.45 | 0.27 | 0.21 | 0.25 | 0.24 | 0.18 | 0.85 | 0.15 | 0.53 |
| RMSE unbiased | 0.44 | 0.26 | 0.21 | 0.26 | 0.22 | 0.17 | 0.62 | 0.12 | 0.22 |
| bias ($mgChla\ m^{-3}$) | 0.01 | 0.01 | -0.10 | -0.03 | -0.08 | -0.07 | 0.57 | 0.09 | 0.48 |
| N | 1264 | 229 | 437 | 154 | 26 | 12 | 12 | 23 | 39 |

Same as in Table 3





**Table 5.** Prokaryotes model *vs. in situ* statistics at Longhurst's provinces

| criteria \ bioms | APLR | ANTA | SANT | SSTC | CHIL | FKLD | EAFR | AUSW | AUSE |
|---|---|---|---|---|---|---|---|---|---|
| MAE ($mgChla\ m^{-3}$) | 0.28 | 0.11 | 0.06 | 0.14 | 0.08 | 0.06 | 0.05 | 0.07 | 0.05 |
| RMSE($mgChla\ m^{-3}$) | 0.64 | 0.19 | 0.13 | 0.22 | 0.13 | 0.08 | 0.05 | 0.08 | 0.05 |
| RMSE unbiased | 0.57 | 0.16 | 0.12 | 0.18 | 0.13 | 0.06 | 0.05 | 0.04 | 0.05 |
| bias ($mgChla\ m^{-3}$) | -0.23 | -0.11 | -0.06 | -0.13 | 0 | -0.06 | -0.02 | -0.06 | -0.01 |
| N | 772 | 42 | 201 | 82 | 21 | 7 | 12 | 39 | 27 |

Same as in Table 3, 4

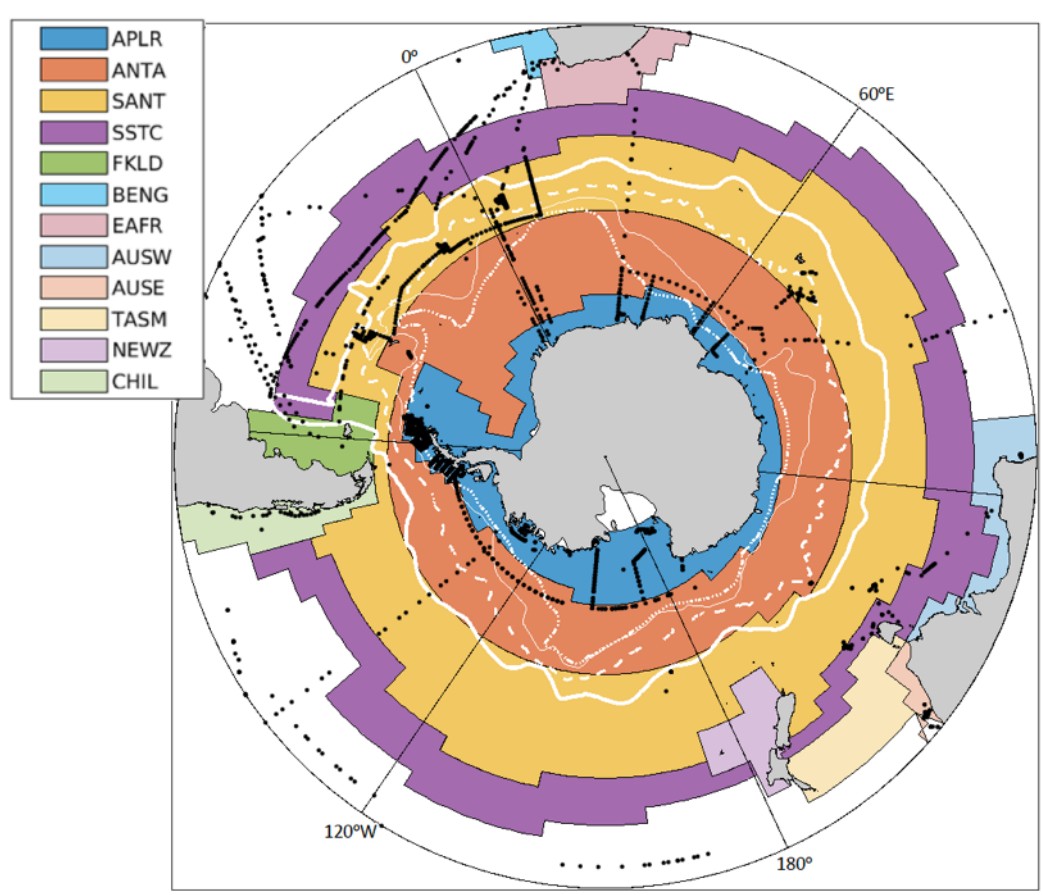

**Figure 1.** Distribution of *in situ* observations (black dots) over the Longhurst's biogeochemical provinces (coloured domains): Austral Polar Province (APLR), Antarctic Province (ANTA), Subantarctic Water Ring Province (SANT), South Subtropical Convergence Province (SSTC), Humbold Current Coastal Province (CHIL), Southwest Atlantic Shelves Province (FKLD), Eastern Africa Coastal Province (EAFR), Australia-Indonesia Coastal Province (AUSW), East Australian Coastal Province (AUSE). The white curves denote the Southern Ocean fronts (Orsi et al., 1995; Orsi and Harris, 2001): the Sub-Antarctic Front (SAF, thick curve); the Polar Front (PF, dashed), the Southern Antarctic Circumpolar Current Front (SACCF, thin curve) and the Southern Boundary of ACC (SBDY, dotted).
**Figure 2.** Surface PFT dominance retrieved by PHYSAT algorithm and simulated with the Darwin-MITgcm version of Dutkiewicz et al. (2015) and current model set up for July 2003 and January 2004. "SCL" represents *Synechococcus* like prokaryotic phytoplankton (not considered in current model version). Simulated haptophytes include coccolithophores and *Phaeocystis*. The our model output is masked by the area with sea ice concentration > 75% during respective month. Model PFT is considered dominant if its Chla fraction of total Chla is more than 55%.

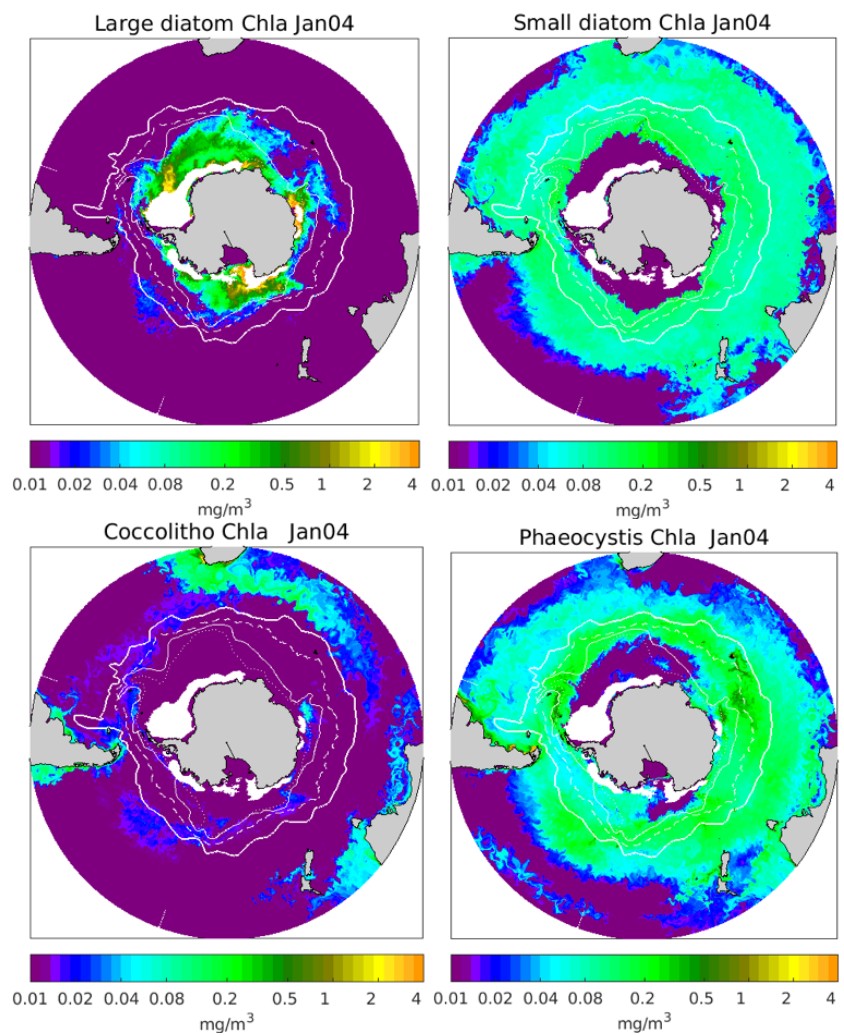

**Figure 3.** Spatial distribution of the model surface chlorophyll "a" for large and small diatoms, coccolithophores and *Phaeocystis*-analogue ("other large") in the Southern Ocean for January 2004 (experiment REF).







**Figure 4.** Meridional (from 75°S to 30°S) distribution of zonally averaged phytoplankton composition for the Atlantic Ocean sector (AOS, 60°W – 18°E), the Indian Ocean sector (IOS, 18°E – 120°E), Australian sector (AST, 120°E – 180°E) and the Pacific Ocean sector (POS, 180°E – 60°W) in February 2008. The red dashed curve represents the upper mixed layer depth (zonally averaged, m). The blue curve represents zonally averaged zooplankton concentration (mgC m$^{-3}$): experiment REF (to the left); experiment PHAEO (to the right). "Cocco" for coccolithophores, "Phaeo" for "other large" (including *Phaeocystis*), "sm dia" for small diatoms and "lg dia" for large diatoms.



**Figure 5.** Surface spatial distribution of the model diatoms (small + large), *Phaeocystis* and coccolithophores chlorophyll "a" in the Southern Ocean for January (upper panels) and March 2004 (experiment PHAEO, middle panels). Lower panes: March 2004 spatial distribution of surface silicon, phosphate and dissolved iron for experiment PHAEO.



**Figure 6.** Spatial distribution of the model Surface particulate inorganic carbon (PIC, upper panels) and particulate organic carbon (POC, lower panels) in the Southern Ocean for two experiments in March 2004: experiment REF (to the left); experiment PHAEO (to the right).



**Figure 7.** The surface PFT dominance simulated with Darwin-MITgcm for 2003/2004 for experiment PHAEO. Pico represents prokaryotes. The our model output is masked by the area with sea ice concentration > 75% during respective month. Model PFT is considered dominant if its Chla fraction of total Chla is more than 55%.


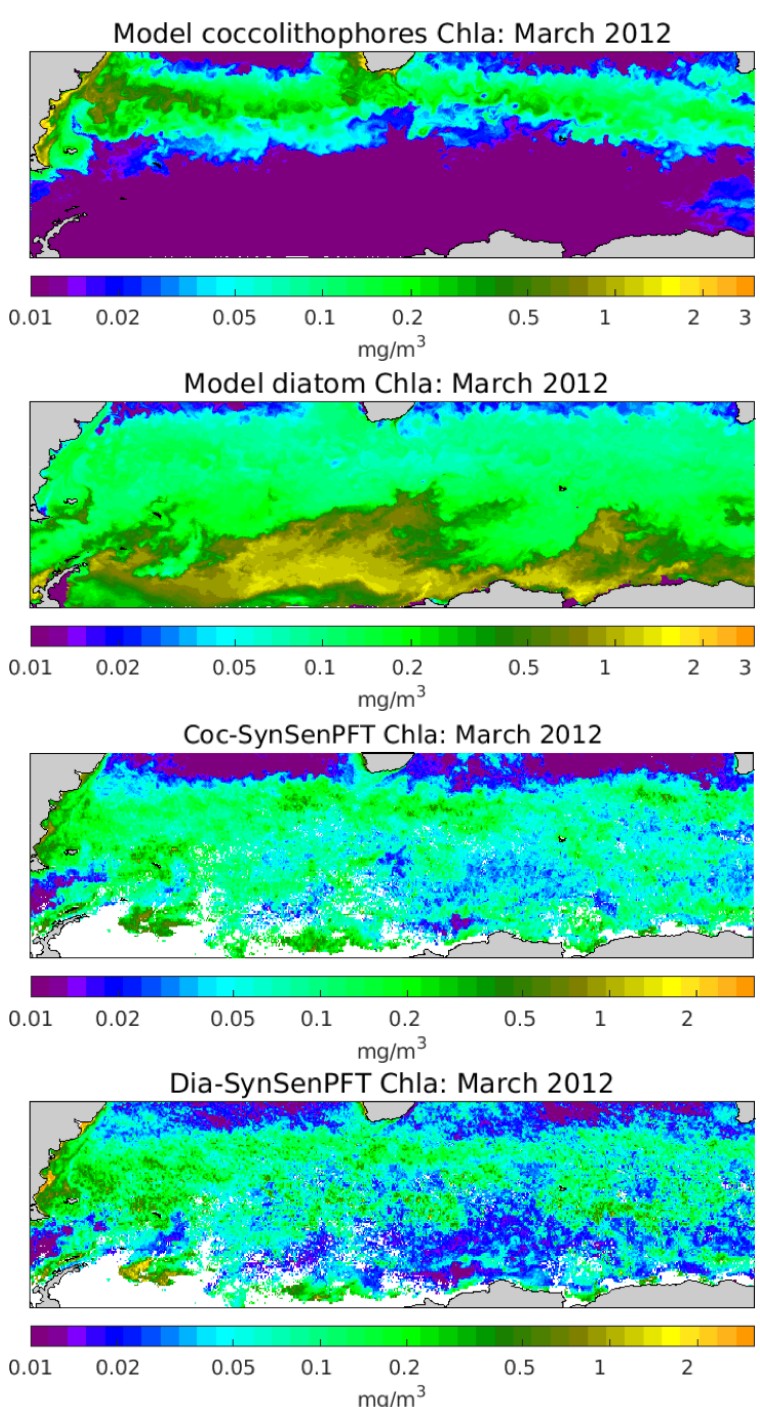

**Figure 8.** The surface Chla for coccolithophores and diatoms as model simulated (two upper panels) and retrieved with SynSenPFT algorithm (two lower panels) distributed over the domain shown by Smith et al. (2017) for March 2012.


**Figure 9.** Simulated surface diatoms Chla shown for several model snapshots and compared against *in situ* observations (circles).





**Figure 10.** Same as Figure 9 but for haptophytes.







**Figure 11.** Same as Figures 9 and 10 but for prokaryotes.