# Peer review of "On modeling the Southern Ocean Phytoplankton Functional Types"

_Biogeosciences, 2019_

## Referee Comment (RC1) · Anonymous Referee #1 · 27 Aug 2019

In this paper the authors use the Darwin-MITgcm to simulate the phytoplankton composition in the Southern Ocean. The paper is focused on the parametrization of the model to improve coccolithophore abundance, include two sizes of diatoms and two life stages of Phaeocystis in the Southern Ocean. The paper is an interesting model development, but I am not sure whether this really fits in the goals of Biogeosciences. The introduction doesn't provide any context or current challenges of why the work is being done. The results and discussion section lacks quantitative assessment of the model. Overall I was left wondering what science advancements or challenges this paper was providing or highlighting. In the introduction the authors seem to switch back and forth between defining PFT as plankton or phytoplankton functional type (see L23 and L 45 for example). The reader is left wondering whether starting at L46, they are

talking about phytoplankton or plankton. This is very confusing and I don't think brings any context to the paper. Along those lines, I thought the whole introduction wasn't very helpful in describing the context and problems tackled by this paper. The intro is mostly about how people have defined phytoplankton functional types when this paper appears to be mostly about the challenges that goes into representing phytoplankton diversity in a model. The introduction also presents only a very marginal portion of the work that has been done in the modeling of phytoplankton communities. Suggesting that this work started off with the paper from Le Quere et al. (2005) and Follows et al. (2007) when this work had started a lot earlier than this. The authors present all the other models in one sentence (L58) summarizing them as only including 2-3 phytoplankton groups and mention one other model that has four. This really comes across as a very narrow view of the work that has been done in this area. The intro would have benefited from expanding on the work and the challenges that have already been learned from the various models out there instead of the classification of PFTs. Furthermore, the view that the Darwin model 'has the highest potential to simulate globally relevant PFTs' is again narrow minded at best especially considering that the authors support this argument by saying that the Darwin allows to represent more than three and up to several thousands of phytoplankton groups (L64). The reader then finds out a few lines later (L85) that the version used here distinguishes only 6 phytoplankton groups (there are several models out there that do this) and not thousands like initially said. This brings the question of why (a) the authors need to state that this model has in fact 'the highest potential' among all models and (b) since they limit their phytoplankton groups to 6 does it really still stand as having the highest potential? The introduction should be focused more on the challenges that the modeling community has faced, the recent advances etc rather than try to convince the reader of why one model is superior to the others (without properly describing their model or the others). Results and discussion: this section lack some quantitative assessment of how well the model does compared to the in situ data. Why not report RMSE, bias etc? everything seems to be based on a few snapshots without a clear description of why the

authors chose those snapshots and a quantitative assessment. It is very hard to know what are the scientific advancements or lessons learned from this paper from the results and discussion section. Supplementary material: as detailed in some of my minor comments it appears that some information in the supplementary material would have benefited to be discussed in detail (and potentially included) in the main text. Similarly, the author sometimes refer to a Figure in the paper and compare it to a figure in the supplementary material which is very hard to follow (L334).

Minor comments: L16: this sentence needs a reference L20: needs a reference L57: three no thee L84: 'The version of the Darwin model used in our study simulates, among total 42 biogeochemical compartments..' change to '. . .among a total of 42. . .'. Btw what do you mean by compartments? As in variables? L85: earlier on this paper it said that the Darwin model had several thousands of phytoplankton groups? How did we end up with 6 only? Methods: what is the spatial resolution of the biogeochemical model? Same as the circulation model? L111: are the CDOM spectral slope used constant values? L128: why do the authors compared to this other model? Seems like a random comparison L150: so the model was spinned up for 6 years only? Was that enough to get stable conditions for the biogeochemistry? did the authors check for that and if so how? L168: define Chla L175: why were these groups not included if they have the observations for it and the model allows to discriminate for them? Section 2.2.1: there should be a 1-sentence description of how they went from pigments to phytoplankton classification Were the in situ data matched for the same day/year as the model run? From my understanding the model was run for the period 1999-2012? Section 2.2.2: The results present some snapshot from various month and year. Why did the authors not compare just a whole climatology for 1999-2012? Or annuals? How did they decide which year to compare? L20: Did the authors look at the full seasonal cycle to conclude this or just the two months that they presented? L236:'to the end of the considered period of time' . What period of time is that? L247-249: how long does it take for the large Phaeocystis to outcompete coccolithophores? L278: 'in the model world'. What does that mean? As in in your model? L287: 'Similar to Figure 2,

Figure 7...' how is Figure 2 similar to Figure 7? One shows all 3 methods while the other compares July and January output from the model. L294: instead of referring throughout the paper to the study by smith et al (2017) refer to it as the in situ dataset. Otherwise the reader is left wondering here for example what that paper is and why we are taking the same area. The first time I read the paper I didn't make the connection that this was the in situ dataset used for comparison. L324-329: this paragraph seems random, doesn't report any of the results yet seems like it should be discussed in the main paper (not appendix) since it contains some quantitative assessment of how well the model does. L331: 'the worst statistics..' use a different word than 'the worst..' L348: "...which indeed might support a biochemical/physiological hypothesis on the coccolithophore distribution...". Which hypothesis is this? Why does this come up for the first time in the conclusion section? This hypothesis wasn't mention anywhere else and it's unclear what it is referring to. L351: how do you define palatability? How did you conclude this from the results presented here? L369: phytoplankton not phytolankton L387: "..the information from these different sources becomes closer..." how can you say it becomes closer? In what way? L396: not only are cruises carried out close to the shelf but they are also mostly during spring and summer introducing another bias. Figure 2: the first column uses the method as the title but the third column's title describes the variable instead

---

## Referee Comment (RC2) · Anonymous Referee #2 · 29 Aug 2019

**Review: „On modeling the Southern Ocean phytoplankton functional types" (Losa et al., 2019)**

**Summary**

In this study, Losa et al. present a version of the DARWIN model, which they modified for the Southern Ocean (SO) application presented in this manuscript. In order to better represent the SO phytoplankton community structure, which mainly consists of silicifying diatoms, calcifying coccolithophores, and colony-forming *Phaeocystis*, the authors have added a second, lightly silicified diatom plankton functional type (PFT) to their model (in addition to a heavily silicified one which was
already included in the model before) and have made small modifications to the parametrization of coccolithophores in a first step (their reference simulation). Subsequently, motivated by problems in keeping both coccolithophores and *Phaeocystis* alive in their reference simulation, the authors have implemented a life cycle switch (based only on the surrounding iron concentrations) for the *Phaeocystis* PFT to simulate both solitary and colonial forms of this phytoplankton type (PHAEO
simulation). In this manuscript, the authors present a comparison of the simulated phytoplankton community structure to those suggested by satellite-based PFT algorithms and pigment data (the latter for the PHAEO simulation only).

In my opinion, the model development study by Losa and co-authors is valuable, as current global models often struggle to correctly represent the SO phytoplankton community. Efforts to improve
upon this are needed, given the importance of this ocean basin for global biogeochemistry and climate. I think the manuscript is in principle suitable for publication in Biogeosciences. However, I cannot recommend the publication in its current form, as I have serious concerns surrounding the model behavior (the extinction of individual PFTs at the end of the reference simulation is worrisome). Furthermore, I think that 1) the chosen PFT parameters and changes done to the model
have to be better motivated in the SO context of this study, 2) the used model parameters and parametrizations need to be better documented throughout the manuscript and limitations need to be discussed (especially surrounding the parametrization of the life stages of *Phaeocystis*), and 3) the impact of the changes and chosen parameters should be more thoroughly assessed by targeted sensitivity simulations.
Below, I first summarize my comments into a few general points and then list all my detailed comments, which should be addressed before the manuscript can be accepted for publication.

**General comments**

Below, I will list my general comments, which should be thoroughly addressed before the manuscript can be published:

1) The "extinction" of either coccolithophores or *Phaeocystis (Antarctica)* in the presented reference simulation deeply worries me. Before this manuscript can be accepted for
publication, the authors should understand where this is coming from and fix it, as I currently do not understand how this can happen, given that (based on observational data) their biogeographies in the SO do not overlap completely in space and time (meaning that there should be room for both to exist). Since this model behavior implies a substantial drift in the biomass distributions in the simulations assessed here, it can be expected to lead to a
substantial sensitivity of the presented results to the chosen analysis year (see also point 7). Furthermore, based on the information included in the current version of the manuscript, I don't understand how the subsequent changes made to the parametrization of *Phaeocystis* (i.e. including life cycle transitions) solved this problem, which should be discussed in more detail by the authors.
2) In the method section, a detailed description of the assumptions surrounding the parameter choices of the different PFTs as well as laboratory studies backing up the chosen numbers (Table 1) is currently lacking. Section 2.1.1 and Table 1 are currently incomplete in their description of the parametrizations and parameters used in this study (i.e. e.g. some parameters are missing, no units are given).

More specifically, regarding the coccolithophores, the authors do currently not motivate why the applied changes to the parametrization (as compared to previous global applications of DARWIN) are justified for the SO (e.g. by relating them to the coccolithophore community in this ocean basin). Regarding *Phaeocystis*, the manuscript could be greatly improved by including a more thorough discussion on the limitations of their current parametrization in the model, as important aspects surrounding their life stage transitions (e.g. light) are currently not accounted for. Additionally, the authors should comment on the usefulness of simulating both life stages within a single model tracer, as this is important information for those wanting to implement *Phaeocystis* into *their own* model.

Furthermore, the manuscript currently lacks a sensitivity analysis assessing e.g. the impact of the changes applied to the coccolithophore parametrization (in order to support what is in my view currently largely a speculation on the drivers of their biogeography in their model as important plots are not shown) or the impact of parameter choices (e.g. regarding those of *Phaeocystis*) on the simulated biogeography.

3) In general, important results (e.g. the change in the simulated phenology when implementing a second diatom PFT or the drivers of the simulated coccolithophore biogeography) are currently getting a bit lost in the manuscript. As these aspects are highly relevant for the modeling community and are the parts for which the manuscript goes beyond a pure model development paper, these aspects deserve more room (in text and figures). Currently, the conclusions drawn by the authors are not fully backed up by the simulations that are discussed and the plots that are shown in the manuscript, making it often impossible for the reader to evaluate what the authors base their arguments on.

4) Throughout the manuscript, the authors use the term "phenology", which typically refers to the annually reoccurring characteristics of the phytoplankton biomass evolution and can be characterized by the timing of e.g. the phytoplankton bloom start or the bloom peak. However, in the current version of the manuscript, "true" phenology is never presented and often only individual months of the simulated biomass fields are shown and discussed, which gives *no* information on the phenology (additionally, a definition of "phenology" and how it is assessed is missing in the method section). In order to e.g. emphasize the importance of including two diatom PFTs in a SO model (whereby the authors claim to have fixed the problem of many models, namely too early blooms), the authors should show the simulated phenology metrics in the revised version of the manuscript (e.g. maps of bloom timing in the "old" model version as compared to the improved setup and those derived from satellites).

5) Throughout the paper, the authors present very little *quantitative* evaluation of the simulated phytoplankton distributions, which should be improved in a revised version of the paper. Currently, the included HPLC data are only used for the PHAEO simulation (by plotting the observational data as scattered dots on top of maps, which is very hard to evaluate for the reader), but should also be included for the "old version" of the model and the reference simulation in order to actually *show* the asserted improvement in model performance. Additionally, the HPLC data can and should also be used for a discussion of the phytoplankton community structure to complement the satellite-derived products. Even though SO data coverage within the MAREDAT data base is limited, the authors should consider evaluating their model output using these phytoplankton carbon biomass data set to complement the currently included HPLC data. Furthermore, in the presentation of the evaluation, the authors often use subjective statements in their description (e.g. "plausible distributions", "skillful enough") which should be avoided as much as possible throughout the manuscript as it is e.g. not clear to me at all when a biomass distribution is "plausible".

6) Overall, I think the introduction in its current form misses a clear focus on the focus area, i.e. the SO. From the title of the paper, I would expect a description of the *observed* SO phytoplankton biogeography somewhere based on available in situ data and satellite algorithms to set up the reader for the assessment of the *simulated* community structure. Additionally, I would expect a summary on what has been done in terms of PFT modeling in the SO specifically, highlighting what gap is filled with the model used here (for this, see e.g. Lancelot et al. (2009), Wang et al. (2011), Le Quéré et al. (2016), Nissen et al. (2018); Note that the list of available studies is much longer than the examples given here!). The introduction in its current form largely focusses on global modeling approaches without an assessment of how they perform in the SO and is thereby of limited use for the goal of the paper.

7) Currently, there is no consistency in the study in what month or even what year is assessed in the different parts of the manuscript (compare e.g. Fig. 3, 4, and 6). In the method section, the authors should clearly state which year(s) and which month(s) of the model output is used in the analysis and why. In this regard, it is e.g. not clear to me why the authors chose to present the ability of the model to represent dominant phytoplankton types in winter, when biomass levels are low. Overall, the figure captions are often incomplete and panel labels are missing entirely. These should be added and referred to in the text to better guide the reader.

**Detailed comments**

**Abstract:**

L. 1: I suggest to make clear in the very first sentence that you're focusing on a single model – otherwise the first sentence sounds like the reader is about to read a review paper on SO PFT modeling. Additionally, I suggest to rephrase to "*under past and* present climate change".

L. 3: By stating "phenology" so prominently in the abstract, you set up the reader for an assessment of the PFT phenology in your model – which you actually never really do (see comments below). Please rephrase here to have a better representation of the content of the paper and/or adapt the content of the result section (see general comments).

L. 8-9: The new model configuration describes the competition and co-occurrence "best" in what regard and compared to what? Please be precise.

L. 9-13: Please specify what "older version" you're referring to here, e.g. by explicitly stating "without the above-mentioned changes, but otherwise identical" (if that is the case).

L. 11-13: In the manuscript, you never actually show a quantitative validation of the model output with the SEM data (no plot at all) or the HPLC data (only in maps for the PHAEO simulation, not for the REF simulation), so that it is hard for the reader to evaluate how the model performance improves with your changes (see comments below). Furthermore, I suggest to not overemphasize the SEM data here in the abstract as this comparison is not a major part of your study.

L. 13: Please rephrase to "SO PFT dominance *patterns*". "agrees well" in what regard? Space? Time? Additionally, the abstract in its current form does not represent how much time you spend in the manuscript on the discussion of dominance patterns as opposed to the validation of chlorophyll concentrations of the individual PFTs. I suggest to rewrite the abstract to more adequately represent the content of the result section.

**Introduction:**

L. 16: Please rephrase "via the sinking of $CO_2$".

L. 17: Please add a reference for the evidence of changes due to on-going climate change.

L. 20: Please add a reference for the impact of phytoplankton community structure on the diversity of higher trophic levels.

L. 21: Please add a reference for the impact of phytoplankton community structure on climate on different temporal and spatial scales.

L. 32: Please add a reference for the impact of *Phaeocystis* on SO export production.

L. 32-35: Why is the description of these types ($N_2$ fixers and pico autotrophs) relevant for a modeling study of the SO? I think you can delete this part to have more room to focus on an introduction of the actual topics, such as what is known on the biogeography (from observations and modeling studies) of the most important types in the SO, namely diatoms, *Phaeocystis*, and coccolithophores.

L. 36-39: I suggest to list the three criteria when first mentioning the division by Le Quéré et al. (2005) in e.g. L. 22. The way it is done currently, the 2nd and 3rd criteria come a bit out of the blue for the reader.

L. 39: Please give an example that is relevant to the SO application in this study.

L. 44-45: I suggest to rephrase to something like "[…] includes also bacteria and zooplankton, but for this study, we use "PFT" to refer to phytoplankton only, in accordance with the definition by the ocean color community".

L. 52-55: The relevance of this statement to the study at hand is not clear to me. Please explain. Additionally, you never really use "PG" throughout the text, it is not clear to me why you introduce it here. I suggest to move the information given here to the only place where you actually use it (section 2.2.2).

L. 56: It is not clear here why you cite Follows et al. (2007) alongside Le Quéré et al. (2005) after spending almost a page on discussing the latter while not introducing the former. Please make clearer.

L. 57: "thee" should be "three"

L. 60: Please see also Krumhardt et al. (2019) for a global model with an explicit representation of coccolithophores and consider adding Nissen et al. (2018) here as well as an example of a regional model with explicit coccolithophores to give a more complete overview on what has been done.

L. 66: Please explain more clearly in the text how the Darwin model offers "the highest potential". For example, does this model generally offer "higher potential" than regional modeling approaches? As I am personally not convinced by this (as it will depend on the question you're trying to answer), I suggest to rephrase this statement to explain more clearly.

L. 70-74: In my view, the goals you list here for the study at hand do not match the content of the result section. For example, the manuscript currently lacks a thorough quantitative (!) assessment of the phytoplankton phenology. What is your conclusion on point 3) here? How can the model complement available in situ observations?

L. 71-72: When is a model "skillful enough" in your opinion? When is a simulated distribution "plausible"? Please specify exactly what you mean by this and avoid subjective judgement whenever possible. Please replace "predict" by "simulate" or similar.

L. 74-75: The statement "When determining […]" is not clear to me. Please be more precise. What do you mean exactly?

**Methods**

L. 80: I suggest to change the title to include the name of the model used in this study.

L. 90: Do you mean lightly silicified? How was the silicification different between these two classes different in the model? How is silicification parametrized? If you introduce a completely new PFT, you need to give more detail on its characteristics.

L. 90-99: Why are these three changes justified for the SO? I suggest to include statements on the reasoning behind e.g. changing the nutrient affinity and grazing parameters for coccolithophores – why does this apply for this SO-focused study and not for global applications of Darwin? Please add a reference regarding the occurrence of lighter silicified diatoms at lower latitudes.

L. 95: Please replace "was presented" by "is represented".

L. 95-99: What sensitivity experiments did you perform here? How did you evaluate what a "realistic co-occurrence of coccolithophores and *Phaeocystis*" is? I think it is important here to briefly sketch the main characteristics of the parametrizations used for *Phaeocystis* if you've actually used those from Popova et al. (2007) and Kaufmann et al. (2017), but see also comment further down (on L. 138 in your manuscript).

L. 101-112: The description of the treatment of light is out of place here as you go back to a description of the PFTs afterwards. Please reorganize the section to make it easier for the reader to follow. Additionally, I am not sure this much detail on the parametrizations surrounding light absorption are needed in the main text. Please consider moving this part to the supplement.

L. 100-117: Here and throughout the text (including e.g. especially Table 1), please make sure you state the units of all variables introduced.

L. 113: Please replace "which is presented" by "which are described by" or similar.

L. 114: According to Table 1, this parameter only applies to Prochlorococcus. I suggest to state that here.

L. 115: I find "biomineralizing function" misleading and would rather say "whether or not they form biominerals such as opal or calcite" (or something along these lines).

L. 115-117: Please rephrase this sentence, it sounds a bit weird to me in its current form.

L. 118: Please rephrase to "The growth of phytoplankton $\mu_j$ (day$^{-1}$)[…]"

L. 123: How are the temperature and nutrient limitation terms calculated? Please add the equations.

L. 124: Alpha PI is missing in Table 1.

L. 125: The phytoplankton-specific light absorption and the maximum quantum yield of carbon fixation are missing in Table 1.

L. 127: Please change to "as opposed to the stud*ies* by X *and* Y". However, I don't understand why you refer to two studies here which are based on a different biogeochemical model (NOBM) than the
one you're using here (DARWIN). Are you using the same function to calculate the temperature limitation as they do? If yes, state that to make your argument clearer. Furthermore, does your statement mean that the growth of $N_2$ fixers is not suppressed at low temperatures? This relates to a comment further down (on Fig. 4) in that I have the impression that your importance of $N_2$ fixers for the SO phytoplankton community is way too high if we take into consideration that their growth
should be limited to regions of temperatures above a certain threshold (e.g. ~18°C, see e.g. Breitbarth et al. (2007) and Luo et al. (2012)) – even though nitrogen fixers have been found more recently in polar waters, I am just not convinced that they make up such a substantial part of the community in terms of biomass in these latitudes. Are you aware of evidence for this?

L. 131: $g_{max}$ and $k_{sat}$ are missing in Table 1. Furthermore, the equation you give has a Holling Type III ingestion term. Are you using Holling Type II or III? Please double-check.

L. 138-145: I have some concerns regarding the way you parametrize *Phaeocystis* here.
- First of all: are you following the parametrizations of Popova et al. (2007) and Kaufmann et
al. (2017) or not? You state this in L. 99, but according to what you state here, I don't think you can say that you use their parametrizations. In both the cited studies, the transition of *Phaeocystis* from single cell to colonies (and back) is a function of a specified maximum colony formation rate, a maximum single cell liberation rate, the single cell biomass concentration (using a threshold concentration to allow for colony formation), the position in
the water column (i.e. light availability, see also Peperzak (1993)), and the nutrient limitation – as opposed to just a fixed iron concentration threshold you seem to have used here (if I understood this correctly). Differences to the cited literature need to be made very clear here as your parametrization appears distinctly different. The effect of neglecting certain aspects and the potential impact on the simulated biogeography should then be at least
discussed somewhere in the manuscript.
- One vs two tracers for *Phaeocystis*: Have I understood correctly that your whole *Phaeocystis* biomass pool just switches back and forth between single cells and colonies based on the iron concentration threshold? I understand that this makes it computationally more efficient, but this might be too simplistic (I am not sure myself). Assuming I understood this correctly,
are you tracking in space and time what "*Phaeocystis* state" the model tracer is in? Based on this tracking: are you confident that you capture the transitions well enough with just the dependency on iron to justify neglecting the other dependencies that have been suggested to be important (such as light levels), meaning that one model tracer is enough to simulate both life cycle stages simultaneously? This would be an important piece of information for
other people wanting to implement *Phaeocystis* into their model. Please discuss this in the manuscript.
- Sensitivity to chosen parameters: I would be curious to see how sensitive your simulated biogeography is to how long *Phaeocystis* is in the colonial form during summer. Have you looked at the sensitivity to the chosen threshold? Additionally, what are the changes in
parameters based on (30% and 25% higher mortality and grazing rate, respectively, as well as 20% lower kFe in single-cell-state, choices seem random)? How sensitive is the simulated biogeography to these choices?

L. 148: What is the horizontal resolution across the SO in the setup you're using here?

L. 151: If you state that your setup was similar to the one in Taylor et al. (2013), I am immediately wondering what is different. Please state this clearly.

L. 160-165: Do you spin up the model in the coupled physical-biogeochemical setup immediately or do you spin up the physics first and only coupled once the circulation in spun up (or close to that)? This is not clear to me right now. I am wondering what impact spinning up both together (what it sounds like based on your manuscript) would have on the simulated biogeographies. Have you looked into this?

L. 160: Please replace "evolved" by "involved".

L. 163-165: How does using model output from a different model compare to initializing with e.g. WOA and satellite derived chlorophyll concentrations (making some further assumptions on C:Chl ratios and the depth profiles)? Do you introduce biases? How does the model used in Taylor et al. (2013) perform in the SO?

L. 168-184: In this section, I am currently lacking a description of what model output you're comparing to the observations. Climatological? Single years? Co-located? Surface only? Please state here, what you're going to present in the result section, as this will help the reader to follow your structure.
As for the comparison with the data by Smith et al. (2017), you need to be clearer here as it is not obvious how you compare the "simulated PFTs" (do you mean the simulated biomass concentrations? Please be precise) to SEM observations (cell counts). Again, do you co-locate? Do you use single year model output? Climatological model output?

L. 186: Similar to above: Please state very clearly what model output you take (and why) for the evaluation. As stated in the comments further down, I find it very confusing as a reader that you currently pick what seems like random months of a random year and are additionally not consistent across the different simulations (compare Fig. 2, which shows July & January, to Fig. 7, which shows June-August and December-February; compare Fig. 4, which shows February 2008, to Fig. 5, which shows March 2004, or to Fig. 8, which shows March 2012). Please rewrite this section accordingly and double-check how you can be consistent in the use of the years.

L. 196: You state "only 0.5°" – how does this compare to your model resolution? (You give an average resolution of 18km, but it wasn't clear to me over what area that is averaged, see further up)

**Results & Discussion**

L. 205: "Improved" compared to what?

L. 206: From the title of the section, the reader expects a discussion of phytoplankton phenology here (i.e. e.g. bloom timing, bloom peak timing, bloom duration), but instead you discuss dominance patterns. Please choose a more appropriate title. In fact, I would suggest to not use "phenology" throughout the text as you currently do not really assess it in a quantitative sense. If you want to keep it (and there is value to that!), you need to introduce this in the method section, where the definition of bloom start etc. is currently missing, and present the simulated phytoplankton phenology and the comparison with e.g. satellite derived phytoplankton phenology.

L. 206-223: You never state in the method section that you will compare model output from a version without the listed changes to the setup which includes the changes. Please add this to the method section.

L. 207: "were misrepresented" – please rephrase to state more clearly what model version/setup/simulation you're referring to here.

L. 208-209: How confident are you in the satellite-derived dominance pattern in austral winter (July)? Additionally, do you really think that for a region like the SO, it is critical how well the model simulates the dominance patterns in winter? Personally, I would have preferred to see the agreement for all summer months (December-February or even March) to additionally get a better feeling for how the model is doing in terms of seasonality.

L. 208: The transition between sentences is confusing for the reader: "[…] in austral summer. In July, […]" First, you set the reader up for hearing more about the summer and then you jump to talk about July. Please rewrite.

L. 210: Related to above, looking at the model performance in a single month does not tell you much about how the model is doing in terms of simulating phenology. Please rephrase.

L. 210-211: Which model are you referring to here? Throughout the text, please add references to panels of the Figures (these need to be added to each Figure!), as this will be very helpful for the reader. Maybe refer also to the HPLC data here? These should support the discussed bias in the community at high latitudes.

L. 211-214: This information belongs into the method section. What exactly do you mean by "in terms of agreement with observed phytoplankton composition"? How did you evaluate this? For completeness, consider adding the reference Trimborn et al. (2015) to the method section 2.2.1. Where do you show the diatom phenology of the model?

L. 218-220: I am curious to what extent the improvement of the model in the SO is at the expense of the model performance on the global scale. Are the simulated patterns still reasonable?

L. 220-223: I cannot follow what you base this conclusion on given the plots you're showing in the manuscript, but I think this is an important point to make. If you really significantly improve the simulated phenology by including two types of diatoms instead of one, this aspect deserves a lot more room than it currently gets in the manuscript in my opinion. Consider including plots of the simulated phenology (e.g. bloom start and bloom peak of total chlorophyll and diatom chlorophyll in "old version", REF and PHAEO) as compared to those derived from satellite products. Consider also adding a reference to the regional SO model used in Nissen et al. (2018) here, as this model simulates too early total chlorophyll/diatom blooms as well, demonstrating that this issue is not restricted to global models.

L. 225-226: Where is this seen? You don't show the biomass patterns for the run without the two diatom classes in the current form of the manuscript.

L: 228-229: In what way is the simulated pattern in agreement with the cited studies? Please be more precise here. Related to earlier comments, how did you evaluate this exactly?

L. 233-236: Consider rephrasing "the model representation of co-existence/competition within the haptophyte group" to something like "the simulated biomass distributions of both coccolithophores and *Phaeocystis* were very sensitive to chosen model parameters, and small changes in […]". What "small changes in the Darwin model physiological parameters" are you referring to here exactly?

What is a small change in this context? And which parameters are you referring to? Can you include more information on these in the supplementary material? Am I understanding it correctly that by the end of your reference simulation, coccolithophores go extinct in your model?

If this is indeed what you mean, I am not entirely sure I understand why this happens, but I certainly find it very worrisome for the evaluation of your reference simulation, as this implies that you have significant drift in your PFT biomass concentrations and/or distributions. Is this the case? This also worries me in that your choice of showing different time periods in the different figures of the manuscript will then have a possibly considerable impact on the biogeographies you show.

In observations, the biogeographies of coccolithophores (mainly in the subantarctic) and *Phaeocystis* (only *P. Antarctica* in the SO, mainly in the high-latitude SO) do generally not fully overlap, so I don't understand how competitive exclusion between these two types of phytoplankton leads to the extinction of one in the model, as I don't see these two types exclusively competing for nutrients.

L. 240: Please clarify: Does the reference simulation already have the changes listed in the method section (in the nutrient affinity and the grazing pressure)? In the method section it sounds like it, here in the result section it does not, I got confused.

L. 245: Why this exact month?

L. 247: Please clarify: By "other large", you mean large diatoms and *Phaeocystis* together? Also, your statement "too low concentrations of coccolithophores south of the PF" is based on what? This statement confuses me due to two reasons: First, Fig. 4 only shows relative contributions to total phytoplankton biomass and does not give any information on absolute biomass levels. Second, I am not aware that one would expect significant concentrations of coccolithophores south of the PF (see e.g. Balch et al., 2016). So what exactly are you referring to here?

L. 249: Similar to above, what do you mean by "more plausible" here? Compared to what? Please be more precise and avoid subjective judgement.

L. 251: I think this statement needs to be rephrased. Smith et al. (2017) state that based on their measurements, coccolithophores made up *maximum* 20% of total chlorophyll concentrations locally, but generally contributed less than 5%. Consequently, I would phrase it more conservatively than saying that simulating 30% of total biomass is in agreement with Smith et al. (2017), which it clearly isn't.

L. 253: I think you're referring to Section S3 here. Do I expect the fraction of coccolithophores to be higher in winter? How is this backed up by observations (e.g. HPLC)? And how relevant is the community structure in SO winter, when biomass levels are generally very low?

L. 254-255: This is an obvious statement. What is the reader to take away from the distribution of zooplankton biomass?

L. 255-260: Here again, what is the "realistic distribution" for you? What are the "other circumstances"? This is a very vague statement. Please be more precise. Have you done a sensitivity simulation in which coccolithophores could not escape the grazing pressure to assess the impact on the biomass distributions and community structure? This would be very interesting to back up your statement. Related to above, in this context it will matter a great deal how different you choose e.g. the maximum grazing rates of zooplankton grazing on coccolithophores as compared to grazing on e.g. diatoms in the model, which is related to what assumptions you make regarding the coccolithophore community you're simulating (all coccolithophore species? *E. huxleyi* only? Please see also comment further up) and prey preferences of the zooplankton PFTs.

Furthermore, I am wondering how high your simulated coccolithophore carbon biomass concentrations are compared to e.g. MAREDAT observations. Taking your ~30% contribution of coccolithophores to total biomass (which seems a bit higher than that suggested by Smith et al. (2017), see above) and a maximum of ~20% in austral summer between 40-50°S in Nissen et al. (2018; their Figure 3), in my view, it is very conceivable to assume that this difference is to a large extent controlled by differences in assumptions surrounding the grazing formulations. Additionally, if one looks at the discussion in e.g. Monteiro et al. (2016), there is a lot that is still not understood with respect to the coccosphere and grazing pressure from zooplankton, which is why I don't think one can per se say that coccolithophores should always escape grazing pressure in models – in the same way as I don't think the reverse can be stated (will be highly dependent on the ecosystem structure at a given location). Therefore, I think it is important to point that out in the manuscript. Additionally, note that Nissen et al. (2018) state that grazing is a major control on the simulated coccolithophore *biogeography* and their biomass concentrations relative to those of diatoms, but they do not comment on the effect of the assumed grazing difference between diatoms and coccolithophores on the simulated phenology of the two in the subantarctic. Please rephrase L. 258 accordingly. Additionally, without the relative grazing advantage of coccolithophores relative to diatoms, the simulated coccolithophore biomass levels in Nissen et al. (2018) increase three-fold between 40-50°S (see their Figure 7), pushing the simulated coccolithophore biomass levels way beyond what MAREDAT observations suggest for this area.

    L. 260: Do you assume the drivers to be the same globally? In my view, one could very well imagine a difference in the relative importance of grazing in controlling coccolithophore bloom phenology, as the competitive success of coccolithophores will
largely depend on 1) which coccolithophores are present (and hence simulated), 2) which other phytoplankton are present, and 3) which grazers are present. I suggest to point this out as a potential limitation of the comparison of a study focusing on the North Atlantic to the one here.

L. 269: Please rephrase in order to avoid subjective statements like "agreed well". Additionally, where is this seen? I suggest to add validation plots to the supplementary material.

    L. 266-278: Why do you show March of 2004 now?

L. 271: Why "potential existence in colony form"? Does that mean you did *not* track when and where *Phaeocystis* was present in the colonial form in your simulations? I think this information would be a useful output to assess where and when the chosen parametrization leads to colony formation and to assess/discuss/speculate what impact neglecting further dependencies of colony formation (light etc., see above) have on the simulated biogeography.

    L. 272-274: I don't see in Figure 5 how the introduction of the high nutrient affinity of coccolithophores causes what you claim here. For that, you would need to show the original biogeography before applying the changes.

L. 274: Replace "depleting" by "depleted".

    L. 275: Where is the Subtropical Front in the plot? The STF is not introduced and the caption of Fig. 5 does not include a definition of the white contours either. Please include this information somewhere.

    L. 275-277: Similar to comment on L. 272, I don't see how Fig. 5 shows this. Again, one would need the plot before the change – otherwise I don't understand how it is possible for the reader to see this. Please clarify.

L. 277-278: Please rephrase "the simulated coccolithophores". This sentence does currently not make a lot of sense. What do you conclude from the fact that you find highest coccolithophore biomass levels (I assume that is what you mean here) where/when silicic acid is depleted? Please discuss shortly what this implies for the competition with diatoms.

L. 279-280: Please revise the grammar of this sentence.

L. 281: Again, why March 2004?

L. 279-284: This whole paragraph is too superficial and lacks the build-up from the introduction and method section, as the impact of different phytoplankton types on POC production/availability in not thoroughly introduced. Additionally, you nowhere state what assumptions you make in DARWIN regarding the routing of biomass losses to POC by the different PFTs. What do you assume for coccolithophores, diatoms, and *Phaeocystis*? Why are the POC concentrations south of the SACCF higher in the PHAEO simulation? I suggest to relate this back to changes in phytoplankton community structure and assumptions in the model, so that the reader can take something away from your statement. Are you showing POC resulting from haptophytes only or from all phytoplankton? You state that you're looking at the impact of haptophytes, but possibly, you're showing all phytoplankton. Please double-check and clarify. Similarly, for PIC, you nowhere state in the method section how calcification by coccolithophores is described in the model. Please add this information. The cited papers by Balch et al. do not comment on POC concentrations, as far as I could see. Please double-check.

L. 288-291: Similar to above, how do you define the "much better agreement" or "even larger agreement"? Try to be quantitative whenever possible. Additionally, in Fig 2 you only show July & January for PHYSAT and the "old" model version, here you make a statement for the months June-August and December-February. Please show all months for PHYSAT and the "old" model version somewhere. And again, I don't understand why you decide on these months now, when before you focused on March 2004. This is very confusing for the reader.

L. 293: "*of* monthly means"

L. 293-298: Why do you reduce the plot to the Atlantic and Indian sector based on Smith et al. (2017)? Why 2012 now? You don't actually show any data from their study so it is not clear to me why you reduce the area shown in the Figure and why you chose a different year all of a sudden.

L. 296: Where is the "smaller belt"? Be precise in your description. What is the latitudinal extent in the model output and the satellite product?

L. 298-306: This is a very nice discussion, but please link it back more explicitly to the "smaller belt" to make the take away message clearer. Same is true for the discussion of the diatom distributions.

L. 310: How were the days of the snapshots chosen?

L. 315: What is "less accurate" in this case? Please be precise.

L. 318: Why "see Vogt et al. (2012)"? This citation here is not obvious to me. Can you clarify for me?

L: 324: Does Fig. S9 only include model output that was collocated with the observations? Please clarify in the text and/or the Figure caption.

L. 331-332: Is a systematic overestimating by 0.5 mg chl m$^{-3}$ really that bad in your view? That's what the writing currently makes it sound like to me.

L. 334: Differ in what way? This is a vague statement.

L. 285-340: Personally, I would suggest to present the validation earlier in the manuscript. I find it a bit unfortunate to have the evaluation as the last result section.

**Conclusions**

L. 342-343: I don't understand the first sentence. How did satellite-derived estimates and in situ observations help to define trait requirements (characteristics? Or simply traits?) of phytoplankton? Can you rephrase?

L. 347-348: The necessity of the inclusion of two diatom classes and the changes to the coccolithophore parametrization have not been sufficiently motivated and the subsequent improvement of the model has not been sufficiently demonstrated, please see comments above (e.g. on L. 220-223 and on L. 275-277 of your manuscript). Furthermore, I don't understand the logic in the sentence in parentheses. Please rephrase to clarify.

L. 349: That temperature is not a driver of the coccolithophore biogeography in your model has not been shown/discussed in your result section. Please include it there or adjust the conclusion section.

L. 350: Please revise the grammar of this sentence ("Neither […]").

L. 350-355: Again, please double-check carefully what in your conclusion section are results that you've actually presented in this manuscript and what are speculations or work not included here. Currently, a lot of the things you say here do strictly not follow from what you've shown.

Additionally, including life stages of *Phaeocystis* allowed for co-existence of the two types where and/or when? Going back to L. 234-236, I think you're referring to the fact that one goes extinct when not accounting for these. I still think this is worrisome and I do not understand at all how the changes to the model then prevent this from happening.

L. 355; Please check the grammar.

L. 359-362: Is this really the case? I would expect the nutrient limitation terms to have a big influence on differences between PFTs as well, given the differences in their half-saturation constants (Table 1). Please double-check. The "realized" growth rate (specific growth rate) is a result of all environmental factors it depends on and the non-linearity of the functions might lead to unexpected results with regards to their impact on the specific growth rate at a given point and time.

L. 362: Please include this information on assumptions surrounding alphaPI in the method section and in Table 1.

L. 367: If your maximum growth rates are likely too high, this should be discussed/mentioned somewhere in the manuscript. Can you plot how your temperature-limited growth rate in the model for *Phaeocystis*, diatoms, and coccolithophores relates to laboratory measurements (see e.g. supplementary material in Le Quéré et al. (2016) for a compilation)? The Tables S5 & S6 do currently not include information on what temperature the reported growth rates are measured at (and you don't specify the temperature dependence used in your model). Plotting the function that is actually used in the model over a range of temperatures together with a range of measurements will help to understand in what temperature ranges the temperature-limited growth rates in your model is too high/too low.

L. 370: And is it a problem if one had to choose different alphaPI for different PFTs?

L. 371-373: Again, this has not actually been shown in your study. Can you back this up with some references? Try to make your language more accurate by including words like "potentially", "possibly", then you would immediately avoid misunderstandings regarding where you speculate and where you refer to things you have actually shown.

L. 379: But you have included CDOM in the model simulations you discuss here, haven't you (see Equation 2)? Then I don't understand what you mean here exactly, as you're talking about possible improvements. Please be precise. What would need to be improved and how?

L. 380: Similar to above: Please be precise on what you think should be improved regarding the algae-sea ice interactions and how you think this would impact the study at hand. Please try to always relate your suggested improvement back to *this* study – there is possibly an endless list of things one *could* improve in your model (and in any other model for that matter), but not all of those things are relevant for modeling PFTs on a basin scale in the SO. Please make very clear why you think the things you suggest to improve are important and how you think they would impact the study at hand.

L. 384-385: Please delete the statement about green algae and dinoflagellates, as this is not relevant here.

L. 386: The information becomes closer? To what? Please revise the logic.

L. 382-403: In my opinion, this whole paragraph is misplaced in the conclusion section. Overall, I think the conclusion section is way too long right now. I would instead suggest to include and "limitations & caveats" section between the results and the conclusions. In such as a section, you can then discuss the difficulties described here, as well as the limitations surrounding the PFT parametrizations and the suggested improvements (L. 355-381). Please focus the conclusion section on the *main take away messages* from your paper.

**Figures/Tables**

Table 1: As mentioned in the detailed comments above, the table is currently incomplete. Please add the missing variables (even if they are the same for the different PFTs, it is important to state that here for important variables such as alphaPI and the maximum grazing rate). Furthermore, please add the units and a short description of each variable to the table. What temperature is the maximum growth rate at? This needs to be specified.

I am also irritated by the three digits of the half-saturation constants of e.g. N – is the model that sensitive to changes in this number? Have you tested this?

Please also add the half-saturation constant of silicic acid by diatoms.

I am also slightly confused by your half-saturation constants of iron. Assuming your reported numbers are in mmol m$^{-3}$, your value for large diatoms (0.028) is e.g. an order of magnitude smaller than those suggested for the SO in Timmermans et al. (2004; 0.19-1.14 nmol L-1 or 0.19*10$^{-3}$-1.14*10$^{-3}$ mmol m$^{-3}$). Regarding the N$_2$ fixers, *Trichodesmium* is typically considered to have a higher iron requirement and half-saturation constant of iron than other phytoplankton PFTs (I suggest you to have a look in e.g. Berman-Frank et al. (2001) and Ward et al. (2013) and check references therein, as I am not an expert myself in nitrogen fixers). Have you tested how your low k$_{Fe}$ for the N$_2$ fixers in your model impacts their relative contribution to the SO phytoplankton community, which is currently quite high (see Fig. 4)?

Table 2: What do "PSC" and "SCM" stand for?

All Figures & Tables:
- Please add panel labels (Figures) and use these in the captions and the text.
- Please double-check that you clearly state in the captions, which year of which simulation you're assessing and what year (Tables 3-5)
- Please make sure you include units for *all* variables in the captions.

Fig. 1: You only plot HPLC observations. Please be precise in caption. Replace "curve" by "contour".

Fig. 2: When you say "Haptophytes" here, do you mean coccolithophores? Or the combination of coccolithophores and *Phaeocystis*? I am confused because both PHYSAT and the Darwin model do
discriminate between the two and you list *Phaeocystis* as a separate class in the Figure legend. Please clarify in the Figure legend as well as in the result section 3.1. Why do you combine the two for the model output? "The our model output" -> please rephrase. What is the basis of choosing 55% as the dominance threshold? This seems random to me. I am surprised to see that there is no area of coexistence, so this means at every grid cell there is always one PFT that contributes more than 55%
to biomass? If I look at the transects in Fig. 4, it does not necessarily look like it. Please double-check.

Fig. 3: What are the white contours? Please add info to caption.

Fig. 4: The caption is incomplete: explain Nfix, Proc, UML... I also suggest to add "REF" and "PHAEO"
directly in the Figure to make clearer immediately what the two columns are.

Fig. 5: panes -> panel. Please change the order of "phosphate" and "iron" in caption to match the order with that in the figure. What are the white contours?

Fig. 6: I suggest to add "REF" and "PHAEO" directly in the Figure to make clearer immediately what the two columns are.

Fig. 7: Please correct "the our model output". Please add a reference to Fig. 2.

Fig. 9: Is "diatoms" large + small here? Be more precise. Add "in situ *HPLC* observations". Which simulation?

Fig. 10 & 11: Please say explicitly "coccolithophores and *Phaeocystis*" (Fig 10). Please state which simulation is shown.

**References**

Berman-Frank, I., Cullen, J. T., Shaked, Y., Sherrell, R. M., & Falkowski, P. G. (2001). Iron availability, cellular iron
quotas, and nitrogen fixation in Trichodesmium. *Limnology and Oceanography*, *46*(6), 1249–1260. https://doi.org/10.4319/lo.2001.46.6.1249

Breitbarth, E., Oschlies, A., & LaRoche, J. (2007). Physiological constraints on the global distribution of *Trichodesmium* – effect of temperature on diazotrophy. *Biogeosciences*, *4*(1), 53–61. https://doi.org/10.5194/bg-4-53-2007

Krumhardt, K. M., Lovenduski, N. S., Long, M. C., Levy, M., Lindsay, K., Moore, J. K., & Nissen, C. (2019).
Coccolithophore Growth and Calcification in an Acidified Ocean: Insights From Community Earth System Model Simulations. *Journal of Advances in Modeling Earth Systems*, *11*, 2018MS001483. https://doi.org/10.1029/2018MS001483

Lancelot, C., de Montety, A., Goosse, H., Becquevort, S., Schoemann, V., Pasquer, B., & Vancoppenolle, M. (2009). Spatial distribution of the iron supply to phytoplankton in the Southern Ocean: a model study. *Biogeosciences*, *6*(12), 2861–2878. https://doi.org/10.5194/bg-6-2861-2009

Le Quéré, C., Buitenhuis, E. T., Moriarty, R., Alvain, S., Aumont, O., Bopp, L., … Vallina, S. M. (2016). Role of zooplankton dynamics for Southern Ocean phytoplankton biomass and global biogeochemical cycles. *Biogeosciences*, *13*(14), 4111–4133. https://doi.org/10.5194/bg-13-4111-2016

Luo, Y.-W., Doney, S. C., Anderson, L. A., Benavides, M., Berman-Frank, I., Bode, A., … Zehr, J. P. (2012). Database of diazotrophs in global ocean: abundance, biomass and nitrogen fixation rates. *Earth System Science Data*, *4*(1), 47–73. https://doi.org/10.5194/essd-4-47-2012

Monteiro, F. M., Bach, L. T., Brownlee, C., Bown, P., Rickaby, R. E. M., Poulton, A. J., … Ridgwell, A. (2016). Why marine phytoplankton calcify. *Science Advances*, *2*(7), e1501822. https://doi.org/10.1126/sciadv.1501822

Peperzak, L. (1993). Daily irradiance governs growth rate and colony formation of Phaeocystis (Prymnesiophyceae). *Journal of Plankton Research*, *15*(7), 809–821. https://doi.org/10.1093/plankt/15.7.809

Timmermans, K. R., Wagt, B. Van Der, & de Baar, H. J. W. (2004). Growth rates, half saturation constants, and silicate, nitrate, and phosphate depletion in relation to iron availability of four large open-ocean diatoms from the Southern Ocean. *Limnology and Oceanography*, *49*(6), 2141–2151. https://doi.org/10.4319/lo.2004.49.6.2141

Wang, S., & Moore, J. K. (2011). Incorporating Phaeocystis into a Southern Ocean ecosystem model. *Journal of Geophysical Research*, *116*(C1), C01019. https://doi.org/10.1029/2009JC005817

Ward, B. A., Dutkiewicz, S., Moore, C. M., & Follows, M. J. (2013). Iron, phosphorus, and nitrogen supply ratios define the biogeography of nitrogen fixation. *Limnology and Oceanography*, *58*(6), 2059–2075. https://doi.org/10.4319/lo.2013.58.6.2059

---

## Author Comment (AC1) · 29 Aug 2019

Dear readers, we would like to draw your attention and remind that, when submitted, the discussed manuscript was supplemented with three videos on the model simulated chlorophyll "a" distribution for diatoms, haptophytes and prokaryotes in the Souther Ocean for the time period from August 2002 to April 2012. These videos also depict comparisons with available in situ observations and clearly illustrate phenology/bloom development for these three phytoplankton types over the aforementioned time period.

Video supplement

Simulated distribution of diatom chlorophyll concentration in the Southern Ocean https://doi.org/10.5446/42871

Simulated distribution of haptophytes chlorophyll concentration in the Southern Ocean https://doi.org/10.5446/42873

Simulated distribution of prokaryotes chlorophyll concentration in the Southern Ocean https://doi.org/10.5446/42872

---

## Referee Comment (RC3) · Anonymous Referee #3 · 6 Sep 2019

General comments:

The paper by Losa et al. (2019) describes marine ecosystem model development in order to better represent the marine phytoplankton community in the Southern Ocean. This is a worthwhile endeavour, and, if done correctly, could lead to major improvements in our understanding of marine ecosystems, and global biogeochemical cycling in the high latitudes. However, unfortunately, the current manuscript has multiple severe shortcomings, that - in my view - preclude a publication in Biogeosciences in its current form. In essence, I have strong reservations about (1) the lack of a scientific purpose of the paper presented, (2) the modelling work itself, which is not following standard protocols in the field with regard to model set-up, testing and quantitative validation, and (3) the interpretation of the results as a result of point (2). Since reviewers

1 and 2 have done an excellent job in pointing out specific shortcomings of the current work already, I would like to highlight my concerns with regard to these three general issues.

1. No scientific hypothesis pursued

As becomes clear from the abstract, and later throughout the entire manuscript, no concrete scientific question is pursued by the paper. Hence, this paper is not suitable for publication in Biogeosciences in its current state. Rather, the current development work should be published in journals such as "Geoscientific Model Development," or "Ecological Modelling" instead. For a successful submission to any of these journals, however, a proper documentation of the model, a documentation of its sensitivity to parameters chosen and assumptions made, and thorough model evaluation and validation would be required.

2. Model set-up characterised by substantial flaws that preclude a conclusive assessment of the main model dynamics and conclusions of this paper

This becomes evident from an under-referenced introduction section, which is limited in scope and does not point out current gaps in marine ecosystem development, a poor, incomplete and flawed methods section that reveals major critical issues with the model set-up, spin-up, the initialisation and documentation of the runs conducted, and major critical issues with model stability and tuning. The paper is further characterised by a complete lack of a suitable model evaluation that would show that the model decisions taken (e.g. two diatoms, Phaeocystis parameterisation, etc.) are sound and robust (how do your nutrient patterns look like compared to observations, what is your NPP and export, what are your zonal and vertical standing stocks of carbon and chlorophyll-a for each of the biological tracers, what is the zonal and vertical biomass of your zooplankton functional types, how sensitive is your model to each of
the parameters of your new tracers, are these values and dynamics realistic?), a profoundly disorganised documentation of model equations and parameters (no units in tables, wrongly declared equations, random selection of topics presented), followed by an erratic presentation of random model results, and finally, as a consequence of the above, a questionable interpretation of the model results that documents a severe lack of understanding of the model behaviour. ██████████████████████████████ ███████████████████████████

An example that will illustrate why I have strong doubts about the methodology used is found in the description of the model set-up described in the Methods section. Here, the authors mention, that

- they use a spin-up period of 6 years for the physical model, which is far too short to equilibrate even the surface layer of the Southern Ocean,

- then initialise biomasses and nutrient fields with results from a model coupled to another, completely different biogeochemical model (Recom-MITgcm) without carefully validating these fields,

- and finally run the coupled model for 13 years only, without spinning up the biogeochemical module first (the authors point out that coccolithophores die out by the end of the simulation, which shows that the model isn't in a stable equilibrium yet for that target region),

- and present results from random months within the first and last few years of the simulation (across all figures we see patterns from: January, June, July, August, December 2003 and January, February, March 2004, February 2008, March 2012), where the biogeochemical model is not even remotely in equilibrium.

Needless to say that common practice in the field is to carefully spin up the physics for multiple decades, then couple to the biogeochemical model, initialise that model

from observations (available as gridded and extrapolated products in standard netcdf format, so no excuses here), spin-up the coupled model for another ten years or so until the biogeochemistry does not drift anymore, and then run the model with varying forcing and finally quantitatively (!!! -> Taylor diagrams, other model evaluation metrics, observational constraints, etc.) analyse the 5- or 10-year averages (or whatever is appropriate to filter out inter-annual and multi-decadal variability in the target region) of the last few decade of the simulation, provided that the point of the study is to present average biogeographic patterns (well, if that was the point of the study, other approaches may hold for other scientific questions). And needless to point out that many of the required observations for the model are actually carefully processed by and available within the team of senior co-authors. ███████████████████

Since the modifications of the biological module are documented even more poorly, there is no quantitative model evaluation, and neither is there a comprehensive documentation of carefully designed sensitivity tests that would allow us to understand the model sensitivity to the new parameterisation, it is impossible to critically evaluate the biological results of this work. Furthermore, the reported model instability with a high sensitivity of model results to parameter choice, as well as the disappearance of major functional groups (coccolithophores) throughout the simulation shows that this model configuration has major stability issues and a huge drift in the biological compartments with likely substantial consequences for productivity, nutrient distribution and biogeochemical cycling in this basin, and is thus unsuitable for publication at this point in time, as it clearly needs to be further tested.

3. Flawed analysis and interpretation of model results

Due to the above issues, any interpretation of the findings in this paper will obviously suffer from major uncertainties due to a lack of conclusive sensitivity tests and a thorough validation of the approach, and thus must remain entirely speculative. In its current form, it is impossible to say whether (1) the implementation of two diatoms is

meaningful and leads to a gain in model performance (most models suffer from an over-estimation of diatom biomass in this ocean basin), (2) the Phaeocystis module is correctly implemented, (3) the modifications of coccolithophore physiology are justified (since the author claims that they die out anyway), and (4) if, in fact, the model produces reasonable biogeography, primary production, depth patterns of biomass, and carbon standing stocks in this basin that would point at an actually improvement of the model compared to previous versions.

As an example for a section that make me doubt the validity of the entire analysis, I would like to point out the following point:

In figure 2, model results are presented for the month of July, i.e. austral winter, where biomass in this basin is clearly very low, as it is dark. Yet, much ado is made about "dominance patterns" of specific PFTs during that time. However, the quantification of dominance on very low background biomass values is meaningless, as we're looking at percentages of zeros, essentially. Furthermore, plankton biogeography in winter is very likely strongly linked to sea ice dynamics and factors not represented in current (global) models (resting spores, etc.) – whereas the authors even discuss spurious dominance patterns for areas clearly covered by ice, e.g. in the Dutkiewicz et al. (2015) set-up (which, apparently, isn't coupled to an ice model in its original configuration). In addition, dominance patterns are compared for a random year of the simulation (2003 and 2004), where biogeography is still reported to contain major drifts (as coccolithophores are reported to die out). This makes the claimed "improvement" of the model quality in terms of phytoplankton biogeography highly questionable. Furthermore, since the MIT-gcm set-up is likely global in scale (we do not know, as this is not described), and since the Dutkiewicz et al. (2015) set-up was likely tuned at the global scale, a comparison with a regionally tuned model is just simply unfitting – a global model will never do as well as a model tuned for a specific region, and it doesn't have to. We do not know, how the current set-up was tuned (I assume the MITgcm is still run in its global set-up, and the rest of the ocean is just not shown on the maps), and how it performs for the

rest of the ocean.

As a marine ecosystem modeller, I am all in favour of seeing further modelling work published that would illuminate the drivers of phytoplankton biogeography, and the respective role of competition, predation and environmental niche dynamics in shaping phytoplankton communities and associated ecosystem services. However, unfortunately, due to the poor quality of the submitted manuscript, my recommendation for this manuscript is forced to be: reject, revise and re-submit.

Note to the author: The posted videos are in no way helpful for a quantitative (!) evaluation of the paper.

———

Specific comments:

Abstract:

Lines 8-14: Revise. Vague and unsubstantiated claims. Be quantitative. Give numbers. What scientific questions would you like to address with your model?

Intro:

General: Does not identify major challenges in the field addressed by this work. Does not introduce Southern Ocean community structure and function. Poorly referenced. Lack of original references. Fairly irrelevant discussion of the multiple meanings of PFT, and the criteria in Le Quéré et al. (2005).

Lines 16-23: Poorly referenced. Give evidence for each of your claims.

Line 28-30: I disagree. Not the main planktonic calcifiers, if we trust modern estimates. The main PHYTOplanktonic calcifiers. See Buitenhuis et al. (2013) and Buitenhuis et al. (2019).

Line 32: References for Phaeocystis contribution to biogeochemical cycles missing.

Lines 32-36: Same thing. Original references for the importance of named groups missing.

Lines 49-50: "..different algorithms... use various approaches..." Be precise. What is relevant in this context. Focus on the issue with carbon to chlorophyll conversions. Your model calculates biomass in carbon units and derived chlorophyll-a biomass. You validate against chlorophyll-based algorithms (e.g. PHYSAT). What are the challenges? How can DARWIN help?

Line 54-55: Do not use multiple names for the same thing. One term – one meaning.

Line 56: No. Le Quéré and Follows were not the only researchers to initiate efforts in PFT modelling. They, as many others, contributed important material and thoughts to an ongoing discussion and effort. Have you read Hood et al. (2006)? Anderson et al. (2005)?

Line 61: NOBM is not the only model with 4 PFTs. In fact many others do. BFM, GFDL, BEC (see e.g. Krumhardt et al. (2019)), etc.

Line 63: No. DARWIN in its 2007 configuration does not contain all PFTs proposed by Le Quéré et al. (2005).

Line 75 ff: What is the point of your paper? You state no scientific purpose. Model development should be published elsewhere.

Methods:

General: Poor and disorganised. Documents strong lack of understanding of model structure and functioning.

Lines 85-876: Use consistent nomenclature to designate the PFTs included in the model. Your statement here lists other groups than those, e.g. represented in Figure 2.

Lines 90-100: You must show all parameters and all equations for novel tracers. Need

to document how diatoms differ, justify coccolithophore modifications, give equations for full Phaeocystis module. Justify each and every parameter, back up with literature values. You also need to show the results of your sensitivity analysis. Any deviation from the conventional PFT equation structure, i.e. the inclusion of life stages needs to be carefully motivated and evaluated. Get Phaeocystis data for the Southern Ocean. Evaluate fraction of biomass in colonial versus single-cell stage. Yes, there is data.

Line 100: Why did you choose to replace other large phytoplankton with Phaeocystis, and not nitrogen fixers? Nitrogen fixers have been shown to only play a very minor role in the Southern Ocean due to their temperature limitation. On the contrary, other large phytoplankton species, such as dinoflagellates, are regularly observed in this ocean basin. This decision seems questionable.

Line 105ff: The light module is completely irrelevant in this context. Move to appendix, or omit.

Line 109; Same holds for kCDOM. You don't ever discuss this. Omit.

Table 1: No units given. No references included. What is mfunc? It does not appear in equations (3) – (6). Use consistent abbreviations and names for all PFTs throughout entire paper.

Line 118: I am 100% positive it was not Geider et al. (1998) who invented the growth functions. This parameterisation is quite similar to what Riley developed in 1946 (!). Please check your referencing.

Lines 119 – 123: Not all parameters defined, all functions named must be given as equations. Incomplete set of equations. For example, what is f(k_sat)? Formulation for Phaeocystis is clearly not as given in (3) – (6).

Line: 130: This is not a Holling II function. Also, you state that DARWIN has two zooplankton types on line 84. Where are the zooplankton traits? Need to report. In general, the role of zooplankton grazing pressure on plankton biogeography is not

discussed in this manuscript. Since zooplankton usually play a vital role in determining the relative biomass fractions and dominance patterns in these models, the role of top-down control must be addressed else in the manuscript.

Line 140ffff: Where does this parameterisation come from. Why did you not choose to follow e.g. Schoemann et al. (2005), the most comprehensive review on Phaeocystis dynamics. What temperature function did you choose, and why? Do you use one or two tracers for Phaeocystis (Phaeo versus Phaeo_cell)? Are there two combined? Do you get realistic fractions of biomass in colonial stage?

Line 144-145: Rewrite. Utterly unclear.

2.1.2. Physics:

6 years is not a spin-up. What are you spinning up? What is the temporal resolution of your simulation? Are you using a global model? Documentation of model set-up is insufficient – do not just say it was "similar" to something else. And, by the way, it was not similar to the model used in Taylor et al. (2013), since these authors used a completely different coupled GCM-biogeochemical model. Why do I care about light penetration, when you haven't even described the ice module? Which ice-model do you use? Is it dynamic?

2.1.3 Biogeochemical tracer initialisation

██████████████ Initialise from observations, not from unvalidated model runs conducted with a completely different model. These observations are generated in your group. Spin up the biogeochemistry. Evaluate (quantitatively) your NPP, export, nutrient fields, total chlorophyll-a before you even start thinking about a reference run. Spin up your biogeochemistry module. Initialisation of biomasses from Recom is absolute nonsense. Below you describe your satellite validation data. Use that as an initialisation. Do not point to Taylor et al. (2013), as this leads me to Losch et al. (2010), and they use a different model. Use new World Ocean Atlas nutrients, for instance. Equilibrate your model. If major Southern Ocean players show a strong drift in their biomass (e.g. your coccolithophores die out, as you write), then there is a major problem in your model.

2.2.1 In situ observation

Cryptic and poorly written. List all data sets used. Which units does the data have, which spatio-temporal resolution, are these surface data, or are they depth-resolved. Pointing at another study is insufficient. How do you treat, bin, grid, quality control this data to be useful for model validation? Do you convert any of this to biomass, in order to evaluate biomasses? What about the vertical pattern? What about the comprehensiveness of this data – how many months, years, seasons, latitude bands, depth levels does your data cover (all this needs to be described in the main text). What do the "measurements" by Smith et al. 2017 comprise.

Line 173: Table 2, in fact, does not contain any useful information.

Figure 1: What kind of observations do you show? Cite the appropriate reference. Why do you show the Longhurst provinces? These are of no relevance in the rest of the paper. They only confuse the reader. Else aggregate and evaluate your model based on these provinces throughout the entire manuscript.

2.2.2 Remote sensing

Why would you use the old 2008 PHYSAT product? There is an updated algorithm using a neural network approach. This should be far better than this outdated version. What type of "abundance" are you referring to, and how can this be compared to the model output? You mention this data is high resolution – did you regrid the data to fit the model grid?

Results & Discussion

General: This section is characterised by a complete chaos in terms of information presented, spatio-temporal scales shown. None of the figures are in any way useful to

evaluate the quality of the model development you presented in your methods section. All you show is colourful surface ocean maps, and most of these maps are useless. Redo entire section from scratch. Does not contain any discussion, as there is no quantitative evaluation and interpretation of the work.

Line 205: You do not discuss phenology in this section.

Lines 207: The satellite estimate you are comparing your results against is not "the truth", merely another algorithm. Be sure to correct. Why, on Earth, would you compare winter values for the Southern Ocean? It's dark in winter, and background biomass is likely very challenging to model, as it will depend on features not included in the current generation of ecosystem models, such as resting spores, overwintering strategies, ice associations, etc. Hence, your entire dominance analysis is severely flawed, see detailed comment above. Why don't you evaluate e.g. a 5-year seasonal average over December – February period (most commonly used)?

Lines 219: This comparison appears flawed. As far as I know, Dutkiewicz et al. (2015) was tuned for a global fit.

Figure 2: Names in legend do not match those given for your PFTs elsewhere in the manuscript. Do not show results for random months and years.

Lines 213-2014: This must be included in the main text, not the supplementary. Along with a thorough discussion of the parameter choices in the methods section.

Lines 220 – 223: This, if true, would deserve a paper on its own. Unfortunately, you do neither quantify nor show phonological patterns. All we see in figure 2 is dominance plots for two selected months in random years.

Line 225: Claimed "augmentation" hasn't been shown. What is augmented? NPP? Export? Nutrient fields? Silicification and calcification rates? Opal export? Relative biomass fractions? None of this has been shown so far.

Line 228: "in agreement with" How is anything shown so far in agreement with anything.

No quantitative evaluation has been provided.

Line 230: "..results are supported.." How, and in which way? Quantitative comparison with data is missing (model evaluation).

Line 233-236: "However, …." This shows that your model is not stable, not in any kind of sensible equilibrium, with likely large consequences for all tracers associated with global biogeochemical cycles, and thus not publishable yet. Absolute game stopper. And by the way, in fig 8 you show us coccolithophore biomass for March 2012, which is "towards the end of your simulation", and thus it looks like coccolithophores are abundantly populating the low latitude Southern Ocean, with a substantially overestimated chlorophyll-a biomass relative to the SynSenPFT estimate.

3.3.

"To cope with the aforementioned chaoticity of the system…."

I stop my detailed review of this section here. All the remainder is speculation. If the model couldn't be tuned to reliably reproduce the biogeography of major players, then this paper shouldn't have been written, but the model should have been developed further.

Figure 3: Why do you show another month? Evaluate model results on same spatio-temporal scales throughout entire manuscript. Compare total model chlorophyll-a estimate to total satellite chlorophyll-a. Compare group-specific chlorophyll-a to your different SynSenPFT etc. algorithms. Quantitatively.

Figure 4: Label your axes. What is the total biomass level in each of these plots? There seem to be far too many nitrogen fixers at 40South. Same for coccolithophore contribution in this area. Are you under- or overestimating total biomass here? Why have you chosen to present a zonal view of the community structure? In figure 1 you show Longhurst biomes. Can you evaluate the depth pattern? What is the link between zooplankton biomass dynamics and the relative fractional contribution of individual groups

to total biomass? Furthermore, Phaeocystis is not usually known to dominate biomass between 60-50S.

Figure 5: Why, again, do you show another month? Compare total chlorophyll-a to observations. Compare the groups to observational estimates from space, or in situ data. Do not just show model nutrients. Compare quantitatively with observational estimates, e.g. from World Ocean Atlas.

Figure 6: Why would you only show modeled PIC? Compare to observational PIC from space. Where is the "Great Calcite Belt"? Do you represent it well? And if your coccolithophores die out throughout your simulation, how does this affect PIC patterns? 2004 does not seem to represent a "typical" year, as there is no typical year in a model with a strong drift.

Figure 7: Same comments as Fig 2. Winter patterns unrepresentative, and very likely very tricky to model. Where is the "Great Calcite Belt"?

Fig 8: Merge with Fig. 5.

Fig. 9 - 11: These figures do not contain any quantitative information. Omit.

4 Concluding remarks and outlook

Same as above. It is impossible for me to evaluate this section. Since the modelling work does not seem to be up to the standard in the field, the model evaluation is missing and the analysis of the results is severely flawed, it is impossible for me to judge the scientific interpretation of the findings. Any conclusions based on this work must remain mere speculations at this point in time.

**Note from Copernicus Publications**: Some parts of this comment have been redacted on 28 April 2020 on request by the BG Executive Editors.

---

## Author Comment (AC2) · 2 Jan 2020

**Response to the comments of reviewer 2**

**Summary**

In this study, Losa et al. present a version of the DARWIN model, which they modified for the Southern Ocean (SO) application
presented in this manuscript. In order to better represent the SO phytoplankton community structure, which mainly consists of
silicifying diatoms, calcifying coccolithophores, and colony-forming Phaeocystis, the authors have added a second, lightly
silicified diatom plankton functional type (PFT) to their model (in addition to a heavily silicified one which was already
included in the model before) and have made small modifications to the parametrization of coccolithophores in a first step
(their reference simulation).

Subsequently, motivated by problems in keeping both coccolithophores and Phaeocystis alive in their reference simulation,
the authors have implemented a life cycle switch (based only on the surrounding iron concentrations) for the Phaeocystis PFT
to simulate both solitary and colonial forms of this phytoplankton type (PHAEO simulation). In this manuscript, the authors
present a comparison of the simulated phytoplankton community structure to those suggested by satellite-based PFT algorithms
and pigment data (the latter for the PHAEO simulation only). In my opinion, the model development study by Losa and co-
authors is valuable, as current global models often struggle to correctly represent the SO phytoplankton community. Efforts to
improve upon this are needed, given the importance of this ocean basin for global biogeochemistry and climate.

I think the manuscript is in principle suitable for publication in Biogeosciences. However, I cannot recommend the publication
in its current form, as I have serious concerns surrounding the model behavior (the extinction of individual PFTs at the end of
the reference simulation is worrisome). Furthermore, I think that 1) the chosen PFT parameters and changes done to the model
have to be better motivated in the SO context of this study, 2) the used model parameters and parametrizations need to be
better documented throughout the manuscript and limitations need to be discussed (especially surrounding the parametrization
of the life stages of Phaeocystis), and 3) the impact of the changes and chosen parameters should be more thoroughly assessed
by targeted sensitivity simulations. Below, I first summarize my comments into a few general points and then list all my
detailed comments, which should be addressed before the manuscript can be accepted for publication.

We thank the reviewer for the constructive comments on the manuscript. Our author's replies are presented in blue, labeled
"R:" and follow each reviewer's comment. The changes in the revised manuscript according to the suggestions are presented
in blue.

**General comments**

Below, I will list my general comments, which should be thoroughly addressed before the manuscript can be published:

1) The "extinction" of either coccolithophores or Phaeocystis (Antarctica) in the presented reference simulation deeply worries me. Before this manuscript can be accepted for publication, the authors should understand where this is coming from and fix it, as I currently do not understand how this can happen, given that (based on observational data) their biogeographies in the SO do not overlap completely in space and time (meaning that there should be room for both to exist). Since this model behavior implies a substantial drift in the biomass distributions in the simulations assessed here, it can be expected to lead to a substantial sensitivity of the presented results to the chosen analysis year (see also point 7).

R: As the reviewer mentioned "based on observational data … that there should be room for both to exist". The question addressed in this paper is what exactly makes/provides this room and how well (if ever) this represented in the model. For experiment REF as well as for other sensitivity experiments (overviewed in the Supplementary Material) there were not sufficiently enough differences between the traits assumed for coccolithophores and "other large" (or *Phaeocystis* analogue). As a result, it took longer for the model to get in a quasi-steady state and finally lead to just one of "similar" PFTs survived (taking over for another PFTs). Thus, in experiment REF coccolithophores do not survive and *Phaeocystis*-analogue indeed represents haptophytes in general. Hence, the experiment REF represents diatoms and haptophytes after reaching a quasi- steady state, but cannot distinguish among haptophytes. In original Darwin-2015 model (Dutkiewicz et al. 2015) "other large" did not survive. In this respect, PHAEO configuration with additional differences introduced in the traits of these PFTs, is the fix.
We explained better in the revised manuscript (L346 – 350).
However, we understand the reviewers concern and realize that it was a mistake to show results such as this without fully explaining the point. We now only show results from after the quasi-steady state.

Furthermore, based on the information included in the current version of the manuscript, I don't understand how the subsequent changes made to the parametrization of Phaeocystis (i.e. including life cycle transitions) solved this problem, which should be discussed in more detail by the authors.

R: The additional differences introduced in the parametrization of *Phaeocystis* makes coccolithophores competitive among phytoplankton of larger cell size (or colonies) that requires higher nutrients concentration to grow and/or among PFTs of similar size (small diatoms and *Phaeocystis* solitary cells) that have of higher palatability factor to be grazed.
This is now more clearly stated in the revised version (L 584 – 589)

2) In the method section, a detailed description of the assumptions surrounding the parameter choices of the different PFTs as well as laboratory studies backing up the chosen numbers (Table 1) is currently lacking.

R: Table 1 contains only the parameters used in the parametrizations crucial to drive the differences/diversity in the considered PFTs traits. Most of the biogeochemical model parameters were taken from Dutkiewicz et al. (2015) and from detailed laboratory studies conducted by (Trimborn et al. 2017). We clarified it in the revised version of the manuscript. In the Supplementary Material (Tables S1-S4) we compile information on the parameters chosen for the various model configurations exploited within this study.

We added the following sentence in the text (L153–155):

"Note that most of the biogeochemical model parameters used in our study are taken from the original study by Dutkiewicz et al. (2015) and from detailed laboratory studies conducted by (Trimborn et al. 2017). Hence, Table 1 contains only the parameters used in the parametrizations crucial to drive the differences/diversity in the considered PFTs traits"

Section 2.1.1 and Table 1 are currently incomplete in their description of the parametrizations and parameters used in this study (i.e. e.g. some parameters are missing, no units are given).

R: Units were provided in the text introducing model parameters in the parametrizations (pages 4 and 5, Section 2.1.1). We revised Table 1 and now it also includes the units.

More specifically, regarding the coccolithophores, the authors do currently not motivate why the applied changes to the parametrization (as compared to previous global applications of DARWIN) are justified for the SO (e.g. by relating them to the coccolithophore community in this ocean basin).

R: Indeed, we first mentioned the parameter modifications in lines 93-95 of the original version (supported by references Nejstgaard et al. (1997), Huskin et al. (2000), Paasche, 2001; Iglesias-Rodríguez et al., 2002) and explained them in more detail in the section 3.3, lines 254 – 265 and 272 – 277 of the original submitted manuscript. The discussed changes in the parameters for coccolithophores such as palatability factor and low half-saturation for nutrients are in consistence with what is, generally, known about this PFT. Moreover, in the study by Monteiro et al. 2016 a version of the Darwin model was applied also globally, and the authors reported and justify, for instance, that grazing protection (introduced via palatability factor) appears to favor coccolithophores in (sub)polar regions.

We improved the text to clarify it (L384-393):

"Our assumptions on low palatability factor of coccolithophores are, nevertheless, backed up by the studies of Nejstgaard et al. (1997), Huskin et al. (2000), Losa et al. (2006) and Monteiro et al. 2016. Based on their laboratory experiments, Nejstgaard et al. (1997) and Huskin et al. (2000) concluded that coccolithophores do not influence the microzooplankton growth due to its "stony" structure. In the study by Losa et al. (2006) on optimized biogeochemical parameters the authors showed that the coccolithophores bloom was associated with low grazing pressure. While the exact mechanisms of how this PFT use the coccolith to protect itself against grazing is not fully understood (Monteiro et al. 2016), the ability of coccolithophores to escape grazing control has "relatively well-supported evidence" (see Monteiro et al. 2016 for review). In addition, the high affinity of coccolithophores for nutrients (for phosphate and iron to a larger extent than for nitrogen, Paasche 2001) makes them strongly competitive in environmental conditions with declining nutrient concentrations (Paasche, 2001; Iglesias-Rodríguez et al., 2002), for instance under strong ocean stratifications or nutrient consumption by other PFTs."
We know also included references to studies by Krumhardt et al. (2017) and Krumhardt et al. (2019).

In the introduction we also added that (L64-71): "Coccolithophores biogeography was investigated globally by Monteiro et al. (2016), Krumhardt et al. (2017) and Krumhardt et al. (2019) and particularly for the Southern Ocean by Nissen et al. (2018). With respect to specific coccolithophores traits, the study emphasized the high nutrient affinity of the coccolithophores (Krumhardt et al. 2017) and high grazing protection of this PFT (Monteiro et al. 2016). Nissen et al. (2018) reported on higher grazing pressure on coccolithophores relative to those on diatoms. While in the study by Krumhardt et al. (2019), the authors used low grazing pressure on coccolithophores relative to those on diatoms. Krumhardt et al. (2019) related the distribution of coccolithophores to a specific temperature function of its growth rate."

In the introduction we also now explicitly state as one of the hypotheses we test in the study (L84-88):
"Distribution of coccolithophores in the Great Calcite Belt is not necessarily controlled by temperature (Smith et al., 2017) but determined by the ability of this PFT to escape grazing because of their exoskeleton (Nejstgaard et al., 1997; Huskin et al., 2000, Monteiro et al., 2016), and to grow under nutrient depleted conditions (especially phosphate and iron) (Paasche, 2001; Iglesias-Rodríguez et al., 2002, Krumhardt et al., 2017). These characteristics of coccolithophores would make them more competitive among other phytoplankton of larger or similar size, small diatoms and *Phaeocystis.*"

Regarding Phaeocystis, the manuscript could be greatly improved by including a more thorough discussion on the limitations of their current parametrization in the model, as important aspects surrounding their life stage transitions (e.g. light) are currently not accounted for.
R: Thank you for the suggestion. In the revised version of the manuscript we introduced a section "Limitation of the study" and extended the discussion on limitations regarding *Phaeocystis* (L528-533), were we state that the light was not considered according the recent findings of Bender et al. (2018):

"*Phaeocystis* colony formation: in this study, we use very simplistic approach to parameterise life cycle transition of *Phaeocystis* given just one model tracer. In our model this transition is triggered only by iron variability (as reported by Bender et al. 2018), but not by light availability (as previously reported by Pererzak, 1993). Since we reported on our first trial, it is worth keeping in mind that the model is expected to be sensitive to the differences we specify for the mortality and grazing rates and iron uptake for colonial and single cell stage. A careful model calibration of these parameters could further improve
     the model performance."

     Additionally, the authors should comment on the usefulness of simulating both life stages within a single model tracer, as this
     is important information for those wanting to implement Phaeocystis into their own model.

R: We were motivated by the necessity to prescribe additional differences in the traits assumed for coccolithophores and
     *Phaeocystis*. It was the simplest approach we came up by following the approach of Popova et al. (2007) and the study by
     Bender et al. (2018).
     We commented on that now in lines 165-168.
     "Note that in the model *Phaeocystis,* independent of the life stage – colonial phase or solitary cells, – is considered as one tracer. However, the assumed morphology and, therefore, physiology (mortality rate, $r_{j,k}$, $ksatF_e$, sinking rate) differ as
     described above. We have not performed any sensitivity experiments with respect to the new parameters. However, we expect
     the model to be sensitive to their specification since it will also determine the competition between *Phaeocystis* and small
     diatoms."

We further commented on limitations (L528-533) as written above.

     Nevertheless, the results shown and discussed allowed us to conclude (L582-589):
     "This parameterization of morphological shifts indeed allows for co-existence of the two types of haptophytes corroborating
     our third hypothesis on the dependence of *Phaeocystis sp.* life stages on iron availability. By considering two life stages of

*Phaeocystis* we introduce additional differences in the traits, which along with assumed physiological parameters for
     coccolithophores makes coccolithophores competitive among phytoplankton of larger cell size requiring higher nutrients
     concentration to grow or/and among PFTs of similar size – small diatoms and *Phaeocystis* solitary cells – but of higher
     palatability factor to be grazed. These additional differences in the traits of distinct haptophytes, coccolithophores and
     *Phaeocystis* allows these groups to coexist (e.g. along the Subantarctic and Polar fronts)."

     Furthermore, the manuscript currently lacks a sensitivity analysis assessing e.g. the impact of the changes applied to the
     coccolithophore parametrization (in order to support what is in my view currently largely a speculation on the drivers of their
     biogeography in their model as important plots are not shown) or the impact of parameter choices (e.g. regarding those of
     Phaeocystis) on the simulated biogeography.

R: Within the scope of testing the formulated hypothesis (now explicitly written in the introduction – see L79-90), several
     sensitivity experiments have been performed. In the Supplementary Materials, we only reported on the simulations that most
     contributed to obtain the concluding results. Moreover, the changes in the coccolithophores physiological parameters are
     strongly backed up by previous studies (please see our responses to the detailed comments). As about the sensitivity to the traits (parameter choices) specified for *Phaeocystis,* experiment REF is considered as one of the sensitivity studies. And the comparison of the final PHAEO experiment to REF illustrate allows to infer on the impact of the assumed traits on the simulated PFT biogeography. (Please see our responses to detailed comments)

3) In general, important results (e.g. the change in the simulated phenology when implementing a second diatom PFT or the drivers of the simulated coccolithophore biogeography) are currently getting a bit lost in the manuscript. As these aspects are highly relevant for the modeling community and are the parts for which the manuscript goes beyond a pure model development paper, these aspects deserve more room (in text and figures).

R: We now try to straighten out the presentation and discussion of our results on consequences of including small diatoms, for instance by explicitly showing and discussing the diatom phenological indices in line with Chla distribution of small and large diatoms.

We present figure R.2.10 depicting spatial distribution of small diatoms at lower latitudes and large diatoms at the higher latitudes of the Southern Ocean (as it was also shown in figure 3 of the original manuscript). Figure R2.8 presents phenological indices for the Southern Ocean diatoms showing, for instance, earlier bloom start date and Chla maximum date for small phytoplankton and later bloom start and maximum date for larger diatoms abundant at higher latitudes. When compare the phenological indices with dominance plot, it is seen that the PFT dominance plots, indeed, to some extent reflects the PFT phenology. It is why when discussion on model deficiencies in reproducing PFT phenology (and PFT composition) and reporting on the main results of sensitivity tests (Supplementary Material) we showed PFT dominance plots. The PFT dominance that we show in the main manuscript and agreed (qualitatively) better with the PHYSAT dominance (among different sensitivity experiments) was only possible to obtained by considering two size classes for the diatoms.

We also would like to emphasize more on the results presented in Figure 5 (Figure 6 in the revised version) in support to the the discussion on coccolithophore biogeography. This figure depicts Southern Ocean spatial distribution of diatoms, coccolithophores and *Phaeocystis* along with silica, iron and phosphate for a particular March 2004 (in the revised version we will show March 2008 or February 2008). We chose to show a particular month of a year (could be any after the steady state) but not climatological monthly mean to clearly show patterns of the distributions: 1) the abundance of coccolithophores in the area with very low phosphate; 2) co-existence of this PFT with small diatoms north of the subantarctic front where silica is presented in lower concentrations than in higher latitudes but still sufficient to support the growth of small diatoms and co-existence coccolithophores with *Phaeocystis* solitary cells north of the Southern Antarctic Circumpolar Current Front in the areas with low iron concentration. In first case coccolithophoes can grow due to high affinity of coccolithophores to phosphate and iron. In second case it survives due to lower palatability factor that makes the coccolithophores competitive with small cells of diatoms *Phaeocystis*.

Currently, the conclusions drawn by the authors are not fully backed up by the simulations that are discussed and the plots that are shown in the manuscript, making it often impossible for the reader to evaluate what the authors base their arguments on.

R: We revised the manuscript to make it clearer. See our responses to the specific comments below.

4) Throughout the manuscript, the authors use the term "phenology", which typically refers to the annually reoccurring characteristics of the phytoplankton biomass evolution and can be characterized by the timing of e.g. the phytoplankton bloom start or the bloom peak. However, in the current version of the manuscript, "true" phenology is never presented and often only individual months of the simulated biomass fields are shown and discussed, which gives no information on the phenology (additionally, a definition of "phenology" and how it is assessed is missing in the method section).

R: We opted to remove the term phenology in the paper since we were using it to refer to the dynamic of the PFTs but without explicitly showing phenological metrics. We actually have calculated the metrics but including these results plus discussion would make this manuscript too long and this subject will be explored in another paper.

Figure R2.8 and R2.9 show the phenological indices.

In order to e.g. emphasize the importance of including two diatom PFTs in a SO model (where by the authors claim to have fixed the problem of many models, namely too early blooms), the authors should show the simulated phenology metrics in the revised version of the manuscript (e.g. maps of bloom timing in the "old" model version as compared to the improved setup and those derived from satellites).

R: As mentioned above, the initial idea of the manuscript was also to include information on the timing of the phytoplankton blooms but we realized that the study would be too complex and diverse on topic to be summarized in one manuscript. We plan a dedicated study on the phenology of the PFTs blooms in the Southern Ocean soon.

5) Throughout the paper, the authors present very little quantitative evaluation of the simulated phytoplankton distributions, which should be improved in a revised version of the paper. Currently, the included HPLC data are only used for the PHAEO simulation (by plotting the observational data as scattered dots on top of maps, which is very hard to evaluate for the reader), but should also be included for the "old version" of the model and the reference simulation in order to actually show the asserted improvement in model performance.

Additionally, the HPLC data can and should also be used for a discussion of the phytoplankton community structure to complement the satellite-derived products.

R: The assessment of experiment PHAEO is also backed up with the statistics of goodness of model-to-data fit presented in tables 3 – 5 (main manuscript) and tables S9-S11 (Supplementary Material).

We did not include the evaluation of the "old version" (Dutkiewicz et al. 2015) against in situ HPLC-based data because: 1) these results were obtained as climatological mean values only, which makes it difficult to get properly (without large representation error) match-ups between model and in situ data; 2) these simulations did not fulfill one of the evaluation criteria which is the agreement with observational PFT dominance. The reference run (REF) agrees sufficiently well with HPLC based haptophytes and diatoms (statistics can be provided), however it does not distinguish between coccolithophores and *Phaeocystis*, so is not adding more information to it, is why we do not show the statistics in the manuscript.

However, we do like this excellent idea to emphasize more on the usefulness of the HPLC data. Thus, in the supplementary material, we provide additional figures depicting seasonal composites of the PFT-Chla derived from the HPLC measurements from August 2002 to April 2004 (Figure R2.1, Figure S12 in the Supplementary Material).

[Figure]

[Figure]

**Figure R3.1. Distribution of seasonal composites of HLPC-Chla (Soppa et al.2017) for diatoms, hyptophytes and prokaryotes. Black counters represent Southern Ocean fronts (as white contours in Figure 1 of the manuscript.)**

R: HPLC-based PFT-Chla date were used for a quantitative assessment of the PHAEO model. To make the original discussion (lines 329-340) more visible we have edited the text (L467-495):

"We have obtained matchup statistics for the comparison of our PHAEO model results against the in situ HPLC-based PFT
Chla observations by Soppa et al. (2017). The mean absolute deviation (mean absolute error, MAE) of collocated model and in situ PFT-Chla over the considered time frame (August 2002 − April 2012) and the entire Southern Ocean is 0.74 mg m$^{-3}$ and 0.22 mg m$^{-3}$ for diatoms and haptophytes, respectively. Tables 4 and 5 present the statistics of model and in situ PFT-Chla comparison at several Longhurst's biogeochemical provinces (Longhurst 1998, see Figure 1). The highest disagreement was obtained for diatoms in the Atlantic Sector of the ANTA province, where the simulated diatom Chla is systematically
overestimated by ~0.5 mg m$^{-3}$. The best agreement with the HPLC based diatom Chla (excluding small provinces, see Figure 1) was obtained at the SSTC and SANT. For the haptophytes, the highest systematic error towards overestimation has been found at two small provinces east of Africa and Australia (EAFR and AUSE) with the bias = 0.57, 0.48 (mg m−3), respectively. The highest random error is (RMSE = 0.62, 0.44 mg m−3) at EAFR and APLR. The lowest differences between predicted and observed haptophytes was at the FKLD, SSTC provinces where haptophytes are mostly presented by coccolithophores, and at the SANT biogeochemical province, where they both co-exist. As additional information on the agreement between model and observations, Figures S9 and S10 in the Supplementary Material present frequency distributions of diatoms and haptophytes Chla for the simulations and measurements as well as the frequency distribution of the model and data differences. The latter shows that statistical criteria, such as MAE and root mean squared error (RMSE) give statistical meaningful metrics with respect to "model minus in situ Chla data" and the evaluation does not necessarily require a logarithmic transformation, as it is often done in ocean colour product validation (Brewin et al., 2010; Losa et al., 2017).

With respect to the agreement between model and observed in situ HPLC-based Chla for prokaryotic pico-phytoplankton depicted in Figure S11 (in Supplementary Material) one can conclude that the frequency distributions of the simulated and observed pico-phytoplankton are different, and the frequency distribution of the differences confirms that MAE and RMSE

given absolute (Table 5) or logarithmically transformed values can hardly provide satisfactory estimates. Nevertheless, it is worth emphasizing that the largest differences between model and observed in situ pico-phytoplankton are located along the Antarctic Peninsula.

However, it is worth noting that these statistical estimates were obtained based on the model and observation match-ups within

$\pm$ 1 week. Moreover, the model does not explicitly represent sea-ice algae and, therefore, might work less well in the region around the sea-ice. In this respect, we have to point out that all the statistics are presented for a qualitative assessment of the model rather than for a quantitative estimates of model uncertainties, since the representation error (Janjic et al., 2018) related to the differences in spatial and temporal scales considered and sampled by the model *vs.* observations as well as to the mismatch in grouping phytoplankton (Bracher et al., 2017) are large."

We now also introduce a discussion on model evaluation with MAREDAT PFT biomass dataset.

Even though SO data coverage within the MAREDAT data base is limited, the authors should consider evaluating their model output using these phytoplankton carbon biomass data set to complement the currently included HPLC data.

R: We show the comparison with the MAREDAT data on diatoms, coccolithophores, *Phaeocystis*, and zooplankton biomass. However, the coverage of the Southern Ocean PFT biomass is, indeed, very limited. Figures R2.1 - R2.2 show distribution of MAREDAT seasonal (summer and spring) composites of diatom, coccolithophores and Phaeocystis biomass, and data vs. model matchups based on monthly MAREDAT and PHAEO climatology. Because of the poor data coverage and large discrepancies in representation temporal scales, differences between the model and data (due to the representation error) are expected to be large. As a result, correlation between model and data PFT biomass is weak but significant (0.23, 0.19 and 0.54 for diatoms, coccolithophores and Phaeocystis, respectively). In general, the model overestimates PFT-carbon biomass in comparison with the data. At the end, showing the quantitative estimates of the data and model agreements, we still give a qualitative assessment. Moreover, MAREDAT measurements are not always collocated for different PFTs, thus, it is not always possible to draw any conclusions on the phytoplankton compositions. However, one, can notice, that diatoms,
coccolithophores and *Phaeocysts* do co-exist in the areas along the subantarctic and polar fronts.

[Figure]

(Figure R2.1 is now in the Supplementary Material, Figure S13)

**Figure R2.1: Climatological sesonal composites of the MAREDAT surface phytoplankton biomass for diatoms (a: for
January – March, b: for October - December), coccolithophores (d: for January – March, e: for October - December)
and Phaeocystis (g: for January – March, h: for October - December); scaterplot of the model vs. MAREDAT**

**matchups based on all surface climatological monthly means: c) for diatoms; f) for coccolithophores; i) for Phaeocystis. (PHAEO model climatology is based on the years 2006 – 2012). Statistics are presented for logtransformed concentrations.**

[Figure]

(Figure R2.2 is now in the Supplementary Material, Figure S14)

**Figure R2.2: Climatological sesonal composites of the MAREDAT surface mesozooplankton (a: for January – March, b: for October - December) and microzooplankton (d: for January – March, e: for October - December); the model**
**total zooplankton vs. MAREDAT zooplankton matchups based on all climatological monthly means: c) for meso-; f) for micro (PHAEO model climatology is based on the years 2006 – 2012). Statistics are presented for logtransformed concentrations.**

Furthermore, in the presentation of the evaluation, the authors often use subjective statements in their description (e.g.
"plausible distributions", "skillful enough") which should be avoided as much as possible throughout the manuscript as it is e.g. not clear to me at all when a biomass distribution is "plausible".

R: We removed subjective statements in the revised version of the manuscript.

6) Overall, I think the introduction in its current form misses a clear focus on the focus area, i.e. the SO. From the title of the paper, I would expect a description of the observed SO phytoplankton biogeography somewhere based on available in situ data and satellite algorithms to set up the reader for the assessment of the simulated community structure. Additionally, I would expect a summary on what has been done in terms of PFT modeling in the SO specifically, highlighting what gap is filled with the model used here (for this, see e.g. Lancelot et al. (2009), Wang et al. (2011), Le Quéré et al. (2016), Nissen et al. (2018); Note that the list of available studies is much longer than the examples given here!). The introduction in its current form largely focusses on global modeling approaches without an assessment of how they perform in the SO and is thereby of limited use for the goal of the paper.

R: We modified the introduction accordantly. In the revised manuscript we focused more on the Southern Ocean, added the information of the expected occurrence of the investigated PFTs (in addition to their importance), wrote about current challenges in modeling phytoplankton groups in the Southern Ocean and added a paragraph that explicitly presents the hypotheses tested in our study.

"The Southern Ocean is one of the most important regions in regulating climate via the uptake of about 40% of the global oceanic anthropogenic $CO_2$ (DeVries, 2014) and at the same time, is a region with the dynamics evidently altered by past and present climate change (Stocker et al., 2013). The climatic changes in the Southern Ocean environmental conditions affect the spatial distribution of phytoplankton (Deppeler and Davidson, 2017). The phenology and dominance of different phytoplankton functional types (PFTs) sustaining the marine food web affect the diversity of higher trophic levels (Edwards and Richardson, 2004). Playing distinct roles in biogeochemical cycling, PFTs may determine how and on which spatial and temporal scales the ocean mediates climate (Wilson et al., 2018).

Major bloom-forming PFTs in the Southern Ocean include the silicifying diatoms, calcifying coccolithophores, and colony-forming *Phaeocystis*. Diatoms, the major phytoplankton silicifiers and primary producers in the Southern Ocean (Rousseaux and Gregg, 2014), have high efficiency of carbon export through grazing, direct sinking of single cells, and through mass sedimentation events (Le Quéré et al., 2005; Kemp et al., 2006). They form large spring blooms in the open nutrient-rich waters in the proximity of the Antarctic Circumpolar Current and Polar Front (Smetacek et al., 2002; Kemp et al., 2006). Coccolithophores, the main phytoplanktonic calcifiers in the world ocean, make a major contribution to the total content of particulate inorganic carbon in the oceans (Ackleson et al., 1988; Milliman, 1993; Rost and Riebesell, 2004; Monteiro et al., 2016) through production and release of calcium carbonate plates (coccoliths), and, therefore, also impact the alkalinity of the ocean. This PFT is abundant along the Great Calcite Belt (Balch et al., 2016) and forms massive blooms along the Patagonian shelf break (Signorini et al., 2006). *Phaeocystis* as a dimethyl sulfide producer alters the atmospheric sulfur cycle and can form dense spring blooms in the seasonal ice zone and Antarctic coastal waters as the Ross Sea and Weddell Sea (El-Sayed et al., 1983; Arrigo et al., 1999; DiTullio et al., 2000; Smith et al., 2012), likely supporting export production (Arrigo et al., 2000;

DiTullio et al., 2000; Wang and Moore, 2011). Modeling studies reported the contribution of diatoms to the total primary production in the Southern Ocean of ~89% (Rousseaux and Gregg, 2014), coccolithophores of ~7-16.5% (Rousseaux and Gregg, 2014; Nissen et al., 2018) and *Phaeocystis* of ~13% (P. antarctica) (Wang and Moore, 2011).

Despite the recognized importance of the PFTs, global biogeochemical models struggle to represent the Southern Ocean phytoplankton community accurately. The difficulties primarily originate from uncertain parameters employed in the parametrizations of, e.g., phytoplankton growth and grazing (Anderson, 2005), that define the differences in the phytoplankton traits. On the other hand, the available observational information is still limited in the Southern Ocean to allow to properly constrain the models.

One of the most investigated regions in the Southern Ocean is the Ross Sea, where many in situ observations on diatoms and *Phaeocystis* have been collected and inspired regional coupled ocean-sea ice-ecosystem modeling activities (Arrigo et al., 2003; Worthen and Arrigo, 2003; Kaufman et al., 2017). Several studies that include *Phaeocystis* in the list of simulated PFTs in the frame of global coupled ocean-biogeochemical models have focused on the Southern Ocean (Lancelot et al., 2009; Wang and Moore, 2011; Le Quéré et al., 2016). These studies specified differences in (photo-)physiological parameters between diatoms and *Phaeocystis*, considering *Phaeocystis* in colony form. In a regional study (Popova et al. 2007, Crozet Islands) within the Southern Ocean, *Phaeocystis* was represented by two different life-stages: colonies and solitary cells. This approach was also successfully used by Kaufman et al. (2017) to examine the influence of climatic changes on the Ross Sea phytoplankton.

Nevertheless, an in-depth evaluation of the model simulations of diatoms and *Phaeocystis* with PFT observations either has not been done (e.g. Lancelot et al. 2009) or has been only performed based on a sparse in situ dataset (Wang and Moore, 2011). A more complete evaluation of these PFTs was presented by Le Quéré et al. (2016) by comparing the dominance of the PFTs to satellite-based dominance retrievals, and to a global dataset of in situ-based integrated PFT biomass within upper 200 m of Alvain et al. (2008) and (Buitenhuis et al., 2013), respectively. In general, as compared to the satellite retrievals, the dominance of diatoms and *Phaeocystis* has been overestimated by Le Quéré et al. (2016), while dominance of coccolithophores was underestimated.

Coccolithophore biogeography has recently been investigated globally by Monteiro et al. (2016), Krumhardt et al. (2017) and Krumhardt et al. (2019), and particularly for the Southern Ocean by Nissen et al. (2018). With respect to specific coccolithophore traits, the study by Krumhardt et al. (2017), Monteiro et al. (2016), as well as earlier studies by Paasche (2001) and Iglesias-Rodríguez et al. (2002), emphasized the high nutrient affinity of the coccolithophores and high grazing protection of this PFT (Monteiro et al., 2016). Nissen et al. (2018) reported on higher grazing pressure on coccolithophores than on diatoms. Krumhardt et al. (2019) used lower grazing pressure on coccolithophores than on diatoms and related the distribution of coccolithophores to a specific temperature function in dependence to its growth rate. However, none of these studies included *Phaeocystis* in their model simulations.

In our study, we improved the representation of key Southern Ocean PFTs, namely diatoms, coccolithophores and *Phaeocystis*,
using the Darwin biogeochemical model coupled to the Massachusetts Institute of Technology (MIT) general circulation model (Darwin-MITgcm). In a first step, we modified the Darwin model to account for two distinct size classes of diatoms and for a high affinity for nutrients and an ability to escape grazing control for coccolithophores. Next, the model was extended to include both solitary and colonial forms of *Phaeocystis*. Observational information from in situ and satellite measurements was used to help to define differences in the PFT traits, to constrain the model, as well as to quantitatively evaluate the model
performance to overall find a representation of the phytoplankton community in the Southern Ocean that is close to observations. We used the optimized Darwin model to test three hypotheses on the factors controlling the biogeography of Southern Ocean phytoplankton groups:

– Size diversity of the diatoms (Queguiner, 2013; Tréguer et al., 2018) leads to the distribution of small diatoms ("slightly silicified and fast growing") at the lower latitudes and large diatoms ("strongly silicified and slowly growing") at higher
latitudes in the Southern Ocean.

– Distribution of coccolithophores in the Great Calcite Belt is not necessarily controlled by temperature (Smith et al., 2017) but determined by the ability of this PFT to escape grazing because of their exoskeleton (Nejstgaard et al., 1997; Huskin et al., 2000; Monteiro et al., 2016), and to grow under nutrient depleted conditions (especially phosphate and iron) (Paasche, 2001; Iglesias-Rodríguez et al., 2002). These characteristics of coccolithophores would make them more competitive among other
phytoplankton of larger or similar size, small diatoms and *Phaeocystis*.

– *Phaeocystis* sp. exists in two life stages, solitary cells and colonies, depending on iron availability (Bender et al., 2018). This additional difference in the traits of distinct haptophytes, coccolithophores and *Phaeocystis*, allows them to co-exist.

The paper is organized as follows. Section 2 describes the numerical model set up, experimental design and observations (in
situ and satellite retrievals) used for model evaluation, Section 3 presents the results and discussion. Section 4 concludes with summary and outlook."

7) Currently, there is no consistency in the study in what month or even what year is assessed in the different parts of the manuscript (compare e.g. Fig. 3, 4, and 6). In the method section, the authors should clearly state which year(s) and which
month(s) of the model output is used in the analysis and why.

R: We provide below (and in the revised version) more details on model evaluation explaining the validation datasets and temporal and spatial representation of the results. A new table (Table 2) is introduced.

"To assess our model results, we compare the simulations to several large in situ and satellite datasets, as detailed below and summarized in Table 2. Where the coverage of the observations is similar in respect to time we use our two-weekly model outputs. Where only monthly climatological or composite data (often from different time periods) are available we use monthly climatological model results for the period of 2006-2012. Where only results for specific months are available from observations we compare our output to these specific months. Table 3 contains the information about the evaluated phytoplankton groups as classified in the model and observations."

In the subsection(s) describing the observational data used to constrain and evaluate the model, we also clarify how we show corresponding model solution.

**Table 2.** Datasets used for model evaluation

| Dataset | reference | PFT product | units | spatial repr. | time repr. | model output | time repr. |
|---|---|---|---|---|---|---|---|
| PHYSAT | Alvain et al. (2008) | dominance | unitless | 1°x1° | monthly climat. (1998-2006) | dominance | 2006–2012** |
| Darwin-15 | Dutkiewicz et al. (2015) | dominance | unitless | 1°x1° | monthly climatology | dominance | 2006–2012** |
| SEM | Smith et al. (2017) | dia *vs.* cocco dominance | % cell counts | in situ | Jan–Feb 2011 Feb-March 2012 | dia *vs.* cocco % C-biomass | Jan–Feb 2011 Feb–Mar 2012 |
| SynSenPFT | Losa et al. (2017) | diatom-Chla | mgChla m$^{-3}$ | 4x4 km* | March 2012 | diatom-Chla | March 2012 |
| | | cocco-Chla | mgChla m$^{-3}$ | 4x4 km* | March 2012 | cocco-Chla | March 2012 |
| PhytoDOAS | Bracher et al. (2017) | diatom-Chla | mgChla m$^{-3}$ | 0.5°x0.5°* | March 2012 | diatom-Chla | March 2012 |
| HPLC | Soppa et al. (2017) | diatom-Chla | mgChla m$^{-3}$ | in situ | Aug2002 – Apr2012 | diatom-Chla | collocated |
| | | hapto-Chla | mgChla m$^{-3}$ | in situ | Aug2002 – Apr2012 | hapto-Chla | collocated |
| | | proka-Chla | mgChla m$^{-3}$ | in situ | Aug2002 – Apr2012 | *Proch*-Chla | collocated |
| MAREDAT | Leblanc et al. (2012) | diatom-C | mgC m$^{-3}$ | in situ | 1933–2009 climat. | diatom-C | 2006–2012** |
| | O'Brien et al. (2013) | cocco-C | mgC m$^{-3}$ | in situ | 1929–2008 climat. | cocco-C | 2006–2012** |
| | Vogt et al. (2013) | *Phaeo*-C | mgC m$^{-3}$ | in situ | 1955–2009 climat. | *Phaeo*-C | 2006–2012** |
| | Buitenhuis et al. (2012) | micro-zoo-C | mgC m$^{-3}$ | in situ | climatology | zoo-C | 2006–2012** |
| | Moriarty et al. (2013) | mezo-zoo-C | mgC m$^{-3}$ | in situ | climatology | zoo-C | 2006–2012** |

$diatom - Chla$ denotes diatom Chla; $cocco - Chla$ is coccolithophore Chla; $hapto - Chla$ is haptophytes Chla; $proka - Chla$ is prokaryotes Chla, $Phaeo - Chla$ is Phaeocystis Chla; $Proch - Chla$ is Prochlorococcus Chla, extension $-C$ denotes carbon biomass; dia vs. cocco is diatom vs. coccolithophores; $zoo$ stands for zooplankton; $repr.$ is representation; climat. is climatology.

\* the data are presented for a reduced Southern Ocean area as in Smith et al. (2017) and Losa et al. (2018).

\*\* model monthly mean climatology over the years 2006 – 2012.

HPLC: "As we can see there and in Table 2, this large dataset gives us the possibility for a quantitative validation of our model results. Two weekly PHAEO model snapshots from August 2002 to April 2012 have been collocated against in situ HPLC-based Chla observations, if available, within a time window ±1 week. We compare the simulated Chla of diatoms (large + small), haptophytes (coccolithophores + *Phaeocystis*) and prokaryotic pico-phytoplankton against HPLC-derived Chla for diatoms, haptophytes and prokaryotes."

MAREDAT: "These datasets are based on a data collection spanning between 55 to 75 years and are provided as climatological monthly composites. Because of the very sparse distribution of these datasets in the Southern Ocean (except for zooplankton), which leads to a large representation error when comparing to the model monthly mean climatology (2006 – 2012), only a qualitative assessment was possible." (While quantitative assessment is also shown)

SEM: "Predicted biomass of diatoms and coccolithophores are additionally compared to diatom and coccolithophore measurements (as cell counts) obtained by scanning electron microscopy in the North Atlantic and Indian Ocean sections of the Southern Ocean (the Great Calcite Belt area) during January – February 2011 and February – March 2012 by Smith et al.

(2017). For qualitative assessment of the simulated diatom and coccolithophore distributions we compare diatom *vs.* coccolithophore dominance to similar estimates by Smith et al. (2017) collocated in space and time."

SynSenPFT: "We chose only the two groups for comparisons because we are using the SynSenPFT results in addition to the in situ SEM based diatom vs. coccolithophores dominance by Smith et al. (2017). Hence, we only use the same areas and time period as in their study for comparisons to the SynSenPFT results."

PHYSAT: "We compare model climatology of Southern Ocean PFT dominance (averaged over the years 2006 – 2012) to the PHYSAT PFT dominance."

Phenological indices: "These indices are calculated based on the REF Chl simulations for diatoms (including small and large) over the year 2007/2008. We chose this particular year because: 1) with the two-weekly model output the phenological indices can be more precisely calculated than based on the two-weekly or monthly mean climatology; 2) it is a typical year over the period 2006 – 2012 with respect to the simulated PFT distribution (after model reached the quasi-steady state) and climate oscillations (Soppa et al., 2016)."

In this regard, it is e.g. not clear to me why the authors chose to present the ability of the model to represent dominant phytoplankton types in winter, when biomass levels are low.

R: We show the simulations also for the winter because even during this period North of the subtropical front the model simulations show not a negligible biomass (please see supplemented video materials). In addition, we wanted to be consistent with what was shown in the study by Dutkiewics et al. (2015), as well as to explicitly illustrate the disagreement between PHYSAT and "old version" (as well as other tested configurations reviewed in the Supplementary Material) and because this disagreement in winter PFT dominance resulted from incorrectly simulated PFT phenology (e.g. very early bloom of diatoms). This was initially a motivation to consider to size classes of diatoms.

Overall, the figure captions are often incomplete and panel labels are missing entirely. These should be added and referred to in the text to better guide the reader.

R: This has been revised for all figures.

**Detailed comments**

**Abstract:**

L. 1: I suggest to make clear in the very first sentence that you're focusing on a single model –otherwise the first sentence sounds like the reader is about to read a review paper on SO PFT modeling. Additionally, I suggest to rephrase to "under past and present climate change".

R: The abstract has been revised accordingly (see comment L. 13).

L. 3: By stating "phenology" so prominently in the abstract, you set up the reader for an assessment of the PFT phenology in your model –which you actually never really do (see comments below). Please rephrase here to have a better representation of the content of the paper and/or adapt the content of the result section (see general comments).

R: The abstract has been revised accordingly (see comment L. 13).

L. 8-9: The new model configuration describes the competition and co-occurrence "best" in what regard and compared to what? Please be precise.

R: Best with respect to considering several "dimensions of phytoplankton diversity" (Dutkiewicz et al., BSD, https://doi.org/10.5194/bg-2019-311)

However, the abstract has been revised accordingly (see comment L. 13).

L. 9-13: Please specify what "older version" you're referring to here, e.g. by explicitly stating "without the above-mentioned changes, but otherwise identical" (if that is the case).

R: The abstract has been revised accordingly (see comment L. 13).

L. 11-13: In the manuscript, you never actually show a quantitative validation of the model output with the SEM data (no plot at all) or the HPLC data (only in maps for the PHAEO simulation, not for the REF simulation), so that it is hard for the reader to evaluate how the model performance improves with your changes (see comments below). Furthermore, I suggest to not overemphasize the SEM data here in the abstract as this comparison is not a major part of your study.

R: A quantitative assessment of the model against HPLC data was provided in three tables (Table 3 − 5) in the main manuscript and 3 Tables in the supplementary material. In Table 2 − 5 we show statistical analysis of the PFT-Chla model-data matchups (RMS, MAE, bias) at several Longhurst's biogeochemical provinces, in table S7-S9 the same but for log-transformed values and table S11 presents more detailed analysis for different sections of the biogeochemical provinces. For qualitative visual evaluation the reader was referred to three supplementary videos. The discussion of the goodness of model to data fit is further extended in the revised version of the manuscript.

In the original version of the manuscript, we compared phytoplankton composition (as meridional distribution of zonally averaged) with respect to co-existing diatoms and coccolithophores with estimates of Smith et al. (2017). Now we show diatom *vs.* coccolithophores dominance collocated in space and time with similar estimates from Smith et al. (2017). Please see figure R2.11 (Figure 5 in the revised version of the manuscript). Nevertheless, the abstract has been revised accordantly, please see below.

L. 13: Please rephrase to "SO PFT dominance patterns". "agrees well" in what regard? Space? Time? Additionally, the abstract in its current form does not represent how much time you spend in the manuscript on the discussion of dominance patterns as opposed to the validation of chlorophyll-a concentrations of the individual PFTs. I suggest to rewrite the abstract to more adequately represent the content of the result section.

R: Thank you for all the comments on the abstract. We have rewritten it based on your comments. We changed the first sentence of the abstract, removed the term phenology and emphasized that the modeled Southern Ocean PFT dominance also agrees well with satellite-based PFT information in terms of spatial and temporal distribution.

"Phytoplankton in the Southern Ocean support important ecosystems and play a key role in the earth's carbon cycle, hence affecting climate. However, current global biogeochemical models struggle to reproduce the dynamics and co-existence of key phytoplankton functional types (PFTs) in this Ocean. Here we explore the traits important to allow three key PFTs (diatoms, coccolithophores and *Phaeocystis*) to have distributions, dominance and composition consistent with observations. In this study we use the Darwin biogeochemical/ecosystem model coupled to the Massachusetts Institute of Technology (MIT) general circulation model (Darwin-MITgcm). We evaluated our model against an extensive synthesis of observations, including in situ microscopy and high-performance liquid chromatography (HPLC), and satellite derived phytoplankton dominance, PFT chlorophyll-a (Chla), and phenology metrics. To capture the regional timing of diatom blooms obtained from satellite required including both a lightly silicified diatom type and a larger and heavy silicified type in the model. To obtain the anticipated distribution of coccolithophores, including the Great Calcite Belt, required accounting for a high affinity for nutrients and an ability to escape grazing control of this PFT. The implementation of two life stages of *Phaeocystis* to simulate both solitary and colonial forms of this PFT (with switching between forms being driven by iron availability) improved the co-existence of coccolithophores and *Phaeocystis* north of the Polar Front. The dual life-stages of *Phaeocystis* allowed it to compete both with other phytoplankton of larger size and/or similar sizes. The evaluation of simulated PFTs showed significant agreement to a large set of matchups with in situ PFT Chl-a data derived from pigment concentrations. Satellite data provided important qualitative comparisons of PFT phenology and PFT dominance. With these newly added traits the model produced the observed >50% coccolithophore contribution to the biomass of biomineralizing PFTs in the Great Calcite Belt. The model together with the large synthesis of observations provides a clearer picture of the Southern Ocean phytoplankton community structure, and new appreciation of the traits that are likely important in setting this structure."

**Introduction:**

L. 16: Please rephrase "via the sinking of CO2".

R: The corresponding sentence was modified.

L. 17: Please add a reference for the evidence of changes due to on-going climate change.

R: We modified the sentence to:

"The Southern Ocean is one of the most important regions in regulating climate via the uptake of about 40% of the global oceanic anthropogenic $CO_2$ (DeVries, 2014) and at the same time, is a region with the dynamics evidently altered by past and present climate change (Stocker et al., 2013)."

L. 20: Please add a reference for the impact of phytoplankton community structure on the diversity of higher trophic levels.

R: We added Edwards and Richardson (2004).

Edwards, M. and Richardson, A. J.: Impact of climate change on marine pelagic phenology and trophic mismatch, Nature, 430, 881, 2004.

L. 21: Please add a reference for the impact of phytoplankton community structure on climate on different temporal and spatial scales.

R: We added Wilson et al. 2014.

Wilson, J. D., Monteiro, F. M., Schmidt, D. N., Ward, B. A., and Ridgwell, A.: Linking Marine Plankton Ecosystems and Climate: A New Modeling Approach to the Warm Early Eocene Climate, Paleoceanography and Paleoclimatology, 33, 1439–1452,635, https://doi.org/10.1029/2018PA003374, 2018.

L. 32: Please add a reference for the impact of Phaeocystis on SO export production.

R: We added Arrigo et al. (2000), DiTullio et al. (2000) and Wang and Morre (2011).

Arrigo, K. R., DiTullio, G. R., Dunbar, R. B., Robinson, D. H., VanWoert, M., Worthen, D. L., and Lizotte, M. P.: Phytoplankton taxonomic variability in nutrient utilization and primary production in the Ross Sea, Journal of Geophysical Research: Oceans, 105, 8827–8846, https://doi.org/10.1029/1998JC000289, 2000.

DiTullio, G., Grebmeier, J., Arrigo, K., Lizotte, M., Robinson, D., Leventer, A., Barry, J., VanWoert, M., and Dunbar, R.: Rapid and early export of Phaeocystis antarctica blooms in the Ross Sea, Antarctica, Nature, 404, 595, 2000.

Wang, S. and Moore, J. K.: Incorporating Phaeocystis into a Southern Ocean ecosystem model, Journal of Geophysical Research: Oceans, 116, https://doi.org/10.1029/2009JC005817, 2011.

L. 32-35: Why is the description of these types (N2fixers and pico autotrophs) relevant for a modeling study of the SO? I think you can delete this part to have more room to focus on an introduction of the actual topics, such as what is known on the biogeography (from observations and modeling studies) of the most important types in the SO, namely diatoms, Phaeocystis, and coccolithophores.

R: We deleted the description of these types (N2fixers and pico autotrophs). However, distribution of these PFTs impacts the abundance of other PFTs. The more accurate they are modeled (accounted for) the better the distribution of other PFTs is simulated (for instance the north edge of the Great Calcite Belt).

L. 36-39: I suggest to list the three criteria when first mentioning the division by Le Quéré et al. (2005) in e.g. L. 22. The way
it is done currently, the 2nd and 3rd criteria come a bit out of the blue for the reader.
R: This part was removed in the revised version of the manuscript (please, check general comments 6).

L. 39: Please give an example that is relevant to the SO application in this study.
R: This part was removed in the revised version of the manuscript (please, check general comments 6).

L. 44-45: I suggest to rephrase to something like "[...] includes also bacteria and zooplankton, but for this study, we use "PFT" to refer to phytoplankton only, in accordance with the definition by the ocean color community".
R: This part was removed in the revised version of the manuscript (please, check general comments 6).

L. 52-55: The relevance of this statement to the study at hand is not clear to me. Please explain.
R: This part was removed in the revised version of the manuscript (please, check general comments 6).

Additionally, you never really use "PG" throughout the text, it is not clear to me why you introduce it here. I suggest to move the information given here to the only place where you actually use it (section 2.2.2).
R: This part was removed in the revised version of the manuscript (please, check general comments 6).

L. 56: It is not clear here why you cite Follows et al. (2007) alongside Le Quéré et al. (2005) after spending almost a page on discussing the latter while not introducing the former. Please make clearer.

R: This part was removed in the revised version of the manuscript (please, check general comments 6).

L. 57: "thee" should be "three"

R: Corrected, but this part was removed in the revised version of the manuscript (please, check general comments 6).

L. 60: Please see also Krumhardt et al. (2019) for a global model with an explicit representation of coccolithophores and
consider adding Nissen et al. (2018) here as well as an example of a regional model with explicit coccolithophores to give a
more complete overview on what has been done.

R: As suggested, we added these references to the introduction.

L. 66: Please explain more clearly in the text how the Darwin model offers "the highest potential". For example, does this
model generally offer "higher potential" than regional modeling approaches? As I am personally not convinced by this (as it
will depend on the question you're trying to answer), I suggest to rephrase this statement to explain more clearly.

R: We meant the ability to consider several dimensions of PFT diversity (Dutkiewicz et al. 2019, BSD), but this part was
removed in the revised version of the manuscript (please, check general comments 6).

Dutkiewicz, S., Cermeno, P., Jahn, O., Follows, M. J., Hickman, A. E., Taniguchi, D. A. A., and Ward, B. A.: Dimensions of
Marine Phytoplankton Diversity, Biogeosciences Discuss., https://doi.org/10.5194/bg-2019-311, in review, 2019.

L. 70-74: In my view, the goals you list here for the study at hand do not match the content of the result section. For example,
the manuscript currently lacks a thorough quantitative (!) assessment of the phytoplankton phenology.

R: As mentioned earlier, the initial idea of the manuscript was also to include information on the timing of the phytoplankton
blooms but we realized that the study would be too complex and diverse on topic to be summarized in one manuscript. We
planed a dedicated study on the phenology of the PFTs blooms in the Southern Ocean soon.
Now, the Chla phenological indices for diatoms are presented in Figure R2.8 (Figure 3a-c in the revised version of the
manuscript), Figure R2.9.

What is your conclusion on point 3) here? How can the model complement available in situ observations?

R: The statement was about the model ability to consider different aspects of differentiation among phytoplankton groups –
biogeochemical role; allometric, photophysiological and optical parameters; accounting for carbon and Chla decoupling. This
ability makes coupled ocean/biogeochemical models a very valuable and skillful instrument that combines the knowledge from
in situ measurements and remote sensing by exploiting various PFT retrievals principles used (separately) in these observations
and relates it to the environmental conditions. However, the introduction has been rewritten.

L. 71-72: When is a model "skillful enough" in your opinion? When is a simulated distribution "plausible"? Please specify exactly what you mean by this and avoid subjective judgement whenever possible.

R: We removed subjective statements in the revised version of the manuscript.

Please replace "predict" by "simulate" or similar.

R: We replaced the term.

L. 74-75: The statement "When determining [...]" is not clear to me. Please be more precise. What do you mean exactly?

R: Within our study, with the available observational satellite and in situ information we constrain the model with respect to PFT traits specified in the model. Nevertheless, this sentence was removed in the revised version of the manuscript (please, check general comments 6).

**Methods**

L. 80: I suggest to change the title to include the name of the model used in this study.

R: Changed to "Darwin-MITgcm numerical models"

L. 90: Do you mean lightly silicified?

R: We meant "slightly silicified" (as in Queguiner, 2013).

How was the silicification different between these two classes different in the model? How is silicification parametrized? If you introduce a completely new PFT, you need to give more detail on its characteristics.

R: We did not introduce a complete new PFT but considered small eukaryotes (as in the original Dutkiewicz et al. 2015 paper) being silicified as specified for large diatoms with the parameters listed in Table 1. Thank you, we have now added $k_{si}$ parameter values for large and small diatoms. The level of silification is parameterized by the cellular Si:C ratio.

L. 90-99: Why are these three changes justified for the SO? I suggest to include statements on the reasoning behind e.g.

changing the nutrient affinity and grazing parameters for coccolithophores –

R: It was motivated by the following studies Paasche, 2001; Iglesias-Rodríguez et al., 2002; Nejstgaard et al., 1997; Huskin et al., 2000, Losa et. 2006, Krumhardt et al. 2017. Now additionally backed up by Monteiro et al. 2016. We state it now in the introduction as one of the hypotheses tested.

Why does this apply for this SO-focused study and not for global applications of Darwin?

R: Indeed, we think this applies also for global applications of Darwin (this was also shown in Monteiro et al. 2016).

Please add a reference regarding the occurrence of lighter silicified diatoms at lower latitudes.

R: The reference to Queguiner (2013) (w.r.t. "slightly silicified diatoms") were provided (L. 93). We also state it now as a hypothesis we test (with additional reference to Tréguer et al. 2018).

L. 95: Please replace "was presented" by "is represented".

R: We replaced as "has been presented".

L. 95-99: What sensitivity experiments did you perform here? How did you evaluate what a "realistic co-occurrence of coccolithophores and Phaeocystis" is?

R: This sentence has been removed/ was rephrased to: "Other nano-phytoplankton (referred to as "other large" in the original Dutkiewicz et al. 2015) has been presented by *Phaeocystis*."

I think it is important here to briefly sketch the main characteristics of the parametrizations used for Phaeocystis if you've actually used those from Popova et al. (2007) and Kaufmann et al. (2017), but see also comment further down (on L. 138 in your manuscript).

R: We indeed introduce two phases of the Phaeocystis life stages (colonies and solitary cells) following Popova et al. (2007) and Kaufman et al. (2017). However, these "two *Phaeocystis* life stages were considered as a function of iron availability (Bender et al. 2018)."

L. 101-112: The description of the treatment of light is out of place here as you go back to a description of the PFTs afterwards. Please reorganize the section to make it easier for the reader to follow.

R: Though it was one of the changes/differences with respect to original Dutkiewicz et al. 2015, we have now moved this description to the Supplementary Material.

Additionally, I am not sure this much detail on the parametrizations surrounding light absorption are needed in the main text. Please consider moving this part to the supplement.

R: We have now moved it to the Supplementary Material.

L. 100-117: Here and throughout the text (including e.g. especially Table 1), please make sure you state the units of all variables introduced.

R: Units were provided in the text introducing model parameters in the parametrizations (pages 4 and 5, Section 2.1.1.) To make it clearer, we now include the parameter units in Table 1. We did not provide unites for the function $\mu_j$ and alpha (also formulated as a function). As suggested, we have added them now.

L. 113: Please replace "which is presented" by "which are described by" or similar.

R: Replaced.

L. 114: According to Table 1, this parameter only applies to Prochlorococcus. I suggest to state that here.

R: We added "applied to Prochlorococcus".

L. 115: I find "biomineralizing function" misleading and would rather say "whether or not they form biominerals such as opal or calcite" (or something along these lines).

R: We were following the expression used as in Smith et al. (2017) but we clarify it accordingly as suggested (L126).

L. 115-117: Please rephrase this sentence, it sounds a bit weird to me in its current form.

R: We rephrased to: "These main differences between specified traits alter the growth rate of particular phytoplankton ($\mu_j$, day$^{-1}$, j = 1, 2,..., 6) and the grazing of phytoplankton by small or micro-zooplanktons ($Gr_{jk}$, k = 1, 2) given the palatability factor ($r_{j,k}$) and sinking rate ($w_{sink}$, m day$^{-1}$)."

L. 118: Please rephrase to "The growth of phytoplankton μj(day-1)[...]"

R: We rephrased as suggested but without the abbreviation and unit since it was introduced earlier in the text.

L. 123: How are the temperature and nutrient limitation terms calculated? Please add the equations.

R: As in the original study by Dutkiewicz et al. (2015), the nutrient limitation is calculated as:

$$\gamma_{\eta_{ij}} = \min\left(\eta_{ij}\right), \eta_{ij} = \frac{\eta_i}{\eta_i + k_{sat_i}}.$$

The temperature limitation is calculated as:

$$\gamma_j^T = \tau_T e^{\left(A_T\left(\frac{1}{T+273.15} - \frac{1}{T_O}\right)\right)},$$

given the coefficient $\tau_T = 0.8$ normalized the maximum value (unitless), the temperature coefficient $A_T = -4000K$, and the reference temperature $T_o = 293.15K$"

We have added this and nutrient limitation functions in the revised version of the manuscript (L135).

L. 124: Alpha PI is missing in Table 1.

R: Alpha is not a parameter but it is calculated by equation (5) (eq. 3 in the revised manuscript) given the phytoplankton-specific light absorption and the maximum quantum yield of carbon fixation.

L. 125: The phytoplankton-specific light absorption and the maximum quantum yield of carbon fixation are missing in Table 1.

R: The phytoplankton-specific light absorption is not a parameter but spectra, we have now added the spectrally averaged phytoplankton-specific light absorption values of the maximum quantum yield of carbon fixation into the table.

(Initially in table 1, we opted to specify parameters, which were different from the configuration of Dutkiewicz et al., 2015).

L. 127: Please change to "as opposed to the studies by X andY". However, I don't understand why you refer to two studies here which are based on a different biogeochemical model (NOBM) than the one you're using here (DARWIN). Are you using the same function to calculate the temperature limitation as they do? If yes, state that to make your argument clearer.

R: We use different (to what is used in NOBM) the tempreture limitation function. However, as opposed to studies using distinct temperature function for different PFTs, we use the same function for all considered PFTs. We revised as following: "The $\gamma_j^T$ function was considered the same for diatom, coccolithophores, *Phaeocystis* and prokaryotes given the coefficient $\tau_T$ = 0.8 normalized the maximum value (unitless), the temperature coefficient $A_T$ = -4000 K, and the reference temperature $T_0$ = 293.15 K." (L142-144).

Furthermore, does your statement mean that the growth of N2fixers is not suppressed at low temperatures?

R: Yes, in this version we did not suppress N2-fixers at low temperatures.

This relates to a comment further down (on Fig. 4) in that I have the impression that your importance of N2fixers for the SO phytoplankton community is way too high if we take into consideration that their growth should be limited to regions of temperatures above a certain threshold (e.g. ~18°C, see e.g. Breitbarth et al. (2007) and Luo et al. (2012)) –even though nitrogen fixers have been found more recently in polar waters, I am just not convinced that they make up such a substantial part of the community in terms of biomass in these latitudes. Are you aware of evidence for this?

R: In PHAEO model simulations, Nfixer shows only north of $45°$S.

The Darwin model has typically not imposed temperature limitation on diazotrophy, as model studies suggest that the majority of diazotrophy happens in warmer waters not so much because of the temperature, but because of the nutrient supply ratios of those waters (see e.g. Monteiro et al 2011; Dutkiewicz et al 2012; Ward et al 2013). We feel this is a reasonable approach in this study especially given the discovery of diazotrophs in colder waters (Zehr, 2011, Fernández-Méndez, 2016), also in Baltic Sea.

Monteiro, F., Dutkiewicz, S., Follows, M. J.: Biogeographical controls on the marine nitrogen fixers. *Global Biogeochemical Cycles*, 25, GB2003, doi:10.1029/2010GB003902, 2011

Dutkiewicz, S., Ward, B.A., Monteiro, F., Follows, M. J.: Interconnection between nitrogen fixers and iron in the Pacfic Ocean: Theory and numerical model. *Global Biogeochemical Cycles* , 26, GB1012, doi:10.1029/2011GB004039, 2012

Ward, B.A., Dutkiewicz, S., Moore, C.M., Follows, M.J. : Iron, phosphorus and nitrogen supply ratios define the biogeography of nitrogen fixation. *Limnology and Oceanography*, 58, 2059-2075, 2013

Zehr, P.: Nitrogen fixation by marine cyanobacteria, *Trends in Microbiology,* 19 (4), 162 – 173, doi:10.1016/j.tim.2010.12.004, 2011

Fernández-Méndez, M., Turk-Kubo, K. A, Buttigieg, P. L., Rapp, J. Z., Krumpen, T, Zehr, J. P., Boetius, A.: Diazotroph Diversity in the Sea Ice, Melt Ponds, and Surface Waters of the Eurasian Basin of the Central Arctic Ocean. Front. Microbiol. 7:1884. doi: 10.3389/fmicb. 2016. 01884, 2016.

L. 131: gmax and ksat are missing in Table 1. Furthermore, the equation you give has a Holling Type III ingestion term. Are you using Holling Type II or III? Please double-check.

R: $g_{max}$ and $k_{sat}$ are used unchanged as in Dutkiewicz et al. (2015). We use Holling Type III to formulate grazing, thank you for pointing out the typing error.

L. 138-145: I have some concerns regarding the way you parametrize Phaeocystis here.

- First of all: are you following the parametrizations of Popova et al. (2007) and Kaufmann et al. (2017) or not? You state this in L. 99, but according to what you state here, I don't think you can say that you use their parametrizations. In both the cited studies, the transition of Phaeocystis from single cell to colonies (and back) is a function of a specified maximum colony formation rate, a maximum single cell liberation rate, the single cell biomass concentration(using a threshold concentration to allow for colony formation), the position in the water column (i.e. light availability, see also Peperzak (1993)), and the nutrient limitation–as opposed to just a fixed iron concentration threshold you seem to have used here (if I understood this correctly). Differences to the cited literature need to be made very clear here as your parametrization appears distinctly different.

  R: We do not use exactly the same parametrizations as in Popova et al. (2007) and Kaufmann et al. (2017). We state that we introduce two phases of the *Phaeocystis* life stages (colonies and solitary cells) following Popova et al. (2007) and Kaufman et al. (2017), but with different implementation: 1) these two *Phaeocystis* life stages were considered only as a function of iron availability as shown in the study by Bender et al. 2018; 2) just one tracer was considered (l. 144 – 145).

The effect of neglecting certain aspects and the potential impact on the simulated biogeography should then be at least discussed somewhere in the manuscript.

R: In this respect we provide the reference to a recent study by Bender et al. 2018 who reported on the role of iron "as a trigger" for colony formation and now state in the introduction that we test the hypothesis that the transition in the *Phaeocystis* life cycle is determined by iron availability.

Besides, Becquevort et al. (2007) and Hassler & Schoemann (2009) also showed that Fe addition had an effect on the morphotype dominance (colonies vs. solitary cells) of *Phaeocystis antarctica* with proportionally more solitary cells under low Fe conditions.

Becquevort, S., Lancelot, C., Schoemann, V.: The role of iron inthe bacterial degradation of organic matter derived from *Phaeocystis antarctica*. Biogeochemistry 83:119–135, doi 2007

Hassler, C. S. and Schoemann, V.: Bioavailability of organically bound Fe to model phytoplankton of the Southern Ocean, Biogeosciences, 6, 2281–2296, https://doi.org/10.5194/bg-6-2281-2009, 2009.

With respect to the influence of light:

Colonial *P. globosa* cells were found to be more effective competitors under high light conditions due to mucus formation, which was suggested to act as an energy drain mechanism storing fixed carbon in the form of polysaccharides inside the mucoid matrix (Riegman and von Boekel 1996). In line with this, colony formation of *P. antarctica* within a natural phytoplankton assemblage of the Ross Sea was favored under a high (52–276 µmol photons $m^{-2}$ $s^{-1}$) relative to a low natural light regime (11–58 µmol photons $m^{-2}$ $s^{-1}$, Feng et al. 2010). Based on Heiden et al. (2019), cell abundance of solitary relative to colonial *P. antarctica* cells as well as the number of colonies was similar between medium and elevated light treatments a, pointing toward a high light tolerance also of the single-celled *P. antarctica*. Similar findings were previously made for a single celled strain when exposing it to increasing irradiances (Trimborn et al. 2017).

Riegman, R., Van Boekel, W. : The ecophysiology of Phaeocystis globose, A review, Journal of Sea Research, 35 (4), 235-242, doi: 10.1016/S1385-1101(96)90750-9, 1996

Feng et al.: Interactive effects of iron, irradiance and CO2on Ross Sea phytoplankton, Deep-Sea Research I, 57, 368–383 doi:10.1016/j.dsr.2009.10.013, 2010

Heiden, J.P., Völkner, C., Jones, E.M., van de, Poll, W.H., Buma, A.G.J., Meredith, M.P., de, Baar, H.J.W., Bischof, K., Wolf-Gladrow, D. and Trimborn, S.: Impact of ocean acidification and high solar radiation on productivity and species composition of a late summer phytoplankton community of the coastal Western Antarctic Peninsula. Limnol. Oceanogr., 64: 1716-1736. doi:10.1002/lno.11147, 2019

Trimborn, S., Thoms, S., Brenneis, T., Heiden, J.P., Beszteri, S. and Bischof, K.: Two Southern Ocean diatoms are more sensitive to ocean acidification and changes in irradiance than the prymnesiophyte *Phaeocystis antarctica*, Physiol Plantarum, 160: 155-170. doi:10.1111/ppl.12539, 2017

- One vs two tracers for Phaeocystis: Have I understood correctly that your whole Phaeocystis biomass pool just switches back and forth between single cells and colonies based on the iron concentration threshold?

  R: Yes, we consider just one tracer (L144 original version). In the revised manuscript (L165) we write: "Note that in the model *Phaeocystis*, independent of the life stage – colonial phase or solitary cells, – is considered as one tracer."

I understand that this makes it computationally more efficient, but this might be too simplistic (I am not sure myself). Assuming I understood this correctly, are you tracking in space and time what "Phaeocystis state" the model tracer is in? Based on this tracking: are you confident that you capture the transitions well enough with just the dependency on iron to justify neglecting the other dependencies that have been suggested to be important (such as light levels), meaning that one model tracer is enough to simulate both life cycle stages simultaneously? This would be an important piece of information for other people wanting to implement Phaeocystis into their model. Please discuss this in the manuscript.

R: We did not track the "*Phaeocystis* state" for the present. We agree that this approach is quite simplistic, but nevertheless it agrees with the recent study by Bender et al. 2018 and allows Phaeo and coccolithophores to co-exist, which was our prior goal. However, we state "there is still room for improvement. For instance, for specifying more precisely the differences in photophysiology and related optical imprints (Moisan and Mitchell, 2018) for *Phaeocystis* in cells and colonies phases." (L355-356 in the original version, L589-591 in the revised manuscript).

- Sensitivity to chosen parameters: I would be curious to see how sensitive your simulated biogeography is to how long
Phaeocystisis in the colonial form during summer. Have you looked at the sensitivity to the chosen threshold? Additionally, what are the changes in parameters based on (30% and 25% higher mortality and grazing rate, respectively, as well as 20% lower kFe in single-cell-state, choices seem random)? How sensitive is the simulated biogeography to these choices?

R: We expect the model to be sensitive to the parameters since it will determine the competition/co-existing also between small diatoms and *Phaeocystis*, which will result in some changes of the PFT distribution. However, at first, we had to test whether this additional differences in the traits for haptophytes would help to get stable solution allowing both (coccolithophores and *Phaeocystis*) to co-exist. Careful calibration of the model with respect to these parameters could further improve the model performance.

L. 148: What is the horizontal resolution across the SO in the setup you're using here?

R: 18 km (L148, Subsection 2.1.2).

L. 151: If you state that your setup was similar to the one in Taylor et al. (2013), I am immediately wondering what is different. Please state this clearly.

R: With respect to the differences we state: "Starting on January 1st, 1992, the model with biogeochemistry was forced until 2012 by 3-hourly atmospheric surface fields of the Japanese 55-year reanalysis (JRA55, Kobayashi et al. 2015). Initially, the model time step had to be decreased to 10 min because of the higher forcing frequency, this constraint was slowly relaxed to 20 min by January 1st 1996. The change in forcing also required an adjustment of some the sea-ice model parameters. The albedos for dry ice, wet ice, dry snow, and wet snow were set to 0.75, 0.71, 0.87, and 0.81, respectively; the simulation did not use the replacement pressure method (Kimmritz et al. 2017)." (L180-185)

L. 160-165: Do you spin up the model in the coupled physical-biogeochemical setup immediately or do you spin up the physics first and only coupled once the circulation in spun up (or close to that)? This is not clear to me right now. I am wondering what impact spinning up both together (what it sounds like based on your manuscript) would have on the simulated biogeographies. Have you looked into this?

R: We do first spin up the physical model and then perform a spinup in the coupled physical-biogeochemical mode.

We now explicitly provide the details on the initialisations (L176-180):

"Initial conditions of the physical model were obtained from a spinup simulation initialised in January 1979 from rest and from temperature and salinity fields derived from the Polar Science Center Hydrographic Climatology (PHC) 3.0 (Steele et al., 2001). In the spinup phase, the model forced until the end of 1991 by 6-hourly atmospheric surface fields derived from the European Centre for Medium-Range Weather Forecasts (ECMWF) 40 year re-analysis (ERA-40) (Uppala et al., 2005). For more details see Losch et al. (2010, Section 3)."

And further (L185-187):

"After spinning up the biogeochemistry for six years (from 1992), during which also the physical simulation is adjusted further to the new forcing, the years 1999 – 2012 are integrated and the period of Aug 2002 – Apr 2012 is used for analysis."

After reaching a guasi-steady state, seasonal cycles of nutrients and PFT biomass are repeated with some interannual variability but no significant drift. Model nutrients climatology agreed with the World Ocean Atlas given correlation coefficient r = {0.92, 0.90, 0.97} and normalized standard deviation STD = {0.67, 1.27, 1.13} for silica, nitrate and phosphate, respectively.

L. 160: Please replace "evolved" by "involved".

R: Changed

L. 163-165: How does using model output from a different model compare to initializing with e.g. WOA and satellite derived chlorophyll concentrations (making some further assumptions on C:Chl ratios and the depth profiles)? Do you introduce biases?

R: With respect to chlorophyll, the model that, actually, has decoupled C and Chla (as well as in REcoM used in Taylor et al., 2013) forgets the initial state within the time from several hours up to several days. There is no need to do any assumptions on C:Chl ratios or the depth profiles. Thus, you do not introduce biases.

How does the model used in Taylor et al. (2013) perform in the SO?

R: The model simulations from Taylor et al. (2013) were evaluated and validated exactly for the Southern Ocean. As stated in Taylor et al. (2013):

„Generally, simulated mean monthly chlorophyll a concentrations (log-transformed) correlated to remote-sensing data at R = 0.62 globally and R = 0.23 for the Southern Ocean. These global correlation values are higher than those presented by other coupled general circulation model (GCM) studies [Schneider et al., 2008; Doney et al., 2009]. Lower correlations appear to be common for polar regions. In a review of both GCM and remote-sensing algorithm models of primary production, the Southern Ocean was found to be an area of highest divergence of estimates [Carr et al., 2006]."

L. 168-184: In this section, I am currently lacking a description of what model output you're comparing to the observations. Climatological? Single years? Co-located? Surface only? Please state here, what you're going to present in the result section, as this will help the reader to follow your structure.

R: We write in Section 3.4.2 "For a more precise comparison of the PHAEO model simulations with in situ information, we collected a series of 2 weekly model snapshots from August 2002 to April 2012 and considered the spatial distribution of Chla for diatoms (large + small), haptophytes and prokaryotes against in situ observations, if available, within a time window ± 1 week."

In general, Section 2.2. has been revised with respect to details on how the model is evaluated given particular data or information. Below we clarify what model output we take (and why) for the evaluation. Additionally, we have introduced a table summarizing the datasets used for PFT evaluation (including spatial and temporal representation) and related model output.

As for the comparison with the data by Smith et al. (2017), you need to be clearer here as it is not obvious how you compare the "simulated PFTs" (do you mean the simulated biomass concentrations? Please be precise) to SEM observations (cell counts). Again, do you co-locate? Do you use single year model output? Climatological model output?

R: we revised this part as following:

"Predicted PFTs were additionally compared to diatom and coccolithophores measurements (as cell counts) reported by Smith et al. (2017). These data were obtained by scanning electron microscopy in the North Atlantic and Indian Ocean sections of the Southern Ocean (the Great Calcite Belt area) during January – February 2011 and February – March 2012. For qualitative assessment of the agreement of the simulated diatom and coccolithophore distributions to these data, we provide estimates of the diatom vs. coccolithophores dominance to compare to the similar estimates by Smith et al. (2017) collocated in space and time."

The evaluation of the coupled model skill with respect to predicted PFT Chla was performed given in situ HPLC-based Chla retrievals for diatoms, haptophytes and prokaryotes (2166, 2388 and 1425 matchups, respectively) over the time period of August 2002 – April 2012 (Soppa et al. 2017). Quantitative assessment of the agreement between model and data collocated matchups were/are provided for several biogeochemical provinces (Tables 3 – 5 of manuscript and Tables S7 – S9 of the Supplementary Material). For the matchups we collected available in situ observations co-located with 2 weekly model snapshots within a time window ± 1 week (as originally stated in Section 3.4.2).

The qualitative assessment for simulated PFTs was performed when the observational data either from satellite or in situ are available in different units as the simulated data. Thus, the qualitative assessment for simulated PFTs was carried out for:

1) the simulated Southern Ocean PFT dominance by comparing to the PHYSAT PFT dominance climatological data product over the years of 1998 – 2006 (Alvain et al. 2008);

2) the diatoms vs. coccolithophores dominance (based on biomass, mmolC m$^{-3}$) in the Great Calcite Belt by comparing with collocated in situ cell counts by Smith et al. 2017 for January – February 2011 and February – March 2012 (keeping in mind that dominance by cell number and dominance by biomass are not equivalent and there is large uncertainty of converting cell counts to biomass);

3) PhytoDOAS coccolithophore fit (Losa et. 2018) and SynSenPFT Chla retrievals for diatoms and haptophytes(coccolithophores) over the same area and time period as shown in Smith et al. (2017);

4) Southern Ocean Diatom phenological indices (see Figure R2.8, added into the revised version of the manuscript as Figure 3a,b,c) compared to Soppa et al. (2016).

Spatial distributions of the simulated nutrients were compared with World Ocean Atlas (Garcia et al. 2014). Originally, the spatial distribution of the simulated surface phosphate, silica and iron were provided for March (2004) in support to discussion on drivers of coccolithophores biogeography (Section 3.3). We added the following information in the manuscript: "In general, the simulated surface nutrient climatology agrees well with the World Ocean Atlas given correlation coefficient of 0.90, 0.97 and normalised standard deviation of 1.27, 1.13 for silicon and phosphate, respectively (see also the Supplementary Material)."

L. 186: Similar to above: Please state very clearly what model output you take (and why) for the evaluation. As stated in the comments further down, I find it very confusing as a reader that you currently pick what seems like random months of a random year and are additionally not consistent across the different simulations (compare Fig. 2, which shows July & January, to Fig. 7, which shows June-August and December-February; compare Fig. 4, which shows February 2008, to Fig. 5, which shows March 2004, or to Fig. 8, which shows March 2012). Please rewrite this section accordingly and double-check how you can be consistent in the use of the years.

R: We apologies, as we now realise how confusing it was to have multiple months/years presented without providing explanations. We now compare to climatologies where possible, but given the importance of interannual variability or specific patterns in the discussed distribution of PFTs and nutrients we still consider a particular month and include comparison of PFT to HPLC matchups and SEM observations from the actual month/year. To make this less confusing, we explicitly state in the text why we chose specific months/years.

In the beginning of Section 2 we now introduce the following overview:

"To assess our model results, we compare the simulations to several large in situ and satellite datasets, as detailed below and summarized in Table 2. Where the coverage of the observations is similar in respect to time we use our two-weekly model outputs. Where only monthly climatological or composite data (often from different time periods) are available we use monthly climatological model results for the period of 2006-2012. Where only results for specific months are available from observations we compare our output to these specific months. Table 3 contains the information about the evaluated phytoplankton groups as classified in the model and observations."

We extended Section 2.2.2 with the following text: "For qualitative assessment of simulated PFTs, we compare model climatology of Southern Ocean PFT dominance (averaged over the years 2006 – 2012) to the PHYSAT PFT dominance climatological data product (1998-2006, Alvain et al. 2008). Comparison of predicted Chla for diatoms and coccolithophores with PhytoDOAS coccolithophore fit (Losa et. 2018) and SynSenPFT Chla retrievals for diatoms and haptophytes (coccolithophores) were carried out for the same area and time period as shown in Smith et al. (2017)".

L. 196: You state "only 0.5°" –how does this compare to your model resolution? (You give an average resolution of 18km, but it wasn't clear to me over what area that is averaged, see further up)

R: The discussion in line 196 refers to the spatial and temporal resolution of PhytoDOAS product as initial input information in the SynSenPFT algorithms, which derived PFT at a daily 4 km resolution (see Losa et al. 2017 for more details). There was no averaging nor projection done with respect to model grid, since it was only foreseen as qualitative evaluation w.r.t the satellite products.

We slightly reformulated this section to improve it:

"Model results are compared to phytoplankton dominating groups from the climatological monthly mean satellite derived product PHYSAT (1998-2006, Alvain et al., 2008). PHYSAT is based on the analysis of normalized water-leaving radiance anomalies, computed after removing the impact of chlorophyll-a variations. Specific water-leaving radiance spectra anomalies (in terms of spectral shapes and amplitudes) have been empirically associated to the presence of dominant phytoplankton groups, based on in situ diagnostic pigment observations. This product is based on the multispectral Sea-Viewing Wide Field-of-View Sensor (SeaWiFS) information and available in http://log.cnrs.fr/Physat-2?lang=fr.

We also evaluated the model simulations (mg m$^{-3}$) against the satellite PFT Chla (mg m$^{-3}$) product SynSenPFT (Losa et al. 2017, https://doi.org/10.1594/PANGAEA.875873). The SynSenPFT product combines the information of two satellite PFT Chla products: one retrieved with the differential optical absorption spectroscopy method (PhytoDOAS, Bracher et al. 2009; Sadeghi et al. 2012) applied to hyperspectral information from the Scanning Imaging Absorption Spectrometer for Atmospheric Chartography (SCIAMACHY, Bracher et al. 2017; https://doi.org/10.1594/ PANGAEA.870486) and the OC-PFT abundance-based approach (Hirata et al. 2011 and refined in Losa et al. 2017) applied to multi-spectral satellite total Chla data from the Ocean Colour Climate Change Initiative (OC-CCI). While the PhytoDOAS products from the SCIAMACHY sensor are only available at 0.5° spatial resolution and monthly means, OC-PFT applied to OC-CCI Chla products can be obtained daily and at 4 km resolution.

PhytoDOAS and PHYSAT satellite products are derived based on phytoplankton absorption properties captured by the satellite sensors and distinguished by the retrieval algorithms either as a particular PFT optical imprint ("finger print") in case of available hyperspectral information (in PhytoDOAS) or as anomalies in a multispectral signal (in PHYSAT). Thus, the PhytoDOAS allows to retrieve quantitatively major PFTs (coccolithophores, diatoms, cyanobacteria), while PHYSAT provides information about five dominant phytoplankton groups: prokaryotes (presented by *Prochloroccocus* and *Synechococcus*-like SCL), diatoms, haptophytes in general and *Phaeocystis* in particular.

We compare model climatology of Southern Ocean PFT dominance (averaged over the years 2006 – 2012) to the PHYSAT PFT dominance (dominance of the modeled PFT is defined if its Chla fraction is more than 55% of the total Chla). In line with the evaluation against the PHYSAT PFT dominance, the simulated PFT dominance are compared to similar estimates obtained in the study by Dutkiewicz et al. (2015). Two SynSenPFT products (at 4 km and daily) – diatoms Chla that combines diatoms

Chla from PhytoDOAS and OC-PFT, and coccolithophores Chla that combines coccolithophores Chla from PhytoDOAS with haptophytes Chla from OC-PFT – are used in addition to the in situ based diatom vs. coccolithophores dominance by Smith et al. (2017). Hence, we only use the same areas and time period as in their study for comparisons to the SynSenPFT results. Here as well the comparison is qualitative as the SynSenPFT products are mostly based on OC-PFT in our study region and the global relationships between Chla and the fraction of PFTs from the OC-PFT algorithm might differ in the Southern Ocean, as shown by Soppa et al. (2014) for diatoms."

**Results & Discussion**

L. 205: "Improved" compared to what?

R: We changed the title of the subsection to "Diversity within diatoms".

L. 206: From the title of the section, the reader expects a discussion of phytoplankton phenology here (i.e. e.g. bloom timing, bloom peak timing, bloom duration), but instead you discuss dominance patterns. Please choose a more appropriate title. In fact, I would suggest to not use "phenology" throughout the text as you currently do not really assess it in a quantitative sense. If you want to keep it (and there is value to that!), you need to introduce this in the method section, where the definition of bloom start etc. is currently missing, and present the simulated phytoplankton phenology and the comparison with e.g. satellite derived phytoplankton phenology.

R: We agree and have changed the title of the subsection to "Diversity within diatoms".

L. 206-223: You never state in the method section that you will compare model output from a version without the listed changes to the setup which includes the changes. Please add this to the method section.

R: We have extended the method section w.r.t. comparison to the version of Dutkiewicz et al. (2015):

"In line with the evaluation against the PHYSAT PFT dominance, the simulated PFT dominance are compared to similar estimates obtained in the study by Dutkiewicz et al. (2015)"

L. 207: "were misrepresented" –please rephrase to state more clearly what model version/setup/simulation you're referring to here.

R: We have added the reference to the Darwin-MITgcm version of Dutkiewicz et al. (2015).

L. 208-209: How confident are you in the satellite-derived dominance pattern in austral winter (July)? Additionally, do you really think that for a region like the SO, it is critical how well the model simulates the dominance patterns in winter? Personally, I would have preferred to see the agreement for all summer months (December-February or even March) to additionally get a better feeling for how the model is doing in terms of seasonality.

R: We now show the PFT dominance for all twelve months in the Supplementary Material. As mentioned before, we show the simulations also for the austral winter because even during this period north of the subtropical front the model simulations show not a negligible biomass (please see supplemented video materials). In addition, we wanted to be consistent with what was shown in the study by Dutkiewics et al. (2015), as well as to explicitly illustrate the disagreement between PHYSAT and "old version" (as well as other tested configurations reviewed in the Supplementary Material) and because this disagreement in winter PFT dominance resulted from incorrectly simulated PFT phenology (e.g. very early bloom of diatoms). This was initially a motivation to consider to size classes of diatoms.

Here (in Figures R2.4 – R2.7) we compile all climatological monthly mean PFT dominance for PHYSAT, Darwin-15, REF and PHAEO (REF and PHAEO climatologies are calculated over the years 2006 - 2012). These four figures are now in the Supplementary Material.

In the revised version of the manuscript (Figure 2) we present PFT dominance for climatological December – January – February and July obtained from PHYSAT, Darwin-2015 (Dutkiewicz et al., 2015) and REF experiment (blue boxes in Figures R2.4 – R2.6).

Note: as seen from the comparison of 2003/2004 monthly mean PFT dominance (presented in the original version of the manuscript for REF and PHAEO) to the climatology, the year 2003/2004 is typical with respect PFT dominance. The outputs are similar to climatology but showing finer spatial structures.

Figure 8 (in the revised manuscript) depicts PHAEO climatological December – January – February and July PFT dominance (blue boxes in Figure R2.7).

In the revised manuscript we edited the text as following (L306-317):

"For complete 12 monthly mean climatologies for PFT dominance as retrieved by PHYSAT and predicted in Dutkiewicz et al. (2015) and REF experiment, the reader is referred to the Supplementary Material (Figures S15 – S17, respectively). In general, the PHYSAT Southern Ocean PFT dominance climatology (over the years 1998 – 2006) shows a strong seasonal variability of PFT compositions and contributions of PFTs to TChla (Alvain et al., 2008). From November to January south of 40°S, the diatom contribution is higher than 50%. This high diatom contribution during the austral spring and summer is associated with large diatom blooms starting in October at lower latitudes and moving towards higher latitudes in December – January. The nano- non-silicified phytoplankton is dominating during the time period from March to October. The Southern Ocean PFT dominance obtained in Dutkiewicz et al. (2015) disagrees with PHYSAT observations: diatoms are underrepresented in comparison to PHYSAT in circumpolar Southern Ocean during January and February, while in July they are over-represented in the Atlantic section of the Subantarctic Zone which is also opposed to the observed dominance of haptophytes. Generally, the model version Dutkiewicz et al. (2015) overestimate the dominance of small non-silicified phytoplankton. These results clearly indicate deficiencies in the Dutkiewicz et al. (2015) model setup and motivated a series of Darwin-MITgcm experiments, with different model configurations with respect to assumed PFTs and their traits described by various physiological parameters."

[Figure]

**Figure R2.4: PHYSAT dominance climatology over 1998 – 2006 (Figure S15 in Supplementary Material).**

[Figure]

**Figure R2.5: DARWIN-2015 (Dutkiewicz et al. 2015) dominance climatology, masked by PHYSAT missing values (Figure S16 in Supplementary Material).**

[Figure]

**Figure R2.6: REF dominance climatology over 2006 – 2012 (Figure S17 in the Supplementary Material). The model output is masked by the area with sea ice concentration > 75% during respective month. White contours denote the Southern Ocean fronts (Orsi et al., 1995; Orsi and Harris, 2001).**

[Figure]

**Figure R2.7: PHAEO dominance climatology over 2006 – 2012 (Figure S18 in Supplementary Material). The model output is masked by the area with sea ice concentration > 75% during respective month. White contours denote the Southern Ocean fronts (Orsi et al., 1995; Orsi and Harris, 2001).**

L. 208: The transition between sentences is confusing for the reader: "[...] in austral summer. In July, [...]" First, you set the reader up for hearing more about the summer and then you jump to talk about July. Please rewrite.

R: We have rewritten as following: "…, while in July simulated diatoms were dominant in the"

L. 210: Related to above, looking at the model performance in a single month does not tell you much about how the model is doing in terms of simulating phenology. Please rephrase.

R: Changed to "indicates model deficiencies in presenting PFT distribution."

L. 210-211: Which model are you referring to here?

R: We added a reference to Darwin-MITgcm – 2015 version (Dutkiewicz et al., 2015).

Throughout the text, please add references to panels of the Figures (these need to be added to each Figure!), as this will be very helpful for the reader.

R: In the revised version we added the references to Figure panels.

Maybe refer also to the HPLC data here? These should support the discussed bias in the community at high latitudes.

R: The REF experiment was not compared to the HPLC derived PFT Chla since it was not our "final" model (did not differentiate among haptophytes).

L. 211-214: This information belongs into the method section. What exactly do you mean by "in terms of agreement with observed phytoplankton composition"?

R: Here we reported (as a summary) on the experiment showing best results among a number of sensitivity experiments (some of them listed in the Supplementary Material) with respect to diatom phenology and general PFT dominance (that reflects also the phenology) but not allowing to distinguish among haptophytes. It is why we just briefly report on the results with the kind of motivation for experiment PHAEO that is evaluated more thoroughly (and also contains small and large diatoms).

How did you evaluate this?

R: The simulated composition/dominance was qualitatively evaluated against PHYSAT and Trimborn et al. (2015) observations.

For completeness, consider adding the reference Trimborn et al. (2015) to the method section

R: The reference to the study by Trimborn et al. (2015) has been added.

2.2.1. Where do you show the diatom phenology of the model?

R: Phenology itself (not phenological indices) can be seen from the supplemented videos showing the dynamics and distribution of the Southern Ocean diatoms, haptophytes and prokaryotes simulated with experiment PHAEO over the time period August 2002 – April 2012.

L. 218-220: I am curious to what extent the improvement of the model in the SO is at the expense of the model performance on the global scale. Are the simulated patterns still reasonable?

R: we have not thoroughly evaluated model performance globally. Some assessments have been performed for the Arctic Ocean. Results showed satisfactory results with respect to TChla, bloom development and nutrients. On a global scale the simulated PFTs show expected distribution of prokaryotic pico-phytoplankton at low latitudes (from 40ºN to 40ºS) and abundance of diatom and haptophytes at high latitudes.

L. 220-223: I cannot follow what you base this conclusion on given the plots you're showing in the manuscript, but I think this is an important point to make.

If you really significantly improve the simulated phenology by including two types of diatoms instead of one, this aspect deserves a lot more room than it currently gets in the manuscript in my opinion.

R: Indeed, we planned a separate paper focusing on the analysis of phenological indices. The dominance plots are a part of model evaluation with available satellite information, but nevertheless reflects the PFTs dynamics (or phenology, if the term "phenology" is used as PFT Chla dynamics in general).

We added the figure below on REF phenology of diatoms (Figure 3 in the revised version of the manuscript). The figure shows the spatial distribution of the REF diatom phenological indices in 2007/2008: bloom start date, chlorophyll-a maximum date, and bloom end date. "We chose this particular year because: 1) with the two-weekly model output the phenological indices can be more precisely calculated than based on the two-weekly or monthly mean climatology; 2) it is a typical year over the period 2006 – 2012 with respect to the simulated PFT distribution (after model reached the quasi-steady state) and climate oscillations (Soppa et al., 2016)."

Consider including plots of the simulated phenology (e.g. bloom start and bloom peak of total chlorophyll and diatom chlorophyll in "old version", REF and PHAEO) as compared to those derived from satellite products.

R: Figures R2.8 and R2.9 depict diatom phenological indices – bloom start date (BSD), chlorophyll maximum date (CMD) and bloom end date (BED) – for experiment REF and PHAEO, respectively. The phenological indices were calculated following (Siegel et al. 2002, Soppa et al. 2016) based on model snapshots of the year 2007/2008. For the "old version" (Dutkiewicz et al. 2015) the outputs are stored only as monthly means, which makes it hardly possible to estimate accurately the phenological indices.

Figure R2.10 show Chla spatial distributions for large and small diatoms for three months of the year 2007/2008. This figure
demonstrates clearly that south of the  Polar front diatom is mostly represented by large cells, while north of the Polar front
the diatom is abundant in small cells showing distinct phenology.

Section 2 has been extended by additional Subsection 2.3 "Diatom phenological indices" with the following text:
"Following Soppa et al. (2016) we evaluate the diatom phenology by calculating phenological indices based on a threshold
method proposed and initially applied for assessing the TChla phenology by Siegel et al. (2002). In particular, we use the
following indices: the Chla maximum date, the bloom start date, and the bloom end date. These indices are calculated based
on the REF Chl simulations for diatoms (including small and large) over the year 2007/2008…"

[Figure]

**Figure R2.8: REF diatom Chla phenological indices: bloom start date (BSD, a), chlorophyll maximum date (CMD, b), bloom end date (BED, c) (Figure 3(a,b,c) in ther revised version of the manuscript).**

[Figure]

**Figure R2.9: PHAEO diatom Chla phenological indices: bloom start date (BSD, a), chlorophyll maximum date (CMD, b), bloom**
**end date (BED, c).**

[Figure]

**Figure R2.10: REF: spatial distribution of Chla for large diatoms (upper panels) and small diatoms (low panels) for December 2007 (a and d), February (b and e) and March (c and f).**

Figure R.210 is similar to Figure 3(d,e,f,g,h,i) of the revised version of the manuscript. (Mind that in the manuscript the figure panels - middle and low – present October 2007, December 2007 and February 2008).

Consider also adding a reference to the regional SO model used in Nissen et al. (2018) here, as this model simulates too early total chlorophyll/diatom blooms as well, demonstrating that this issue is not restricted to global models.
R: We now include the reference to the study by Nissen et al. (2018).

L. 225-226: Where is this seen? You don't show the biomass patterns for the run without the two diatom classes in the current form of the manuscript.
R: We refer to the runs without considering two diatom classes in the supplementary material. The runs shown in the main manuscript clearly show the distribution of large diatoms at high latitudes and small diatoms at low latitudes (see Figure 3 and

Figure R2.10). Figure 3 has been replotted for the year 2007/2008 and also includes now the phenological indices for the same time period (see also Figure R.2.8).

L: 228-229: In what way is the simulated pattern in agreement with the cited studies? Please be more precise here. Related to earlier comments, how did you evaluate this exactly?

R: We evaluate this qualitatively by comparing our plots to those presented by Signorini et al. (2006), Balch et al. (2016) and Smith et al.(2017) in their papers. We removed this sentence in the revised version of the manuscript.

L. 233-236: Consider rephrasing "the model representation of co-existence/competition within the haptophyte group" to something like "the simulated biomass distributions of both coccolithophores and Phaeocystis were very sensitive to chosen model parameters, and small changes in [...]".

R: Thanks, rephrased as suggested.

What "small changes in the Darwin model physiological parameters" are you referring to here exactly? What is a small change in this context? And which parameters are you referring to?

Can you include more information on these in the supplementary material?

R: 5% (or less) changes in palatability factor or mortality rate of coccolithophores or *Phaeosystis* could lead to the situation when one overcompetes the other. This was documented in the sensitivity tests overviewed in the Supplementary Material (S1.1.2 – S1.1.4).

Am I understanding it correctly that by the end of your reference simulation, coccolithophores go extinct in your model? If this is indeed what you mean, I am not entirely sure I understand why this happens, but I certainly find it very worrisome for the evaluation of your reference simulation, as this implies that you have significant drift in your PFT biomass concentrations and/or distributions. Is this the case?

R: In revised version we write that "after reaching a quasi-steady state, in experiment REF" and show REF results after reaching this quasi-steady state. In the revise version of the manuscript we more carefully state the issues of the long-term decline in coccolithophores.

"For instance, in experiment REF after reaching a quasi-steady state, coccolithophores did not survive. It happened because there were not sufficient differences between the traits assumed for coccolithophores and "other large" (or *Phaeocystis*-analogue). As a result, it took longer for the model to get in a quasi-steady state and finally lead to just one of the haptophytes survived (taking over for another). Hence, the experiment REF represents diatoms and haptophytes after reaching a quasi-steady state, but cannot distinguish among haptophytes. In original Darwin-2015 model (Dutkiewicz et al. 2015) "other large" did not survive"

This also worries me in that your choice of showing different time periods in the different figures of the manuscript will then have a possibly considerable impact on the biogeographies you show.

R: All the figure shown in the original version of the manuscript (except for figure 3 and 6, the panels depicting exp. REF related to distinguishing among haptophytes) were representative for the considered period of evaluation time period August 2002 – April 2012 (see figure R1.1 in our response to reviewer 1). In the revised version of the manuscript we do not show the year 2003/2004.

The issue we stressed here with REF results is about the model behavior in the presence of the biochemical module structure
with two similar (with respect to the assumed traits) PFTs leading to that just one of these two exists. In experiment REF we did not differentiate sufficiently enough between the traits assumed for coccolithophores and "other large" (or Phaeo), which lead to a longer integration period of time to reach a quasi-steady state and to just one of "similar" PFTs survived. It means that in experiment REF *Phaeo*-analogue indeed represents haptophytes in general (taking over for coccolithophores). In the original Darwin-2015 model (Dutkiewicz et al. 2015) "other large" did not survive.

We make this clearer in the text as mentioned above.

In observations, the biogeographies of coccolithophores (mainly in the subantarctic) and Phaeocystis (only P. Antarcticain the SO, mainly in the high-latitude SO) do generally not fully overlap, so I don't understand how competitive exclusion between these two types of phytoplankton leads to the extinction of one in the model, as I don't see these two types exclusively
competing for nutrients.

R: In model configuration REF (as in Darwin-2015), however, the traits initially considered in the way that "other large" and coccolithophores compete for the same resources. By considering two life stages of *Phaeocystis* we introduce additional differences in the traits, which along with changed physiological parameters for coccolithophores makes coccolithophores competitive among phytoplankton of larger cell size (or colonies) that requires higher nutrients concentration to grow and/or
among PFTs of similar size – small diatoms and *Phaeocystis* solitary cells – that have higher palatability factor to be grazed. We make this clearer in the text (L582-587).

L. 240: Please clarify: Does the reference simulation already have the changes listed in the method section (in the nutrient affinity and the grazing pressure)? In the method section it sounds like it, here in the result section it does not, I got confused.
R: The reference simulation (REF) contains the changes with respect to the nutrient affinity and the grazing pressure as listed in the Method section. The discussed in line 240 is the introduced "two distinct life stages of Phaeocystis ant. (colonies and solitary cells) in which its morphological features and physiology depend on iron availability (Bender et al., 2018)."

L. 245: Why this exact month?
R: February is chosen as one of the austral summer months (in Figure S8 of the supplementary material, we present also results for different months). The year 2008 is typical (as explained above). We clarify in the text:

"Figure 4 presents these meridional PFT distributions of the different PFTs in February 2008 (one of the months discussed in the previous subsection, Figure 3)"

L. 247: Please clarify: By "other large", you mean large diatoms and Phaeocystis together?

R: "Other large", in terms of the study by Dutkiewicz et al. (2015), means non-silicified nano-phytoplankton, in our case *Phaeocystis*-analogue, not strictly however (see table 2 of the original version, table 3in the revised version).

The text is revised accordingly:

"One can see that in experiment REF, "other large" (Dutkiewicz et al. 2015, in our case non-silicified nano-phytoplankton including Phaeocystis, but not strictly) outcompetes coccolithophores leading to too low concentrations of coccolithophores north of the Polar Front…"

Also, your statement "too low concentrations of coccolithophores south of the PF" is based on what? This statement confuses me due to two reasons: First, Fig. 4 only shows relative contributions to total phytoplankton biomass and does not give any information on absolute biomass levels. Second, I am not aware that one would expect significant concentrations of coccolithophores south of the PF (see e.g. Balch et al., 2016). So what exactly are you referring to here?

R: Thank you for this comment. The correct statement should be "too low concentrations of coccolithophores **north** of the PF" for experiment REF based on what is known from observations in the Great Calcite Belt (Smith et al.,2017; Balch et al., 2016)

L. 249: Similar to above, what do you mean by "more plausible" here? Compared to what? Please be more precise and avoid subjective judgement.

R: "more plausible" with respect to the coccolithophores fractions gradually increasing in the direction to the north of the subantarctic front, compared to experiment REF that could not simulate specifically coccolithophores in the Great Calcite Belt (Balch et al. 2016, Smith et al. 2017) since hardly distinguished between coccolithophores and Phaeocystis.

However, we have changed the text as to not using subjective statements.

"One can see that in experiment REF, "other large" (Dutkiewicz et al. 2015, in our case non-silicified nano-phytoplankton including *Phaeocystis,* but not strictly) outcompetes coccolithophores leading to too low concentrations of coccolithophores north of the Polar Front, while small diatoms exist in both experiments (however, in different percentages). In experiment PHAEO, meridional distributions of the phytoplankton composition reveal that the coccolithophores fraction gradually increases to the north of the Subantarctic Front… This result is comparable to the estimates of Smith et al. (2017) obtained in AOS and IOS  for late summer (January – February – March) of the years 2011 and 2012."

L. 251: I think this statement needs to be rephrased. Smith et al. (2017) state that based on their measurements, coccolithophores made up maximum20% of total chlorophyll concentrations locally, but generally contributed less than 5%. Consequently, I would phrase it more conservatively than saying that simulating 30% of total biomass is in agreement with Smith et al. (2017), which it clearly isn't.

R: We agree, our estimates of PHAEO coccolithophores contribution to total biomass exceed 20% reported by Smith et al.
2017 based on *in situ* SEM (but in much better agreement than those from experiment REF). Moreover, in general, our PHAEO results confirm what is clearly stated in this study by Smith et al. (2017): in the Great Calcite Belt, coccolithophores are "an important contributor to phytoplankton biomass" and can contribute more than 50% into the biomineralizing phytoplankton. To avoid any further confusions, we now present diatom *vs.* coccolithophores dominance as shown in Smith et al. 2017. We now provide the following figure (Figure R2.8, Figure 5 in the revised version of the manuscript) depicting diatom *vs.*
coccolithophores dominance obtained in experiment PHAEO…

"…collocated in space and time with observations of Smith et al. (2017). Although our estimates have been obtained based on phytoplankton biomass (mmolC m$^{-3}$), but not on cell counts as in Smith et al. (2017), our results agree to their higher concentrations and dominance of diatoms in the SBDY and SACCF, while north of the Polar Front coccolithophores become
more abundant. However, as compared with Smith et al. 2017 (their figure 2), in the Atlantic section, the dominance of simulated coccolithophores (55%) is shifted northward of the subantarctic front leading to underestimation of the coccolithophore dominance along the polar front and south of SAF front and overestimation north of SAF."

[Figure]

[Figure]

**Figure R2.11: PHAEO Diatom *vs.* coccolithophores dominance averaged over January-February 2011 (a) February - March 2012 (b). The size of the circles is relative to phytoplankton carbon content (mmolC/m³). The largest size of the circle corresponds to 3.12 (mmolC/ m³). (Figure 5 in the revised version of the manuscript)**

L. 253: I think you're referring to Section S3 here.

R: Right we refer to Section S3.

Do I expect the fraction of coccolithophores to be higher in winter? How is this backed up by observations (e.g. HPLC)?

And how relevant is the community structure in SO winter, when biomass levels are generally very low?

R: North to subtropical front the model simulations show not negligible biomass (please see supplemented video materials). However, we cannot confirm that it is expected that fraction of coccolithophores to be higher in winter, since there is no *in situ* HPLC observation except for August 2006. However, based on satellite retrievals Alvain et al. (2008) reported on decreased fractions of diatom from March to September and mostly non-silicified nano-phytoplankton contributing to the

TChla.

L. 254-255: This is an obvious statement. What is the reader to take away from the distribution of zooplankton biomass?

R: Sorry, in a pre-version, there were a following sentence on comparison of the distribution of zooplankton biomass (mmolC m⁻³) to the MAREDAT data (in the context of shown ranges). Thus, we wanted first to comment on the agreement with the

MAREDAT zooplankton data and continue with zooplankton grazing pressure.

In the revised version we write:

"The largest differences between REF and PHAEO zooplankton is shown between 75°S and 50°S. However, for both experiments, REF and PHAEO, simulated zooplankton is within 0 to 20 mgC m⁻³, which agrees with in situ observations reported by Moriarty and O'Brien (2013) and shown in Dutkiewicz et al. (2015) and in Supplementary Material (Section 5.2)"

L. 255-260: Here again, what is the "realistic distribution" for you? What are the "other circumstances"? This is a very vague statement. Please be more precise.

R: Rephrased as following:

"The discussed distribution of coccolithophores have been obtained under the assumption of lower palatability function
(leading to lower grazing pressure) in comparison with what is assumed for other PFTs"

By "other circumstances" we meant other assumptions on the PFT traits made in the current model set up. We delete the words in the revised manuscript

Have you done a sensitivity simulation in which coccolithophores could not escape the grazing pressure to assess the impact
on the biomass distributions and community structure? This would be very interesting to back up your statement.

R: Indeed, our choice of particular model configuration is based on several sensitivity tests with the assumed PFT palatability factor. However, it was also supported by literature.

Related to above, in this context it will matter a great deal how different you choose e.g. the maximum grazing rates of
zooplankton grazing on coccolithophores as compared to grazing on e.g. diatoms in the model, which is related to what assumptions you make regarding the coccolithophore community you're simulating (all coccolithophore species? E. huxleyi only?

Please see also comment further up and prey preferences of the zooplankton PFTs.

R: In other words, depends on assumed palatability factor for diatoms, *Phaeocystis* and coccolithophores (0.8, 0.78, 0.58,
respectively) regarding to coccolithophores community but considering still *E. huxleyi* as dominant (Krumhardt et al. 2017, Smith et al. 2017).

The "palatability matrix" defines the zooplankton preferences for different phytoplankton types. For most phytoplankton this is 1. Maximum grazing rate is multiplied by this factor.

Furthermore, I am wondering how high your simulated coccolithophore carbon biomass concentrations are compared to e.g. MAREDAT observations.

R: Compared to MAREDAT observation, we overestimate coccolithophores carbon biomass (Figure R2.1). However, it is worth keeping in mind, that estimamted uncertainties for MAREDAT coccolithophores due to the convertion from cell counts to biomass are several 100%.

Taking your ~30% contribution of coccolithophores to total biomass (which seems a bit higher than that suggested by Smith et al. (2017), see above) and a maximum of ~20% in austral summer between 40-50°S in Nissen et al. (2018; their Figure 3), in my view, it is very conceivable to assume that this difference is to a large extent controlled by differences in assumptions surrounding the grazing formulations.

R: We agree, our PHAEO estimates of the coccolithophores contribution to total biomass a bit higher than 20% estimates reported by Smith et al. 2017 (but in much better agreement than those from experiment REF). However, in general, our PHAEO results confirm what is clearly stated in this study by Smith et al. (2017): in the Great Calcite Belt, coccolithophores are "an important contributor to phytoplankton biomass" and can contribute more than 50% (up to 78%) into the biomineralizing phytoplankton. To be more consistence with the study by (Smith et al. 2017) we now present diatom *vs.*

coccolithophores dominance (Figure R2.11). We agree, that we could still calibrate the model and get lower value with slightly enlarged grazing pressure on coccolithophores, but it should be still less than for diatom and *Phaeocystis*.

Monteiro et al. (2016) reported that, in subpolar regions, the cocccolithophore contribution to total phytoplankton biomass can increase up to 40% under bloom conditions.

Additionally, if one looks at the discussion in e.g. Monteiro et al. (2016), there is a lot that is still not understood with respect to the coccosphere and grazing pressure from zooplankton, which is why I don't think one can per se say that coccolithophores should always escape grazing pressure in models –in the same way as I don't think the reverse can be stated (will be highly dependent on the ecosystem structure at a given location). Therefore, I think it is important to point that out in the manuscript.

R: We agree with the reviewer, that the distribution of coccolithophores is not always explained only by grazing protection but a combination of different factors (e. g. immune to light, high affinity to nutrients) in different area, we emphasize this in the manuscript, we now include the references to the study by Krumhardt et al. (2017) and Monteiro et al. (2016).
It is worth mentioning, that in the study by Monteiro et al. (2016), the authors stated that

– "the reduction in grazing pressure might have been the likely initial reason for why coccolithophores calcify" and
there are other benefits (associated with calcification)

–  "grazing protection appears to favor coccolithophores in (sub)polar, coastal, and equatorial areas" (their Fig. 4)

– the initial benefits "associated with grazing protections have relatively well-supported evidence"

W.r.t. what is still not understood, the authors of the study by Monteiro et al. (2016) mentioned that studies investigating the
protective role of coccolith based on comparison of direct grazing on different clones (calcified and noncalcified, representing different phase) of same coccolithophore species reported on slower, the same or faster grazing on calcified cells in comparison with non-calcified coccolithophore cells, but these results might be obtained because of other independent of calcification conditions.

In the text we write:

"The discussed distribution of coccolithophores have been obtained under the assumption of lower palatability function (leading to lower grazing pressure) in comparison with what is assumed for other PFTs. This contradicts the study by Nissen et al. (2018), who reported on an increased (relative to diatoms) grazing of coccolithophores as a factor controlling the coccolithophore biogeography in the Southern Ocean. Our assumption on low palatability factor of coccolithophores are, nevertheless, backed up by studies by Nejstgaard et al. (1997), Huskin et al. (2000), Losa et al. (2006) and Monteiro et al. 2016. In the study by Losa et al. (2006) on optimized biogeochemical parameters, it was shown that coccolithophore blooms are associated with low grazing pressure. Based on laboratory experiments, Nejstgaard et al. (1997) and Huskin et al. (2000) concluded that coccolithophores (due to its "stony" structure) do not influence the microzooplankton growth. While the exact mechanisms of how this PFT use the coccolith to protect itself against grazing is not fully understood (Monteiro et al. 2016), the ability of coccolithophores to escape grazing control has "relatively well-supported evidence" (see Monteiro et al. 2016 for review)."

Additionally, note that Nissen et al. (2018) state that grazing is a major control on the simulated coccolithophore biogeography and their biomass concentrations relative to those of diatoms, but they do not comment on the effect of the assumed grazing difference between diatoms and coccolithophores on the simulated phenology of the two in the subantarctic. Please rephrase L. 258 accordingly.

R: Rephrased accordingly (the word "phenology" has been removed).

Additionally, without the relative grazing advantage of coccolithophores relative to diatoms, the simulated coccolithophore biomass levels in Nissen et al. (2018) increase three-fold between 40-50°S (see their Figure 7), pushing the simulated coccolithophore biomass levels way beyond what MAREDAT observations suggest for this area.

R: We are not sure it is a good argument. It could be just a compensation for other model deficiencies. Moreover, the aforementioned representation error in the MAREDAT data are large.

Besides, in the studies by Krumhardt et al. (2019), Monteiro et al. (2016) the grazing pressure (palatability factor) for coccolithophores was also considered lower than for diatoms.

L. 260: Do you assume the drivers to be the same globally? In my view, one could very well imagine a difference in the relative importance of grazing in controlling coccolithophore bloom phenology, as the competitive success of coccolithophores will largely depend on 1) which coccolithophores are present (and hence simulated), 2) which other phytoplankton are present, and

3) which grazers are present.

R: We agree with the reviewer, moreover, in our discussion we provide two examples.

1) "The simulated abundance of coccolithophores north of the Subtropical Front (STF) – where phosphate occurs in very low concentrations – is explained by the introduced high affinity of this PFT to phosphate (small half-saturation rate) allowing coccolithophores to grow in nutrient depleted conditions.

2) "in the region between the Subtropical and Subantarctic Fronts the occurrence of coccolithophores is more evidently linked to low grazing pressure on this PFT, due to its much lower palatibility for zooplankton in comparison with small diatoms or *Phaeocystis* presented by single solitary cells."

I suggest to point this out as a potential limitation of the comparison of a study focusing on the North Atlantic to the one here.

R: Our assumptions are based not only on studies focusing on the North Atlantic, they are supported also by the study by
Paasche, 2001, Iglesias-Rodríguez et al (2002), Rost, B. and Riebesell (2004), Krumhardt et al. (2017) provided global overview on coccolithophores as well as backed up by the study by (Monteiro et al. 2016).

L. 269: Please rephrase in order to avoid subjective statements like "agreed well". Additionally, where is this seen? I suggest to add validation plots to the supplementary material.

R: We provide the statistics and we rephrase the text as following.

"In general, the simulated surface nutrient climatology agrees well with the World Ocean Atlas (Garcia et al., 2014) with correlation coefficient of 0.90 and 0.97 and normalised standard deviation of 1.27 and 1.13 for silicon and phosphate, respectively".

L. 266-278: Why do you show March of 2004 now?

R: In the revised version we show the year 2008 (a typical year, it does not change anything with respect to discussion and conclusion). However, it worth emphasizing that we show a particular summer month (of the year 2008) March (or February as in the revised version of the manuscript) but not a climatological month to see much clearly patterns of the depicted distributions. We make the reason behind this choice clear in the revised text so as not to cause the confusion that the original
version caused.

"We present this particular summer month of a typical year to show much clearly the patterns of the depicted and discussed distribution which could not be obviously seen on seasonal or climatological mean maps"

L. 271: Why "potential existence in colony form"? Does that mean you did not track when and where Phaeocystis was present
in the colonial form in your simulations? I think this information would be a useful output to assess where and when the chosen parametrization leads to colony formation and to assess/discuss/speculate what impact neglecting further dependencies of colony formation (light etc., see above) have on the simulated biogeography.

R: We did not trace it explicitly for the outputs, since the priority was to get *Phaeocystis* and coccolithophores co-existing. However, we agree that this would be interesting and we will do this in future studies.

L. 272-274: I don't see in Figure 5 how the introduction of the high nutrient affinity of coccolithophores causes what you claim here. For that, you would need to show the original biogeography before applying the changes.

R: Indeed, it is seen from the depicted distribution of PFT Chla (e.g. coccolithophores) in line with obtained distribution of nutrients (e.g. phosphate). We write,

"The spatial distribution of silicon, dissolved iron and phosphate is discussed in line with the simulated PFT Chla biogeography. … Thus Figure 6 shows that the simulated abundance of coccolithophores north of the South Subtropical Convergence province (SSTC) – where phosphate occurs in very low concentrations – is explained by the introduced high affinity of this PFT to phosphate (small half-saturation rate in the $\gamma_\eta$ function) allowing coccolithophores to grow in nutrient depleted conditions. … "

L. 274: Replace "depleting" by "depleted".

R: Replaced.

L. 275: Where is the Subtropical Front in the plot? The STF is not introduced and the caption of Fig. 5 does not include a
definition of the white contours either. Please include this information somewhere.

R: We now show the position of the Subtropical Front in Figure 1.

L. 275-277: Similar to comment on L. 272, I don't see how Fig. 5 shows this. Again, one would need the plot before the change – otherwise I don't understand how it is possible for the reader to see this. Please clarify.

R: Please see our response to the comments on L. 272-274, here we continue

"…However, between the Subtropical and Subantarctic Fronts, where there are still nutrients (other than phosphate) to support the growth of small diatoms and *Phaeocystis,* the occurrence of coccolithophores is more evidently linked to low grazing pressure on this PFT due to its much lower palatibility for zooplankton in comparison with small diatoms or *Phaeocystis*
presented by single solitary cells."

L. 277-278: Please rephrase "the simulated coccolithophores". This sentence does currently not make a lot of sense.

R: Thank you. Rephrased as "As in the study by Smith et al. 2017 reported on observed coccolithophores biogeography in the Great Calcite Belt, our simulated coccolithophore Chla is distributed in the silica-depleted area, where small diatom cells, if
could still compete for other nutrients, have higher palatability for grazers."

What do you conclude from the fact that you find highest coccolithophore biomass levels (I assume that is what you mean here) where/when silicic acid is depleted? Please discuss shortly what this implies for the competition with diatoms.

R: Thank you for the comment, indeed, it implies for the co-existence/competition with diatoms presented in small cells, since,
even if small diatom could compete on phosphate, nutrients or iron, it can be easier grazed than coccolithophores. Coccolithophores do not compete with small diatoms on silica resources and might survive due to its lower palatability factor.

Moreover, in this area silica limited diatoms can grow slower allowing coccolithophores for early access to other not used by diatoms macronutrients and iron (Dutkiewicz et al. 2019). We edited the text as written above (L412-416).

L. 279-280: Please revise the grammar of this sentence.

R: Revised as following "Figure 7 illustrates the implication of the differences among the haptophytes on the carbon cycle as carbon contributes differently into inorganic and organic, particulate and, consequently, dissolved pools."

L. 281: Again, why March 2004?

We can show climatological March or February (summer month) or March/February 2008 (summer month of a typical year, as discussed above). See figures R2.13 and R2.14. It does not change the point. By showing a particular month we just benefit with respect to finely shown and resolved patterns of the distribution.

[Figure]

**Figure R2.13: PHAEO surface distribution of the PIC, POC and PIC:POC ratio for March 2008**

[Figure]

**Figure R2.14: REF surface distribution of the POC and PIC:POC ratio for March 2008**

L. 279-284: This whole paragraph is too superficial and lacks the build-up from the introduction and method section, as the impact of different phytoplankton types on POC production/availability in not thoroughly introduced.

R: The purpose of presenting the figures was to emphasize the importance of distinguishing among haptophytes when providing any carbon stock estimates. We did not introduce POC equations/parameterisations since for the full set of Darwin equations we refer(red) the reader to the study by Dutkiewicz et al. (2015).

Additionally, you nowhere state what assumptions you make in DARWIN regarding the routing of biomass losses to POC by the different PFTs. What do you assume for coccolithophores, diatoms, and Phaeocystis?

R: As in the study by Dutkiewicz et al. (2015, their equation A12), POC is a prognostic variable presenting integrated total particulate dead organic carbon (that makes it different from what is measured).

The model parameterizes a fraction of dead cells, and non-assimilated grazed cells to either a DOC or a detrital (i.e. POC) pool. This fraction (0.5) is the same for large diatom, coccolithophores and *Phaecystis*. For *Prochlorococcus*-like prokaryotic pico-phytoplankton and Nfixer the fraction is 0.2.

Why are the POC concentrations south of the SACCF higher in the PHAEO simulation? I suggest to relate this back to changes in phytoplankton community structure and assumptions in the model, so that the reader can take something away from your statement.

R: Experiment PHAEO reveal different distribution and composition of PFTs (higher biomass of diatoms, for instance), which results in higher POC concentrations (see Figure 4, original version).

Are you showing POC resulting from haptophytes only or from all phytoplankton? You state that you're looking at the impact of haptophytes, but possibly, you're showing all phytoplankton. Please double-check and clarify.

R: We show total particular dead organic carbon. We clarify it in the text.

Similarly, for PIC, you nowhere state in the method section how calcification by coccolithophores is described in the model. Please add this information.

R: As in Dutkiewicz et al. (2015), PIC is produced by coccolithophores (no other type produces PIC) in accordance to their equation A15 given the PIC dissolution rate, PIC sinking rate and ratio of inorganic carbon of organic phosphate specified as $0.0033(day^{-1})$, $10\ (mday^{-1})$ and $0.8\ (mmolC/mmolP)$.

We did not introduce this parameterization in the method, since did not opt to focus too much on the PIC/POC but rather on PFTs and provide an (obvious probably) example how the PFT distribution and composition alter the particulate carbon pool.

The cited papers by Balch et al. do not comment on POC concentrations, as far as I could see. Please double-check.

R: Balch et al. (2005) presented PIC, Balch et al. (2016) looked at PIC/POC.

L. 288-291: Similar to above, how do you define the "much better agreement" or "even larger agreement"? Try to be quantitative whenever possible.

R: we apologies for not being quantitative enough. We now tried to reduce the number of subjective terms used. However, in this particular case, only qualitative assessment (in terms of distribution) was performed, partly because of the mismatch in definition of POC in the model and observations.

We changed the text as following:

"Figure 7 illustrates the importance of distinguishing among haptophytes on the carbon cycling as carbon distributed into different inorganic and organic, particulate and, consequently, dissolved pools. Shown are the particulate inorganic carbon (PIC, panel a) produced by coccolithophores (see Dutkiewicz et al. 2015, their eq. A15) and ratio of PIC to total particulate dead organic matter (POC, Dutkiewicz et al. 2015, their eq. A12), PIC:POC (panel b), for the experiments PHAEO in February 2008. Due to the improved representation of the coccolithophores and, therefore PIC (see Balch et al. 2005) in the experiment PHAEO, the depicted PHAEO PIC:POC ratio (opposed to those in REF, Figure 7c) clearly indicates that north of the SAF the value can be from 0.4 up to 1 (on the Patagonian Shelf) which is comparable with PIC:POC export ratio presented in Balch et al. (2016), even though there is a mismatch in how POC is presented in the model and how it is measured. As in the study by

Balch et al. (2016) the PIC:POC ratio is lower than 0.05 south of the Polar front."

Additionally, in Fig 2 you only show July & January for PHYSAT and the "old" model version, here you make a statement for the months June-August and December-February. Please show all months for PHYSAT and the "old" model version somewhere.

R: Here we show all months for PHYSAT and "old" model version (Figure R2.6 and Figure R2.7). The figures have been added to the Supplementary Material (Figure S15 and S16).

And again, I don't understand why you decide on these months now, when before you focused on March 2004. This is very confusing for the reader.

R: We just wanted to provide more information about PFT dominance obtained for experiment PHAEO that is evaluated in more details. It is why we showed December, January and February. That was actually what the reviewer expressed in reviewer's comment on 208 – 209: "Personally, I would have preferred to see the agreement for all summer months (December-February or even March) to additionally get a better feeling for how the model is doing in terms of seasonality". The results could be compared with the monthly PHYSAT dominance climatology published in (Alvain et al. 2008).

We apologies once again for the confusion. We have remedied this by showing now PHAEO monthly mean climatology (over 2006 - 2012) for December – January – February to compare with the revised version of figure 2. All climatological monthly mean PFT dominance obtained in PHAEO are shown in the supplementary material (Figure S18)

L. 293: "of monthly means"

R: Corrected

L. 293-298: Why do you reduce the plot to the Atlantic and Indian sector based on Smith et al. (2017)? Why 2012 now? You don't actually show any data from their study so it is not clear to me why you reduce the area shown in the Figure and why you chose a different year all of a sudden.

R: The observations shown and discussed in Smith et al. (2017) were collected in the year 2011 (January - February) and 2012 (February - March). We do not show any data from Smith et al (2017). However, our results could be compared against their data as well as to hyperspectral satellite coccolithophore retrievals shown in Losa et al. (2018, for the same domain as in Smith et al. 2017). It is why we reduce the area and chose 2012 (could be also 2011). We clarify it in the revised manuscript.

L. 296: Where is the "smaller belt"? Be precise in your description. What is the latitudinal extent in the model output and the satellite product?

R: We write:

"Figure 9 presents the monthly mean spatial distribution of simulated surface Chla for coccolithophores and diatoms over the region from 30°S to 70°S and from 70°W to 120°E as shown in the study by Smith et al. (2017). These model results are compared with Chla obtained for the same domain and time with SynSenPFT algorithm (Losa et al., 2017). The simulated coccolithophore distribution reveals the calcite belt around 35°S to <50°S, which in comparison with SynSenPFT is well agreeing considering the northern boundary. The results are supported by the PhytoDOAS PFT retrievals from hyper-spectral information presented in the study by Losa et al. (2018, https://oceanopticsconference.org/extended/Losa_Svetlana.pdf) for the related region and time frame. But opposed to these satellite products the predicted calcite belt is not extending further south of the Polar Front."

L. 298-306: This is a very nice discussion, but please link it back more explicitly to the "smaller belt" to make the take away message clearer.

R: We now continue:

"In this respect, it is worth emphasizing that SynSenPFT product at the latitudes higher than 60°S is mostly influenced by OC-PFT estimates because of much less available SCIAMACHY information (see Supplementary Material, Section S2) and the OC-PFT retrievals (Losa et al., 2017) contain information generally on haptophytes (not specifically on coccolithophores). Moreover, PhytoDOAS coccolithophore retrievals are based on coccolithophore specific absorption spectrum that is, indeed, very similar to the specific absorption spectrum of *Phaeocystis*. Model simulations, as seen from Figures 4 and 6, support the evidence of *Phaeocystis* dominance among haptophytes at these latitudes. Thus, SynSenPFT more likely overestimates coccolithophore Chla at the latitudes higher than 60˚S".

(The result is, somehow, in line with independent study by Holligan et al. (2010) concluded that current satellite algorithms may significantly overestimate PIC in cold waters of the Southern Ocean.)

Holligan, P.M. Charalampopoulou, A., Hutson, R.: Seasonal distributions of the coccolithophore, *Emiliania huxleyi*, and of particulate inorganic carbon in surface waters of the Scotia Sea, *Journal of Marine Systems*, 82 (4), 195 – 205, doi: 10.1016/j.jmarsys.2010.05.007, 2010.

Same is true for the discussion of the diatom distributions.

R: We edited the text as following:

"For diatoms, modeled Chla exceeds SynSenPFT estimates south of the Antarctic Circumpolar Current Front. However, SynSenPFT diatom Chla is known to be underestimated for the Antarctic Province (see Losa et al. 2017). At the same time, diatom Chla estimates obtained with PhytoDOAS are higher (see Supplementary Material, Section S2) despite the low coverage of the product, which can indicate that predicted diatom Chla could be a bit less overestimated than it is suggested by comparison with SynSenPFT."

L. 310: How were the days of the snapshots chosen?

R: Two-weekly snapshots over the period of time from August 2002 to April 2012.

L. 315: What is "less accurate" in this case? Please be precise.

R: We clarify: "our simulated Chla for Phaeocystis as haptophytes in Ross Sea are underrepresented in comparison with HPLC-derived estimates."

"However, the comparison of *Phaeocystis* biomass to the MAREDAT dataset (Vogt et al., 2012) revealed quite a good agreement (see Subsection 3.3.3)": coefficient correlation r = 0.54, MAE = -0.61 for log-transformed biomass (see Figure S13, in the Supplementary Material)

L. 318: Why "see Vogt et al. (2012)"? This citation here is not obvious to me. Can you clarify for me?

R: We edited the text as mentioned above and clarify in subsection 3.3.3.

L: 324: Does Fig. S9 only include model output that was collocated with the observations? Please clarify in the text and/or the Figure caption.

R: It is clarified now that Figure S9 shows collocated with the observations model matchups.

L. 331-332: Is a systematic overestimating by 0.5 mg chl m-3 really that bad in your view? That's what the writing currently makes it sound like to me.

R: What we wanted to point out that this level of systematic model-from-observation deviation was the highest in comparison with bias estimates obtained for other biogeochemical provinces.

We make this clearer:

"The highest disagreement was obtained for diatoms in the Atlantic Sector of the ANTA province, where the simulated diatom Chla is systematically overestimated by ~0.5 mg m$^{-3}$. The best agreement with the HPLC based diatom Chla (excluding small provinces, see Figure 1) was obtained at the SSTC and SANT. For the haptophytes, the highest systematic error towards overestimation has been found at two small provinces east of Africa and Australia (EAFR and AUSE) with the bias = 0.57,

0.48 (mg m$^{-3}$), respectively. The highest random error is (RMSE = 0.62, 0.44 mg m$^{-3}$) at EAFR and APLR. The lowest differences between predicted and observed haptophytes was at the FKLD, SSTC provinces where haptophytes are mostly presented by coccolithophores, and at the SANT biogeochemical province, where both coccolithophores and *Phaeocystis* co-exist…"

L. 334: Differ in what way? This is a vague statement.

R: Rephrased: "…frequency distributions of the simulated and observed prokaryotic pico-plankton are different…"

L. 285-340: Personally, I would suggest to present the validation earlier in the manuscript. I find it a bit unfortunate to have the evaluation as the last result section.

R: The story we wanted initially to tell the reader is how leveraging satellite estimates and in situ observations allowed us to define the trait requirements for capturing phytoplankton biogeography in the Southern Ocean. And then evaluated the model set up with the specified traits allowing also to test the hypothesis … We now lay this out in the introduction so that this process is clearer.

**Conclusions**

L. 342-343: I don't understand the first sentence. How did satellite-derived estimates and in situ observations help to define trait requirements (characteristics? Or simply traits?) of phytoplankton? Can you rephrase?

R: We clarify that "an extensive synthesis of available observations on the Southern Ocean PFTs allowed us to better understand their biogeography. This information was used to infer which types should coexist in which regions, and, therefore, to constrain the model." In other words, this gave us a basis on which to define which traits would allow these regional co-existences.

L. 347-348: The necessity of the inclusion of two diatom classes and the changes to the coccolithophore parametrization have not been sufficiently motivated and the subsequent improvement of the model has not been sufficiently demonstrated, please see comments above (e.g. on L. 220-223 and on L. 275-277 of your manuscript).

R: Please see above our response to the aforementioned comments. Nevertheless, we only change parameter values for coccolihophers (as listed in Table 1), not the underlying parameterization.

This part of the conclusions has been revised as following:

"Our results support the hypothesis that introducing two size classes of diatoms in biogeochemical models is a prerequisite to simulate the observed diatom phenology and PFT distribution in general. We have also shown that the simulated biogeography of coccolithophores is not controlled by temperature itself as reported by Smith et al. (2017), since we did not use a specific for coccolithophores temperature limitation function. It was directly explained by phosphate depleting as well as by low palatability of this PFT for grazers. This confirms our second hypothesis. Nevertheless, we found that the simulation of co-occurrence of coccolithophores and *Phaeocystis* required additional model developments to account for changes in assumed life stage of Phaeocystis (Popova et al., 2007; Kaufman et al., 2017) subject to iron availability (Bender et al., 2018). This parameterization of morphological shifts indeed allows for co-existence of the two types of haptophytes corroborating our third hypothesis on the dependence of *Phaeocystis sp.* life stages on iron availability. By considering two life stages of *Phaeocystis* we introduce additional differences in the traits, which along with assumed physiological parameters for coccolithophores makes coccolithophores competitive among phytoplankton of larger cell size requiring higher nutrients concentration to grow or/and among PFTs of similar size – small diatoms and *Phaeocystis* solitary cells – but of higher palatability factor to be grazed. These additional differences in the traits of distinct haptophytes, coccolithophores and

Phaeocystis allows these groups to co-exist (e.g. along the Subantarctic and Polar Fronts). However, there is still room for improvement,…"

Furthermore, I don't understand the logic in the sentence in parentheses. Please rephrase to clarify.

R: The sentence in parentheses has been removed.

L. 349: That temperature is not a driver of the coccolithophore biogeography in your model has not been shown/discussed in your result section. Please include it there or adjust the conclusion section.

R: In a way we refer to the fact that we do not use a specific for coccolithophores temperature limitation function.

L. 350: Please revise the grammar of this sentence ("Neither[...]").

R: Revised

L. 350-355: Again, please double-check carefully what in your conclusion section are results that you've actually presented in this manuscript and what are speculations or work not included here. Currently, a lot of the things you say here do strictly not follow from what you've shown.

R: We wrote that "The simulation of co-occurrence of coccolithophores and *Phaeocystis* required additional model developments. Thus, as a first trial, the Darwin model was augmented to account for changes in assumed life stage of *Phaeocystis ant.* (Popova et al., 2007; Kaufman et al., 2017) subject to iron availability (Bender et al., 2018). This parameterization of morphological shifts did indeed allow for co-existence of the two types of haptophytes."

That is what we have done and shown.

Additionally, including life stages of Phaeocystis allowed for co-existence of the two types where and/or when?

R: in the revised version we have extended the sentence mentioned above as following: "(e.g. along the Subantarctic and Polar fronts)."

Going back to L. 234-236, I think you're referring to the fact that one goes extinct when not accounting for these. I still think this is worrisome and I do not understand at all how the changes to the model then prevent this from happening.

R: As we explain above: we are dealing with model behavior in the presence of the biochemical module structure including two similar (with respect to the assumed traits) PFTs leading to that just one of these two exists. In experiment REF we did not differentiate sufficiently enough between the traits assumed for coccolithophores and "other large" (or *Phaeocystis*), which lead to a longer integration period of time to reach a quasi-steady state and to just one of "similar" PFTs survived. It means that in experiment REF *Phaeocystis*-analogue indeed represents haptophytes in general (taking over for coccolithophores). In the original Darwin-2015 model (Dutkiewicz et al. 2015) "other large" did not survive. Thus, the introduction of additional diversity/differences in the PFT traits is crucial.

It helps to make coccolithophores competitive among phytoplankton of larger cell size (or colonies) requiring higher nutrients concentration to grow or/and among PFTs of similar size – small diatoms and *Phaeocystis* solitary cells – but of higher palatability factor to be grazed. We make this clearer in the conclusion of the revised text.

L. 355; Please check the grammar.

R: Thanks. Checked and corrected

L. 359-362: Is this really the case? I would expect the nutrient limitation terms to have a big influence on differences between PFTs as well, given the differences in their half-saturation constants (Table 1). Please double-check.

R: We agree that the nutrient limitation terms also affect differences between PFTs. The discussed was (should be) eq. 3 and 5 in particular (equation number is corrected in the revised version, it is now 1 and 3) with respect to photophysiological parameters $P_{max}^c$ and $\alpha$.

The "realized" growth rate (specific growth rate) is a result of all environmental factors it depends on and the non-linearity of the functions might lead to unexpected results with regards to their impact on the specific growth rate at a given point and time.

R: We agree, what we wanted to emphasize is the importance of careful specification of $\alpha$.

L. 362: Please include this information on assumptions surrounding alphaPI in the method section and in Table 1.

R: We do not prescribe alpha by a number explicitly, it is calculated in eq. 5 (see also Dutkiewicz et al. 2015). We now add this reference.

L. 367: If your maximum growth rates are likely too high, this should be discussed/mentioned somewhere in the manuscript. Can you plot how your temperature-limited growth rate in the model for Phaeocystis, diatoms, and coccolithophores relates to laboratory measurements (see e.g. supplementary material in Le Quéré et al. (2016) for a compilation)?

R: We did not use the approach used in Le Quéré et al. (2016) for deriving a temperature limiting functions and allowing to suppress the PFT distribution to particular latitudes. All phytoplankton types have the same temperature function (unitless) as done in Dutkiewicz et al. (2015).

The Tables S5 & S6 do currently not include information on what temperature the reported growth rates are measured at (and you don't specify the temperature dependence used in your model).

R: We included in the tables information extracted from the literature (including growth rate at specific T if available). In our model as in Darwin-2015 (Dutkiewicz et al. 2015), we use the following formulation:

$$\gamma_j^T = \tau_T e^{\left( A_T \left( \frac{1}{T+273.15} - \frac{1}{T_O} \right) \right)},$$

given the coefficient $\tau_T = 0.8$ normalized the maximum value (unitless), the temperature coefficient $A_T = -4000\,K$, and the optimal temperature $T_o = 293.15\,K$.

Plotting the function that is actually used in the model over a range of temperatures together with a range of measurements will help to understand in what temperature ranges the temperature-limited growth rates in your model is too high/too low.

R: In this respect we would like to state the following: 1) we started from the parameter values used in the original study by Dutkievicz et al. (2015) changed them not too much, mostly to account for relative differences reported from the lab experiments (Tables S5, S6); 2) and there is still importance of alpha (Losa et al. 2004, 2006); 3) as mentioned above we did not use the approach used in Le Quéré et al. (2016) for deriving PFT specific temperature limiting functions and allowing to suppress the PFT distribution to particular latitudes; 4) in equation 4, there is another function $\gamma_\eta = f(k_{sat_\eta}, \eta)$, in our case $k_{sat_\eta}$ depends on the assumed phytoplankton size (Ward et al. 2012, 2017).

L. 370: And is it a problem if one had to choose different alphaPI for different PFTs?

R: It is not a problem, the alpha should be different (Losa et al. 2004, 2006). The discussed is eq. (5) – (6), used to calculate the alpha (Dutkiewicz et al. 2015) given the Phytoplankton specific absorption spectra and $\phi_{max}$.

L. 371-373: Again, this has not actually been shown in your study. Can you back this up with some references?

R: It is a result of one of the sensitivity experiments included in the Supplementary Material (Figure S4, Section S1.1.4)

Try to make your language more accurate by including words like "potentially", "possibly", then you would immediately avoid misunderstandings regarding where you speculate and where you refer to things you have actually shown.

R: Thank you for pointing this out. In the revised manuscript we are careful to use more accurate words for things we have actually shown, and these less specific ones for speculation.

L. 379: But you have included CDOM in the model simulations you discuss here, haven't you (see Equation 2)? Then I don't understand what you mean here exactly, as you're talking about possible improvements. Please be precise. What would need to be improved and how?

R: We pointed out on results of our sensitivity test with respect to including/excluding CDOM in/from the model. Results of this experiment indicate that the model is sensitive to parameterisations of remineralization processes. We state this as a study limitation.

L. 380: Similar to above: Please be precise on what you think should be improved regarding the algae-sea ice interactions and how you think this would impact the study at hand.

R: We raise this issue for the discussion because current set up of the light penetration module (as in Taylor et al. 2013) equation (12) and (13) of the original version of the manuscript leads to diatom blooming if the sea ice concentration is less 90%. With no sea-ice-algae specified in the model it might lead to overestimation of the simulated diatom Chla in the marginal ice zone. At least we would like to mention this issue as a limitation of the study.

Please try to always relate your suggested improvement back to this study –there is possibly an endless list of things one could improve in your model (and in any other model for that matter), but not all of those things are relevant for modeling PFTs on a basin scale in the SO. Please make very clear why you think the things you suggest to improve are important and how you think they would impact the study at hand.

R: All discussed aspects for improvement were proposed based only on the experience/sensitivity tests with the model that were carried out but not yet shown in the manuscript since required more thorough evaluation and presentations. Nevertheless, a sensitivity experiment with changed $\phi$ leading to changing $\alpha$ (different for diatom and *Phaeocystis*) showed that it is possible to further improve representation of diatom and *Phaeocystis* in the Ross Sea. Experiments with excluding or including CDOM

effected PFT compositions (via altering the remineralisation processes and, therefore, nutrient distributions).

L. 384-385: Please delete the statement about green algae and dinoflagellates, as this is not relevant here.

R:  Deleted as suggested.

L. 386: The information becomes closer? To what? Please revise the logic.

R: we deleted this expression.

L. 382-403: In my opinion, this whole paragraph is misplaced in the conclusion section. Overall, I think the conclusion section is way too long right now. I would instead suggest to include and "limitations & caveats" section between the results and the conclusions. In such as a section, you can then discuss the difficulties described here, as well as the limitations surrounding the PFT parametrizations and the suggested improvements (L. 355-381).

R: As suggested, we have introduced a new subsection: "Limitations of the study"

Please focus the conclusion section on the main take away messages from your paper.

R: Thank you, we have done this now in the revised version.

Figures/Tables Table 1: As mentioned in the detailed comments above, the table is currently incomplete. Please add the missing variables (even if they are the same for the different PFTs, it is important to state that here for important variables such as alphaPI and the maximum grazing rate).

R: In the original version we did not include the alphaPI in Table 1 as a parameter since it was considered as a function calculated with eq. 3, and maximum grazing rate we kept unchanged as in Dutkiewicz et al. (2015): for large zooplankton - $gmax_{jk} = \{1; 0.1\}$ on large and small phytoplankton respectively; for small zooplankton - $gmax_{jk} = \{0.1; 1\}$ on large and small phytoplankton respectively. We added the reference in the text (L139, L148, L153).

Furthermore, please add the units and a short description of each variable to the table.

R: Added.

What temperature is the maximum growth rate at? This needs to be specified.

R: at $30°$, it is now specified in table 1.

I am also irritated by the three digits of the half-saturation constants of e.g. N –is the model that sensitive to changes in this number? Have you tested this?

R: These parameters we calculated using empirical allometric relationships (Ward et al. 2012. 2017, Dutkiewicz et al, BGD, in review). It is why the values are so precise. We have not performed a sensitivity test with respect to the precision required.

Dutkiewicz, S., Cermeno, P., Jahn, O., Follows, M. J., Hickman, A. E., Taniguchi, D. A. A., and Ward, B. A.: Dimensions of Marine Phytoplankton Diversity, Biogeosciences Discuss., https://doi.org/10.5194/bg-2019-311, in review, 2019.

Please also add the half-saturation constant of silicic acid by diatoms.

R: We added the half-saturation constant of silicic acid in table 1.

I am also slightly confused by your half-saturation constants of iron. Assuming your reported numbers are in mmol m-3, your value for large diatoms (0.028) is e.g. an order of magnitude smaller than those suggested for the SO in Timmermans et al. (2004; 0.19-1.14 nmol L-1 or 0.19*10-3-6701.14*10-3mmol m-3). Regarding the N2fixers, Trichodesmium is typically considered to have a higher iron requirement and half-saturation constant of iron than other phytoplankton PFTs (I suggest you to have a look in e.g. Berman-Frank et al. (2001) and Ward et al. (2013) and check references therein, as I am not an expert myself in nitrogen fixers). Have you tested how your low kFef or the N2fixers in your model impacts their relative contribution to the SO phytoplankton community, which is currently quite high (see Fig. 4)?

R: In our study half-saturation constant of iron for N-fixers is higher than for all other PFTs (except for large diatoms).

In the version of the model in Ward et al. (2013), Monod kinetic are used. As such growth is given as $\mu_{max} * \frac{R}{R-K_R}$, where $R$ is nutrient, and $K_R$ is the half-saturation for growth. In general, most lab/field observations are given in terms of nutrient uptake (not growth):

$v_{max} * \frac{R}{R-k_R}$, where $k_R$ is half saturation for nutrient uptake.

$K_R$ and $k_R$ are related but not the same. In fact, $K_R$ is often about an order of magnitude lower than $k_R$. (See discussions in

Ward et al, JPR, 2014; Verdy et al, L&O, 2009; Dutkiewicz et al, BGD, in review). In our study as well as in Dutkiewicz et al. 2015, the growth half saturations were parameterized based on a nominal size for each phytoplankton and using empirical allometeric relationships for $k_R$ and the other relevant parameters needed to calculate $K_R$. These are more fully explained in Dutkiewicz et al (BGD, in review).

Table 2: What do "PSC" and "SCM" stand for?

R: These stand to specify size class as referred in the observations (Phytoplankton Size Class, PSC) and Darwin-2015 model (Size Class in Model, SCM), explained in the caption.

**All Figures& Tables:**

Please add panel labels (Figures) and use these in the captions and the text.

R: The panel labels have been added

Please double-check that you clearly state in the captions, which year of which simulation you're assessing and what year (Tables 3-5)

R: Checked.

Please make sure you include units for all variables in the captions.

R: Checked. Included

Fig. 1: You only plot HPLC observations. Please be precise in caption.

R: We modified the caption.

Replace "curve" by "contour".

R: Replaced

**Fig. 2**

When you say "Haptophytes" here, do you mean coccolithophores? Or the combination of coccolithophores and Phaeocystis?

I am confused because both PHYSAT and the Darwin model do discriminate between the two and you list Phaeocystisas a separate class in the Figure legend. Please clarify in the Figure legend as well as in the result section 3.1.

R: "Haptophytes" is a sum of coccolithophores and *Phaeocystis*. We clarify it in the figure caption.

Why do you combine the two for the model output?

R: Experiment REF as well as Darwin-2015 presented in Figure 2 could not distinguish between coccolithophores and *Phaeocystis*. We emphasize this in section 3.1.

"The our model output" -> please rephrase.

R: Thank you, rephrased.

What is the basis of choosing 55% as the dominance threshold? This seems random to me.

R: Indeed, it was just our choice.

I am surprised to see that there is no area of coexistence, so this means at every grid cell there is always one PFT that contributes more than 55% to biomass?

R: The area of co-existence is depicted in the green colour (mixed).

If I look at the transects in Fig. 4, it does not necessarily look like it. Please double-check.

R: In the transect, small and large diatoms, coccolithophores and *Phaeocystis* are shown separately. In the dominance plot, they are combined in diatoms and haptophytes, respectively.

Fig. 3

What are the white contours? Please add info to caption.

R: We have added that the white contours represent the Southern Ocean fronts (Orsi at al., 2005, as in Figure 1):

the Sub-Antarctic Front (SAF, thick contour); the Polar Front (PF, dashed), the Southern Antarctic Circumpolar Current Front (SACCF, thin contour) and the Southern Boundary of ACC (SBDY, dotted).

Fig. 4

The caption is incomplete: explain Nfix, Proc, UML... I also suggest to add "REF" and "PHAEO" directly in the Figure to make clearer immediately what the two columns are.

R: The caption is complete now, explaining 'UML' for the upper mixed layer, 'Nfix' for nitrogen fixers, 'Proc' for

Prochlorococcus. We have added "REF" and "PHAEO" directly in the Figure.

**Fig. 5:**

panes ->panel.

R: Corrected.

Please change the order of "phosphate" and "iron" in caption to match the order with that in the figure.

R: Changed, thank you for pointing this out (now figure 6).

What are the white contours?

R: We have added that the white contours represent the Southern Ocean fronts (Orsi at al., 2005, as in Figure 1).

**Fig. 6:**

I suggest to add "REF" and "PHAEO" directly in the Figure to make clearer immediately what the two columns are.

R: Added as suggested.

**Fig. 7:**

Please correct "the our model output".

R: Corrected. Now figure 8 in the revised version of the manuscript.

Please add a reference to Fig. 2.

R: Added.

"Figure 8. Surface PFT dominance simulated with Darwin-MITgcm for 2007/2008 for experiment PHAEO (see figure 2 for comparison). Model PFT is considered dominant if its Chla fraction of total Chla is more than 55%. The model output is masked by the area with sea ice concentration > 75% during respective month."

**Fig. 9:**

Is "diatoms" large + small here? Be more precise.

R: Figure 9 is replaced by a reference to a supplemented video. The description is provided with link to the supplementary videos.

https://doi.org/10.5446/42871

Add "in situ HPLC observations".

R: Added when providing the link to the supplementary videos.

Which simulation?

R: "PHAEO", it is stated in the section title and when providing the link to the supplementary videos.

**Fig. 10 & 11**

Please say explicitly "coccolithophores and Phaeocystis" (Fig 10).

R: Done with the link provided to video, since, as for diatoms, these figures were removed in the revised version of the manuscript.

Link for haptophytes (coccolithophores and Phaeocystis):
https://av.tib.eu/media/42873

Link for haptophytes prokaryotes:
https://av.tib.eu/media/42872

Please state which simulation is shown.

R: The shown is PHAEO. The figures are replaced by a reference by a video.

**References**

Berman-Frank, I., Cullen, J. T., Shaked, Y., Sherrell, R. M., & Falkowski, P. G.: Iron availability, cellular iron quotas, and
nitrogen fixation in Trichodesmium. Limnology and Oceanography, 46(6), 1249–1260, https://doi.org/10.4319/lo.2001.46.6.1249720, 2001.

Breitbarth, E., Oschlies, A., & LaRoche, J.: Physiological constraints on the global distribution of Trichodesmium– effect of temperature on diazotrophy. Biogeosciences, 4(1), 53–61, https://doi.org/10.5194/bg-4-53-2007, 2007

Krumhardt, K. M., Lovenduski, N. S., Long, M. C., Levy, M., Lindsay, K., Moore, J. K., & Nissen, C.: Coccolithophore
Growth and Calcification in an Acidified Ocean: Insights From Community Earth System Model Simulations. Journal of Advances in Modeling Earth Systems, 11, 2018MS001483, https://doi.org/10.1029/2018MS001483, 2019.

Lancelot, C., de Montety, A., Goosse, H., Becquevort, S., Schoemann, V., Pasquer, B., & Vancoppenolle, M.: Spatial distribution of the iron supply to phytoplankton in the Southern Ocean: a model study. Biogeosciences, 6(12), 2861–2878, https://doi.org/10.5194/bg-6-2861-2009730, 2009

Le Quéré, C., Buitenhuis, E. T., Moriarty, R., Alvain, S., Aumont, O., Bopp, L., ... Vallina, S. M.: Role of zooplankton dynamics for Southern Ocean phytoplankton biomass and global biogeochemical cycles. Biogeosciences, 13(14), 4111–4133, https://doi.org/10.5194/bg-13-4111-2016, 2016.

Luo, Y.-W., Doney, S. C., Anderson, L. A., Benavides, M., Berman-Frank, I., Bode, A., ... Zehr, J. P.: Database of diazotrophs in global ocean: abundance, biomass and nitrogen fixation rates. Earth System Science Data, 4(1), 47–73,
https://doi.org/10.5194/essd-4-47-2012, 2012

Monteiro, F. M., Bach, L. T., Brownlee, C., Bown, P., Rickaby, R. E. M., Poulton, A. J., ... Ridgwell, A.: Why marine phytoplankton calcify. Science Advances, 2(7), e1501822, https://doi.org/10.1126/sciadv.1501822, 2016.

Peperzak, L.: Daily irradiance governs growth rate and colony formation of Phaeocystis (Prymnesiophyceae). Journal of Plankton Research, 15(7), 809–821, https://doi.org/10.1093/plankt/15.7, 1993

Timmermans, K. R., Wagt, B. Van Der, & de Baar, H. J. W.: Growth rates, half saturation constants, and silicate, nitrate, and phosphate depletion in relation to iron availability of four large open-ocean diatoms from the Southern Ocean. Limnology and Oceanography, 49(6), 2141–2151, https://doi.org/10.4319/lo.2004.49.6.2141745, 2004

Wang, S., & Moore, J. K.: Incorporating Phaeocystis into a Southern Ocean ecosystem model. Journal of Geophysical Research, 116(C1), C01019, https://doi.org/10.1029/2009JC005817, 2011

Ward, B. A., Dutkiewicz, S., Moore, C. M., & Follows, M. J.: Iron, phosphorus, and nitrogen supply ratios define the biogeography of nitrogen fixation. Limnology and Oceanography, 58(6), 2059–2075, https://doi.org/10.4319/lo.2013.58.6.2059, 2014.

---

## Author Comment (AC3) · 2 Jan 2020

**Response to the comments of reviewer 1**

**Anonymous Referee #1**

In this paper the authors use the Darwin-MITgcm to simulate the phytoplankton composition in the Southern Ocean. The paper is focused on the parametrization of the model to improve coccolithophore abundance, include two sizes of diatoms and two life stages of Phaeocystis in the Southern Ocean. The paper is an interesting model development, but I am not sure whether this really fits in the goals of Biogeosciences.

R: We thank the reviewer for the constructive comments on the manuscript. Our author's replies are presented in blue, labelled "R:" and follow each reviewer's comment.

We have revised the paper in response to the insightful comments of all three reviewers. The revised version is clearer why this is a novel paper and fits the goals of Biogeosciences: it provides unique synthesis (of in situ measurements and satellite observations) on the Southern Ocean phytoplankton communities, and addresses some of the challenges in modelling them.

INTRO: The introduction doesn't provide any context or current challenges of why the work is being done.

R: The introduction has been improved substantially. We have further considered in the introduction the challenges that the model community is facing despite recent advances and that this has motivated us to adjust a coupled model to improve representation of the dominating phytoplankton groups in the Southern Ocean.

"The Southern Ocean is one of the most important regions in regulating climate via the uptake of about 40% of the global oceanic anthropogenic $CO_2$ (DeVries, 2014) and at the same time, is a region with the dynamics evidently altered by past and present climate change (Stocker et al., 2013). The climatic changes in the Southern Ocean environmental conditions affect the spatial distribution of phytoplankton (Deppeler and Davidson, 2017). The phenology and dominance of different phytoplankton functional types (PFTs) sustaining the marine food web affect the diversity of higher trophic levels (Edwards and Richardson, 2004). Playing distinct roles in biogeochemical cycling, PFTs may determine how and on which spatial and temporal scales the ocean mediates climate (Wilson et al., 2018).

Major bloom-forming PFTs in the Southern Ocean include the silicifying diatoms, calcifying coccolithophores, and colony-forming Phaeocystis. Diatoms, the major phytoplankton silicifiers and primary producers in the Southern Ocean (Rousseaux and Gregg, 2014), have high efficiency of carbon export through grazing, direct sinking of single cells, and through mass sedimentation events (Le Quéré et al., 2005; Kemp et al., 2006). They form large spring blooms in the open nutrient-rich waters in the proximity of the Antarctic Circumpolar Current and Polar Front (Smetacek et al., 2002; Kemp et al., 2006).

Coccolithophores, the main phytoplanktonic calcifiers in the world ocean, make a major contribution to the total content of particulate inorganic carbon in the oceans (Ackleson et al., 1988; Milliman, 1993; Rost and Riebesell, 2004; Monteiro et al., 2016) through production and release of calcium carbonate plates (coccoliths), and, therefore, also impact the alkalinity of the ocean. This PFT is abundant along the Great Calcite Belt (Balch et al., 2016) and forms massive blooms along the Patagonian shelf break (Signorini et al., 2006). Phaeocystis as a dimethyl sulfide producer alters the atmospheric sulfur cycle and can form dense spring blooms in the seasonal ice zone and Antarctic coastal waters as the Ross Sea and Weddell Sea (El-Sayed et al., 1983; Arrigo et al., 1999; DiTullio et al., 2000; Smith et al., 2012), likely supporting export production (Arrigo et al., 2000; DiTullio et al., 2000; Wang and Moore, 2011). Modeling studies reported the contribution of diatoms to the total primary production in the Southern Ocean of ~89% (Rousseaux and Gregg, 2014), coccolithophores of ~7-16.5% (Rousseaux and Gregg, 2014; Nissen et al., 2018) and Phaeocystis of ~13% (P. antarctica) (Wang and Moore, 2011).

Despite the recognized importance of the PFTs, global biogeochemical models struggle to represent the Southern Ocean phytoplankton community accurately. The difficulties primarily originate from uncertain parameters employed in the parametrizations of, e.g., phytoplankton growth and grazing (Anderson, 2005), that define the differences in the phytoplankton traits. On the other hand, the available observational information is still limited in the Southern Ocean to allow to properly constrain the models.

One of the most investigated regions in the Southern Ocean is the Ross Sea, where many in situ observations on diatoms and *Phaeocystis* have been collected and inspired regional coupled ocean-sea ice-ecosystem modeling activities (Arrigo et al., 2003; Worthen and Arrigo, 2003; Kaufman et al., 2017). Several studies that include *Phaeocystis* in the list of simulated PFTs in the frame of global coupled ocean-biogeochemical models have focused on the Southern Ocean (Lancelot et al., 2009; Wang and Moore, 2011; Le Quéré et al., 2016). These studies specified differences in (photo-)physiological parameters between diatoms and *Phaeocystis*, considering *Phaeocystis* in colony form. In a regional study (Popova et al. 2007, Crozet Islands) within the Southern Ocean, *Phaeocystis* was represented by two different life-stages: colonies and solitary cells. This approach was also successfully used by Kaufman et al. (2017) to examine the influence of climatic changes on the Ross Sea phytoplankton.

Nevertheless, an in-depth evaluation of the model simulations of diatoms and *Phaeocystis* with PFT observations either has not been done (e.g. Lancelot et al. 2009) or has been only performed based on a sparse in situ dataset (Wang and Moore, 2011). A more complete evaluation of these PFTs was presented by Le Quéré et al. (2016) by comparing the dominance of the PFTs to satellite-based dominance retrievals, and to a global dataset of in situ-based integrated PFT biomass within upper 200 m of Alvain et al. (2008) and (Buitenhuis et al., 2013), respectively. In general, as compared to the satellite retrievals, the dominance of diatoms and *Phaeocystis* has been overestimated by Le Quéré et al. (2016), while dominance of coccolithophores was underestimated.

Coccolithophore biogeography has recently been investigated globally by Monteiro et al. (2016), Krumhardt et al. (2017) and Krumhardt et al. (2019), and particularly for the Southern Ocean by Nissen et al. (2018). With respect to specific coccolithophore traits, the study by Krumhardt et al. (2017), Monteiro et al. (2016), as well as earlier studies by Paasche, (2001) and Iglesias-Rodríguez et al. (2002), emphasized the high nutrient affinity of the coccolithophores and high grazing protection of this PFT (Monteiro et al., 2016). Nissen et al. (2018) reported on higher grazing pressure on coccolithophores than on diatoms. Krumhardt et al. (2019) used lower grazing pressure on coccolithophores than on diatoms and related the distribution of coccolithophores to a specific temperature function in dependence to its growth rate. However, none of these studies included *Phaeocystis* in their model simulations.

In our study, we improved the representation of key Southern Ocean PFTs, namely diatoms, coccolithophores and *Phaeocystis*, using the Darwin biogeochemical model coupled to the Massachusetts Institute of Technology (MIT) general circulation model (Darwin-MITgcm). In a first step, we modified the Darwin model to account for two distinct size classes of diatoms and for a high affinity for nutrients and an ability to escape grazing control for coccolithophores. Next, the model was extended to include both solitary and colonial forms of *Phaeocystis*. Observational information from in situ and satellite measurements was used to help to define differences in the PFT traits, to constrain the model, as well as to quantitatively evaluate the model performance to overall find a representation of the phytoplankton community in the Southern Ocean that is close to observations. We used the optimized Darwin model to test three hypotheses on the factors controlling the biogeography of Southern Ocean phytoplankton groups:

– Size diversity of the diatoms (Queguiner, 2013; Tréguer et al., 2018) leads to the distribution of small diatoms ("slightly silicified and fast growing") at the lower latitudes and large diatoms ("strongly silicified and slowly growing") at higher latitudes in the Southern Ocean.

– Distribution of coccolithophores in the Great Calcite Belt is not necessarily controlled by temperature (Smith et al., 2017) but determined by the ability of this PFT to escape grazing because of their exoskeleton (Nejstgaard et al., 1997; Huskin et al., 2000; Monteiro et al., 2016), and to grow under nutrient depleted conditions (especially phosphate and iron) (Paasche, 2001; Iglesias-Rodríguez et al., 2002). These characteristics of coccolithophores would make them more competitive among other phytoplankton of larger or similar size, small diatoms and *Phaeocystis*.

– *Phaeocystis* sp. exists in two life stages, solitary cells and colonies, depending on iron availability (Bender et al., 2018). This additional difference in the traits of distinct haptophytes, coccolithophores and *Phaeocystis*, allows them to co-exist.

The paper is organized as follows. Section 2 describes the numerical model set up, experimental design and observations (in situ and satellite retrievals) used for model evaluation, Section 3 presents the results and discussion. Section 4 concludes with summary and outlook."

RESULTS and DISCUSSION: The results and discussion section lacks quantitative assessment of the model.

R: The quantitative assessment of the model comparing simulated PFTs concentration against *in situ* PFTs concentration was provided by three tables in the main manuscript (Table 3 - 5) and three tables (S7-S9) and three figures in the supplementary material. We edited the text to make it more visible now.

"We have obtained matchup statistics for the comparison of our PHAEO model results against the in situ HPLC-based PFT Chla observations by Soppa et al. (2017). The mean absolute deviation (mean absolute error, MAE) of collocated model and in situ PFT-Chla over the considered time frame (August 2002 – April 2012) and the entire Southern Ocean is 0.74 mg m$^{-3}$ and 0.22 mg m$^{-3}$ for diatoms and haptophytes, respectively. Tables 4 and 5 present the statistics of model and in situ PFT-Chla comparison at several Longhurst's biogeochemical provinces (Longhurst 1998, see Figure 1). The highest disagreement was obtained for diatoms in the Atlantic Sector of the ANTA province, where the simulated diatom Chla is systematically overestimated by ~0.5 mg m$^{-3}$. The best agreement with the HPLC based diatom Chla (excluding small provinces, see Figure 1) was obtained at the SSTC and SANT. For the haptophytes, the highest systematic error towards overestimation has been found at two small provinces east of Africa and Australia (EAFR and AUSE) with the bias = 0.57, 0.48 (mg m$-3$), respectively.

The highest random error is (RMSE = 0.62, 0.44 mg m$-3$) at EAFR and APLR. The lowest differences between predicted and observed haptophytes was at the FKLD, SSTC provinces where haptophytes are mostly presented by coccolithophores, and at the SANT biogeochemical province, where they both co-exist. As additional information on the agreement between model and observations, Figures S9 and S10 in the Supplementary Material present frequency distributions of diatoms and haptophytes Chla for the simulations and measurements as well as the frequency distribution of the model and data differences. The latter shows that statistical criteria, such as MAE and root mean squared error (RMSE) give statistical meaningful metrics with respect to "model minus in situ Chla data" and the evaluation does not necessarily require a logarithmic transformation, as it is often done in ocean colour product validation (Brewin et al., 2010; Losa et al., 2017).

With respect to the agreement between model and observed in situ HPLC-based Chla for prokaryotic pico-phytoplankton depicted in Figure S11 (in Supplementary Material) one can conclude that the frequency distributions of the simulated and observed pico-phytoplankton are different, and the frequency distribution of the differences confirms that MAE and RMSE given absolute (Table 5) or logarithmically transformed values can hardly provide satisfactory estimates. Nevertheless, it is worth emphasizing that the largest differences between model and observed in situ pico-phytoplankton are located along the Antarctic Peninsula.

However, it is worth noting that these statistical estimates were obtained based on the model and observation match-ups within ± 1 week. Moreover, the model does not explicitly represent sea-ice algae and, therefore, might work less well in the region around the sea-ice. In this respect, we have to point out that all the statistics are presented for a qualitative assessment of the model rather than for a quantitative estimates of model uncertainties, since the representation error (Janjic et al., 2018) related to the differences in spatial and temporal scales considered and sampled by the model *vs.* observations as well as to the mismatch in grouping phytoplankton (Bracher et al., 2017) are large."

We show now the comparison with the MAREDAT data on diatoms, coccolithophores, Phaeocystis, and zooplankton biomass.

"The representation error is even larger for the comparison of PHAEO monthly mean climatology of the diatom, coccolithophores and *Phaeocystis* biomass (mgC m$^{-3}$) with monthly composites of in situ PFT biomass measurements from the MAREDAT dataset. Figure S13 shows the distribution of MAREDAT seasonal (summer and spring) composites of diatom, coccolithophores and *Phaeocystis* biomass data vs. model matchups based on monthly MAREDAT and PHAEO monthly climatology for diatoms (panel c), coccolithophores (f) and *Phaeocystis* (i). Because of the poor data coverage and large discrepancies in the represented temporal and spatial scales, differences between the model and in situ data are expected to be large. As a result, correlations between model and data PFT biomass from MAREDAT datasets are weak but significant (0.23, 0.19 and 0.54 for diatoms, coccolithophores and *Phaeocystis*, respectively). In general, the model overestimates PFT-carbon biomass in comparison with the *in situ* data. At the end, showing the quantitative estimates of the data and model agreements (MAE = {-0.38, -1.03, -0.61}, RMSE = {0.88, 1.13, 1.04}) for log-transformed biomass of diatoms, coccolithophores and

*Phaeocystis*, respectively), we still make a qualitative assessment. MAREDAT measurements are not always collocated for different PFTs, thus, it is not always possible to draw conclusions on the phytoplankton compositions. However, one, can notice, that diatoms, coccolithophores and *Phaeocystis* do co-exist in the areas along the subantarctic and polar fronts (see Figure S13, in the Supplementary Material)."

CONCLUSION: Overall, I was left wondering what science advancements or challenges this paper was providing or highlighting.
R: We improved the conclusion to make clear the science advancements provided by our study.

"… Our results support the hypothesis that introducing two size classes of diatoms in biogeochemical models is a prerequisite to simulate the observed diatom phenology and PFT distribution in general. We have also shown that the simulated biogeography of coccolithophores is not controlled by temperature itself as reported by Smith et al. (2017). It was directly explained by phosphate depleting as well as by low palatability of this PFT for grazers. This confirms our second hypothesis. Nevertheless, we found that the simulation of co-occurrence of coccolithophores and Phaeocystis required additional model developments to account for changes in assumed life stage of Phaeocystis (Popova et al., 2007; Kaufman et al., 2017) subject to iron availability (Bender et al., 2018). This parameterization of morphological shifts indeed allows for co-existence of the two types of haptophytes corroborating our third hypothesis on the dependence of Phaeocystis sp. life stages on iron availability. By considering two life stages of Phaeocystis we introduce additional differences in the traits, which along with assumed physiological parameters for coccolithophores makes coccolithophores competitive among phytoplankton of larger cell size requiring higher nutrients concentration to grow or/and among PFTs of similar size – small diatoms and Phaeocystis
solitary cells – but of higher palatability factor to be grazed. These additional differences in the traits of distinct haptophytes,
coccolithophores and Phaeocystis allows these groups to co-exist….”

INTRO: In the introduction the authors seem to switch back and forth between defining PFT as plankton or phytoplankton
functional type (see L23 and L 45 for example). The reader is left wondering whether starting at L46, they are talking about
phytoplankton or plankton. This is very confusing and I don't think brings any context to the paper.

R: We considered this issue important because of existing mismatch between grouping and "dimension of diversity" used in
different observational techniques and models. Moreover, we wanted to emphasize that focus of the study is phatoplankton
(not plankton in general). But we agree with the reviewer that we indeed added to much information on how PFTs are defined.
In the revised version of the manuscript we removed the discussion on the definition PFTs.

INTRO: Along those lines, I thought the whole introduction wasn't very helpful in describing the context and problems tackled
by this paper.

R: The introduction has been revised accordantly. Please see above the reply to your second comment.

INTRO: The intro is mostly about how people have defined phytoplankton functional types when this paper appears to be
mostly about the challenges that goes into representing phytoplankton diversity in a model.

R: The introduction has been revised accordantly and in the revised version of the manuscript we removed the discussion on
the definition PFTs. Please see above the reply to your second comment (paragraphs 3 to 6).

INTRO: The introduction also presents only a very marginal portion of the work that has been done in the modeling of
phytoplankton communities. Suggesting that this work started off with the paper from Le Quere et al. (2005) and Follows et
al. (2007) when this work had started a lot earlier than this.

R: The introduction has been revised accordantly; please see above the reply to your second comment (paragraphs 4 to 6).

INTRO: The authors present all the other models in one sentence (L58) summarizing them as only including 2-3 phyto-
plankton groups and mention one other model that has four. This really comes across as a very narrow view of the work that
has been done in this area.

R: The introduction has been revised accordantly and we removed this sentence; please see above the reply to your second
comment.

INTRO: The intro would have benefited from expanding on the work and the challenges that have already been learned from
the various models out there instead of the classification of PFTs.

R: The introduction has been revised accordantly; please see above the reply to your second comment (now paragraphs 3 to 6).

INTRO: Furthermore, the view that the Darwin model 'has the highest potential to simulate globally relevant PFTs' is again narrow minded at best especially considering that the author support this argument by saying that the Darwin allows to represent more than three and up to several thousands of phytoplankton groups (L64). The reader then finds out a few lines later (L85) that the version used here distinguishes only 6 phytoplankton groups (there are several models out there that do this) and not thousands like initially said. This brings the question of why (a) the authors need to state that this model has in fact 'the highest potential' among all models and (b) since they limit their phytoplankton groups to 6 does it really still stand as having the highest potential? The introduction should be focused more on the challenges that the modeling community has faced, the recent advances etc rather than try to convince the reader of why one model is superior to the others (without properly describing their model or the others).

R: We did not attend to emphasize the "superiority" of Darwin model (w.r.t. number of simulated PFTs), but rather used it as a good example of a model allowing to consider different aspects of phytoplankton diversity (dimensions of the diversity, Dutkiewicz et al., 2019) or differentiation among phytoplankton groups - biogeochemical role, allometric, photo-physiological and optical parameters. But we agree with the reviewer that the paragraph can be wrongly interpreted and for this reason removed it from the introduction and revised the introduction accordantly; please see above the reply to your second comment.

RESULTS and DISCUSSION: this section lack some quantitative assessment of how well the model does compared to the in situ data. Why not report RMSE, bias etc? Everything seems to be based on a few snapshots without a clear description of why authors chose those snapshots and a quantitative assessment.

R: Thank you for the comment. The RMSE and bias estimates between model and in situ HPLC-based PFT data over the period of time Aug 2002 – Apr 2012 were presented in tables 3, 4 and 5, also partitioned for the different Longhurst's provinces. We improved the text so that the readers do not miss out the Tables. The complete set of two-weekly snapshots over the period Aug 2002 – Apr 2012 was/is presented in the video supplement.

It is very hard to know what are the scientific advancements or lessons learned from this paper from the results and discussion section.

R: We substantially revised the manuscript to present the results in line with the hypotheses highlighted in the introduction and what we say about these in the conclusion. It is now much clearer summarized in the abstract.

"…We evaluated our model against an extensive synthesis of observations, including *in situ* microscopy and high-performance liquid chromatography (HPLC), and satellite derived phytoplankton dominance, PFT chlorophyll-a (Chla), and phenology metrics. To capture the regional timing of diatom blooms obtained from satellite required including both a lightly silicified diatom type and a larger and heavy silicified type in the model. To obtain the anticipated distribution of coccolithophores, including the Great Calcite Belt, required accounting for a high affinity for nutrients and an ability to escape grazing control of this PFT. The implementation of two life stages of *Phaeocystis* to simulate both solitary and colonial forms of this PFT (with switching between forms being driven by iron availability) improved the co-existence of coccolithophores and *Phaeocystis* north of the Polar Front. The dual life-stages of *Phaeocystis* allowed it to compete both with other phytoplankton of larger size and/or similar sizes. The evaluation of simulated PFTs showed significant agreement to a large set of matchups with *in situ* PFT Chl-a data derived from pigment concentrations. Satellite data provided important qualitative comparisons of PFT phenology and PFT dominance. With these newly added traits the model produced the observed >50% coccolithophore contribution to the biomass of biomineralizing PFTs in the Great Calcite Belt. The model together with the large synthesis of observations provides a clearer picture of the Southern Ocean phytoplankton community structure, and new appreciation of the traits that are likely important in setting this structure."

Supplementary material: as detailed in some of my minor comments it appears that some information in the supplementary material would have benefited to be discussed in detail (and potentially included) in the main text. Similarly, the author sometimes refers to a Figure in the paper and compares it to a figure in the supplementary material which is very hard to follow (L334).

R: The paper is kept shorter in order to be more readable and for the readers who are more interested in the technical details we provide more information at the supplement.

Minor comments:

L16: this sentence needs a reference

R: We added DeVries (2014) and Stocker et al. (2013).

"The Southern Ocean is one of the most important regions in regulating climate via the uptake of about 40% of the global oceanic anthropogenic CO2 (DeVries, 2014) and at the same time, is a region with the dynamics evidently altered by past and present climate change (Stocker et al., 2013)."

DeVries, T.: The oceanic anthropogenic CO2 sink: Storage, air-sea fluxes, and transports over the industrial era, Global Biogeochemical Cycles, 28, 631–647, https://doi.org/10.1002/2013GB004739, 2014.

Stocker, T., Qin, D., Plattner, G.-K., Tignor, M., Allen, S., Boschung, J., Nauels, A., Xia, Y., Bex, V., and Midgley, P. e.: Climate change 2013: the physical science basis: Working Group I contribution to the Fifth assessment report of the Intergovernmental Panel on Climate Change,

Cambridge University Press, Cambridge, United Kingdom and New York, NY, USA, https://doi.org/doi:10.1017/CBO9781107415324, 2013.

L20: needs a reference

R: We added Wilson et al. (2018).

"Playing distinct roles in biogeochemical cycling, PFTs may determine how and on which spatial and temporal scales the ocean mediates climate (Wilson et al., 2018)."

Wilson, J. D., Monteiro, F. M., Schmidt, D. N., Ward, B. A., and Ridgwell, A.: Linking Marine Plankton Ecosystems and
Climate: A New Modeling Approach to the Warm Early Eocene Climate, Paleoceanography and Paleoclimatology, 33, 1439–1452, https://doi.org/10.1029/2018PA003374, 2018.

L57: three no thee

R: The sentence was removed in the revised version of the manuscript.

L84: 'The version of the Darwin model used in our study simulates, among total 42 biogeochemical compartments..' change to '...among a total of 42...'.

R: Changed.

Btw what do you mean by compartments? As in variables?

R: "Compartments", "Components", "model variables", we changed to components.

W.r.t. model components: except for the reduced number of phytoplankton groups (from nine to six) as biomass and Chla, model variables are identical to those in Dutkiewicz et al. (2015).

L103-104 in the revised version we state: "Starting from this reduced with respect to the number of PFTs Dutkiewicz et al.
(2015) Darwin configuration…"

L85: earlier on this paper it said that the Darwin model had several thousands of phytoplankton groups? How did we end up with 6 only?

R: We used a version with six PFTs since it handles the PFTs we are interested in and at the same time it has an "affordable complexity" given the spatial and temporal resolution integration time period and, therefore, computational expanses.

The model Darwin can be considered as a biogeochemical model framework allowing to simulate different number of phytoplankton types given specific applications. We have removed the sentence about 1000 plankton as this is not germane to the paper.

Methods: what is the spatial resolution of the biogeochemical model? Same as the circulation model?

R: Yes, the Darwin biogeochemical model is coupled to MITgcm and running on the same grid. The spatial resolution is 18 km as mentioned in Subsection 2.1.2. We made it clearer in the revised manuscript (Lines 90-91).

L111: are the CDOM spectral slope used constant values?

R: Yes, in lines 110-111 (in the original version), we wrote that "$s_{cdom}$ is the CDOM spectral slope (the value is taken from the study by Kitidis et al., 2006), but in the revised paper we omitted the part of the light penetration in the ocean.

L128: why do the authors compared to this other model? Seems like a random comparison

R: The list of the reference to the studies using distinct temperature limitation function for different PFTs (to suppress the PFT distribution to particular latitudes) could be extended. However, in the revised manuscript, we delete the sentence.

L150: so the model was spinned up for 6 years only?

R: The model was spun up for more than 6 years. We improved the text to explicitly explain how the model initialization in 1992 was carried out, including details on the initial physical model spin up starting in 1979 (L173-187).

"The simulation includes a dynamic sea-ice model with a viscous-plastic rheology and a zero-layer thermodynamic submodel (Losch et al., 2010). Penetrating light is attenuated within sea ice with an exponential law (Taylor et al. 2013).

Initial conditions of the physical model were obtained from a short spin-up simulation initialised in January 1979 from rest and from temperature and salinity fields derived from the Polar Science Center Hydrographic Climatology (PHC) 3.0 (Steele et al., 2001). In the spin-up phase, the model is forced until the end of 1991 by 6-hourly atmospheric surface fields derived from the European Centre for Medium-Range Weather Forecasts (ECMWF) 40 year re-analysis (ERA-40) (Uppala et al., 2005). More details can be found in Losch et al. 2010 (section 3). Starting on January 1st, 1992, the model with biogeochemistry is forced until 2012 by 3-hourly atmospheric surface fields of the Japanese 55-year reanalysis (JRA55, Kobayashi et al. 2015). Initially, the model time step had to be decreased to 10 min because of the higher forcing frequency, this constraint was slowly relaxed to 20 min by January 1$^{st}$ 1996. The change in forcing also required an adjustment of some the sea-ice model parameters. The albedos for dry ice, wet ice, dry snow, and wet snow were set to 0.75, 0.71, 0.87, and 0.81, respectively; the simulation does not use the replacement pressure method (Kimmritz et al., 2017). After spinning up the biogeochemistry for six years, during which also the physical simulation adjusts to the new forcing, the years 1999 – 2012 are integrated and the period of Aug 2002 – Apr 2012 is used for analysis."

Was that enough to get stable conditions for the biogeochemistry? Did the authors check for that and if so how?

R: Yes, with these years (since January 1992 – August 2002) we obtained enough stable conditions for the biogeochemistry as we checked the model solution with respect to reaching the quasi-steady (as example see the figure below depicting the temporal evolution of PFT Chla averaged over Longhurst's biogeochemical provinces). In general, the model forgets the biogeochemical initial conditions quite fast, however it takes from 3 to 5 years for the model to adjust/adapt to the specified biogeochemical parameters.

We have also edited the text of the manuscript to state the following (L198-202):

"As in previous studies using the ecosystem model (e.g. Dutkiewicz et al. 2015; Clayton et al. 2017) the phytoplankton establish a repeating seasonal cycle after about 3 years such that we can assume a "quasi-steady state" by year 2002. Surface nutrients are also in quasi-steady state. Longer term drift in deep nutrient distributions do not significantly change the results for the rest of the period that we consider here. It is not computationally possible to reach a totally adjusted system, and the ecological questions we address in this paper do not require such adjustments."

[Figure]

Figure R1: Temporal evolution of PFT chlorophyll-a concentrations averaged over the APLR, ANTA, SANT and SSTC biogeochemical provinces (Longhurst, 1998): ("other large" – is a *Phaeo* - analogue).

L168: define Chla

R: Chla is the chlorophyll-*a* concentration, was defined earlier (L83 of the original version)

L175: why were these groups not included if they have the observations for it and the model allows to discriminate for them?

R: We decided to focus on diatoms and haptophytes because overall these are the most important phytoplankton groups in the Southern Ocean/Antarctic and there are *in situ* data for evaluation of our simulations. There were no *in situ* data of PFT-Chla derived from HPLC pigments for these PFTs apart from prokaryotes (L211-212, 221-225). The same for the satellite SynSenPFT product.

"It is worth mentioning that DPA allows also to retrieve other PFTs – like dinoflagellates, cryptophytes and green algae – however, they were not included in this referenced dataset, originally generated for the evaluation of satellite retrievals of diatoms, coccolithophores (haptophytes) and prokaryotes"

Section2.2.1: there should be a 1-sentence description of how they went from pigments to phytoplankton classification.

R: We extended the methods including the following information (L216-221): "The phytoplankton groups were derived using the Diagnostic Pigment Analysis (DPA) following Vidussi et al. (2001) and Uitz et al. (2006) and modified as in Hirata et al. (2011) and Brewin et al. (2015) and adapted to a much larger data set. Briefly, different PFTs have different and specific pigments (marker pigments, e.g. fucoxanthin – diatoms) that allow to distinguish the different phytoplankton groups. The biomass of a specific PFT can be quantified by determining the contribution of the corresponding diagnostic pigment to total phytoplankton biomass (represented by the weighting sum of the diagnostic pigment)".

Section 2.2.2: The results present some snapshot from various month and year. Why did the authors not compare just a whole climatology for 1999-2012? Or annuals? How did they decide which year to compare?

R: Originally years 2004 and 2008 (January and March) were shown because we considered them as typical with respect to the obtained dominance climatology and we wanted to demonstrate the consistency of model simulations covering the time period of interest. Several two-weekly model PFT Chla snapshots overlaid by collocated HPLC-derived estimates were shown as an example of model matchups to these observational dataset (Soppa et al. 2017) available over the time period of August 2002 – April 2012. (The statistics of all available model to HPLC-based observations matchups – compiled in three video supplements – were presented in Tables 3-5 S7-S9).

In the revised version we reiterated the presentations of the datasets used for evaluation and corresponded model outputs (its temporal and spatial representation). A new table (Table 2) is introduced.

"To assess our model results, we compare the simulations to several large *in situ* and satellite datasets, as detailed below and summarized in Table 2. Where the coverage of the observations is similar in respect to time we use our two-weekly model outputs. Where only monthly climatological or composite data (often from different time periods) are available we use monthly climatological model results for the period of 2006-2012. Where only results for specific months are available from observations we compare our output to these specific months. Table 3 contains the information about the evaluated phytoplankton groups as classified in the model and observations."

Thus, in the revised version we present the PFT dominance in the climatological context, but the agreement with in situ HPLC derived observations (Soppa et al. 2017) and diatom and coccolithophores cell counts (Smith et al. 2017) observations is assessed given collocated matchups over the time period of August 2002 – April 2012 (HPLC) and over particularly January – February 2011 and February – March 2012 (the time period of available cell counts for diatoms and coccolithophores by Smith et al., 2017) .  We have further extended the model evaluation with comparison of model PFT and zooplankton (carbon) biomass with available MAREDAT datasets. Since the MAREDAT data are presented as climatological monthly composites, this evaluation has been carried out based on model climatology (over the years 2006 – 2012).

**Table R1.1 (Table 2 in the revised manuscript): Data sets used for model evaluation.**

| Dataset | reference | PFT product | units | spatial repr. | time repr. | model output | time repr. |
|---|---|---|---|---|---|---|---|
| PHYSAT | Alvain et al. (2008) | dominance | unitless | 1°x1° | monthly climat. (1998-2006) | dominance | 2006–2012** |
| Darwin-15 | Dutkiewicz et al. (2015) | dominance | unitless | 1°x1° | monthly climatology | dominance | 2006–2012** |
| SEM | Smith et al. (2017) | dia *vs.* cocco dominance | % cell counts | in situ | Jan–Feb 2011 Feb-March 2012 | dia *vs.* cocco % C-biomass | Jan–Feb 2011 Feb–Mar 2012 |
| SynSenPFT | Losa et al. (2017) | diatom-Chla cocco-Chla | mgChla m$^{-3}$ mgChla m$^{-3}$ | 4x4 km* 4x4 km* | March 2012 March 2012 | diatom-Chla cocco-Chla | March 2012 March 2012 |
| PhytoDOAS | Bracher et al. (2017) | diatom-Chla | mgChla m$^{-3}$ | 0.5°x0.5°* | March 2012 | diatom-Chla | March 2012 |
| HPLC | Soppa et al. (2017) | diatom-Chla hapto-Chla proka-Chla | mgChla m$^{-3}$ mgChla m$^{-3}$ mgChla m$^{-3}$ | in situ in situ in situ | Aug2002 – Apr2012 Aug2002 – Apr2012 Aug2002 – Apr2012 | diatom-Chla hapto-Chla *Proch*-Chla | collocated collocated collocated |
| MAREDAT | Leblanc et al. (2012) | diatom-C | mgC m$^{-3}$ | in situ | 1933–2009 climat. | diatom-C | 2006–2012** |
|  | O'Brien et al. (2013) | cocco-C | mgC m$^{-3}$ | in situ | 1929–2008 climat. | cocco-C | 2006–2012** |
|  | Vogt et al. (2013) | *Phaeo*-C | mgC m$^{-3}$ | in situ | 1955–2009 climat. | *Phaeo*-C | 2006–2012** |
|  | Buitenhuis et al. (2012) | micro-zoo-C | mgC m$^{-3}$ | in situ | climatology | zoo-C | 2006–2012** |
|  | Moriarty et al. (2013) | mezo-zoo-C | mgC m$^{-3}$ | in situ | climatology | zoo-C | 2006–2012** |

*diatom − Chla* denotes diatom Chla; *cocco − Chla* is coccolithophore Chla; *hapto − Chla* is haptophytes Chla; *proka − Chla* is prokaryotes Chla, *Phaeo − Chla* is *Phaeocystis* Chla; *Proch − Chla* is *Prochlorococcus* Chla, extension −*C* denotes carbon biomass; *dia vs. cocco* is diatom *vs.* coccolithophores; *zoo* stands for zooplankton; *repr.* is representation; climat. is climatology.

\* the data are presented for a reduced Southern Ocean area as in Smith et al. (2017) and Losa et al. (2018).

\*\* model monthly mean climatology over the years 2006 – 2012.

L20: Did the authors look at the full seasonal cycle to conclude this or just the two months that they presented?

R: We looked at the full seasonal cycle, more precisely, a series of seasonal cycles. In figure 2 we show the PFT dominance plots for four months but in the supplement we show the whole climatology for each dataset: PHYSAT (Fig. S15), Darwin-15 (Fig. S16), REF (Fig. S17) and PHAEO (Fig. S18).

In the revised manuscript we write (L306-317):

"For complete 12 monthly mean climatologies for PFT dominance as retrieved by PHYSAT and predicted in Dutkiewicz et al. (2015) and REF experiment, the reader is referred to the Supplementary Material (Figures S15 – S17, respectively). In general, the PHYSAT Southern Ocean PFT dominance climatology (over the years 1998 – 2006) shows a strong seasonal variability of PFT compositions and contributions of PFTs to TChla (Alvain et al., 2008). From November to January south of 40˚S, the diatom contribution is higher than 50%. This high diatom contribution during the austral spring and summer is associated with large diatom blooms starting in October at lower latitudes and moving towards higher latitudes in December – January. The nano- non-silicified phytoplankton is dominating during the time period from March to October. The Southern Ocean PFT dominance obtained in Dutkiewicz et al. (2015) disagrees with PHYSAT observations: diatoms are underrepresented in comparison to PHYSAT in circumpolar Southern Ocean during January and February, while in July they are over-represented in the Atlantic section of the Subantarctic Zone which is also opposed to the observed dominance of haptophytes. Generally, the model version Dutkiewicz et al. (2015) overestimate the dominance of small non-silicified phytoplankton. These results clearly indicate deficiencies in the Dutkiewicz et al. (2015) model setup and motivated a series of Darwin-MITgcm experiments, with different model configurations with respect to assumed PFTs and their traits described by various physiological parameters…"

The seasonal cycle(s) over the time period Aug 2002 – Apr 2012 are also shown in the three added videos (Simulated distribution of diatom (small + large) chlorophyll concentration in the Southern Ocean, https://doi.org/10.5446/42871; Simulated distribution of haptophytes chlorophyll concentration in the Southern Ocean, https://doi.org/10.5446/42873; Simulated distribution of prokaryotes chlorophyll concentration in the Southern Ocean, https://doi.org/10.5446/42872).

L236: 'to the end of the considered period of time'. What period of time is that?

R: To avoid any confusion, we now write "after reaching a quasi-steady state" (L349 in revised manuscript).

L247-249: how long does it take for the large Phaeocystis to outcompete coccolithophores?

R: Four to five years.

L278: 'in the model world'. What does that mean? As in in your model?

R: Yes, we changed to "in our model".

L287: 'Similar to Figure 2, Figure 7...' how is Figure 2 similar to Figure 7? One shows all 3 methods while the other compares July and January output from the model.

R: Similar in terms of depicting the PFTs dominance. We removed the term "Similar to Figure 2" to avoid confusion.

L294: instead of referring throughout the paper to the study by smith et al (2017) refer to it as the in situ dataset. Otherwise the reader is left wondering here for example what that paper is and why we are taking the same area. The first time I read the paper I didn't make the connection that this was the in situ dataset used for comparison.

R: Thank you for the comment. We indeed referred to the Smith et al. (2017) study and results they presented given their data. We have now rephrased the sentence: "For a more precise evaluation of the PHAEO results by comparing to the study by Smith et al. (2017), we show diatom vs. coccolithophores dominance collocated in space and time with observations of Smith et al. (2017) which agrees well to their higher concentrations and dominance of diatoms in the SBDY and SACCF, while north of the Polar Front coccolithophores become more abundant (better seen in Fig. 9). …"

L324-329: this paragraph seems random, doesn't report any of the results yet seems like it should be discussed in the main paper (not appendix) since it contains some quantitative assessment of how well the model does.

R: As pointed out above (where? I think it is in the third comment), we now more clearly present the quantitative assessment of our model configuration.

L331: 'the worst statistics..' use a different word than 'the worst..'

R: We replaced by "the highest disagreement".

L348: "...which indeed might support a biochemical/physiological hypothesis on the coccolithophore distribution...". Which
hypothesis is this? Why does this come up for the first time in the conclusion section? This hypothesis wasn't mention anywhere else and it's unclear what it is referring to.

R: In the original text the hypothesis was mentioned in Sections 3.1, 3.2, 3.3. In the revised version of the manuscript we state it clearer in the Introduction.

"We used the optimized Darwin model to test three hypotheses on the factors controlling the biogeography of Southern Ocean phytoplankton groups:

– Size diversity of the diatoms (Queguiner, 2013; Tréguer et al., 2018) leads to the distribution of small diatoms ("slightly silicified and fast growing") at the lower latitudes and large diatoms ("strongly silicified and slowly growing") at higher latitudes in the Southern Ocean.

– Distribution of coccolithophores in the Great Calcite Belt is not necessarily controlled by temperature (Smith et al., 2017) but determined by the ability of this PFT to escape grazing because of their exoskeleton (Nejstgaard et al., 1997; Huskin et al., 2000; Monteiro et al., 2016), and to grow under nutrient depleted conditions (especially phosphate and iron) (Paasche, 2001; Iglesias-Rodríguez et al., 2002). These characteristics of coccolithophores would make them more competitive among other phytoplankton of larger or similar size, small diatoms and *Phaeocystis*.

– *Phaeocystis* sp. exists in two life stages, solitary cells and colonies, depending on iron availability (Bender et al., 2018). This additional difference in the traits of distinct haptophytes, coccolithophores and *Phaeocystis*, allows them to co-exist."

L351: how do you define palatability? How did you conclude this from the results presented here?

R: Palatability is one of the model parameters (see eq. 7 in the original version and table 1). When specifying the PFT traits,
we assume lower palatability for coccolithophores than for diatoms, other small PFT and *Phaeocystis* (solitary cells), as well as lower half saturation nutrient (phosphate) uptake. The results shown were obtained under these assumptions. From Figure 5 (6 in the revised version) it is seen that (L407-412) "… the simulated abundance of coccolithophores north of the Subtropical Front (STF) – where phosphate occurs in very low concentrations – is explained by the introduced high affinity of this PFT to phosphate (small half-saturation rate in $\gamma_\eta$ function) allowing coccolithophores to grow in nutrient depleted conditions.
However, in the region between the Subtropical and Subantarctic Fronts the occurrence of coccolithophores is more evidently linked to low grazing pressure on this PFT, due to its much lower palatability for zooplankton in comparison with small diatoms or *Phaeocystis* presented by single solitary cells."

L369: phytoplankton not phytoplankton
R: Corrected.

L387: "..the information from these different sources becomes closer..." how can you say it becomes closer? In what way?

R: We have deleted this expression.

L396: not only are cruises carried out close to the shelf but they are also mostly during spring and summer introducing another bias.

R: We agree and also pointed out in the manuscript the bias due to the fact the expeditions are carried out usually during spring and summer months in the Southern Ocean.

Figure 2: the first column uses the method as the title but the third column's title describes the variable instead

R: We modified the titles of panels to: PHYSAT dominance, Darwin (2015) dominance and REF PFT dominance.

---

## Author Comment (AC4) · 2 Jan 2020

**Response to the comments of reviewer 3**

**General comments:**

The paper by Losa et al. (2019) describes marine ecosystem model development in order to better represent the marine phytoplankton community in the Southern Ocean. This is a worth while end eavour, and, if done correctly, could lead to major improvements in our understanding of marine ecosystems, and global biogeochemical cycling in the high latitudes. However, unfortunately, the current manuscript has multiple severe shortcomings, that - in my view - preclude a publication in Biogeosciences in its current form. In essence, I have strong reservations about (1) the lack of a scientific purpose of the paper presented, (2) the modelling work itself, which is not following standard protocols in the field with regard to model set-up, testing and quantitative validation, and (3) the interpretation of the results as a result of point (2).

Since reviewer 1 and 2 have done an excellent job in pointing out specific shortcomings of the current work already, I would like to highlight my concerns with regard to these three general issues

We thank the reviewer for the comments on the manuscript. Our replies are presented in blue, labeled "R:" and follow each reviewer comment. (For few additional figures we refer the reviewer to our responses to reviewer 2)

1. No scientific hypothesis pursued

As becomes clear from the abstract, and later throughout the entire manuscript, no concrete scientific question is pursued by the paper. Hence, this paper is not suitable for publication in Biogeosciences in its current state. Rather, the current development work should be published in journals such as "Geoscientific Model Development," or "Ecological Modelling" instead. For a successful submission to any of these journals, however, a proper documentation of the model, a documentation of its sensitivity to parameters chosen and assumptions made, and thorough model evaluation and validation would be required.

R: The purpose of this study is to understand some of the traits that leads to observed distribution of the key phytoplankton in the Southern Ocean: the silicifying diatoms, calcifying coccolithophores, and colony-forming *Phaeocystis*. We provide a synthesis of observations, and then use a model to explore what traits need to be incorporated to allow these observed distributions. In the manuscript we provide details on model development and on the evaluation of the simulations with in situ und satellite information. The use of as much as possible observational information to constrain the model allowed to define the required differences in the PFT traits, therefore, to test the hypothesis on the factors controlling the biogeography of key

Southern Ocean phytoplankton groups.

In the manuscript we write:

"Observational information from in situ and satellite measurements was used to help to define differences in the PFT traits, to constrain the model, as well as to quantitatively evaluate the model performance to overall find a representation of the phytoplankton community in the Southern Ocean that is close to observations. We used the optimized Darwin model to test three hypotheses on the factors controlling the biogeography of Southern Ocean phytoplankton groups:"

In particular, the following three hypotheses have been tested which are also supported by literature:

- Size diversity of the diatoms (Queguiner, 2013; Tréguer et al., 2018) leads to the distribution of small diatoms ("slightly silicified and fast growing") at lower latitudes and large diatoms ("strongly silicified and slowly growing") at higher latitudes in the Southern Ocean.

– Distribution of coccolithophores in the Great Calcite Belt is not necessarily controlled by temperature (Smith et al., 2017) but determined by the ability of this PFT to escape grazing because of their exoskeleton (Nejstgaard et al., 1997; Huskin et al., 2000; Monteiro et al., 2016), and to grow under nutrient depleted conditions (especially phosphate and iron) (Paasche, 2001; Iglesias-Rodríguez et al., 2002). These characteristics of coccolithophores would make them more competitive among other phytoplankton of larger or similar size, small diatoms and *Phaeocystis* solitary cells.

– Phaeocystis sp. exists in two life stages, solitary cells and colonies, depending on iron availability (Bender et al., 2018). This additional difference in the traits of distinct haptophytes , coccolithophores and *Phaeocystis*, allows them to co-exist.

These hypotheses were formulated in Section 3.1 and 3.3. In the revised version, we state them in the introduction to make it clearer for the reader. We have also revised the manuscript following the comments of all three reviewers.

2. Model set-up characterized by substantial flaws that preclude a conclusive assessment of the main model dynamics and conclusions of this paper. It is evident in every section that a non-modeller with very little background on marine ecosystem model development, testing, and evaluation has conducted the current analysis. This becomes evident from an under-referenced introduction section, which is limited in scope and does not point out current gaps in marine ecosystem development, a poor, incomplete and flawed methods section that reveals major critical issues with the model set-up, spin-up, the initialisation and documentation of the runs conducted, and major critical issues with model stability and tuning.

R: The manuscript was revised accordantly. In the introduction we show that current biogeochemical models focusing on the SO phytoplankton community are either regional (e.g. Ross Sea, Crozet Islands), struggle to represent more different PFTs or are not well validated against available in situ/satellite datasets. The difficulties primarily originate from uncertain parameters employed in the parameterizations of e.g. phytoplankton growths and grazing that on the other hand define the differences in the phytoplankton traits.

We added the following paragraphs to the introduction (list of references can be found in the end of this document):

"Despite the recognized importance of the PFTs, global biogeochemical models struggle to represent the Southern Ocean phytoplankton community accurately. The difficulties primarily originate from uncertain parameters employed in the parametrizations of, e.g., phytoplankton growth and grazing (Anderson, 2005), that define the differences in the phytoplankton traits. On the other hand, the available observational information is still limited in the Southern Ocean to allow to properly
constrain the models.

One of the most investigated regions in the Southern Ocean is the Ross Sea, where many in situ observations on diatoms and *Phaeocystis* have been collected and inspired regional coupled ocean-sea ice-ecosystem modeling activities (Arrigo et al., 2003; Worthen and Arrigo, 2003; Kaufman et al., 2017). Several studies that include *Phaeocystis* in the list of simulated PFTs
in the frame of global coupled ocean-biogeochemical models have focused on the Southern Ocean (Lancelot et al., 2009; Wang and Moore, 2011; Le Quéré et al., 2016). These studies specified differences in (photo-)physiological parameters between diatoms and *Phaeocystis*, considering *Phaeocystis* in colony form. In a regional study (Popova et al. 2007, Crozet Islands) within the Southern Ocean, *Phaeocystis* was represented by two different life-stages: colonies and solitary cells. This approach was also successfully used by Kaufman et al. (2017) to examine the influence of climatic changes on the Ross Sea
phytoplankton.

Nevertheless, an in-depth evaluation of the model simulations of diatoms and *Phaeocystis* with PFT observations either has not been done (e.g. Lancelot et al. 2009) or has been only performed based on a sparse in situ dataset (Wang and Moore, 2011). A more complete evaluation of these PFTs was presented by Le Quéré et al. (2016) by comparing the dominance of the PFTs
to satellite-based dominance retrievals, and to a global dataset of in situ-based integrated PFT biomass within upper 200 m of Alvain et al. (2008) and (Buitenhuis et al., 2013), respectively. In general, as compared to the satellite retrievals, the dominance of diatoms and *Phaeocystis* has been overestimated by Le Quéré et al. (2016), while dominance of coccolithophores was underestimated.

Coccolithophore biogeography has recently been investigated globally by Monteiro et al. (2016), Krumhardt et al. (2017) and Krumhardt et al. (2019), and particularly for the Southern Ocean by Nissen et al. (2018). With respect to specific coccolithophore traits, the study by Krumhardt et al. (2017), Monteiro et al. (2016), as well as earlier studies by Paasche, (2001) and Iglesias-Rodríguez et al. (2002), emphasized the high nutrient affinity of the coccolithophores and high grazing protection of this PFT (Monteiro et al., 2016). Nissen et al. (2018) reported on higher grazing pressure on coccolithophores
than on diatoms. Krumhardt et al. (2019) used lower grazing pressure on coccolithophores than on diatoms and related the distribution of coccolithophores to a specific temperature function in dependence to its growth rate. However, none of these studies included *Phaeocystis* in their model simulations."

We also added more details here and in the manuscript concerning the methods to show that our work is consistent. We provide detailed information about model spin up as well as detailed description of how the used MITgcm configuration was span up since 1979 (in original version we referred the reader to the study by Tailor et al., 2013, which is a correct reference as well as the reference to the study by Losch et al. 2010). To initialise (in 1992) the biogeochemical model variables, we used the results of the study by Taylor et al. (2013) that used a similar MITgcm configuration coupled with the Regulated Ecosystem Model (REcoM, Schartau et al. 2007) to examine the mechanisms behind the phytoplankton bloom in the Antarctic seasonal ice zone. Since their REcoM-MITgcm simulations were validated for the SO and the variables involved in the cycling of N, C, Fe, Si (including inorganic and organic particular and dissolved pools) and chlorophyll-a (decoupled from carbon) are presented in both Darwin-MITgcm and REcoM-MITgcm models, we use correspondent REcoM-based model states as initial conditions for these variables. The model variables describing the P cycle were initialised given N-based variables and the Redfield N:P ratio. The REcoM-based phytoplankton and zooplankton biomasses from Taylor et al. (2013) were distributed equally between six and two Darwin PFTs and zooplankton groups, respectively.

As in previous studies using the ecosystem model (e.g. Dutkiewicz et al 2015; Clayton et al 2017) the plankton establish a repeating seasonal cycle after about 3 years such that we can assume a "quasi-steady state" by year 2002. Surface nutrients are also in quasi-steady state. Longer term drift in deep nutrient distributions do not significantly change the results for the rest of the period that we consider here. It is not computationally possible to reach a totally adjusted system, and the ecological questions we address in this paper do not require such adjustments."

W added the following paragraphs to the manuscript:
"The biogeochemical model is coupled to a global configuration of the Massachusetts Institute of Technology general circulation model (MITgcm, 2012) on a cubed-sphere grid (Adcroft et al., 2004) with a mean horizontal grid spacing of 18 km and 50 vertical levels with the resolution ranging from 10 m near the surface to 450 m deep ocean (Menemenlis et. al., 2005, Losch et al. 2010). The simulation includes a dynamic sea-ice model with a viscous-plastic rheology and a zero-layer thermodynamic submodel (Losch et al., 2010). Penetrating light is attenuated within sea ice with an exponential law (Taylor et al. 2013, Appendix A2).

Initial conditions of the physical model were obtained from a short spin-up simulation initialised in January 1979 from rest and from temperature and salinity fields derived from the Polar Science Center Hydrographic Climatology (PHC) 3.0 (Steele et al., 2001). In the spin-up phase, the model forced until the end of 1991 by 6-hourly atmospheric surface fields derived from the European Centre for Medium-Range Weather Forecasts (ECMWF) 40 year re-analysis (ERA-40) (Uppala et al., 2005). More details can be found in Losch et al. 2010 (section 3). Starting on January 1st, 1992, the model with biogeochemistry is forced until 2012 by 3-hourly atmospheric surface fields of the Japanese 55-year reanalysis (JRA55, Kobayashi et al. 2015). Initially, the model time step had to be decreased to 10 min because of the higher forcing frequency, this constraint was slowly relaxed to 20 min by January 1st 1996. The change in forcing also required an adjustment of some the sea-ice model parameters.

The albedos for dry ice, wet ice, dry snow, and wet snow were set to 0.75, 0.71, 0.87, and 0.81, respectively; the simulation does not use the replacement pressure method (Kimmritz et al., 2017). After spinning up the biogeochemistry for six years, during which also the physical simulation adjusts to the new forcing, the years 1999 – 2012 are integrated and the period of Aug 2002 – Apr 2012 is used for analysis."

The paper is further characterised by a complete lack of a suitable model evaluation that would show that the model decisions taken (e.g. two diatoms, Phaeocystis parameterisation, etc.) are sound and robust (how do your nutrient patterns look like compared to observations, what is your NPP and export, what are your zonal and vertical standing stocks of carbon and chlorophyll-a for each of the biological tracers, what is the zonal and vertical biomass of your zooplankton functional types, how sensitive is your model to each of the parameters of your new tracers, are these values and dynamics realistic?), a profoundly disorganised documentation of model equations and parameters (no units in tables, wrongly declared equations, random selection of topics presented), followed by an erratic presentation of random model results, and finally, as a consequence of the above, a questionable interpretation of the model results that documents a severe lack of understanding of the model behaviour. In my view, the senior model developers that are co-authors on this paper would need to guide and advise the lead author(s) on what is standard practice in our field.

R: We disagree with the reviewer's assessment. We have evaluated the model against the best available data. We now also include evaluation against MAREDAT. However, this data is sparse in space and time, and especially in vertical. Thus, we would be unable to perform all the metric that the reviewer suggests. However, we performed for our model even more specific evaluation. Our interests are ecological, thus we consider specifically community structure. The data we use are:
The evaluation of the PFTs was performed with the best available information:

- satellite PFT-Chla (SynSenPFT, Losa et al. 2017) and phytoplankton dominance retrievals (PHYSAT, Alvain et al. 2008) in terms of spatial/temporal distribution;
    - quantitatively using *in sit*u HPLC-based PFT-Chla (large dataset from August 2002 – April 2012, Soppa et al. 2017), diatom vs. coccolithophores dominance based on cell counts sampled in the Great Calcite Belt (during January - February 2011 and February – March 2012, Smith et al. (2017). The timing of the satellite and in situ observations was considered when choosing the time period to be shown in the results. In the revised version we also include a statistical analysis of matchups between model PFT (carbon) biomass and MAREDAT observational data (as climatological monthly composites) for diatoms, *Phaeocystis* and coccolithophores. However, these data are rather sparse (not evenly distributed over the Southern Ocean).

As about nutrients, in the original manuscript we indeed show the nutrients distribution, and also report on (visual) agreement with the Garcia et al. (2014). In revised version of the manuscript we provide quantitative estimates of the nutrient assessment (see also our reply to reviewer's specific comments).

However, prior to provide any NPP or export information and any other standing stocks, one has to evaluate model performance with respect to observed PFTs, since these estimates mentioned by the reviewer as well as "…bulk properties such as total chlorophyll" depends on the PFT distribution itself (Anderson, 2005). Thus, new primary and export productions are beyond the scope of the current study.

The total chlorophyll-a concentration, nevertheless, agreed with OC-CCI total Chla product with a correlation coefficient r = 0.67 and mean absolute error MAE = 0.21 mgChla m$^{-3}$.

Zonally averaged zooplankton biomass showing the range from 0 - 20 (mgC m$^{-3}$) was presented in Figure 4 and Figure S8 (Supplementary Material). The agreement with MAREDAT data (Moriarty and O'Brien, 2013) shown in Dutkiewicz et al. (2015) is now explicitly introduced in the revised version of the manuscript and supplementary material.

In the method section we provided all the modification made with respect to original Darwin-2015 module (Dutkiewicz et al. 2015) (some of the listed modifications the reviewer proposed to remove). The units of the biogeochemical parameters were given in the text where they were introduced and in the revised manuscript they were also added in the tables. We also added few more parameters in Table 1 as suggested by the reviewer 2. Sensitivity experiments are provided in the Supplementary Material and we thank the reviewer for finding the typo in the name of the function "Holling III" used to parameterize grazing.

We now explain the logic underpinning the presentation of our results. Figures 9 – 11 depicting selected ("random") model snapshots against *in situ* observations were presented as an example of the modeled and in situ HPLC data agreement, with the link to video materials showing 2 weekly PFT spatial distributions and available *in situ* observations over the 10 year period (August 2002 – April 2012). The statistical analysis of the model and data matchups between PHAEO PFT-Chla and HPLC derived PFT-Chla were provided in Tables 3 to 5 (and Tables S7-S9 in the Supplementary Material). To avoid repeating information presented in the tables 3-5, we do not include the three figures corresponding to the model/*in situ* comparison.

An example that will illustrate why I have strong doubts about the methodology used is found in the description of the model set-up described in the Methods section. Here, the authors mention, that

- they use a spin-up period of 6 years for the physical model, which is far too short to equilibrate even the surface layer of the Southern Ocean,- then initialise biomasses and nutrient fields with results from a model coupled to another, completely different biogeochemical model (Recom-MITgcm) without carefully validating these fields,

R: We hope that with the details on the model spin up provided above and in the revised manuscript we prove that the model spin up was properly performed (w.r.t REcoM-MITgcm results of Taylor et al. (2013) validated for the Southern Ocean). The reviewer may be more used to low resolution biogeochemistry models that are, for instance used to study carbon cycling. Such models are indeed spun up for long periods (often thousands of years) such that physics and biogeochemistry reach steady state. Such long spin ups are not possible with higher resolution models. Often initialization choices are important such the physics transitions smoothly. Similarly, when the questions are more of ecological (e.g. community structure) and not biogeochemical (air-sea fluxes of carbon), long spin ups needed to adjust the deep ocean are not required and likely not helpful since it is aimed to maintain an intermediate depth nutrient profile similar to the real ocean. Short time spin-ups (10 years) are what have been done in many of the recent Darwin model studies (Follows et al, 2007; Barton et al, 2010; Monteiro et al, 2016; Dutkiewicz et al. 2015; Clayton et al. 2017). Our tests running longer periods show that the ecological results are not significantly affected.

Barton, A. D., Dutkiewicz, S., Flierl, G., Bragg, J., Follows M. J.: Patterns of diversity in marine phytoplankton, Science, 327 (5972), 1509-1511, doi: 10.1126/science.1184961, 2010

- and finally run the coupled model for 13 years only, without spinning up the biogeochemical module first (the authors point
out that coccolithophores die out by the end of the simulation, which shows that the model isn't in a stable equilibrium yet for that target region),
R: The model experiment PHAEO (the main results we present in the manuscript) was in the quasi-steady state by the time of August 2002 – April 2012, which was used in our study (after physical model spin up since 1978 and biogeochemical module since 1992 over 10 years). Seasonal cycles of nutrients and PFT biomass are clearly present with some interannual variability
but no significant drift. (Figure R3.3 and Figure R1.1) In addition, we only show results after this spin-up period.

However, for experiment REF as well as for other sensitivity experiments (overviewed in the Supplementary Material) there were no sufficient differences between the traits assumed for coccolithophores and "other large" (or Phaeo) phytoplankton. As a result, it took longer for the model to get in a quasi-steady state and finally just one of "similar" PFTs survived (taking over
for another PFTs). Thus, in the experiment REF coccolithophores do not survive and Phaeo-analogue indeed represents haptophytes in general. In original Darwin-2015 model (Dutkiewicz et al. 2015) "other large" phytoplankton did not survive either. In this respect, we do acknowledge, that the REF simulations we showed for the year 2003/2004 still included slow drift towards dying coccolithophores (but nutrients and other PFTs did not drift).

We apologize as in the original version of this paper we presented some of the results in a confusing way. We realize that it was confusing to show REF output during this drift, though in some way we did want to make the point that the "generic other" needed to have significant different traits to enable both haptophytes to co-exist. This was the reason to add the bimorphic *Phaecystsis* traits. We wanted also to emphasize the problem of the instability of a complicated ecosystem model as a non-linear system in case of many variables and uncertain parameters.

To avoid any confusions, in the revised version of the manuscript we edited the sentence in L236 (original version) replacing "to the end of the considered period of time" by "after reaching a quasi-steady state" and only show results from both simulations for 2006-2012 when comparing PHAEO and REF.

- and present results from random months within the first and last few years of the simulation (across all figures we see patterns from: January, June, July, August, December 2003 and January, February, March 2004, February 2008, March 2012), where the biogeochemical model is not even remotely in equilibrium.

In the experiment PHAEO (the main experiment we present, evaluate and use for testing our hypothesis), the model is in quasi-steady state by the year 2002 and we no longer show results from 2003 and 2004 for the experiment REF as explained above.

As to the "random" months: these were chosen to match with the HPLC observations. As this was not clear also for the other reviewers we now show results from climatological months (2006-2012) instead. The specific model snapshots (video supplement) and matchup statistics (Tables 3 to 6) are however still presented and movies show the interannual variability and the visual matchups from year to year.

Needless to say that common practice in the field is to carefully spin up the physics for multiple decades, then couple to the biogeochemical model, initialise that model from observations (available as gridded and extrapolated products in standard netcdfformat, so no excuses here), spin-up the coupled model for another ten years or sountil the biogeochemistry does not drift anymore, and then run the model with varying forcing  and  finally quantitatively (!!!  -> Taylor diagrams, other model  evaluation metrics, observational constraints, etc.)  analyse the 5- or 10-year averages (or what-ever is appropriate to filter out inter-annual and multi-decadal variability in the target region) of the last few decades of the simulation, provided that the point of the study is to present average biogeographic patterns (well, if that was the point of the study, other approaches may hold for other scientific questions).  And needless to point out that many of the required observations for the model are actually carefully processed by and available within the team of senior co-authors. I have hardly ever seen such a botch job.

R: As we explained above we did carefully spin up the physical model (accordingly to the application). As about the biogeochemical model initialization we can add the following: in the common practice, any initialization is worth to be done from a realistic and physically-consistent state of the system. In an ideal case, it could be a state estimation obtained via observational data assimilation into the model. Indeed, initialization just by pure observations can result in so-called "initialization shock". From the other side, normally biogeochemical ocean models forget fast their state in the upper ocean. Plankton biomass and detrital matter relatively quickly reach quasi-steady state, and the model results are relatively insensitive to the initial conditions. For phytoplankton, zooplankton and detritus the biogeochemical models could be initialized by standard profiles (Losa et al. 2006). For nutrients and proper carbon cycling simulations, it is more crucial to use nutrient initial states as close to observations as possible (but consistent with the used physics). With respect to nutrients state in the interior ocean, therefore, the utilization of the Taylor et al. (2013) validated run (which was initialized with the state accounting for the World Ocean information) is a consistent way to initialize our model with a similar physical and identical grid configuration. The system is more inertial and has longer memory in the deep ocean.

In the revised version of the manuscript we provide additional information on the nutrient evaluation. For phosphate, that in our case was initialized given nitrate solution of Taylor et al. (2013) and Redfield N:P ratio, our climatology agrees well with WOA18 given the correlation coefficient 0.97 and normalized standard deviation 1.13 (one can consider these statistical criteria as inputs for Taylor diagram). We added the following information in the revised manuscript:

"In general, the simulated surface nutrient climatology agrees well with the World Ocean Atlas (Garcia et al., 2014) with correlation coefficient of 0.90 and 0.97 and normalised standard deviation of 1.27 and 1.13 for silicon and phosphate, respectively."

With respect to the observational data processed and produced in the Phytooptics group in Alfred-Wegener-Institute, for the Southern Ocean the satellite retrievals possess a number of limitations also discussed in the original version of this manuscript (see lines 392 - 403):

"In situ measurements in the Southern Ocean are sparse in space and time and only provide a fraction of the information obtained by the model. Satellite observations cover larger areas frequently but only cloud-free scenes which leads to a temporal bias in the often cloud-covered Southern Ocean. In addition, they are limited to only observe the first optical depth, which often limits the detection of the chlorophyll maximum. The development of algorithms for deriving PFT information requires a large in situ dataset with homogeneous temporal and spatial distribution. Nevertheless, scientific cruises in the Southern Ocean are often carried out close to the continents/ice shelf or in regions with high phytoplankton concentration (Figure 1). The diagnostic pigment analysis used to estimate PFTs from HPLC pigments assumes that different PFTs have different marker pigments, but it is known that they can also have pigments in common (Hirata et al. 2011). This ambiguity leads to uncertainties in the in situ database which is, on the one hand, needed as fundamental input for the algorithms of PFT retrievals and, on the other hand, used for direct comparison with model output here. Concerning spectral based methods applied to either in situ or satellite data, it is difficult to distinguish the specific absorption spectra of PFTs (e.g., coccolithophores and Phaeocystis). These and more limitations are well discussed by Sathyendranath (2014) and Bracher et al. (2017)."

One purpose of the study is to get the model set up that would allow (in future) to complement the observations either as additional information, or by simulating (if possible) the initial measurements and the retrieval algorithm itself (Dutkiewicz et al., 2017, 2018) and by assimilating directly the measurements.

Since the modifications of the biological module are documented even more poorly, there is no quantitative model evaluation, and neither is there a comprehensive documentation of carefully designed sensitivity tests that would allow us to understand the model sensitivity to the new parameterisation, it is impossible to critically evaluate the biological results of this work.

R: We provided all the modification made with respect to original Darwin-2015 (Dutkiewicz et al. 2015) module and equations in section 2.1.1. In the revised version we added few more parameters in Table 1 as suggested by reviewer 2. The sensitivity experiments have been added to the Supplementary Material.

Furthermore, the reported model instability with a high sensitivity of model results to parameter choice, as well as the disappearance of major functional groups (coccolithophores) throughout the simulation shows that this model configuration has major stability issues and a huge drift in the biological compartments with likely substantial consequences for productivity, nutrient distribution and biogeochemical cycling in this basin, and is thus unsuitable for publication at this point in time, as it clearly needs to be further tested.

R: As explained above, the model configuration used in the evaluated experiment (PHAEO) does not reveal any "huge drift" and we no longer show REF output prior to 2005.

3. Flawed analysis and interpretation of model results

Due to the above issues, any interpretation of the findings in this paper will obviously suffer from major uncertainties due to a lack of conclusive sensitivity tests and a thorough validation of the approach, and thus must remain entirely speculative. In its current form, it is impossible to say whether (1) the implementation of two diatoms is meaningful and leads to a gain in model performance (most models suffer from a nover-estimation of diatom biomass in this ocean basin), (2) the Phaeocystis module is correctly implemented, (3) the modifications of coccolithophore physiology are justified (since the author claims that they die out anyway), and (4) if, in fact, the model produces reasonable biogeography, primary production, depth patterns of biomass, and carbon standing stocks in this basin that would point at an actually improvement of the model compared to previous versions.

R: (1) We now straighten out the presentation and discussion of our results on consequences of including small diatoms, for instance by explicitly showing and discussing the diatom phenological indices in line with Chla distribution of small and large diatoms. The adjusted model reveals spatial distribution of small diatoms at lower latitudes and large diatoms at the higher latitudes of the Southern Ocean (as it was also shown in figure 3 of the original manuscript). "To capture the regional timing of diatom blooms obtained from satellite, it was required including both a lightly silicified diatom type and a larger and heavy silicified type in the model." Thus, our results support the hypothesis that introducing two size classes of diatoms in biogeochemical models is a prerequisite to simulate the observed diatom phenology and PFT distribution in general.

(2) What is the correct implementation of the "Phaeocystis module"? To consider *Phaeocystis* in two life stages was initially proposed by Popova et al. (2006), later used by Kaufman et al. (2017). We use similar approach, but with different implementation: these two *Phaeocystis* life stages are considered only as a function of iron availability as shown in the study by Bender et al. 2018; just one tracer is considered. We are motivated by this recent study by Bender et al. 2018 who reported on the role of iron "as a trigger" for colony formation and now state in the introduction that we test the hypothesis that the transition in the Phaeocystis life cycle is determined by iron availability. Becquevort et al. (2007) and Hassler & Schoemann (2009) also showed that Fe addition had an effect on the morphotype dominance (colonies vs. solitary cells) of *Phaeocystis* 340 *antarctica* with proportionally more solitary cells under low Fe conditions.)

In the manuscript we state:

"The implementation of two life stages of *Phaeocystis* to simulate both solitary and colonial forms of this PFT (with switching between forms being driven by iron availability) improved the co-existence of coccolithophores and *Phaeocystis* north of the Polar Front."

We also write that:

"In this study, we use very simplistic approach to parameterize life cycle transition of *Phaeocystis* given just one model tracer. In our model this transition is triggered only by iron variability (as reported by Bender et al. 2018), but not by light availability (as previously reported by Pererzak, 1993). Since we reported on our first trial, it is worth keeping in mind that the model is expected to be sensitive to the differences we specify for the mortality and grazing rates and iron uptake for colonial and single 350 cell stage. A careful model calibration of these parameters could further improve the model performance."

Hassler, C. S. and Schoemann, V.: Bioavailability of organically bound Fe to model phytoplankton of the Southern Ocean, Biogeosciences, 6, 2281–2296, https://doi.org/10.5194/bg-6-2281-2009, 2009.

Becquevort, S., Lancelot, C., and Schoemann, V.: The role of iron in the bacterial degradation of organic matter derived from *Phaeocystis Antarctica*, Biogeochemistry, 83, 119–135, doi:10.1007/s10533-007-9079-1, 2007.

(3) Coccolithophores do exist in the final (PHAEO) model set up presented and evaluated. The modifications of coccolithophore physiology are justified by validation with an extensive synthesis of datasets (considered in this study), our 360 previous experience and sensitivity tests. These modifications are consistent with and backed up by Nejstgaard et al. (1997), Huskin et al. (2000), Paasche (2001), Iglesias-Rodríguez et al. (2002), Losa et al. (2006), Monteiro et al. (2016), Krumhardt et al. (2017).

(4) Reasonable biogeography of the Southern Ocean PFTs has been validated with available in situ and satellite datasets. Since 365 our interest is primarily ecological, precise estimates of carbon standing stocks are beyond this study.

In figure 2, model results are presented for the month of July, i.e. austral winter, where biomass in this basin is clearly very low, as it is dark. Yet, much ado is made about "dominance patterns" of specific PFTs during that time. However, the quantification of dominance on very low background biomass values is meaningless, as we're looking at percentages of zeros, 370 essentially.

R: Chlorophyll concentrations north of the Subantarctic Front are not necessarily small in austral winter (see supplementary videos) and that is the reason for showing the results in Figure 2 for July. A similar figure panel was also presented in Dutkiewicz et al. (2015). Note that in the revised version of the manuscript Figure 2 shows the climatological values for all three datasets: PHYSAT, Dutkiewicz et al. (2015) and this study.

Furthermore, plankton biogeography in winter is very likely strongly linked to sea ice dynamics and factors not represented in current (global)models (resting spores, etc.) – whereas the authors even discuss spurious dominance patterns for areas clearly covered by ice, e.g. in the Dutkiewicz et al. (2015) set-up (which, apparently, isn't coupled to an ice model in its original configuration).

R: The authors referred to the patterns not covered by ice, we made that clearer in the text. Even though the original Darwin-2015 was not coupled to any ice model, it did have an ice mask. However, not the entire Southern Ocean is ice covered in the winter and it is informative to look at the ice-free water standing stocks at lower latitudes.

In addition, dominance patterns are compared for a random year of the simulation (2003 and 2004), where biogeography is
still reported to contain major drifts (as coccolithophores are reported to die out). This makes the claimed "improvement" of the model quality interms of phytoplankton biogeography highly questionable.

R: Now we present the climatological monthly mean PFT dominance for the period between 2006 and 2012 (see Figure R2.6 and R2.7 in our responses to reviewer 2), which confirms that the dominance we showed for the year 2003/2004 in the former version of the manuscript were typical.

Furthermore, since the MIT-gcm set-up is likely global in scale (we do not know, as this is not described), and since the Dutkiewicz et al. (2015) set-up was likely tuned at the global scale, a comparison with a regionally tuned model is just simply unfitting – a global model will never do as well as a model tuned for a specific region, and it doesn't have to. We do not know, how the current set-up was tuned (I assume the MITgcm is still run in its global set-up, and the rest of the ocean is just not
shown on the maps), and how it performs for the rest of the ocean.

R: The current coupled Darwin-MITgcm also runs globally. We agree though that the Darwin-MITgcm model configuration used in Dutkiewicz et al. (2015) was developed for global applications, and make this statement in the revised manuscript (lines 118-119). The problems we highlight and use as motivation for the present study are independent of the differences between Darwin-2015 and current configurations: 1) too early diatom bloom north of the Subantarctic Front, which results in
diatom dominance north of the Subantarctic Front; 2) the survival only of one of two similar (w.r.t size class and traits) phytoplankton types ("other large" did not survive in Dutkiewicz et al.,2015).

As a marine ecosystem modeller, I am all in favour of seeing further modelling work published that would illuminate the drivers of phytoplankton biogeography, and the respective role of competition, predation and environmental niche dynamics in shaping phytoplankton communities and associated ecosystem services. However, unfortunately, due to the poor quality of the submitted manuscript, my recommendation for this manuscript is forced to be: reject, revise and resubmit. Note to the author: The posted videos are in no way helpful for a quantitative (!) evaluation of the paper.

R: The videos were supported by three tables (Tables 3 – 5) in the main text of the manuscript and four tables (Tables S7 – S9) and three figures (Figures S9 – S11) in the supplement presenting the quantitative assessment of the model simulations against in situ observations over the period of time August 2002 – April 2012, which the reviewer seems to have misinterpreted.

**Specific comments:**

**Abstract:**

Lines 8-14: Revise. Vague and unsubstantiated claims. Be quantitative. Give numbers.

R: We did not feel that details on quantitative assessment belonged in the abstract. The abstract has been re-written to be tighter, and to focus on the novel traits added.

What scientific questions would you like to address with your model?

R: We added the overall aim of this study in the Abstract and the three hypotheses we investigate in the Introduction (L80-90).

We have rewritten the abstract as following:

"Phytoplankton in the Southern Ocean support important ecosystems and play a key role in the earth's carbon cycle, hence affecting climate. However, current global biogeochemical models struggle to reproduce the dynamics and co-existence of key phytoplankton functional types (PFTs) in this Ocean. Here we explore the traits important to allow three key PFTs (diatoms, coccolithophores and *Phaeocystis*) to have distributions, dominance and composition consistent with observations. In this study we use the Darwin biogeochemical/ecosystem model coupled to the Massachusetts Institute of Technology (MIT) general circulation model (Darwin-MITgcm). We evaluated our model against an extensive synthesis of observations, including in situ microscopy and high-performance liquid chromatography (HPLC), and satellite derived phytoplankton dominance, PFT chlorophyll-a (Chla), and phenology metrics. To capture the regional timing of diatom blooms obtained from satellite required including both a lightly silicified diatom type and a larger and heavy silicified type in the model. To obtain the anticipated distribution of coccolithophores, including the Great Calcite Belt, required accounting for a high affinity for nutrients and an ability to escape grazing control of this PFT. The implementation of two life stages of *Phaeocystis* to simulate both solitary and colonial forms of this PFT (with switching between forms being driven by iron availability) improved the co-existence of coccolithophores and *Phaeocystis* north of the Polar Front. The dual life-stages of *Phaeocystis* allowed it to compete both with other phytoplankton of larger size and/or similar sizes. The evaluation of simulated PFTs showed significant agreement to a large set of matchups with in situ PFT Chl-a data derived from pigment concentrations. Satellite data provided important qualitative comparisons of PFT phenology and PFT dominance. With these newly added traits the model produced the observed >50% coccolithophore contribution to the biomass of biomineralizing PFTs in the Great Calcite Belt. The model together with the large synthesis of observations provides a clearer picture of the Southern Ocean phytoplankton community structure, and new appreciation of the traits that are likely important in setting this structure."

**Intro:**

General: Does not identify major challenges in the field addressed by this work. Does not introduce Southern Ocean community structure and function. Poorly referenced. Lack of original references.

R: The introduction has been revised and improved substantially considering all the comments from the reviewers. We have further considered in the introduction the challenges that the model community is facing despite recent advances and that this has motivated us to adjust a coupled model to predict more realistically the dominating phytoplankton groups in the Southern Ocean.

Fairly irrelevant discussion of the multiple meanings of PFT, and the criteria in Le Quéré et al. (2005).

R: We considered this issue important because of existing mismatch between grouping and "dimension of diversity" used in different observational techniques and models. But we agree with the reviewers that we indeed added to much information on how PFTs are defined. In the revised version of the manuscript we removed the discussion on the definition PFTs.

Lines 16-23: Poorly referenced. Give evidence for each of your claims.

R: The paragraph was revised and references added to it.

"The Southern Ocean is one of the most important regions in regulating climate via the uptake of about 40% of the global oceanic anthropogenic $CO_2$ (DeVries, 2014) and at the same time, is a region with the dynamics evidently altered by past and present climate change (Stocker et al., 2013). The climatic changes in the Southern Ocean environmental conditions affect the spatial distribution of phytoplankton (Deppeler and Davidson, 2017). The phenology and dominance of different phytoplankton functional types (PFTs) sustaining the marine food web affect the diversity of higher trophic levels (Edwards and Richardson, 2004). Playing distinct roles in biogeochemical cycling, PFTs may determine how and on which spatial and temporal scales the ocean mediates climate (Wilson et al., 2018)."

Line 28-30: I disagree. Not the main planktonic calcifiers, if we trust modern estimates. The main PHYTOplanktonic calcifiers. See Buitenhuis et al. (2013) and Buitenhuis etal. (2019).

R: Corrected to "phytoplankton calcifiers".

Line 32: References for Phaeocystis contribution to biogeochemical cycles missing.

R: References added (Arrigo et al., 2000; DiTullio et al., 2000; Wang and Moore, 2011).

Lines 32-36:  Same thing.   Original references for the importance of named groups missing.

R: In the revised version of the manuscript we removed the discussion on the definition PFTs.

Lines 49-50: "..different algorithms...use various approaches..." Be precise. What is relevant in this context. Focus on the issue with carbon to chlorophyll conversions. Your model calculates biomass in carbon units and derived chlorophyll-a biomass. You validate against chlorophyll-based algorithms (e.g. PHYSAT). What are the challenges? How can DARWIN help?

R: In the model carbon and Chla are decoupled in accordance to Geider (1998). Besides, PHYSAT is not a chlorophyll-based algorithm, but spectral radiance based method. Darwin helps due to considering different aspects/several dimensions of phytoplankton diversity (Dutkiewicz et al., BSD). Nevertheless, we removed this paragraph in the revised version of the manuscript.

Line 54-55: Do not use multiple names for the same thing. One term – one meaning.

R: In this respect, we have to admit this a general situation with the terms "phytoplankton functional types" and "phytoplankton group". Nevertheless, we removed this paragraph in the revised version of the manuscript.

Line 56: No. Le Quéré and Follows were not the only researchers to initiate efforts in PFT modelling. They, as many others, contributed important material and thoughts to an ongoing discussion and effort.  Have you read Hood et al. (2006)?  Anderson et al. (2005)?

R: The introduction as revised accordantly.

Line 61: NOBM is not the only model with 4 PFTs. In fact many others do. BFM, GFDL, BEC (see e.g. Krumhardt et al. (2019)), etc.

R: The introduction as revised accordantly.

Line 63: No. DARWIN in its 2007 configuration does not contain all PFTs proposed by Le Quéré et al. (2005).

R: Agree, we referred here to Darwin (2010, 2015). Again, the introduction as revised accordantly and this sentence was removed.

Line 75 ff: What is the point of your paper? You state no scientific purpose. Model development should be published elsewhere.

R: Please, check our reply to your comment 2.

**Methods:**

General: Poor and disorganised. Documents strong lack of understanding of model structure and functioning.

R: For the model description and detailed equations and parameterizations the reader is referred to the study by Dutkiewicz et al (2015). In the current study, we introduce the modifications and changes carried out for our study (relative to their configuration) and list the parameterizations and parameters specifying the differences in the PFT traits. In the revised version we include few more parameters as suggested by reviewer 2 (the maximum quantum yield of carbon fixation; the spectrally averaged phytoplankton-specific light absorption and mortality rate). We also edited section 2.1.1 explaining how we introduce and parameterized the two different life stages for the *Phaeocystis* (L157-168).

Lines 85-876: Use consistent nomenclature to designate the PFTs included in the model. Your statement here lists other groups than those, e.g. represented in Figure 2.

R: Indeed, the grouping is done differently in the observations (PHYSAT, PhytoDOAS, HPLC-Chla), original Darwin-2015 model and current version. It is why we introduced Table 2 and, in the introduction, put lots of attention on the differences in the nomenclature used in modeling and satellite and in situ observation.

Lines 90-100: You must show all parameters and all equations for novel tracers. Need to document how diatoms differ, justify coccolithophore modifications, give equations for full Phaeocystis module. Justify each and every parameter, back up with literature values. You also need to show the results of your sensitivity analysis. Any deviation from the conventional PFT equation structure, i.e. the inclusion of life stages needs to be carefully motivated and evaluated. Get Phaeocystis data for the

Southern Ocean. Evaluate fraction of biomass in colonial versus single-cell stage. Yes, there is data.

R: All parameters are shown in Table 1. We do not change equations *per se* except for the modification that are listed in 2.1.1, Equations 10 and 11. The detailed literature parameter values for photophysiological parameters are presented in section S1 (Supplementary Material). In the Supplementary Material we also provide an analysis of a number of parameters (traits) determined by specific sensitivity tests.

We did not perform any sensitivity analysis to the parameters mortality rate and grazing pressure for *Phaeocystis* solitary cell. But we expect that the model is sensitive to these parameters. We introduce a discussion on this issue in the revised version (L167-168):

"We have not performed any sensitivity experiments with respect to the new parameters. However, we expect the model to be sensitive to their specification since it will also determine the competition between *Phaeocystis* and small diatoms."

We also state this in the new subsection "Perspectives and limitations of the study" and additionally write (L532-533):

"A careful model calibration of these parameters could further improve the model performance."

The inclusion of *Phaeocystis* life stages from our side was motivated by necessity to introduce additional differences in the haptophytes traits. While in general it was introduced by previous studies by Popova et al. (2006), Kaufman et al. (2016). It
would be interesting later on to evaluate the simulated fractions of *Phaeocystis* in colonial *vs.* single-cells with available observational data. However, at this point we did not trace the particular stage of modeled *Phaeocystis* and leave this topic for future publications.

Line 100: Why did you choose to replace other large phytoplankton with Phaeocystis, and not nitrogen fixers? Nitrogen fixers
have been shown to only play a very minor role in the Southern Ocean due to their temperature limitation. On the contrary, other large phytoplankton species, such as dinoflagellates, are regularly observed in this ocean basin. This decision seems questionable.

R: We were motivated by the following reasons:

1. "Other large" did not survive in the original Darwin 2015 version.
2. Distributions of N-fixer as well as prokaryotic picophytoplankton determine the extent and abundance of small phytoplankton and coccolithophores north of the Subantarcic and Suptropical Fronts. Additionally, we wanted to maintain a reasonably good performance of the model globally. Thus, we kept N-fixer.
3. We cannot strictly state that the *Phaeocystis*-analogue considered is pure *Phaeocystis* sp., it could be also other misrepresented nano-PFTs.

We clarify in the text (L.114-120):

" …Thus, in the modified Darwin version the following six PFTs are considered: large and small diatoms, *Phaeocystis* and coccolithophores, *Proclorococcus*-like and N-fixers. Although later two PFTs only play a very minor role in the Southern Ocean, their distributions determine the extent and abundance of small phytoplankton and coccolithophores north of the
Subantarcic and Suptropical Fronts. Hence, we keep N-fixer and *Proclorococcus*-like prokarytes in our version (it would also allow to maintain a reasonably good performance of the model globally). *Phaeocystis* are considered as adjusted (with respect to the traits) "other large", since "other large" did not survive in the original (Dutkiewicz et al., 2015) version that was developed for the global ocean. However, we cannot strictly state that the *Phaeocystis*-analogue considered is pure *Phaeocystis sp.*, it could be also other misrepresented nano-PFTs."

Line 105ff: The light module is completely irrelevant in this context. Move to appendix or omit.

R: Moved to the Supplementary Material.

Line 109; Same holds for kCDOM. You don't ever discuss this. Omit.

R: This light module, as well as correspondent kCDOM, was introduced as differences from the original Dutkiewicz et al. (2015) Darwin configuration.

Table 1: No units given. No references included.

R: The units were provided in the text in the revised version of the manuscript and also in the table.

What is mfunc? It does not appear in equations (3) – (6).

R: "mfunc" was/is determined in line 115. We explain it now in the table footnote "as a biomineralizing function".

Use consistent abbreviations and names for all PFTs throughout entire paper.

R: Sometimes it was not possible it is why Table 2 was introduced.

Line 118: I am 100% positive it was not Geider et al. (1998) who invented the growth functions. This parameterisation is quite similar to what Riley developed in 1946 (!). Please check your referencing.

R: With the reference to Geider et al. (1998) we refer to the set of the equations (3) – (6) accounting for decoupling between carbon and Chla.

Lines 119 – 123:  Not all parameters defined, all functions named must be given as equations. Incomplete set of equations. For example, what is f(k_sat)?

R: For the complete set of the equations the reader is referred to the study by Dutkiewicz et al. (2015). The equations (3) - (11) were provided to emphasize where the main differences between considered PFT traits were specified.

Formulation for Phaeocystis is clearly not as given in (3) – (6).

R: Indeed, it is as presented in equations (3) – (6) plus equations (10) and (11).

Line:  130:  This is  not a  Holling  II  function.

R: It is Holling III function

Also, you state that  DARWIN  has  two zooplankton types  on  line 84.  Where are  the  zooplankton  traits? Need to report.

R: The palatability parameters $r_{ij}$ also determine the zooplankton traits, for rest the reader is referred to Dutkiewicz et al. (2015).

In general, the role of zooplankton grazing pressure on plankton biogeography is not discussed in this manuscript. Since zooplankton usually play a vital role in determining the relative biomass fractions and dominance patterns in these models, the role of top-down control must be addressed else in the manuscript.

R: This crucial issue was discussed in part 3.3 (L256-262, in the revised version in L 366-376), where the reviewer "stopped detailed review".

"The discussed distribution of coccolithophores have been obtained under the assumption of lower palatability function (leading to lower grazing pressure) in comparison with what is assumed for other PFTs. This contradicts the study by Nissen et al. (2018), who reported on an increased (relative to diatoms) grazing of coccolithophores as a factor controlling the coccolithophore biogeography in the Southern Ocean. Our assumptions on low palatability factor of coccolithophores are, nevertheless, backed up by studies by Nejstgaard et al. (1997), Huskin et al. (2000), Losa et al. (2006) and Monteiro et al. (2016). In the study by Losa et al. (2006) on optimised biogeochemical parameters, it was shown that coccolithophore blooms are associated with low grazing pressure. Based on laboratory experiments, Nejstgaard et al. (1997) and Huskin et al. (2000) concluded that coccolithophores (due to its "stony" structure) do not influence the microzooplankton growth. While the exact mechanisms of how this PFT uses the coccolith to protect itself against grazing is not fully understood (Monteiro et al., 2016), the ability of coccolithophores to escape grazing control has "relatively well-supported evidence" (see Monteiro et al. 2016 for review)."

And later (L274-277, in the revised version in L409-412):

"…in the region between the Subtropical and Subantarctic Fronts the occurrence of coccolithophores is more evidently linked to low grazing pressure on this PFT due to its much lower palatability for zooplankton in comparison with small diatoms or *Phaeocystis* presented by single solitary cells."

Line 140ffff: Where does this parameterisation come from. Why did you not choose to follow e.g. Schoemann et al. (2005), the most comprehensive review on Phaeocystis dynamics.

R: Agree that the study by Schoemann et al. (2005) nicely reviews the *Phaeocystis* distribution, also suggesting a specific temperature limitation function for *Phaeocystis*.

We, nevertheless, decided to follow Popova et al. (2007) and Kaufman et al. (2017) who proposed to consider *Phaeocystis* in two different life stages and recent study by Bender et al. (2018).

What temperature function did you choose, and why?

As in Dutkiewicz et al. (2015), the original study we are based on:

$\gamma_j^T = \tau_T e^{\left( A_T \left( \frac{1}{T+273.15} - \frac{1}{T_O} \right) \right)}$, given the coefficient $\tau_T = 0.8$ normalized the maximum value (unitless), the temperature coefficient $A_T = -4000K$, and the optimal temperature $T_o = 293.15K$.

We have added it the manuscript.

Do you use one or two tracers for Phaeocystis (Phaeo versus Phaeo_cell)? Are there two combined?

R: We use one tracer.

Lines 144 – 145 (original version): "Note, that in the model *Phaeocystis* is still the same variable/array, but the assumed morphology and, therefore, physiology is different for the different life stages of *Phaeocystis*".

Do you get realistic fractions of biomass in colonial stage?

R: We did not trace it explicitly for the outputs, since the priority was to get *Phaeocystis* and coccolithophores co-existing. However, we agree that this would be interesting and we will do this in future studies.

Line 144-145: Rewrite. Utterly unclear.

R: We revised as following: "Note, that in the model *Phaeocystis*, independently on the life stage – colonial phase or solitary cells – is considered as one tracer. However, the assumed morphology and, therefore, physiology (mortality rate, grazing pressure, $k_{satFe}$ ,sinking rate) differ as described above" (L165-168).

2.1.2. Physics: 6 years is not a spin-up. What are you spinning up?

R: Please see our response to general comments above. Section 2.1.2 has been revised accordantly

Are you using a global model? Documentation of model set-up is insufficient – do not just say it was "similar" to something else.

R: We are using global model. Please see our response to general comments above. Section 2.1.2 has been revised accordantly.

And, by the way, it was not similar to the model used in Taylor et al. (2013), since these authors used a completely different coupled GCM-biogeochemical model.

R: The study by Taylor et al. (2013) used the same MITgcm sea ice – ocean model configuration (except for external forcing and some of the sea ice model parameters). The similarity in the configuration also includes the initialization (1992) of physical state based on the model spin up from the year 1979. Please see our response to general comments above.

What is the temporal resolution of your simulation?

R: The time step of model integration is 20 min (L181-183):

"Initially, the model time step had to be decreased to 10 min because of the higher forcing frequency. This constraint was slowly relaxed to 20 min by January 1st, 1996."

The model output is stored as two-weekly model snapshots. We clarify this in the manuscript (L227-228): "Two-weekly PHAEO model snapshots from August 2002 to April 2012"

Why do I care about light penetration, when you haven't even described the ice module?

Which ice-model do you use? Is it dynamic?

R: We have to care about the light penetration since it is one of the factors controlling the phytoplankton dynamics (particularly, in the marginal ice zone). In the revised version we explicitly provide information about the dynamical sea-ice model used.

Please see our response to general comments above.

**2.1.3 Biogeochemical tracer initialization**

Botch job. Really. Initialise from observations, not from unvalidated model runs con-ducted with a completely different model. These observations are generated in your group.

R: Initialization with the satellite observation: the satellite data provide only surface information, interpolation to the model grid will introduce representation errors even for the surface layer since the satellite values are provided for the first optical depth which varies in accordance to the variation of the optical constituents. Satellite Chla is available only for haptophytes (problem with distinguishing between Phaeo and coccolithophores), diatoms, and prokaryotes.

Spin up the biogeochemistry. Evaluate (quantitatively) your NPP, export, nutrient fields, total chlorophyll-a before you even start thinking about a reference run. Spinup your biogeochemistry module.

R: The biogeochemistry had been spun up over the period 1992 – 1998 and, further, 1998 – 2002. Evaluation of nutrients fields had been performed, the nutrients were validated and the results are presented (above and in the revised version). As about evaluation of NPP, export production, we would state opposite: before estimating NPP, export production or any other standing stock, one has to validate PFTs (Anderson, 2005).

(Nevertheless, the model climatological TChla agreed with OC-CCI TChla given correlation coefficient 0.67, mean absolute error 0.21 mgChla/m$^3$, rmse = 0.4 mgChla/m$^3$ (0.33 in log10 scales) and normalized standard deviation 0.69)

Initialisation of biomasses from Recom is absolute nonsense. Below you describe your satellite validation data. Use that as an initialisation.

R: Please see our responses to the two comments above and also to the general comments. Taylor et al. (2013, which showed its evaluation) provided Chla 3D distribution consistent with the physical model dynamics and grid. Moreover, the model "forgets" quite fast the initial Chla and biomass conditions (distributed mostly in upper ocean).

Do not point to Taylor et al. (2013), as this leads me to Losch et al. (2010), and they use a different model. Use new World Ocean Atlas nutrients, for instance.

R: In the study by Losch et al. (2010) the same physical model configuration was used to provide initial conditions for 1992 after spinning up the model from 1979).

Equilibrate your model. If major Southern Ocean players show a strong in their biomass (e.g. your coccolithophores die out, as you write), then there is a major problem in your model.

R: After biogeochemical spin up via model integration from 1992 to 1998 and further, the model reaches for the experiment REF steady state in 2004 – 2005 and in 2002 for the experiment PHAEO. Please see our response to the general comments above.

**2.2.1 In situ observation**

Cryptic and poorly written. List all data sets used.

R: We have edited and extended the description of section 2.2.1 (see below)

Which units does the data have, which spatio-temporal resolution, are these surface data, or are they depth-resolved. Pointing at another study is insufficient.

R: The details are included in the revised version as written below.

How do you treat, bin, grid, quality control this data to be useful for model validation?

Do you convert any of this to biomass, in order to evaluate biomasses?

R: The surface HPLC measurements used for PFT Chla retrievals do not require any conversion to be directly compared with model PFT Chla (PFT Chla is also a prognostic model variable due to Chla decoupling following Geider et al., 1998).

What about the vertical pattern?

R: The in situ dataset are taken at the surface, within the first 12 m.

What about the comprehensiveness of this data – how many months, years, seasons, latitude bands, depth levels does your data cover (all this needs to be described in the main text).

R: We added information into the section and edited this the text as following:

"A quantitative assessment of the model has been carried out using in situ observation from a large global and quality controlled dataset of in situ chlorophyll-a concentrations (mg m⁻³) of diatoms, haptophytes and prokaryotes derived from high precision liquid chromatography (HPLC) phytoplankton pigments (Soppa et al. 2017, https://doi.pangaea.de/10.1594/PANGAEA.875879). The dataset is composed **surface (first 12 m)** measurements collected by different expeditions in the Southern Ocean (south of 30°S, see Figure 1) over the time period August 2002 – April 2012, sampled mostly during austral spring and summer months (see supplemental video materials). The phytoplankton groups were derived using the Diagnostic Pigment Analysis (DPA) following Vidussi et al. (2001) and Uitz et al. (2006) and modified as in Hirata et al. (2011) and Brewin et al. (2015) and adapted to a much larger data set. Briefly, different PFTs have different and specific pigments (marker pigments, e.g. fucoxanthin – diatoms) that allow distinguishing the PFTs. The biomass of a specific PFT can be quantified by determining the contribution of the corresponding diagnostic pigment to total phytoplankton biomass (represented by the weighted sum of the diagnostic pigments). It is worth mentioning that DPA allows also to retrieve other phytoplankton groups – like dinoflagellates, cryptophytes and green algae – however, they were not included in the referred data set originally generated for the evaluation of satellite retrievals of diatoms, coccolithophores (haptophytes) and prokaryotes. For more details on the method and data quality control of this in-situ data set, we refer the reader to the study by Losa et al. (2017, Supplementary Material, Sections 1 and 3).

Figure 1 shows the locations of the available *in situ* data in the Southern Ocean. As we can see there and in Table 2" (table 2 is a newly introduced table), "this large dataset gives us the possibility for a quantitative validation of our model results. Two weekly PHAEO model snapshots from August 2002 to April 2012 have been collocated against in situ HPLC-based Chla observations, if available, within a time window +-1 week. We compare the simulated Chla of diatoms (large + small), haptophytes (coccolithophores + Phaeocystis ) and prokaryotic pico-phytoplankton against HPLC-derived Chla for diatoms, haptophytes and prokaryotes. The matchup statistics is presented for several biogeochemical provinces (Longhurst, 1998) distributed over the Southern Ocean (Figure 1): Austral Polar Province (APLR), Antarctic Province (ANTA), Subantarctic Water Ring Province (SANT), South Subtropical Convergence Province (SSTC), Humbold Current Coastal Province (CHIL), Southwest Atlantic Shelves Province (FKLD), Eastern Africa Coastal Province (EAFR), Australia-Indonesia Coastal Province (AUSW), East Australian Coastal Province (AUSE). In the Supplementary Material (Figure S12) we also present the distribution of the HPLC-derived Chla dataset (Soppa et al., 2017) as seasonal climatological PFT composites."

Figure R3.1 (Figure S12 in the Supplementary Material) depicts distribution of the in situ HPLC based Chla (HPLC-Chla) for diatoms, haptophytes and prokaryotes (Soppa et al., 2017) available over the period of time August 2002 – April 2012 and composed for different seasons.

[Figure]

[Figure]

**Figure R3.1. Distribution of seasonal composites of HLPC-Chla (Soppa et al.2017) for diatoms, hyptophytes and prokaryotes. Black counters represent Southern Ocean fronts (as white contours in Figure 1 of the manuscript).**

What do the "measurements" by Smith et al. 2017 comprise.

R: We write "Predicted PFTs were additionally compared to diatom and coccolithophores measurements (as cell counts) reported by Smith et al. (2017). These data were obtained by scanning electron microscopy in the North Atlantic and Indian Ocean sections of the Southern Ocean during the time period of January – February 2011 and February – March 2012."
Additionally, for qualitative assessment of the agreement of the simulated diatom and coccolithophore distributions to these data, we provide now estimates of the diatom *vs.* coccolithophores dominance (Figure R3.2) to compare to the similar presentation of estimates by Smith et al. (2017).

[Figure]

[Figure]

**Figure R3.2: PHAEO Diatom *vs.* coccolithophores dominance averaged over January-February 2011 (a) February - March 2012 (b). The size of the circles is relative to phytoplankton carbon content (mmolC/m³). The largest size of the circle corresponds to 3.12 (mmolC/ m³). (Figure 5 in the revised version of the manuscript)**

Line 173: Table 2, in fact, does not contain any useful information.

R: Table 2 contains the information about the evaluated phytoplankton groups as classified in the model and from observations, which based on different phytoplankton grouping.

Figure 1: What kind of observations do you show? Cite the appropriate reference.

R: Figure 1 depicts the locations of HPLC measurements collected over the period of time August 2002 – April 2012 (Soppa et al. 2017). The figure caption has been extended accordingly in the revised version of the manuscript.

Why do you show the Longhurst provinces? These are of no relevance in the rest of the paper. They only confuse the reader. Else aggregate and evaluate your model based on these provinces throughout the entire manuscript.

R: The Longhurst provinces are depicted in this figure, since quantitative assessment of the model simulated PFT Chla against in situ HPLC based estimates were performed for each of these provinces (Tables 3 – 5 and Tables S7 – S9).

**2.2.2 Remote sensing**

Why would you use the old 2008 PHYSAT product? There is an updated algorithm using a neural network approach. This should be far better than this outdated version.

R: We use the PHYSAT version that is freely available. We tried to contact Severine Alvain several times but we received no response and therefore could not obtain the neural network results.

What type of "abundance" are you referring to, and how can this be compared to the model output?

R: Abundance-based approach is the term used by the ocean colour community to classify algorithms derived to estimate either PFTs or phytoplankton size classes based on observed relationships between some measure of abundance of phytoplankton and their type or size structure (IOCCG 2014). The one we mention in the paper is the OC-PFT of Hirata et al. (2011) and it gives the information of the Chla in mg/m3 of each PFT. Note that we did not compare the OC-PFT product directly to our simulations but we used the SynSenPFT product which combines the information of OC-PFT and PhytoDOAS (spectral-based algorithm).

This part of the manuscript was revised to make it clearer for the reader:

"Model results are compared to phytoplankton dominating groups from the climatological monthly mean satellite derived product PHYSAT (1998-2006, Alvain et al., 2008). PHYSAT is based on the analysis of normalized water-leaving radiance anomalies, computed after removing the impact of chlorophyll-a variations. Specific water-leaving radiance spectra anomalies (in terms of spectral shapes and amplitudes) have been empirically associated to the presence of dominant phytoplankton groups, based on in situ diagnostic pigment observations. This product is based on the multispectral Sea-Viewing Wide Field-of-View Sensor (SeaWiFS) information and available in http://log.cnrs.fr/Physat-2?lang=fr.

We also evaluated the model simulations (mg m$^{-3}$) against the satellite PFT Chla (mg m$^{-3}$) product SynSenPFT (Losa et al. 2017, https://doi.org/10.1594/PANGAEA.875873). The SynSenPFT product combines the information of two satellite PFT Chla products: one retrieved with the differential optical absorption spectroscopy method (PhytoDOAS, Bracher et al. 2009; Sadeghi et al. 2012) applied to hyperspectral information from the Scanning Imaging Absorption Spectrometer for Atmospheric Chartography (SCIAMACHY, Bracher et al. 2017; https://doi.org/10.1594/ PANGAEA.870486) and the OC-PFT abundance-based approach (Hirata et al. 2011 and refined in Losa et al. 2017) applied to multi-spectral satellite total Chla data from the Ocean Colour Climate Change Initiative (OC-CCI). While the PhytoDOAS products from the SCIAMACHY sensor are only available at 0.5° spatial resolution and monthly means, OC-PFT applied to OC-CCI Chla products can be obtained daily and at 4 km resolution.

PhytoDOAS and PHYSAT satellite products are derived based on phytoplankton absorption properties captured by the satellite sensors and distinguished by the retrieval algorithms either as a particular PFT optical imprint ("finger print") in case of available hyperspectral information (in PhytoDOAS) or as anomalies in a multispectral signal (in PHYSAT). Thus, the PhytoDOAS allows to retrieve quantitatively major PFTs (coccolithophores, diatoms, cyanobacteria), while PHYSAT provides information about five dominant phytoplankton groups: prokaryotes (presented by *Prochloroccocus* and *Synechococcus*-like SCL), diatoms, haptophytes in general and *Phaeocystis* in particular.

We compare model climatology of Southern Ocean PFT dominance (averaged over the years 2006 – 2012) to the PHYSAT PFT dominance (dominance of the modeled PFT is defined if its Chla fraction is more than 55% of the total Chla). In line with the evaluation against the PHYSAT PFT dominance, the simulated PFT dominance are compared to similar estimates obtained in the study by Dutkiewicz et al. (2015). Two SynSenPFT products (at 4 km and daily) – diatoms  Chla that combines diatoms Chla from PhytoDOAS and OC-PFT, and coccolithophores Chla that combines coccolithophores Chla from PhytoDOAS with haptophytes Chla from OC-PFT – are   used in addition to the in situ based diatom vs. coccolithophores dominance by Smith et al. (2017). Hence, we only use the same areas and time period as in their study for comparisons to the SynSenPFT results. Here as well the comparison is qualitative as the SynSenPFT products are mostly based on OC-PFT in our study region and the global relationships between Chla and the fraction of PFTs from the OC-PFT algorithm might differ in the Southern Ocean, as shown by Soppa et al. (2014) for diatoms."

You mention this data is high resolution – did you regrid the data to fit the model grid?

R: If the reviewer refers to SynSenPFT product, the data were gridded only for the visualization since, so far, being used for qualitative assessment because of product limitations as discussed in lines 392 – 403.

**Results & Discussion**

General:  This section is characterised by a complete chaos in terms of information presented, spatio-temporal scales shown. None of the figures are in any way useful to evaluate the quality of the model development you presented in your methods section.

R: Below we provide our motivation/explain why our results are presented in the way that the reviewer criticizes and what has been changed in the revised version.

All you show is colourful surface ocean maps, and most of these maps are useless. Redo entire section from scratch. Does not contain any discussion, as there is no quantitative evaluation and interpretation of the work.

R: The colourful figures (as well as video supplementary) were presented for qualitative assessment or to support the discussion on the hypothesis. Quantitative evaluation was also presented in several tables and figures. We now extend the discussion on quantitative assessment previously provided (Tables 3-5, S7-S9) and additionally performed. As a summary on the model evaluation performed is presented below.

The evaluation of the coupled model skill with respect to predicted PFT Chla was performed given *in situ* HPLC-based Chla retrievals for diatoms, haptophytes and prokaryotes (2166, 2388 and 1425 matchups, respectively) over the time period of August 2002 – April 2012 (Soppa et al. 2017). Quantitative assessment of the agreement between model and data were/are provided for several biogeochemical provinces (Tables 3 – 5). Three additional tables and three figures depicting probability density of the predicted and observed PFT-Chla and their differences were provided in the Supplementary Material (Tables S7-S9, Figures S9 - S11).

Qualitative assessment for simulated PFTs was possible for:

1) simulated Southern Ocean PFT dominance when comparing to the PHYSAT PFT dominance climatological data product (Alvain et al. 2018);

2) diatoms vs. coccolithophores dominance in the Great Calcite Belt (compared with in situ cell counts by Smith et al. 2017) for January – February 2011 and February – March 2012;

3) satellite data based PhytoDOAS coccolithophore fit (Losa et al. 2018) and SynSenPFT Chla results for diatoms and haptophytes(coccolithophores) over the same area and time period as shown in Smith et al. (2017);

4) Southern Ocean Diatom phenological indexes (we additionally present here for a typical year 2007/2008) compared to Soppa et al. (2016).

"We chose this particular year because: 1) with the two-weekly model output the phenological indices can be more precisely calculated than based on the two-weekly or monthly mean climatology; 2) it is a typical year over the period 2006 – 2012 with respect to the simulated PFT distribution (after model reached the quasi-steady state) and climate oscillations (Soppa et al., 2016)."

The spatial distributions of the PHAEO simulated March 2004 nutrients were shown in Figure 5 (compared, agreed visually, with World Ocean Atlas (WOA) climatological March, Garcia et al. 2014). These nicely spatially resolved nutrients distributions along with PFTs distributions were shown to support the discussion on the factor controlling the Southern Ocean Biogeography. For this purpose, we did want to see fine structures of these distributions, thus we do not performed larger temporal scale (over several years) averaging. We now provide quantitative assessment of the model – data nutrients agreement in terms of correlation and normalized standard deviation based on model climatological means (averaged over the years 2006 – 2012) and WOA data. Temporal evolution of the simulated surface phosphate, silica and iron averaged over several biogeochemical provinces are presented in figure R3.5.

Simulated zooplankton biomass was shown in Figure 4 and Figure S8 to be compared with MAREDAT (Moriarty and O'Brien, 2013) data also depicted in Dutkiewicz et al. (2015, their Figure 12, panel i). We now explicitly add quantitative assessment of the agreement between the model and MAREDAT observations with statistical analysis of model zooplankton climatology matchups with climatological monthly composites of the MAREDAT micro- and mesozooplankton (Figures R2.2, responses to reviewer 2; in the Supplementary Material, Figure S14).

PIC:POC we only discuss in a context of a qualitative agreement with previously reported studies (Balch et al., 2005, 2016) and only in the context of importance of distinguishing among haptophytes (Figures R3.7).

Line 205: You do not discuss phenology in this section.

R: The term "phenology" is used here as PFT Chla dynamics in general, but the reviewer is right and we removed the term phenology and now call this section as "Diversity within diatoms".

Lines 207: The satellite estimate you are comparing your results against is not "the truth", merely another algorithm. Be sure to correct.

R: We have never stated that the satellite estimate is "the truth" but an additional information with possible limitations and uncertainties (see lines 392 – 403 of the original version).

Why, on Earth, would you compare winter values for the Southern Ocean? It's dark in winter, and background biomass is likely very challenging to model, as it will depend on features not included in the current generation of ecosystem models, such as resting spores, overwintering strategies, ice associations, etc. Hence, your entire dominance analysis is severely flawed, see detailed comment above. Why don't you evaluate e.g. a 5-year seasonal average over December – February period (most commonly used)?

R: The months shown were chosen in consistency to the PFT dominance presented in original Dutkiewicz et al. (2015). In the revised version we show dominance plots in climatological context (given these climatological maps, it is seen, however, that the year 2003/2004 shown in the original version is representative). In the supplementary material (Figures S15-S18), we now show climatological monthly mean of the PFT dominance as retrieved PHYSAT (1998-2006) and obtained in Darwin-2015 (Dutkiewicz et al. 2015), REF (2006-2012) and PHAEO (2006-2012). Please see these figures included in our responses to reviewer 2 (R2.4 – R2.7). In the manuscript we show monthly climatology for December, January and February (Figure R3.3). However, we would like to show PFT dominance for July since it appeared a kind of diagnostic month indicating whether the model has deficiencies in reproducing observed phytoplankton dynamics. We agree that current biogeochemical models still underrepresent some processes or phenomena. One of these phenomena is the size diversity within diatoms. After evaluation of our adjusted model against the extensive synthesis of observations (including *in situ* SEM and HPLC, and satellite derived phytoplankton dominance, PFT-Chla, and phenology metrics) we confirm that including both a lightly silicified diatom type and a larger and heavy silicified type is required to capture the regional timing of observed diatom blooms and, therefore, seasonality of the observed phytoplankton composition.

[Figure]

| mixed | *Phaeo.* | diatoms | SCL | *Prochl.* | hapto | small euk |

**Figure R3.3. Climatology of surface PFT dominance retrieved by PHYSAT algorithm (1998-2006, upper), simulated with the Darwin-MITgcm version of Dutkiewicz et al. (2015) (middle) and the current model set up REF (bottom) (2006 - 2012). "SCL" represents Synechococcus-like prokaryotic phytoplankton (not considered in the current model version). Simulated haptophytes include coccolithophores and *Phaeocystis*. Model PFT is considered dominant if its Chla fraction of total Chla is more than 55%. The model output (REF) is masked by the area with sea ice concentration > 75% during respective month. Darwin-15 is masked by PHYSAT missing values. White contours denote the Southern Ocean fronts (Orsi et al., 1995; Orsi and Harris, 2001) as in Figure 1**

Lines 219: This comparison appears flawed. As far as I know, Dutkiewicz et al. (2015) was tuned for a global fit.

R: The current version also runs globally, moreover, the problem discussed is independent from that whether the model is tuned globally or regionally (see our response above).

Figure 2: Names in legend do not match those given for your PFTs elsewhere in the manuscript. Do not show results for random months and years.

R: The names in legend are given as in Alvain et al. (2008). The names "diatom" and "hapto" match our diatom and haptophytes. We extend the figure capture to better clarify the correspondence of legend names to Darwin-2015 and the current Darwin version (please also see Table 2).

"Climatology of surface PFT dominance retrieved by PHYSAT algorithm (Alvain et al. 2008) (left), simulated with the Darwin-MITgcm version of Dutkiewicz et al. (2015) (middle) and current model set up REF (right) for 2006 – 2012. "SCL" represents *Synechococcus*-like prokaryotic phytoplankton (retrieved by PHYSAT, but not considered in current model version); "*Procl"* denotes *Prochlorococcus*, "small euk" is other small eukaryotes. Simulated haptophytes (hapto) include coccolithophores and *Phaeocystis*. Model PFT is considered dominant if its Chla fraction of total Chla is more than 55%."

Lines 213-2014: This must be included in the main text, not the supplementary. Along with a thorough discussion of the parameter choices in the methods section.

R: We opted to have the detailed protocol of the experiments as supplement since it would make the manuscript extremely long. Moreover, today it is an established practice to use supplementary (including videos) to provide additional information along with the details the study is focusing on in the main manuscript. It is common practice that the supplementary materials are read as well.

Lines 220 – 223: This, if true, would deserve a paper on its own. Unfortunately, you do neither quantify nor show phonological patterns. All we see in figure 2 is dominance plots for two selected months in random years.

R: We indeed have performed the analysis and we planned a separate paper focusing on the analysis of phenological indices. As mentioned above, we now compare dominance monthly mean climatology instead of PFT dominance for specific years; and show diatom phenological indices estimated for the year 2007/2008.

[Figure]

**Figure R3.4: REF diatom Chla phenological indices: bloom start date (BSD, a), chlorophyll maximum date (CMD, b), bloom end date (BED, c) (**Figure 3(a,b,c) **in ther revised version of the manuscript).**

Section 2 has been extended by additional Subsection 2.3 "Diatom phenological indices" with the following text:

"Following Soppa et al. (2016) we evaluate the diatom phenology by calculating phenological indices based on a threshold method proposed and initially applied for assessing the TChla phenology by Siegel et al. (2002). In particular, we use the following indices: the Chla maximum date, the bloom start date, and the bloom end date. These indices are calculated based on the REF Chl simulations for diatoms (including small and large) over the year 2007/2008…"

Line 225: Claimed "augmentation" hasn't been shown. What is augmented? NPP?Export? Nutrient fields? Silicification and calcification rates? Opal export? Relative biomass fractions? None of this has been shown so far.

R: Augmentation is mentioned here with respect to model set up. The model was augmented/extended by considering two size classes of diatoms and two life stages of *Phaeocystis*.

Line 228: "in agreement with" How is anything shown so far in agreement with anything. No quantitative evaluation has been provided.

R: The discussed is the spatial distribution and dominance of co-existing small diatom and coccolithophores with results reported by Signorini et al. (2006), Balch et al. (2016) and Smith et al. (2017). Qualitative comparison is still an assessment.

We, nevertheless, remove this sentence.

Line 230: "..results are supported.." How, and in which way? Quantitative comparison with data is missing (model evaluation).

R: Direct quantitative comparison with the PhytoDOAS PFT retrieved fit factor (Losa et al. 2018) is not possible simply due to different units. However, qualitative assessment of the coccolithophores distribution in the Great Calcite Belt, with respect to its co-occurrence with diatoms north of the Subantarcic Front and to the position of the northern edge of the Great Calcite Belt, is still possible.

Line 233-236: "However,...." This shows that your model is not stable, not in any kind of sensible equilibrium, with likely large consequences for all tracers associated with global biogeochemical cycles, and thus not publishable yet. Absolute game
stopper.

R: We carefully checked the model solution in different regions with respect to reaching a quasi-steady state (see figure R3.5 below). The silica is still drifting slightly, but does not affect the results. The issue we stressed here is about the biochemical

[Figure]

**Figure R3.5: Temporal evolution of monthly mean nutrient concentrations averaged over the APLR, ANTA, SANT and SSTC biogeochemical provinces (Longhurst, 1998).**

And by the way, in fig 8 you show us coccolithophore biomass for March 2012, which is "towards the end of your simulation",
and thus it looks like coccolithophores are abundantly populating the low latitude Southern Ocean, with a substantially overestimated chlorophyll-a biomass relative to the SynSenPFT estimate.

R: Figure 8 depicts results of experiment PHAEO (the experiment where both Phaeocystis and coccolithophores co-exist).

3.3. "To cope with the aforementioned chaoticity of the system...."
I stop my detailed review of this section here. All the remainder is speculation. If the model couldn't be tuned to reliably reproduce the biogeography of major players, then this paper shouldn't have been written, but the model should have been developed further.

R: Please see our response to the "first general issue".

Figure 3:  Why do you show another month?  Evaluate model results on same spatio-temporal scales throughout entire manuscript.

R: We now introduce a new table (Table 2). This table presents the datasets used to constrain the model, including information on spatial and temporal representation of these data and corresponding model output (please Table R2 in our responses to reviewer 2).

The purpose of showing Figure 3 was to, in particular, illustrate the distribution of small *vs.* large diatoms depicting also fine spatial scales (which would be vanished if presented as climatological mean). The January was chosen as a month showing more prominently (compare to the months) diatoms dominance in the Southern Ocean.

In the revised version we replace this figure with a figure depicting diatom Chla phenological indices (for the year 2007/2008, Figure R3.4) in line with the distribution of large *vs.* small diatoms for few months of this typical year (to explore this distribution within different bloom periods, Figure 3d-f: large diatom, Figure 3g-i: small diatom).

Compare total model chlorophyll-a estimate to total satellite chlorophyll-a. Compare group-specific chlorophyll-a to your
different SynSenPFT etc. algorithms. Quantitatively.

R: The quantitative evaluation with SynSenPFT was not performed because of the limitations of the SynSenPFT product for the Southern Ocean (as discussed in lines 298 – 304). The model climatological TChla agreed with OC-CCI TChla given correlation coefficient 0.67, mean absolute error 0.21 mgChla/m$^3$, rmse = 0.4 mgChla/m$^3$ (0.33 in log10 scales) and normalized standard deviation 0.69. However our prior goal is the PFT evaluation (Anderson, 2005).

As about comparison to SynSenPFT, we clarify above and in the revised version of the manuscript:

"Two SynSenPFT products (at 4 km and daily) – diatoms  Chla that combines diatoms Chla from PhytoDOAS and OC-PFT, and coccolithophores Chla that combines coccolithophores Chla from PhytoDOAS with haptophytes Chla from OC-PFT – are used in addition to the in situ based diatom *vs.* coccolithophores dominance by Smith et al. (2017). Hence, we only use the same areas and time period as in their study for comparisons to the SynSenPFT results. Here as well **the comparison is**
**qualitative** as the SynSenPFT products are mostly based on OC-PFT in our study region and the global relationships between Chla and the fraction of PFTs from the OC-PFT algorithm might differ in the Southern Ocean, as shown by Soppa et al. (2014) for diatoms."

Figure 4: Label your axes.
R: We label the axes of the revised version of the figure.

What is the total biomass level in each of these plots?

R: We do not provide the biomass level yet (since no total carbon biomass data available), instead we now show a more precise evaluation of predicted carbon biomass of diatom, coccolithophores and *Phaeosystis* against available MAREDAT datasets.

"In addition, simulations are also compared to the global MAREDAT *in situ* datasets of diatoms (Leblanc et al. 2012, https://doi.org/10.1594/PANGAEA.777384), coccolithophores (O'Brien et al. 2013, https://doi.org/10.1594/PANGAEA.785092), Phaeocystis spp. (Vogt et al. 2013, https://doi.org/10.1594/PANGAEA.779101) …These datasets are based on a data collection spanning between 55 to 75

years and are provided as climatological monthly composites."

[Figure]

[Figure]

**Figure R3.6: Climatological sesonal composites of the MAREDAT surface phytoplankton biomass for diatoms (a: for January – March, b: for October - December), coccolithophores (d: for January – March, e: for October - December) and Phaeocystis (g: for January – March, h: for October - December); scaterplot of the model vs. MAREDAT matchups based on all surface climatological monthly means: c) for diatoms; f) for coccolithophores; i) for Phaeocystis. (PHAEO model climatology is based on the years 2006 – 2012). Statistics are presented for logtransformed concentrations. Black counters represent Southern Ocean fronts (as white contours in Figure 1 of the manuscript).**

However, the coverage of the Southern Ocean PFT biomass is, indeed, very limited (see figure R3.6, same as S13).

"Figure S13 shows the distribution of MAREDAT seasonal (summer and spring) composites of diatom (panels a and b), coccolithophores (panels d and e) and *Phaeocystis* (panels g and h) biomass data vs. PHAEO monthly climatology matchups to MAREDAT monthly climatology for diatoms (panel c), coccolithophores (panel f) and *Phaeocystis* (i). Because of the poor data coverage and large discrepancies in representation temporal scales, differences between the model and data (due to the representation error) are expected to be large. As a result, correlation between model and data PFT biomass is weak but significant (0.23, 0.19 and 0.54 for diatoms, coccolithophores and *Phaeocystis*, respectively). In general, the model overestimates PFT-carbon biomass in comparison with the data. At the end, showing the quantitative estimates of the data and model agreements, we still give a qualitative assessment. Moreover, MAREDAT measurements are not always collocated for different PFTs, thus, it is not always possible to draw any conclusions on the phytoplankton compositions. However, one, can notice, that diatoms, coccolithophores and *Phaeocysts* do co-exist in the areas along the subantarctic and polar fronts."

There seem to be far too many nitrogen fixers at 40 South.

R: We agree with the reviewer, however unfortunately, for this area the observational information is rather sparse.

Same for coccolithophore contribution in this area. Are you under- or overestimating total biomass here?

R: We cannot *per se* conclude whether we over- or underestimate total carbon biomass (since there is no data available), but biomass of diatom, *Phaeocystis* and coccolithophores. In respect to MAREDAT coccolithophores data base (however, over the Southern Ocean there are not many observations, see Fig. R3.6 d,e), our climatological mean biomass (mgC/m$^3$) are higher (Fig. R.3.6 f), especially in the Atlantic Ocean Section north of the South Subtropical Convergence Province (SSTC), where the seemingly simulated Great Calcite Belt is shifted northward. So we do have overestimation for coccolithophores there. However, it is worth mentioning that the MAREDAT data error because of the conversion from cell counts to carbon biomass is several 100%.

Nevertheless, as we write in the revised version of the manuscript (L369-376):

"For a more precise evaluation of the PHAEO results with the study by Smith et al. (2017), we show diatom *vs.* coccolithophores dominance collocated in space and time with observations of Smith et al. (2017) (Figure 5). Even though our estimates have been obtained based on phytoplankton biomass (mmol C m$^{-3}$), but not on cell counts as in Smith et al. (2017), our results agree well to their higher concentrations and dominance of diatoms in the SBDY and SACCF, while north of the

Polar Front coccolithophores become more abundant (better seen in Fig. 9). As compared with Smith et al. 2017 (their figure 2), in the Atlantic section, the dominance of simulated coccolithophores (55% ) is shifted northward of the Subantarctic Front leading to underestimation of the coccolothophore dominance along the polar front and south of SAF and overestimation north of SAF."

Why have you chosen to present a zonal view of the community structure?

R: As seen, the distribution of simulated and observed PFTs reveal prominent zonal features (one can also notice zonally determined Longhurst provinces over the Southern Ocean, defined Southern Ocean Fronts, zonal gradients in hydrography). The authors find, that is a nice representative way of showing latitudinal changes in the PFT composition.

In figure 1 you show Longhurst biomes. Can you evaluate the depth pattern?

R: Figure 1 shows *in situ* surface HPLC-Chla (Soppa et al., 2017) distributed over the Longhurst's biogeochemical provinces. For these provinces the quantitative assessment of the agreement between model and HPLC-based Chla are presented. Is there any reason to evaluate the depth pattern/topography in this context or context of the manuscript? If the reviewer refers to an evaluation of PFT-Chla vertical distribution, we have to admit that it is not currently possible since the dataset (Soppa et al.,

2017) contains surface information.

What is the link between zooplankton biomass dynamics and the relative fractional contribution of individual groups to total biomass? Furthermore, Phaeocystis is not usually known to dominate biomass between 60-50S.

R: In experiment PHAEO (right panels of Figure 4) *Phaeocystis* is not the dominating biomass between 60-50S.

Figure 5: Why, again, do you show another month? Compare total chlorophyll-a to observations. Compare the groups to observational estimates from space, or in situ data.

R: Figure 5 (6 in the revised version) depicts PFT and nutrients distribution for a typical summer month February and typical year 2008 to back up the discussion on the co-existence and distribution of the Southern Ocean key phytoplankton groups and drivers of the PFT biogeography. The particular month (but not climatological one) was chosen also to show the resolved spatial scales of the discussed distribution of limiting nutrients and PFTs.

As we clarify it in the text (L394-416):

"Figure 6 depicts the Chla spatial distribution for diatoms (a), *Phaeocystis* (b) and coccolithophores (c) for February 2008 from PHAEO. We present this particular summer month of a typical year to clearly show the patterns of the depicted distribution, which could not be very obviously seen on seasonal or climatological mean maps. One can notice co-existence of simulated PHAEO diatoms and *Phaeocystis* south of the Polar Front and the co-occurrence of diatoms and coccolithophores in the Subantarctic Zone north of the Subantarctic Front. This agrees to (Smith et al., 2017) and is supported by the PhytoDOAS PFT retrievals from SCIAMACHY hyper-spectral information within the same time frame and region in Losa et al. (2018) and Smith et al. (2017).

Figure 6 presents the spatial distribution of silicon (d), dissolved iron (f) and phosphate (g) in February 2008 from PHAEO. … The spatial distribution of silicon, dissolved iron and phosphate is discussed in line with the simulated PFT Chla biogeography. The regions with high iron concentrations (in the Ross Sea, along the Western Antarctic Peninsula, around the Falkland, South Georgia and South Sandwich, Crozet and Kerquen Islands) indicate the area of Phaeocystis potential existence in colonial form. Thus Figure 6 shows that the simulated abundance of coccolithophores north of the Subtropical Front (STF) – where phosphate occurs in very low concentrations – is explained by the introduced high affinity of this PFT to phosphate (small half-saturation rate in $\gamma_{\eta_-}$ function) allowing coccolithophores to grow in nutrient depleted conditions. However, in the region between the Subtropical and Subantarctic Fronts the occurrence of coccolithophores is more evidently linked to low grazing pressure on this PFT due to its much lower palatibility for zooplankton in comparison with small diatoms or Phaeocystis presented by single solitary cells. As in the study by Smith et al. (2017) reported biogeography of observed coccolithophores in the Great Calcite Belt, our simulated coccolithophore Chla is distributed in the silica-depleted area, where small diatom cells, even if they could still compete for other nutrients, have higher palatability for grazers. Coccolithophores do not compete with small diatoms on silica resources and might survive due to its lower palatability factor. It could also be that in this area silica limited diatoms slowly grow allowing coccolithophores for earlier access to other (not used yet by diatoms) macronutrients and iron."

Showing total Chla in this respect would not help even though it agrees with OC-CCI TChla (as already mentioned above) with a correlation coefficient r = 0.67 and mean absolute error MAE = 0.21 mgChla m$^{-3}$. The comparison of model PFT Chla with in situ HPLC-Chla (Soppa et al., 2017) over the time period of August 2002 – April 2012 are shown in three supplementary video materials and tables 3-5, S7-S9.

Do not just show model nutrients. Compare quantitatively with observational estimates, e.g. from World Ocean Atlas.

R: Figure 5 (6 in the revised version) is used to support the discussion on the drivers of nano-phytoplankton distribution in the Great Calcite Belt. A quantitative assessment of climatological mean surface nutrients against the World Ocean Atlas is presented below (as correlation coefficient r and normalized standard deviation). We add this information into the text of the manuscript:

"In general, the simulated surface nutrient climatology agrees well with the World Ocean Atlas (Garcia et al., 2014) with correlation coefficient of 0.90, 0.92 and 0.97 and normalised standard deviation of 1.27, 0.67 and 1.13 for silicon, nitrate and phosphate, respectively."

Figure R3.5 (panels a, b and d, e) depicts WOA and modelled seasonal evolution of phosphate and silica averaged over the APLR, ANTA, SANT and SSTC biogeochemical provinces (Longhurst, 1998).

Figure 6: Why would you only show modeled PIC? Compare to observational PIC from space.

R: Figure 6 (7 in the revised version) was presented to emphasize the importance of distinguishing among haptophytes before estimating any carbon standing stocks (as it was suggested by the reviewer). Comparing with PIC from space model would show underestimation of PIC south of the Polar Front in the line also with the qualitative comparison with PhytoDOAS. We think that the quantitative estimates of the differences will not add anything to the conclusion. However, the results are also in line with the study by Holligan et al. (2010) concluded that current satellite algorithms may significantly overestimate PIC in cold waters of the Southern Ocean.

Holligan, P.M. Charalampopoulou, A., Hutson, R.: Seasonal distributions of the coccolithophore, *Emiliania huxleyi*, and of particulate inorganic carbon in surface waters of the Scotia Sea, *Journal of Marine Systems*, 82 (4), 195 – 205, doi: 10.1016/j.jmarsys.2010.05.007, 2010.

Where is the "Great Calcite Belt"? Do you represent it well? And if your coccolithophores die out throughout your simulation, how does this affect PIC patterns?

R: Coccolithophores do not die in experiment PHAEO and do not have any strong drift. The Great Calcite Belt can be seen from figure R.3.7 depicting PIC:POC ratio for experiment PHAEO (here for March 2008, in the revised version for February 2008).

[Figure]

**Figure R3.7: Spatial distribution of PIC, POC the ratio of the model surface particulate inorganic carbon to particulate organic carbon (PIC:POC) for experiment PHAEO in March 2008.**

does not seem to represent a "typical" year, as there is no typical year in a model with a strong drift.

R: Indeed, in experiment REF for the year 2004 PIC/POC was slightly affected by still slow drift shifting towards *Phaeosystis* (we no longer show 2003/2004 results for REF). However, it is worth mentioning that for the year 2004, as well as for the following years, REF shows unrealistically low PIC:POC ratio since this model configuration does not distinguish among haptophytes.

The results for typical year 2008 (long after the model solution reached quasi-steady state) for experiment REF are presented in figures R3.4. Please mind, that there was/is no strong drift in the experiment PHAEO that was/is finally evaluated.

[Figure]

**Figure R3.8: REF surface distribution of the POC and PIC:POC ratio for March 2008**

[Figure]

**Figure 7 in the revised manuscript: Spatial distribution of the model surface particulate inorganic carbon (PIC, mmol m$^{-3}$) for experiment PHAEO (left panel), ratio of PIC to total particulate (dead) organic carbon (PIC:POC) for experiment PHAEO (middle panel) and PIC:POC for experiment REF (right panel) in February 2008. White contours denote the Southern Ocean fronts (Orsi et al., 1995; Orsi and Harris, 2001) as in Figure 1.**

Figure 7: Same comments as Fig 2. Winter patterns unrepresentative, and very likely very tricky to model. Where is the "Great Calcite Belt"?

R: You can expect to see the "Great Calcite Belt" depicted in Figure 7 that shows dominance among haptophytes, diatoms and prokaryotes. The simulated winter PFT Chla patterns are clearly seen (but hardly being evaluated with available data) in the supplemented videos.

Fig 8: Merge with Fig. 5.

R: Figures 8 and 5 can hardly be merged since illustrate different aspects. While Figure 8 shows the PHAEO Chla distribution for diatom and coccolithophores compared to SynSenPFT retrievals in support to the discussed diatom *vs.* coccolithophores dominance as predicted by Darwin-MITgcm and observed by Smith et al. (2017), Figure 5 (6 in the revised version) is introduced to discuss, in particular, on drivers of nano-phytoplankton biogeography in the Great Calcite Belt and limitations of the satellite retrievals.

Fig. 9 - 11: These figures do not contain any quantitative information. Omit.

R: The figures are replaced by the reference to three supplemented videos (Simulated distribution of diatom (small + large) chlorophyll concentration in the Southern Ocean, https://doi.org/10.5446/42871; Simulated distribution of haptophytes chlorophyll concentration in the Southern Ocean, https://doi.org/10.5446/42873; Simulated distribution of prokaryotes chlorophyll concentration in the Southern Ocean, https://doi.org/10.5446/42872). Although this video material does not provide quantitative assessment, we still think it a valuable evaluation. It clearly shows the simulated distribution and Chla phenology of the key Southern Ocean PFTs against the observed distribution. Moreover, the statistical analysis of model to data matchups was also, performed, shown and discussed.

In the text we write:

"Although these videos only allow visual comparison, they do show that the *in situ* observations (indicated by circles) match well the model Chla of diatoms and haptophytes in the area close to the Antarctic Peninsula and in the Southwest Atlantic Shelves biogeochemical province (FKLD, Longhurst, 1998), which illustrates a good agreement between the model and observations. In the Ross Sea, however, the model performance is less accurate: our simulated Chla for Phaeocystis as haptophytes in Ross Sea are underrepresented in comparison with HPLC-derived estimates."

The shown model and HPLC-based PFT Chla matchups were/are supported by statistical analysis. The matchup statistics were/are presented in Tables 3-5 and Tables S7-S9. We have slightly restructured and edited the text so that the reader does not miss the provided quantitative assessment:

"We have obtained matchup statistics for the comparison of our PHAEO model results against the *in situ* HPLC-based PFT

Chla observations by Soppa et al. (2017). The mean absolute deviation (mean absolute error, MAE) of collocated model and *in situ* PFT-Chla over the considered time frame (August 2002 – April 2012) and the entire Southern Ocean is 0.74 mg m$^{-3}$ and 0.22 mg m$^{-3}$ for diatoms and haptophytes, respectively. Tables 4 and 5 present the statistics of model and in situ PFT-Chla comparison at several Longhurst's biogeochemical provinces (Longhurst 1998, see Figure 1). The highest disagreement was obtained for diatoms in the Atlantic Sector of the ANTA province, where the simulated diatom Chla is systematically overestimated by ~0.5 mg m$^{-3}$. The best agreement with the HPLC based diatom Chla (excluding small provinces, see Figure 1) was obtained at the SSTC and SANT. For the haptophytes, the highest systematic error towards overestimation has been found at two small provinces east of Africa and Australia (EAFR and AUSE) with the bias = 0.57, 0.48 (mg m$^{-3}$), respectively. The highest random error is (RMSE = 0.62, 0.44 mg m$^{-3}$) at EAFR and APLR. The lowest differences between predicted and observed haptophytes was at the FKLD, SSTC provinces where haptophytes are mostly presented by coccolithophores, and at the SANT biogeochemical province, where both coccolithophores and *Phaeocystis* co-exist. As additional information on the agreement between model and observations, Figures S9 and S10 in the Supplementary Material present frequency distributions of diatoms and haptophytes Chla for the simulations and measurements as well as the frequency distribution of the model and data differences. The latter shows that statistical criteria, such as MAE and root mean squared error (RMSE) give statistical meaningful metrics with respect to "model minus *in situ* Chla data" and the evaluation does not necessarily require a logarithmic transformation, as it is often done in ocean colour product validation (Brewin et al., 2010; Losa et al., 2017). With respect to the agreement between model and observed *in situ* Chla for prokaryotic pico-phytoplankton (Soppa et. al 2017) depicted in Figure S11 (Supplementary Material) one can conclude that the frequency distributions of the simulated and observed pico-phytoplankton are different, and the frequency distribution of the differences confirms that MAE and RMSE given absolute (Table 6) or logarithmically transformed values can hardly provide satisfactory estimates. Nevertheless, it is worth emphasizing that the largest differences between model and observed in situ prokaryotic pico-phytoplankton are located along the Antarctic Peninsula.

It is worth mentioning that the statistical estimates between model and observation PFT-CHla were carried out using matchups within $\pm 1$ week. Moreover, the model does not explicitly represent sea-ice algae and, therefore, might work less well in the region around the sea-ice. In this respect, we have to point out that all the statistics are presented for a qualitative assessment of the model rather than for a quantitative estimate of model uncertainties, since the representation error (Janjič et al., 2018) related to the differences in spatial and temporal scales considered and sampled by the model vs. observations as well as to the mismatch in grouping phytoplankton (Bracher et al., 2017) are quite large."

**4 Concluding remarks and outlook**

Same as above. It is impossible for me to evaluate this section. Since the modeling work does not seem to be up to the standard in the field, the model evaluation is missing and the analysis of the results is severely flawed, it is impossible for me to judge the scientific interpretation of the findings. Any conclusions based on this work must remain mere speculations at this point in time.

R: Our conclusions were drawn based on the experiment PHAEO evaluated with available in situ and satellite observations quantitative (when possible) and qualitatively. The required by the reviewer additional model evaluation is provided in the revised version but does not change the conclusion.

In a sense it is not a standard study because we look on biogeochemical modeling of phytoplankton diversity from observational measurement perspective: towards a synergy between different types of observations (given their assumptions, principles and limitations) and numerical models which have now a possibility to account for different measurement principles, different aspects of the diversity. Such models might guide the further retrieval algorithm developments that in turn would allow to get observations for constraining better the biogeochemical models, since there are still lots of uncertainties in the model parametrizations and parameters representing differences in plankton traits (e.g. phytoplankton growth and lost).